

SciPost Phys. Lect. Notes 67 (2023)

# Topological insulators and geometry of vector bundles

**Alexander S. Sergeev**

M.V. Lomonosov Moscow State University, Moscow, Russia

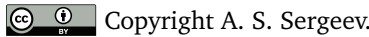

a.sergeev@physics.msu.ru

## Abstract

For a long time, band theory of solids has focused on the energy spectrum, or Hamiltonian eigenvalues. Recently, it was realized that the collection of eigenvectors also contains important physical information. The local geometry of eigenspaces determines the electric polarization, while their global twisting gives rise to the metallic surface states in topological insulators. These phenomena are central topics of the present notes. The shape of eigenspaces is also responsible for many intriguing physical analogies, which have their roots in the theory of vector bundles. We give an informal introduction to the geometry and topology of vector bundles and describe various physical models from this mathematical perspective.

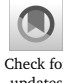

# Preface

## Why topological insulators?

Topological insulators are unique materials that do not conduct electricity in the bulk, but support metallic states on their boundary. These states have unusual dispersion relation and cannot be removed from the surface, unless the material transforms into a topologically trivial phase. There is no order parameter responsible for these properties, and topological insulators fall outside the Landau theory of phase transitions. Instead, they are characterized by topological invariants, which take integral values and remain unchanged during any variations of the Hamiltonian that do not close the bulk gap and preserve symmetry. These invariants and associated observable properties are studied by the topological band theory. More broadly, the field of topological matter includes insulators, superconductors, semimetals, metamaterials, and other systems. The theory of topological insulators was developed first and is now well-understood, which makes it a good place to start learning about topological physics.

There are several aspects that make topological phases exceptional, even among other exotic states of matter:

- *New property of matter*: Topological properties provide a new classification scheme, which gives an unexpectedly rich landscape of phases. Entries in crystallographic databases now include the topological class of a material along with its symmetry group and the value of the band gap.

- *Cross-discipline field*: Theory of topological phases has numerous links with high-energy physics and gauge theories. Examples include Dirac and Weyl fermions appearing in the momentum space of a crystal; Wilson loop operator used to diagnose topology of Bloch eigenstates; chiral anomaly in topological semimetals, and many more.

- *Accessible physics*: Topological properties of crystals are often captured by simple models based on single-particle Hamiltonians in the tight-binding approximation. This makes topological band theory a rare topic, in which understanding the physical content of recent research articles often requires little more than the undergraduate-level background in solid state theory.

- *Advanced mathematics*: In the literature on topological physics, one can encounter concepts from differential geometry, algebraic topology, theory of characteristic classes, and other fields of mathematics, which may lie well beyond the typical physicist's curriculum. In a sense, topological matter provides a physical realization of many subtle mathematical phenomena.

- *Reality*: Topological materials exist in nature. Somewhat unusual for condensed matter physics, the field is theory-driven: a research cycle often starts with a theoretical idea, which is then corroborated by *ab initio* calculations and finally is confirmed by experiments with real materials. The experimental realizations of the initial idea can use various platforms, such as metamaterials or artificial crystals made of ultracold atoms.

- *Universality*: The topological physics was discovered in the quantum Hall experiments studying disordered 2D electron gas in a strong magnetic field; then it was developed for the electronic states in perfect crystals, giving rise to the topological band theory; later these ideas were applied to other periodic systems with band structures, such as photonic and acoustic metamatrials; today, the range of topological systems is ever wider.

### Why vector bundles?

Due to the cross-discipline nature of the field, there are many learning paths leading to the topological band theory. Depending on the background, one can prefer to understand topological matter in terms of the Dirac equation or on the basis of the quantum Hall effect. In any case, the understanding will be largely based on analogies between various physical phenomena. However, often it is not immediately clear, why a given analogy appears and to what extent does it hold. In many cases, the source of the physical analogy is a mathematical construction shared by physical models. To illustrate this point, we consider an analogy between a topological insulator, the Dirac monopole, and a closed two-dimensional surface, which opens a seminal review article on topological insulators [1] and is widely used in the literature.

A paradigmatic example of a topological phase is Chern insulator. Imagine a two-dimensional crystal described by the Bloch Hamiltonian $\hat{H}_{\boldsymbol{k}}$, where $\boldsymbol{k} = (k_x, k_y)$ is the crystal momentum. Suppose that there is a single occupied band with the corresponding Bloch eigenstate $|\psi_{\boldsymbol{k}}\rangle$. Then one can compute the Berry potentials $A_j = \langle \psi_{\boldsymbol{k}} | \partial_{k_j} \psi_{\boldsymbol{k}} \rangle$ and the Berry curvature $f = \partial_{k_x} A_y - \partial_{k_y} A_x$. It turns out that the integral of the Berry curvature over the Brillouin zone is an integer multiple of $2\pi$. This integer is called the Chern number:

$$c = \frac{1}{2\pi} \int_{BZ} f \, dk_x dk_y \,.$$

A Chern insulator is characterized by a non-zero Chern number. Since $c$ is an integer, it cannot change under smooth deformations of the Hamiltonian, unless some steps in the algorithm described above become ill-defined. This happens if the bulk gap closes at some $\boldsymbol{k}$. At this point, the bands become degenerate, so one cannot uniquely prescribe an eigenstate $|\psi_{\boldsymbol{k}}\rangle$ to the point of degeneracy.

What is the physical meaning of the Chern number and why is it an integer? One way to understand this is to compare the momentum-space picture of the Chern insulator with the magnetic monopole introduced by Dirac. Suppose that a source of the magnetic field is contained inside a sphere $S^2$. Then, given the vector potential $\boldsymbol{A}$ on the sphere, one computes the field strength $\boldsymbol{B}$ and the flux through the sphere. By considering the wave function of a quantum particle moving near the source of the field, Dirac showed that the flux must be quantized in units of $\Phi_0 = \frac{h}{e}$, where $h$ is the Planck constant and $e$ is the electric charge of the particle. Thus, the number

$$n = \frac{1}{\Phi_0} \int_{S^2} \boldsymbol{B} \cdot d\boldsymbol{S} \,,$$

is an integer showing how many magnetic monopoles are contained inside the sphere $S^2$. This number will remain constant for any variations of the field configuration, provided that none of the monopoles moves through the surface of the sphere. Returning to the context of the band theory, we note that the Berry potential changes under the gauge transformations

$$|\psi_{\boldsymbol{k}}\rangle \rightarrow e^{i\beta(\boldsymbol{k})}|\psi_{\boldsymbol{k}}\rangle \,,$$

in the same way as the magnetic vector potential does. Thus, the Berry curvature is analogous to the magnetic field strength, and the Chern number plays the role of the number of magnetic monopoles inside the sphere. This allows one to interpret the Berry curvature as a "magnetic field in the momentum space", and the Dirac quantization condition explains the integrality of the Chern number. On the other hand, the character of relationship between the quantities in the two examples remains unclear.

Moreover, both the Chern insulator and the magnetic monopole share some similarities with a purely geometrical situation. Locally, a smooth two-dimensional surface can be characterized by the Gaussian curvature $\kappa$, which is computed from the radii of curvature of certain

sections of the surface at a given point. If the surface is closed and orientable, one can count the number of holes in it, or find the genus $g$ of the surface. For example, a torus $T^2$ has a single hole, so $g = 1$, and for the sphere $g = 0$. The Gauss–Bonnet theorem states that one can find the genus of the surface $\mathcal{B}$ by integrating the Gaussian curvature over the whole surface:

$$1 - g = \frac{1}{4\pi} \int_{\mathcal{B}} \kappa \, dS \, .$$

If we smoothly deform the surface, the local values of the curvature will change, but the genus will remain the same. As in the examples above, we have an integer-valued topological invariant, which is computed by integration of a local quantity. This example does not clarify the relationship between the first two; but it indicates that we are on to something even deeper. If the three formulas for topological invariants are indeed comparable, then the similarity between the Chern insulator and the Dirac monopole has in fact geometric nature.

There is a mathematical object, yet invisible, which connects all three examples: a vector bundle. These objects first appeared in physics in the context of gauge fields describing fundamental interactions. Later it was realized that vector bundles play an important role in many other physical situations. In fact, adjectives in "geometric phase" and "topological insulator" refer to certain characteristics of the shape of a vector bundle. Differential geometry studies the local shape of vector bundles, which is described in terms of connection and its curvature. This is the mathematical setting of the classical gauge theories, in which the curvature is associated with the field strength. Topology of vector bundles focuses on their global properties, which are characterized by topological invariants. These quantities are unchanged under smooth deformations and can be computed from the local geometric data. The theory of vector bundles provides the precise language for the situations in the examples above, which allows one to identify their common features and to understand the limitations of the analogies. With this language, one can put the topological band theory into a wider perspective combining geometry of surfaces studied by Gauss and certain aspects of gauge theories in high-energy physics.

Unfortunately, the theory of vector bundles is an advanced mathematical topic. Looking up the definition of vector bundle in Wikipedia [2], one finds:[1]

> A real vector bundle consists of:
>
> 1. topological spaces $\mathcal{B}$ and $M$
> 2. a continuous surjection $\pi : M \to \mathcal{B}$
> 3. for every point $p \in \mathcal{B}$, the structure of a finite-dimensional real vector space on $\pi^{-1}(p)$,
>
> where the following compatibility condition is satisfied: for every point $p$ in $\mathcal{B}$, there is an open neighborhood $U \subseteq \mathcal{B}$, a natural number $k$, and a homeomorphism
>
> $$\varphi : U \times \mathbb{R}^k \to \pi^{-1}(U),$$
>
> such that for all $p \in U$:
>
> - $(\pi \circ \varphi)(p, \boldsymbol{v}) = p$ for all vectors $\boldsymbol{v}$ in $\mathbb{R}^k$, and
> - the map $\boldsymbol{v} \mapsto \varphi(p, \boldsymbol{v})$ is a linear isomorphism between the vector spaces $\mathbb{R}^k$ and $\pi^{-1}(p)$.

Perhaps, learning the mathematics of vector bundles to understand the topological band theory would lead us too far astray! Even when studying a physicist-oriented textbook on differential geometry, it takes considerable time to familiarize oneself with the formalism. Such expositions often focus on the applications in gauge theories and General relativity. This requires discussion of differential forms, Lie groups, and principal bundles, which are not necessarily needed in the context of condensed matter physics.

---

[1]The notation is slightly altered to match with the one used in the main text.

## About these notes

Abstract mathematical definitions like the one quoted above describe the essence of an object and allow one to prove strong statements. They appear as a result of a process in which one removes unnecessary details and reduces the number of assumptions. It can also be useful to go in the opposite direction and to look at a specific example of a mathematical concept. Imagine a unit sphere $S^2 \subset \mathbb{R}^3$ in the three-dimensional space. At each point of the sphere, there is a tangent plane. The collection of all such planes is called the tangent bundle $TS^2$ of the sphere, which is an example of vector bundle. The example is, in fact, very specific: the sphere $S^2$ is a differential manifold, and not just a topological space; the geometry of the bundle is closely related to the shape of the sphere; the bundle has natural metric and connection induced from the ambient space $\mathbb{R}^3$. A mathematician might see these details as weakening of the hypothesis. For a physicist, however, not only is this picture more tangible, but also in some ways more relevant than the general definition. With a slight modification, $TS^2$ can be interpreted as a complex line bundle, that is, a collection of complex one-dimensional vector spaces. Then it can be used to illustrate a number of mathematical phenomena, which appear in physical models. Examples include covariant differentiation and curvature in gauge theories, geometric phase in quantum systems, and topological quantization in electromagnetism. All these concepts play an essential role in the theory of topological insulators. In this way, one can understand the mathematics of the topological band theory without invoking the abstract formalism. This is the motivation behind the present work. It has the following goals:

- To give an informal introduction to geometry and topology of vector bundles, and to discuss their physical applications with a primary focus on condensed matter physics.

- To present basic results of the topological band theory with an emphasis on the underlying mathematical structures.

- To treat physical and mathematical concepts on an equal footing, thus providing a stereoscopic view of the subject.

- To convey general ideas by a careful examination of simple examples.

- To provide an entry point and motivation for further studies of the topics discussed in the notes.

The text is an extended version of the lecture notes for a one-semester course taught by the author at Moscow State University. The audience of the course included beginning graduate students and advanced undergraduates. The specialties of the students varied from theoretical and experimental condensed matter physics to high-energy theory. Formal mathematical prerequisites for the course include vector analysis and basic linear algebra. In the physical part, the reader should be familiar with electromagnetism and non-relativistic quantum mechanics. Knowledge of basic solid state theory will be helpful, but is not necessary, as we will introduce all needed concepts. The text contains exercises, which are an integral part of the course. It is important to solve or at least attempt all of them before moving to the next topic. If an exercise is referenced later in the text, it contains a list of links to the corresponding exercises or sections (the latter are marked with § symbol). The **boldface** font is used to indicate the first mention of a term, accompanied by its definition.

The notes are organized as follows. In Sections 1–4, we give a brief survey of geometry and topology of vector bundles, focusing on complex line bundles. Although mathematical in style, our discussion will be very far from being rigorous. The goal is to learn just enough mathematical language to be able to spot common patterns in physical models and to construct analogies while understanding their origin and limitations. In Sections 1 and 2, we introduce

the language of vector bundles and basic notions of differential geometry, such as covariant derivative, parallel transport, and curvature of connection. Physical examples include Foucault pendulum and a quantum particle in the electromagnetic field, which are described in a unified formalism based on complex line bundles. In Sec. 3, we first use the geometric picture of the electromagnetic field to consider two systems, in which magnetic flux affects the physics in the field-free region. Then we introduce a new type of complex line bundle, which arises as a collection of eigenspaces of a Hamiltonian, and discuss the concept of geometric phase. This phase results from the parallel transport and is similar to the daily rotation angle of the Foucault pendulum. We provide a detailed dictionary between the two settings. In Sec. 4, we define the Chern number, a global topological invariant of a complex line bundle, which describes the topological quantization in the context of magnetic monopoles and provides a basic classification of topological phases of matter. We will see that these physical concepts are closely related to the fact that one cannot define a smooth, nowhere-vanishing field of tangent vectors on the sphere $S^2$. Any such field must contain singularities; we will learn how to compute the number of the singularities and will use this result to sketch a proof of Gauss–Bonnet theorem.

In the second part of the notes, Sections 5–10, we discuss how geometric and topological properties of vector bundles manifest themselves in condensed matter physics. The presentation is based on the "minimal working examples" described by simple two-band models, in parallel with our focus on complex line bundles in the first part. In Sec. 5, we introduce the tight-binding formalism, which is used in what follows. Sec. 6 is devoted to the modern theory of electric polarization, which relates a macroscopic observable property with geometry of an eigenspace vector bundle. We show that the polarization is most naturally defined in terms of currents rather than of the static charge distribution, and then use Wannier functions to translate this definition into the language of quantum mechanics. In Sec. 7, we consider a process of charge pumping, in which the polarization of a one-dimensional crystal changes under a periodic variation of the parameters of the model. Owing to its inherent ambiguity, the polarization can change monotonically in a periodic process. We characterize the corresponding vector bundle by a topological invariant, and examine peculiar features of the spectrum, which appear in the case of the open boundary conditions. In Sec. 8, we discuss a situation, in which the process of charge pumping happens inside the momentum space of a crystal, giving rise to the concept of two-dimensional Chern insulator. We apply our results from the previous section to this new physical context and obtain the bulk-boundary correspondence for two-band Chern insulators. We also discuss how these findings fit into a wider perspective of topological classification of matter. We briefly consider this classification in three spatial dimensions, which becomes especially rich for the case of two-band models.

The section on Chern insulators is the central point of the story. The last two sections show directions of how these results can be generalized. In Sec. 9, we touch upon the topic of topological classification of systems with symmetries. Due to complexity of the models, often one cannot write an explicit expression for a relevant topological invariant, and the problem calls for different methods. First, we use symmetry-adapted Wannier functions in inversion-symmetric crystals to illustrate the idea of topological quantum chemistry. Then we briefly describe the classification of time-reversal invariant topological insulators in terms of the surface states. In Sec. 10, we relax the condition that the crystal must be gapped. We categorize gapless two-band systems according to their dimension and dimension of the space of available Hamiltonians. Then we introduce nodal line semimetals and Weyl semimetals, and discuss their characteristic surface states. Finally, we use the language of real vector bundles to describe nodal points in systems with certain symmetry constraints.

### Sources and further reading

The material discussed in the mathematical part of the notes is loosely based on a textbook [3], which provides a physicist-oriented introduction to the machinery of differential geometry and basic algebraic topology. This is a good place to start learning about manifolds and differential forms, disguised as surfaces and certain well-behaved integrands in our notes. Another friendly exposition of these topics is given in Ref. [4]. The standard reference [5] covers additional material, but is more formal and rather advanced. For a discussion of vector bundles aimed at mathematicians, see Refs. [6,7].

The main sources of the material on topological band theory are textbooks [8,9]. Our discussion shares with them some parts of the general story line (such as the sequence "polarization → charge pumps → Chern insulators"), but differs in many details. Ref. [8] provides a concise example-driven presentation of the theory of topological insulators. In the textbook [9], discussion revolves around the concept of Berry phase and includes wide range of topics, combining analytical and computational perspectives. Our notes aim to complement these excellent sources with the mathematical point of view. The reader may also consult an introductory article [10], which uses the language of vector bundles. The review article [1], besides a pedagogical exposition, provides an overview of early theoretical developments and experimental results on topological insulators and superconductors. Further physical insights can be found in an online course [11] featuring video mini-lectures by pioneers of the field. A thorough research-level presentation of the subject is given in a textbook [12]. Further references to more specialized review articles will be given in the main text.

Today, topological aspects of condensed matter have become too vast a subject to be covered in a single textbook or a lecture course. Our choice of topics is shaped by the mathematical perspective, and some important physical ideas fall outside the scope of these notes. Below is an (incomplete) list of such omissions:

- Systems with spin-orbit coupling, which must have at least four bands (while we focus on the two-band case). We give only a brief discussion of the quantum spin Hall insulators in Sec. 9.4.

- A connection between topological band theory and Dirac equation. This perspective is developed in lecture notes [13], an introductory article [14], and a textbook [15].

- The integer quantum Hall effect, which, historically, is a common ancestor of all topological materials. We refer the interested reader to the comprehensive lecture notes [16] and references therein.

- Axion electrodynamics and quantized magnetoelectric response of topological insulators. For an introduction to these topics, see Ref. [17].

- Non-Hermitian topology in quantum and classical systems, which is reviewed in Ref. [18].

- Symmetry-protected topological phases, which arise from the interplay of topology, symmetry, and strong correlations. For an overview, see Ref. [19].

### Acknowledgments

I warmly thank my colleagues O.G. Kharlanov, K.V. Antipin, and A.A. Markov for many fruitful discussions. I am deeply grateful to the three anonymous referees for their numerous remarks and suggestions, which largely influenced the contents and organization of the notes. I thank B. Mera, R.-J. Slager, and M.A. Martin-Delgado for informative and encouraging correspondence regarding the first version of the manuscript. I would like to thank all students who

attended the lectures for their tricky questions and keen interest in the subject. I thank the organizers of Topological Quantum Matter conference at Nordita, Stockholm in August 2019 for their hospitality. The travel support by the Foundation for the Advancement of Theoretical Physics and Mathematics "BASIS", grant No. 19-36-013, is gratefully acknowledged. I would like to thank J. van Wezel and A. Bouhon for insightful discussions during the conference. The work was partially supported by RFBR grant No. 19-02-00828 A.

# 1 Connection on a vector bundle

Vector field is one of the fundamental mathematical objects used throughout all areas of physics. A vector field $\boldsymbol{v}$ defined over a physical space is commonly described by a vector-valued function, which associates to a point $p$ an element of a fixed vector space $V$. However, there is no physical reason to think that this vector space is "the same" for all points $p$. Instead, one can imagine that each point $p$ has its own copy $V_p$ of $V$, which gives rise to the concept of vector bundle. This may look like an unnecessary complication of a simple situation. For instance, taking a derivative of a vector field becomes problematic: in a vector bundle, all vector spaces $V_p$ are independent, and so are their bases. Thus, the usual component-wise differentiation loses its meaning in this context. This problem is solved by introducing an additional mathematical structure, called connection on a vector bundle. Surprisingly, this structure appears in models of many physical phenomena, as we will see below.

In this section, we introduce the language of vector bundles and develop a basis-independent way to differentiate vector fields. Then we consider fields that are constant along a given curve, and discuss their physical realizations in classical mechanics.

## 1.1 From vector fields to vector bundles

### 1.1.1 Idea of vector bundle

Consider a velocity vector field $\boldsymbol{v}$ of a thin layer of fluid flowing on a plane $\mathbb{R}^2$, as shown in Fig. 1.1 on the left. At each point $p \in \mathbb{R}^2$, the velocity is given by the vector $\boldsymbol{v}(p)$. One can add such velocities for two flows, $\boldsymbol{v}_1(p) + \boldsymbol{v}_2(p)$. A scalar multiple $\lambda \boldsymbol{v}(p)$ is also a value of velocity at $p$ for some flow. Thus, all possible velocities of a point moving through $p \in \mathbb{R}^2$ form together a vector space. We will call it the **tangent space** to the plane at $p$ and denote it $T_p \mathbb{R}^2$. Note that it does not make sense to add velocities at different points, so the tangent spaces $T_p \mathbb{R}^2$ and $T_q \mathbb{R}^2$ are independent for $p \neq q$. The collection of all tangent spaces

$$T\mathbb{R}^2 = \{T_p \mathbb{R}^2 \mid p \in \mathbb{R}^2\}, \tag{1.1}$$

is called the **tangent bundle** of the plane $\mathbb{R}^2$.

There are several ways to generalize this construction. First, we replace the plane with another surface $\mathcal{B}$ and obtain the tangent bundle of this surface

$$T\mathcal{B} = \{T_p \mathcal{B} \mid p \in \mathcal{B}\}, \tag{1.2}$$

which contains velocity fields of points moving on the surface $\mathcal{B}$. Next, note that a velocity field is a very specific type of a field. For a general vector field, vectors at $p$ form a vector space $V_p$, which is not related to the shape of the surface $\mathcal{B}$ near $p$. This gives us the notion of **vector bundle** $V$ over $\mathcal{B}$:

$$V = \{V_p \mid p \in \mathcal{B}\}, \tag{1.3}$$

which can be though of as a collection of vector spaces attached to the points of $\mathcal{B}$. There are special terms for the parts of a vector bundle:

- One calls $\mathcal{B}$ the **base space**, or simply the base, of the bundle $V$. The base can be any smooth geometric object, like a curve, a surface, or their higher-dimensional generalizations.

- The vector space $V_p$ is known as a **fiber** at $p \in \mathcal{B}$. As the point $p$ moves on $\mathcal{B}$, the fibers $V_p$ must vary smoothly. In particular, the fibers have the same dimension for all $p \in \mathcal{B}$.

- A **global section** $\boldsymbol{v}$ of the bundle $V$ is a smooth choice of an element $\boldsymbol{v}(p)$ in each fiber. In other words, it is a function $\boldsymbol{v} : \mathcal{B} \to V$ such that $\boldsymbol{v}(p) \in V_p$, generalizing the concept of vector field. When defined only over some region $\Sigma \subset \mathcal{B}$, the section is said to be **local**.

The reader may wonder, what a smooth variation of vector spaces $V_p$ can possibly mean. The precise answer is given by the mathematical definition of smooth vector bundle, which lies outside the scope of these notes.[2] However, there will be an appropriate notion of smoothness for each vector bundle we will encounter. For example, the tangent bundle $T\mathcal{B}$ to a surface $\mathcal{B}$ can be thought of as a collection of planes orthogonal to the surface normals $\boldsymbol{n}$. Then the planes vary smoothly over the surface in the sense that the vector field $\boldsymbol{n}$ is smooth.

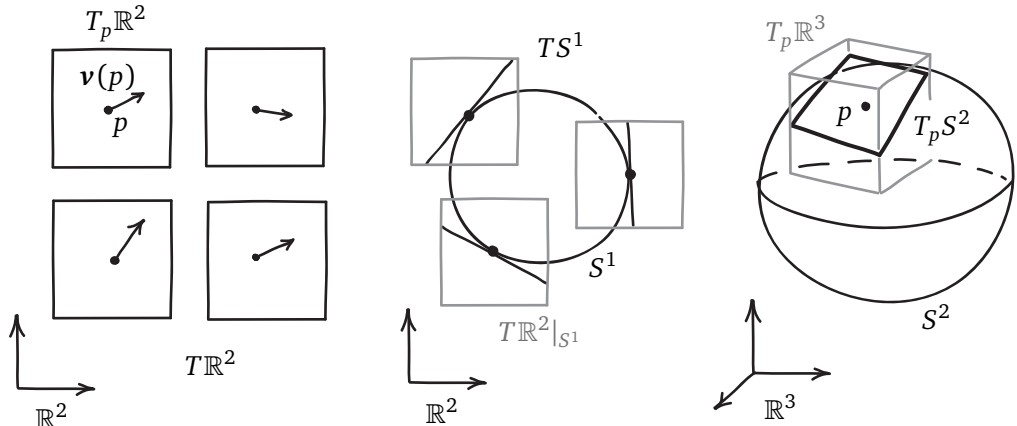

Figure 1.1: Examples of vector bundles. Line segments, rectangles and boxes represent vector spaces. LEFT: The tangent bundle $T\mathbb{R}^2$ of the plane $\mathbb{R}^2$. Arrows show a section $\boldsymbol{v}$ of $T\mathbb{R}^2$. The value of the section $\boldsymbol{v}$ at the point $p \in \mathbb{R}^2$ is the vector $\boldsymbol{v}(p) \in T_p\mathbb{R}^2$ in the tangent space at this point. MIDDLE: The tangent bundle $TS^1$ of the circle $S^1 \subset \mathbb{R}^2$ as a subbundle of the restriction $T\mathbb{R}^2|_{S^1}$. RIGHT: The tangent bundle $TS^2$ of the sphere $S^2 \subset \mathbb{R}^3$ as a subbundle of the restriction $T\mathbb{R}^3|_{S^2}$.

Given a vector bundle, we can obtain new bundles by specifying subsets of the base space or by selecting vector subspaces of the fiber. Suppose that we have a circle $S^1 \subset \mathbb{R}^2$ on the plane. Then we can **restrict** the bundle $T\mathbb{R}^2$ to the circle by considering only those fibers that are attached to the points of $S^1$:

$$T\mathbb{R}^2|_{S^1} = \{T_p\mathbb{R}^2 \mid p \in S^1\}. \tag{1.4}$$

One can further choose a vector subspace in each fiber, thus defining a **subbundle**. For example, select in each fiber of $T\mathbb{R}^2|_{S^1}$ a line that is tangent to the circle. The result is the tangent bundle of the circle $TS^1 \subset T\mathbb{R}^2|_{S^1}$, as shown in the middle panel of Fig. 1.1. Now let us add

---

[2]A smooth vector bundle is an object described by the definition given in Preface, which carries an extra layer of data: a differential structure.

a new dimension to this picture: consider a two-dimensional sphere $S^2 \subset \mathbb{R}^3$. The tangent bundle of the sphere $TS^2$ is then a subbundle of the restriction $T\mathbb{R}^3|_{S^2}$ (Fig. 1.1, right). If a vector bundle $V$ is a subbundle of $W$, we will call the latter an **ambient bundle** for $V$.

### 1.1.2 Basis sections and bundle metric

We will call **basis sections** of a vector bundle a set of sections whose values $\{e_i(p)\}$ form a basis in the fiber $V_p$ for all $p$ in the region where the sections are defined. A smooth choice of an inner product $\langle \cdot, \cdot \rangle_V$ in the fibers is known as **bundle metric**. For a tangent bundle $T\mathcal{B}$, it is called simply a **metric** on $\mathcal{B}$. Here, we specify basis sections and metrics that will be used in what follows.

We start with the tangent bundle $T\mathbb{R}^2$. The plane $\mathbb{R}^2$ is itself a vector space with the standard inner product. To each point $p \in \mathbb{R}^2$ with coordinates $(x_1, x_2)$, one associates the position vector[3]

$$\boldsymbol{r} = \boldsymbol{f}_i x_i \,, \tag{1.5}$$

where $\boldsymbol{f}_1 = (1,0)^T$ and $\boldsymbol{f}_2 = (0,1)^T$ form the standard orthonormal basis for $\mathbb{R}^2$. We use the coordinate system $(x_1, x_2)$ to define the basis vectors for $T_p\mathbb{R}^2$ as

$$\boldsymbol{e}_i(p) = \frac{\partial \boldsymbol{r}}{\partial x_i} \equiv \partial_i \boldsymbol{r} \,, \quad i = 1, 2 \,, \tag{1.6}$$

where $\boldsymbol{r}$ is the position vector at $p$. Then we declare $\{\boldsymbol{e}_1(p), \boldsymbol{e}_2(p)\}$ to be orthonormal, which defines an inner product in $T_p\mathbb{R}^2$ and gives $\mathbb{R}^2$ the **standard** metric $\langle \cdot, \cdot \rangle_{T\mathbb{R}^2}$. A similar construction works for the Euclidean space $\mathbb{R}^n$ and its tangent bundle.

Next, we define a metric and basis sections for the tangent bundle of the sphere $TS^2$. This can be done by using the ambient space $\mathbb{R}^3$ with its standard metric. Note that at a point $p \in S^2 \subset \mathbb{R}^3$, the tangent space $T_pS^2$ is a subspace of $T_p\mathbb{R}^3$. Thus, any vector $\boldsymbol{v}(p) \in T_pS^2$ can also be thought of as a three-dimensional vector, which we will denote by $\boldsymbol{v}^a(p) \in T_p\mathbb{R}^3$. We define the inner product in $T_pS^2$ as

$$\langle \boldsymbol{v}(p), \boldsymbol{w}(p) \rangle_{TS^2} = \langle \boldsymbol{v}^a(p), \boldsymbol{w}^a(p) \rangle_{T\mathbb{R}^3} \,. \tag{1.7}$$

One says that the resulting metric on $S^2$ is **induced** from the metric on $\mathbb{R}^3$. More generally, let $\mathcal{B}$ be a two-dimensional surface inside $\mathbb{R}^3$. We define the induced metric on $\mathcal{B}$ in a similar way, by considering the tangent bundle $T\mathcal{B}$ as a subbundle of $T\mathbb{R}^3|_{\mathcal{B}}$.

To introduce basis sections for $TS^2$, we consider $S^2$ as a sphere of radius $r$ centered at the origin of the space $\mathbb{R}^3$. A point $p \in S^2$ is described by the position vector

$$\boldsymbol{r} = \begin{pmatrix} r \sin\theta \cos\varphi \\ r \sin\theta \sin\varphi \\ r \cos\theta \end{pmatrix} \,. \tag{1.8}$$

We define the basis sections $\{\boldsymbol{e}_\theta, \boldsymbol{e}_\varphi\}$ for $TS^2$ as the normalized derivatives $\partial_\alpha \boldsymbol{r}$ for $\alpha = \theta, \varphi$:

$$\boldsymbol{e}_\theta^a = \begin{pmatrix} \cos\theta \cos\varphi \\ \cos\theta \sin\varphi \\ -\sin\theta \end{pmatrix} \,, \qquad \boldsymbol{e}_\varphi^a = \begin{pmatrix} -\sin\varphi \\ \cos\varphi \\ 0 \end{pmatrix} \,, \tag{1.9}$$

which are also orthogonal at each point. These sections are defined everywhere on the sphere except at the poles.

---

[3]Here and throughout the notes, we use the Einstein summation convention: there is a sum over each pair of repeating indices.

### 1.1.3 How to differentiate a section?

Our next goal is to take a directional derivative of a section of a vector bundle at some point $p$ of the base space. To fix a direction at $p$, we specify a smooth curve $\mathcal{T}$ passing through this point.

We start with a familiar case of the planar vector field $\boldsymbol{v}$ on the plane $\mathbb{R}^2$, that is, a section $\boldsymbol{v}$ of $T\mathbb{R}^2$. Define a curve $\mathcal{T}$ parametrically as a pair of functions $x_1(\tau), x_2(\tau)$, where $x_i$ are Cartesian coordinates on the plane and $\tau$ is the parameter of the curve. Then the section $\boldsymbol{v}$ restricts to a smooth vector field along the curve, and we wish to differentiate this field with respect to $\tau$ at the point $p$. To this end, we decompose $\boldsymbol{v}$ in terms of the Cartesian basis sections (1.6) as $\boldsymbol{v} = \boldsymbol{e}_i v_i$. Thus, the section $\boldsymbol{v}$ is represented by a function $v : \mathbb{R}^2 \to \mathbb{R}^2$ sending a point of the plane to the pair of components of the field,

$$v(p) = \begin{pmatrix} v_1(p) \\ v_2(p) \end{pmatrix}. \tag{1.10}$$

Then we compute the derivative component-wise:

$$\partial_\tau \boldsymbol{v} = \partial_\tau (\boldsymbol{e}_i v_i) = \boldsymbol{e}_i (\partial_\tau v_i), \tag{1.11}$$

Crucially, we used in the last equality that

$$\partial_\tau \boldsymbol{e}_i = 0, \quad \text{or} \quad \boldsymbol{e}_i = \text{const}. \tag{1.12}$$

This need not hold for other basis sections. For example, the sections $\{\boldsymbol{e}_r, \boldsymbol{e}_\varphi\}$ corresponding to the polar coordinate system are not constant. When computing the derivative of a field $\boldsymbol{v}$ expressed in terms of $\{\boldsymbol{e}_r, \boldsymbol{e}_\varphi\}$, one needs to take into account the change of basis vectors. This is the starting point of the vector calculus in curvilinear coordinates.

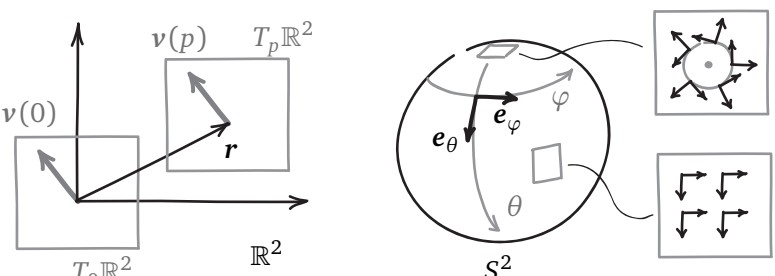

Figure 1.2: LEFT: The identification between the elements of the fibers of $T\mathbb{R}^2$ described by Eq. (1.13). The tangent space at the origin $T_0\mathbb{R}^2$ is identified with that at the point $p$ with the position vector $\boldsymbol{r}$. RIGHT: Basis sections of $TS^2$ given by Eq. (1.9) and their characteristic shapes near the equator and near the poles.

The choice of the Cartesian basis sections as constant ones matches with our intuition of the parallel translation of a vector on the plane. In fact, it can be justified formally by using the vector space structure of the base space $\mathbb{R}^2$. First, note that the elements of the tangent space $T_0\mathbb{R}^2$ at the origin are naturally identified with the points of the plane $\mathbb{R}^2$: a velocity vector $\boldsymbol{v}(0) \in T_0\mathbb{R}^2$ defines a trajectory $\boldsymbol{r} = t\boldsymbol{v}(0)$, where $t$ is the time. So, we can associate the velocity vector with the point of the plane where one arrives at $t = 1$. Further, this allows us to connect the tangent spaces at distinct points. Let $\boldsymbol{r}$ be the position vector of a point $p$. Then the elements of the tangent spaces $T_0\mathbb{R}^2$ and $T_p\mathbb{R}^2$ are identified by

$$\boldsymbol{v}(0) \quad \longleftrightarrow \quad \boldsymbol{v}(p) = \partial_t(\boldsymbol{r} + t\boldsymbol{v}(0))|_{t=0}, \tag{1.13}$$

as shown in Fig. 1.2 on the left. The summation on the right is possible because of the vector space structure of $\mathbb{R}^2$ and the identification between the points and velocities described above. Globally, this gives a notion of a uniform vector field, or a constant section of $T\mathbb{R}^2$. Cartesian basis sections (1.6) are such constant sections corresponding to the standard basis $\{\boldsymbol{f}_1, \boldsymbol{f}_2\}$ of $\mathbb{R}^2$. Note that the identification (1.13) does not depend on the coordinate system used on the plane.

Now consider a tangent vector field on the sphere $S^2$, or a section $\boldsymbol{v}$ of $TS^2$. Here, the situation is different in two important aspects. First, the section $\boldsymbol{v}$ cannot be described by a global function like that given by Eq. (1.10). Indeed, the basis sections $\{\boldsymbol{e}_\theta, \boldsymbol{e}_\varphi\}$ are not defined at the poles. The presence of such singularities is a general feature of $TS^2$. As we will see in Sec. 4, there is a topological constraint, which forces any global section of this bundle to vanish at some point. Hence, unit sections (that is, sections with values of unit norm) cannot be global, and there is no global choice of basis. This constraint makes the section $\boldsymbol{v}$ of $TS^2$ into something very different from a function $v : S^2 \to \mathbb{R}^2$, which does not have to vanish anywhere.

Second, one cannot sensibly define the differentiation in terms of the components of a vector in the fashion of Eq. (1.11). For example, let us decompose $\boldsymbol{v}$ in terms of the basis sections $\{\boldsymbol{e}_\theta, \boldsymbol{e}_\varphi\}$ given by Eq. (1.9). But are these basis sections constant? Since the sphere looks locally like a plane, we can apply our intuitive understanding of the derivative on a flat surface. Fig. 1.2, right, shows two typical situations. Near the equator, $\{\boldsymbol{e}_\theta, \boldsymbol{e}_\varphi\}$ resemble Cartesian basis sections, but near the poles they correspond to the polar coordinate system and cannot be declared to be constant. Unlike the plane $\mathbb{R}^2$, the sphere $S^2$ is not a vector space, so the correspondence (1.13) is of no use here. The tangent spaces at the different points are independent, and there is no natural way to compare their elements. We have to endow $TS^2$ with an additional structure that will allow us to define a locally constant section along a curve and to differentiate a general section $\boldsymbol{v}$.

## 1.2 Covariant derivative

We need to construct a basis-independent differential operator acting on sections of $TS^2$, which will behave like the ordinary directional derivative (1.11) on the plane. In particular, the derivative must act "inside" $TS^2$, that is, produce a two-component tangent vector. This constraint is responsible for *covariance*: the components of the derivative must transform like vector components under change of basis. One way to define such an operator is to use the embedding $S^2 \subset \mathbb{R}^3$, as we did when defining the induced metric in Sec. 1.1.2.

### 1.2.1 Projection from ambient space

Consider a section $\boldsymbol{v}$ of $TS^2$ as a section $\boldsymbol{v}^a$ of $T\mathbb{R}^3|_{S^2}$. As such, it can be decomposed as $\boldsymbol{v}^a = \boldsymbol{e}_i v_i^a$, where $i = 1, 2, 3$ and $\{\boldsymbol{e}_i\}$ are the Cartesian basis sections for $T\mathbb{R}^3|_{S^2}$. Let $\theta(\tau), \varphi(\tau)$ describe a curve $\mathcal{T}$ passing through a point $p \in S^2$. As an intermediate step, we introduce the derivative of a section $\boldsymbol{v}$ as a three-dimensional vector field:

$$\partial_\tau^a \boldsymbol{v} = \boldsymbol{e}_i \partial_\tau v_i^a, \qquad i = 1, 2, 3. \tag{1.14}$$

However, the resulting vector need not belong to $TS^2$; this can be fixed by using projection. Define the following operator:

$$\nabla_\tau \boldsymbol{v} = \text{Proj}(\partial_\tau^a \boldsymbol{v}), \tag{1.15}$$

where $\text{Proj} : T_p\mathbb{R}^3 \to T_p S^2$ is the orthogonal projection to the tangent space at the point $p$ where we compute the derivative. We will call $\nabla_\tau$ the **covariant derivative** of the section $\boldsymbol{v}$ along the curve $\mathcal{T}$ with respect to the parameter $\tau$. For the coordinate curves on the sphere,

this gives us the derivatives $\nabla_\theta$ and $\nabla_\varphi$. Note that the covariant derivative $\nabla_\tau$ is manifestly independent of the basis choice for $TS^2$. However, it requires a constant basis and a metric in the ambient bundle $T\mathbb{R}^3$.

In a similar way, the covariant derivative can be defined on the tangent bundle $T\mathcal{B}$ of any two-dimensional surface $\mathcal{B} \subset \mathbb{R}^3$. For example, on a plane $\mathbb{R}^2 \subset \mathbb{R}^3$ the projection operator acts identically, and the formula for $\nabla_\tau$ reduces to the ordinary derivative (1.11). On a curved surface $\mathcal{B}$, the value of $\nabla_\tau v$ is affected by the geometry of the surface via the variation of the tangent planes along the curve.

Since the projection is a linear operator, the covariant derivative $\nabla_\tau$ on the surface $\mathcal{B}$ inherits two important properties of the derivative $\partial_\tau^a$:

$$\text{Linearity:} \quad \nabla_\tau(v + \lambda w) = \nabla_\tau v + \lambda \nabla_\tau w\,, \tag{1.16}$$

$$\text{Leibniz rule:} \quad \nabla_\tau(f v) = (\partial_\tau f) v + f \nabla_\tau v\,, \tag{1.17}$$

where $v$ and $w$ are sections of $T\mathcal{B}$, $f$ is a scalar function on $\mathcal{B}$, and $\lambda$ is a constant.

### 1.2.2 Connection coefficients

In practice, it is often useful to have a coordinate expression for $\nabla_\tau$. Let $\{e_1, e_2\}$ be the basis sections for the tangent bundle $T\mathcal{B}$ of a two-dimensional surface. To compute $\nabla_\tau v$ at the point $p \in \mathcal{B}$, we decompose $v$ in terms of the basis sections and then use linearity and the Leibniz rule:

$$\nabla_\tau v = \nabla_\tau(e_\alpha v_\alpha) = e_\alpha \partial_\tau v_\alpha + (\nabla_\tau e_\alpha) v_\alpha\,, \qquad \alpha = 1, 2\,. \tag{1.18}$$

Now note that the covariant derivatives $\nabla_\tau e_\alpha$ of the basis vectors belong to the tangent space $T_p\mathcal{B}$. Hence, $\nabla_\tau e_\alpha$ can be decomposed in terms of the same basis, which gives

$$\nabla_\tau e_\alpha = e_\beta (\nabla_\tau e_\alpha)_\beta \equiv e_\beta \omega_{\tau\alpha}^\beta\,, \tag{1.19}$$

where $\omega_{\tau\alpha}^\beta$ are functions on $\mathcal{B}$ called **connection coefficients** with respect to the basis $\{e_1, e_2\}$. It should be clear from the definition that the connection coefficients depend on the basis choice. Note the different roles of the indices: $\tau$ corresponds to the differentiation along a curve on the base space, while $\alpha$ and $\beta$ refer to the basis vectors in the fiber. Finally, we obtain the following expression for the covariant derivative:

$$\nabla_\tau v = e_\beta (\partial_\tau v_\beta + \omega_{\tau\alpha}^\beta v_\alpha)\,. \tag{1.20}$$

Let us find the connection coefficients along the coordinate curves of $TS^2$ with respect to the basis sections $\{e_\theta, e_\varphi\}$ defined in Eq. (1.9). Since the basis is orthonormal, the projection of a vector $w$ to the tangent plane is given by

$$\text{Proj}(w) = e_\beta \langle e_\beta, w \rangle_{T\mathbb{R}^3}\,. \tag{1.21}$$

**Exercise 1.1.** Compute $\nabla_\tau e_\alpha$ for $\tau = \theta, \varphi$ and $\alpha = \theta, \varphi$. Find the corresponding connection coefficients $\omega_{\tau\alpha}^\beta$.

By results of the exercise, the only non-zero connection coefficients with respect to the basis $\{e_\theta, e_\varphi\}$ are

$$\omega_{\varphi\theta}^\varphi = \cos\theta\,, \qquad \omega_{\varphi\varphi}^\theta = -\cos\theta\,. \tag{1.22}$$

With these functions at hand, one need not compute projections to find the covariant derivative of a section $v$ of $TS^2$.

### 1.2.3 Complex plane notation

One can further simplify the expression (1.20) for the covariant derivative on $T\mathcal{B}$ by considering its fibers as complex vector spaces.

Let $\boldsymbol{v}$ be an element of a two-dimensional real vector space $V$. We know how to multiply $\boldsymbol{v}$ by a real number $\lambda \in \mathbb{R}$, obtaining a new vector $\lambda\boldsymbol{v} \in V$. Once a basis is chosen, $\boldsymbol{v}$ can be represented as a column of components $(v_1, v_2)^T$ with $v_i \in \mathbb{R}$. Now consider a complex one-dimensional vector space, or a **complex line** $W$. Vectors in $W$ can be multiplied by complex scalars. A choice of the basis vector $\boldsymbol{1}$ allows one to identify any vector $\boldsymbol{w} \in W$ with a complex number: $\boldsymbol{w} = w\boldsymbol{1}$, where $w \in \mathbb{C}$. We wish to describe the real plane as a complex line. While their (real) dimensions coincide, the latter carries extra structure: multiplication by the imaginary unit $i$. In real terms, this is given by a linear operator $I$ that squares to the minus identity and can be described as a $\frac{\pi}{2}$ rotation. We only need to specify the sense of rotation, which is fixed by choosing an orientation of the real plane (which defines the term "clockwise"). Then we define the action of the complex number $a + ib$ on the vector $\boldsymbol{v}$ as

$$(a + ib)\boldsymbol{v} = a\boldsymbol{v} + bI(\boldsymbol{v}). \tag{1.23}$$

One says that the choice of $I$ endows $V$ with a **complex structure**.

For example, let us introduce a complex structure on a fiber $T_pS^2$. The basis vectors $\{\boldsymbol{e}_\theta(p), \boldsymbol{e}_\varphi(p)\}$ given by Eq. (1.9) are orthonormal. Define the operator $I$ as the $\frac{\pi}{2}$ counter-clockwise rotation when viewed from outside the sphere. We choose $\boldsymbol{e}_\theta(p)$ as the complex basis vector in the plane $T_pS^2$, understood now as a complex line. The real basis vectors become linearly dependent as complex vectors: $\boldsymbol{e}_\varphi(p) = i\boldsymbol{e}_\theta(p)$. A pair of real components $(v_\theta, v_\varphi)^T$ of a section $\boldsymbol{v}$ turns into a single complex function:

$$\boldsymbol{v} = \boldsymbol{e}_\theta v_\theta + \boldsymbol{e}_\varphi v_\varphi = (v_\theta + iv_\varphi)\boldsymbol{e}_\theta. \tag{1.24}$$

The choice of the complex structure turns $TS^2$ into a **complex line bundle**, that is, a bundle with complex one-dimensional spaces as fibers. We will use

$$\boldsymbol{1} \equiv \boldsymbol{e}_\theta, \tag{1.25}$$

as the standard complex basis section for $TS^2$.

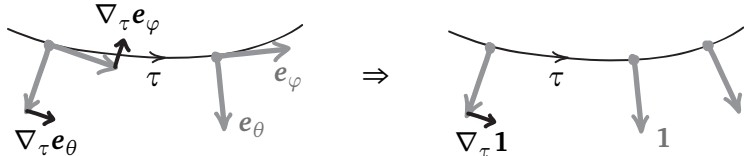

Figure 1.3: LEFT: Covariant derivatives of the orthonormal basis vectors $\{\boldsymbol{e}_\theta, \boldsymbol{e}_\varphi\}$ are orthogonal to the basis vectors and have equal magnitude. RIGHT: The same situation simplified by the complex plane notation. The pair of basis sections is replaced by a single section $\boldsymbol{1} = \boldsymbol{e}_\theta$.

Consider the covariant derivative $\nabla_\tau$ along a curve on the sphere, and return for a moment to the real basis sections $\{\boldsymbol{e}_\theta, \boldsymbol{e}_\varphi\}$. The connection coefficients are determined by the covariant derivatives of the basis vectors. Since the basis is orthonormal, the only degree of freedom it has inside the fiber is the rotation. Thus the covariant derivatives $\nabla_\tau \boldsymbol{e}_\alpha$ of the basis vectors

are perpendicular to them and have equal magnitude, as shown in Fig. 1.3. Let us denote this magnitude by $\omega_\tau$. We have

$$\nabla_\tau \boldsymbol{e}_\theta = \omega_\tau \boldsymbol{e}_\varphi, \qquad \nabla_\tau \boldsymbol{e}_\varphi = -\omega_\tau \boldsymbol{e}_\theta, \tag{1.26}$$

which explains the simple form of Eqs. (1.22). In the complex notation with $\boldsymbol{e}_\theta = \mathbf{1}$ and $\boldsymbol{e}_\varphi = i\mathbf{1}$ this becomes

$$\nabla_\tau \mathbf{1} = \omega_\tau i\mathbf{1} = i\omega_\tau \mathbf{1}, \qquad \nabla_\tau(i\mathbf{1}) = -\omega_\tau \mathbf{1}. \tag{1.27}$$

It follows that $\nabla_\tau(i\mathbf{1}) = i\nabla_\tau \mathbf{1}$, so that $\nabla_\tau$ is a *complex linear* differential operator. We conclude that the restriction to the orthonormal basis sections allows us to replace four functions $\omega_{\tau\alpha}^\beta$ with a single **connection coefficient** $\omega_\tau$ defined by

$$\nabla_\tau \mathbf{1} = i\omega_\tau \mathbf{1}. \tag{1.28}$$

In a similar fashion, one can consider the tangent bundle $T\mathcal{B}$ of an orientable surface[4] $\mathcal{B} \subset \mathbb{R}^3$ as a complex line bundle. A section $\boldsymbol{v}$ of $T\mathcal{B}$ becomes a complex vector field, which is described in terms of a basis section $\mathbf{1}$ by a complex function $v$ on the base space $\mathcal{B}$:

$$\boldsymbol{v} = v\mathbf{1}. \tag{1.29}$$

For any curve on the surface, we define the corresponding complex connection coefficient $\omega_\tau$ with respect to the basis $\mathbf{1}$. The expression (1.18) for the covariant derivative along the curve takes the form

$$\nabla_\tau \boldsymbol{v} = (\partial_\tau v)\mathbf{1} + v\nabla_\tau \mathbf{1} = (\partial_\tau v)\mathbf{1} + i\omega_\tau \boldsymbol{v}. \tag{1.30}$$

This can be written more succinctly if we define the action of the ordinary derivative $\partial_\tau$ on the section $\boldsymbol{v}$ by $\partial_\tau \boldsymbol{v} = (\partial_\tau v)\mathbf{1}$:

$$\nabla_\tau \boldsymbol{v} = (\partial_\tau + i\omega_\tau)\boldsymbol{v}. \tag{1.31}$$

While the last expression does not contain the basis section $\mathbf{1}$ explicitly, one should keep in mind that both $\omega_\tau$ and $\partial_\tau \boldsymbol{v}$ depend on the basis choice.

### 1.2.4 Transformation laws for connection coefficients

The basis sections $\{\boldsymbol{e}_\theta, \boldsymbol{e}_\varphi\}$ of $TS^2$ are closely related to the coordinate system $(\theta, \varphi)$ on the sphere. In general, this need not be so: one can choose basis sections and coordinates independently. Here, we examine what happens to the connection coefficients under both types of transformations.

First, consider the change of basis. We start with the connection coefficient $\omega_\tau$ defined with respect to the basis section $\mathbf{1}$. Consider another basis section $\mathbf{1}'$ related to $\mathbf{1}$ by a position-dependent phase rotation: $\mathbf{1}' = e^{i\beta(\tau)}\mathbf{1}$, as shown in the left panel of Fig. 1.4. Then

$$\nabla_\tau \mathbf{1} = i\omega_\tau \mathbf{1}, \qquad \nabla_\tau \mathbf{1}' = i\omega_\tau' \mathbf{1}'. \tag{1.32}$$

On the other hand,

$$\nabla_\tau \mathbf{1}' = \nabla_\tau(e^{i\beta}\mathbf{1}) = i(\partial_\tau \beta)e^{i\beta}\mathbf{1} + e^{i\beta}\nabla_\tau \mathbf{1} = i(\omega_\tau + \partial_\tau \beta)e^{i\beta}\mathbf{1}, \tag{1.33}$$

by the Leibniz rule. It follows that the transformation law for connection coefficients has the form

$$\omega_\tau' = \omega_\tau + \partial_\tau \beta. \tag{1.34}$$

---

[4]A surface is orientable if its tangent bundle admits a uniform choice of orientation of the fibers.

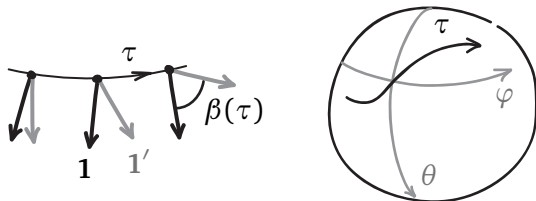

Figure 1.4: LEFT: New basis section $\mathbf{1}'$ related to the original section $\mathbf{1}$ by a position-dependent rotation through an angle $\beta(\tau)$. RIGHT: The curve parameterized by $\tau$ and the coordinate curves.

Another important transformation law is associated with the change of the coordinates on the base space. For convenience, here we consider the bundle $TS^2$, but the result can be generalized to other cases. First, we express the connection coefficient $\omega_\tau$ along a curve $\mathcal{T}$ in terms of connection coefficients $\omega_\theta$ and $\omega_\varphi$ along the coordinate curves. Recall that the curve on the sphere is defined parametrically by the two functions $\theta(\tau), \varphi(\tau)$. From the definition of the covariant derivative (1.15), we have

$$\nabla_\tau \mathbf{1} = \mathrm{Proj}\big(\partial_\tau^a \mathbf{1}(\theta(\tau),\varphi(\tau))\big) = \mathrm{Proj}\left(\frac{d\varphi}{d\tau}\partial_\varphi^a \mathbf{1} + \frac{d\theta}{d\tau}\partial_\theta^a \mathbf{1}\right). \tag{1.35}$$

It follows that the connection coefficients are related as

$$\omega_\tau = \frac{d\varphi}{d\tau}\omega_\varphi + \frac{d\theta}{d\tau}\omega_\theta. \tag{1.36}$$

More generally, one can use the functions $\theta(x_1, x_2), \varphi(x_1, x_2)$ to define new coordinates $(x_1, x_2)$ on the sphere. Indeed, for a fixed value of the second coordinate $x_2 = x_2^0$, the pair of functions $\theta(x_1, x_2^0)$ and $\varphi(x_1, x_2^0)$ defines a coordinate curve parameterized by $x_1$. The connection coefficients along the new coordinate curves are given by

$$\omega_i = (\partial_i \alpha)\omega_\alpha, \tag{1.37}$$

where $i = 1, 2$ and we sum over $\alpha = \theta, \varphi$.

## 1.3 Parallel transport

Recall that our first attempt to differentiate a section of $TS^2$ failed because we could not find an appropriate constant basis. Now that we have defined the covariant derivative $\nabla_\tau$, which is manifestly basis-independent, we can find the corresponding constant field. Consider a vector field $\mathbf{v}$ over a curve $\mathcal{T}$ on a surface $\mathcal{B}$, which satisfies the following equation:

$$\nabla_\tau \mathbf{v} = 0. \tag{1.38}$$

If we fix some vector $\mathbf{v}_0 \in T_p\mathcal{B}$ at the starting point $p$ of the curve as a boundary condition for Eq. (1.38), the solution is unique. Such vector field is called the **parallel transport** of the vector $\mathbf{v}_0$ along the curve and will be denoted $\mathbf{v}_{PT}$.

To understand the parallel transport equation geometrically, consider a section $\mathbf{v}$ of the tangent bundle $T\mathcal{B}$ along a curve $\mathcal{T}$. Suppose that the length of vectors is constant along the curve. Then the three-dimensional derivative $\partial_\tau^a \mathbf{v}$ is orthogonal to $\mathbf{v}$ and can be decomposed into the tangential and normal components, as shown in Fig. 1.5. By the definition (1.15),

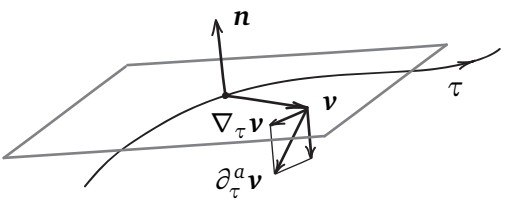

Figure 1.5: Decomposition of the three-dimensional derivative $\partial_\tau^a \boldsymbol{v}$ into the tangential component $\nabla_\tau \boldsymbol{v}$ and the normal component.

the tangential part equals the covariant derivative $\nabla_\tau \boldsymbol{v}$. Thus, if the vector $\boldsymbol{v}$ is parallel transported, it does not rotate around the surface normal $\boldsymbol{n}$, and vice versa:

$$\nabla_\tau \boldsymbol{v} = 0 \quad \Leftrightarrow \quad \text{angular velocity of } \boldsymbol{v} \text{ around } \boldsymbol{n} \text{ is zero, and } |\boldsymbol{v}| = \text{const.} \tag{1.39}$$

### 1.3.1 Parallel transport in $TS^2$

Let us consider the parallel transport of a vector in $TS^2$ along the coordinate curves. To this end, we express the covariant derivatives $\nabla_\varphi$ and $\nabla_\theta$ in the complex notation introduced in Sec. 1.2.3. From Eq. (1.22) we know that there are only two non-zero connection coefficients with respect to the basis $\{\boldsymbol{e}_\theta, \boldsymbol{e}_\varphi\}$. They merge into a single complex connection coefficient $\omega_\varphi$, so we have

$$\omega_\varphi = \cos\theta, \quad \omega_\theta = 0, \tag{1.40}$$

with respect to the basis section $\boldsymbol{1} = \boldsymbol{e}_\theta$.

Since the connection coefficient along a meridian vanishes, the covariant derivative $\nabla_\theta$ of a section $\boldsymbol{v} = v\boldsymbol{1}$ reduces to the ordinary derivative of its component:

$$\nabla_\theta \boldsymbol{v} = (\partial_\theta v)\boldsymbol{1}. \tag{1.41}$$

Thus, in order to perform the parallel transport of a vector along a meridian, one simply has to keep constant its complex component $v$ with respect to the basis section $\boldsymbol{1} = \boldsymbol{e}_\theta$. In the real terms, this means that the parallel transported vector has the constant length and the constant angle with the meridian.

Now consider the parallel transport along a circle $\mathcal{C}$ of a constant latitude $\theta$. Let $\boldsymbol{v} = e^{i\alpha(\varphi)}\boldsymbol{1}$ be a vector field along $\mathcal{C}$, where $\alpha(\varphi)$ is some smooth function with the initial condition $\alpha(0) = 0$. The field $\boldsymbol{v}$ satisfies the equation of the parallel transport (1.38) if

$$(\partial_\varphi + i\omega_\varphi)e^{i\alpha(\varphi)} = 0. \tag{1.42}$$

It follows that

$$\partial_\varphi \alpha = -\omega_\varphi \quad \Rightarrow \quad \alpha(\varphi) = \int_0^\varphi (-\omega_{\varphi'}) d\varphi' = -\varphi\cos\theta. \tag{1.43}$$

We conclude that the parallel transport of the vector $\boldsymbol{v}_0 = \boldsymbol{1}(\theta, 0)$ along the circle of latitude $\theta$ is given by

$$\boldsymbol{v}_{PT} = e^{-i\varphi\cos\theta}\boldsymbol{1}. \tag{1.44}$$

On the equator, $\theta = \frac{\pi}{2}$, the field $\boldsymbol{v}_{PT}$ coincides with the basis section $\boldsymbol{1}$. At the other circles of latitude, the vectors $\boldsymbol{v}_{PT}$ rotate with respect to $\boldsymbol{1}$ as $\varphi$ changes, and the "angular velocity" of the rotation depends on $\theta$. This suggests that one cannot define a section $\boldsymbol{v}$ that would simultaneously satisfy $\nabla_\varphi \boldsymbol{v} = 0$ and $\nabla_\theta \boldsymbol{v} = 0$ over some region of the sphere. We will see that this is indeed the case in Sec. 2.2.2.

Note that, away from the poles, the basis section **1** satisfies the equation of the parallel transport along the coordinate curves that are great circles, the equator and meridians:

$$\nabla_\varphi \mathbf{1}(\tfrac{\pi}{2}, \varphi) = 0, \quad \nabla_\theta \mathbf{1}(\theta, \varphi) = 0. \tag{1.45}$$

One can see this directly from the definition of the covariant derivative (1.15). Along the equator, the derivative $\partial_\varphi^a \mathbf{1}$ vanishes; the derivative $\partial_\theta^a \mathbf{1}$ along the meridians is always orthogonal to the tangent plane.

### 1.3.2 Parallel transport in classical mechanics

The parallel transport of a vector in $TS^2$ along a circle of latitude has a famous physical realization: the **Foucault pendulum**. This device was constructed in 1851 by L. Foucault to provide a direct experimental evidence for the daily rotation of the Earth [20]. The key feature of the Foucault pendulum is that it can oscillate freely in any direction, so there is no preferred swing plane. The simplicity of the requirement is deceptive: see Ref. [21] for a recent overview of experimental difficulties and ways they can be overcome.

For a moment, suppose that the Earth does not rotate. If we deflect the pendulum and launch it with zero initial velocity, it will oscillate in the plane containing the vertical axis. Let us describe this plane by its normal vector $\boldsymbol{w}$, as shown in Fig. 1.6, left. Since all forces acting on the mass belong to this plane, the direction of $\boldsymbol{w}$ will remain constant.

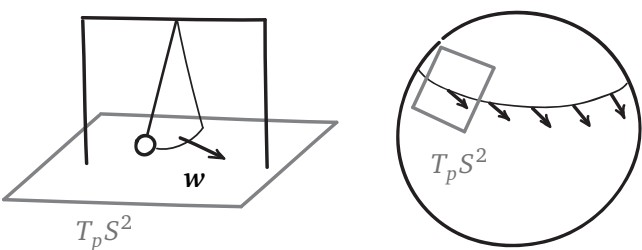

Figure 1.6: LEFT: Foucault pendulum located at the point $p$ of the Earth's surface. The vector $\boldsymbol{w}$ is normal to the swing plane of the pendulum and belongs to the tangent space $T_p S^2$. RIGHT: Evolution of the swing plane normal $\boldsymbol{w}$ due to the rotation of the Earth, which moves the Foucault pendulum along a circle of latitude.

Now consider the Earth rotation with the angular velocity $\Omega$. At the moment $t = 0$, we launch the pendulum located at the intersection of the Greenwich meridian, $\varphi = 0$, with the circle of latitude $\theta$. The longitude of the pendulum is then given by $\varphi(t) = \Omega t$ (note that the coordinate system is stationary and does not rotate with the surface of the Earth). Since $\Omega$ is much smaller than the frequency of the pendulum, the latter will adjust its motion to the slowly changing direction of gravity. Thus, each swing can be approximated by a motion in a plane, which contains the vertical axis. We are interested in the behavior of the plane normal $\boldsymbol{w}$ as a function of time. Since the vector $\boldsymbol{w}$ is tangent to the surface of the Earth, the time derivative of $\boldsymbol{w}$ as a three-dimensional vector does not vanish in general. But there is still no reason for $\boldsymbol{w}$ to rotate around the vertical axis $\boldsymbol{n}$ (as seen from an inertial reference frame). We conclude that the vector $\boldsymbol{w}$ undergoes the parallel transport, as described by (1.39), along the circle of latitude [22]. An observer standing on the surface of the Earth and watching the pendulum will notice the rotation of the swing plane around the vertical axis (Fig. 1.6, right). Indeed, for such an observer, the basis vector $\mathbf{1} = \boldsymbol{e}_\theta$ will not change. From Eq. (1.44), we

find that the angular velocity of the swing plane rotation is $-\Omega \cos\theta$, which agrees with the experimental data for the Foucault pendulum.

Is it possible to realize the parallel transport experimentally without using gravity? Foucault constructed another device, based on the conservation of the angular momentum [20]. The device consisted of a heavy disk on an axle, which was supported by a system of gimbals that allowed the axle to move freely. Once the disk was spinning, the axle maintained its direction in the inertial reference frame. Accordingly, the axle rotated in the non-inertial laboratory frame. Foucault named this device the **gyroscope**, where the root "scope" referred to the observation of the rotation of the Earth.

Note that in contrast with the pendulum, the gyroscope demonstrates the parallel transport in $T\mathbb{R}^3|_{S^2}$ rather than in $TS^2$. It turns out that a gyroscope-based system can model the parallel transport of a two-dimensional vector as well. Using a feedback loop to control gimbals, one can build a device, which has zero angular velocity around a given axis. If the direction of the axis is altered, the gyroscopic device will change its orientation due to the effect of the parallel transport. This phenomenon was studied as an unwanted feature of the inertial navigation systems based on such gyroscopic devices [23].

The constraint (1.39) can also be realized in a conceptually simpler mechanical system. Consider a disk, which can rotate around an axle without friction. Place this system on any surface $\mathcal{B} \subset \mathbb{R}^3$, orient the axle along the surface normal $\boldsymbol{n}$ and make sure that the disk is at rest with respect to an inertial frame. Mark some point of the disk by a vector $\boldsymbol{w}$ and move the system along a curve on the surface, while keeping the axle aligned with the surface normal. Since the friction is absent, one cannot transfer to the disk any angular momentum around the axle. Thus its angular velocity around the axle will remain zero, and the evolution of the vector will correspond to the parallel transport of the vector $\boldsymbol{w}$ in the tangent bundle $T\mathcal{B}$.

### 1.3.3 Parallel transport angle

An important characteristic of the Foucault pendulum is its daily rotation angle. The angle of the swing plane rotation $\alpha$ as a function of $\varphi$ is given by Eq. (1.43). Thus, the rotation angle $\Delta\alpha$ for the full circle of latitude is

$$\Delta\alpha = -2\pi \cos\theta \,. \tag{1.46}$$

This result shows that in general, the parallel transported vector need not return to itself after traveling along a closed curve. In mathematics, this phenomenon is known as **holonomy**. As we will see in Sec. 2, it underlies the concept of the field strength in gauge theories. The reader can experience the effects of parallel transport in the following physical exercise:

**Exercise 1.2.** Take a pen and rise your arm vertically. Direct the pen so that it looks forwards. The arm represents a radius of a sphere and the pen is a tangent vector at the north pole. Perform the parallel transport of the vector along a triangle formed by arcs of great circles: move along a meridian, then along the equator, and return along another meridian to the starting point. Observe that the pen has rotated (despite the fact that it has zero angular velocity around the axis of the arm during the process). How is the angle of rotation related to the shape of the triangle?

Now consider a more general case of the parallel transport of a tangent vector on a surface $\mathcal{B}$. Let $\mathcal{C} \subset \mathcal{B}$ be a closed oriented contour parameterized by $\tau$ (which increases in the positive direction). Assume that some smooth basis section $\boldsymbol{1}$ is chosen in the fibers of $T\mathcal{B}$ over the contour, and let $\omega_\tau$ be the corresponding connection coefficient. We define the **parallel**

**transport angle** associated with the contour $\mathcal{C}$ as

$$\Delta\alpha(\mathcal{C}) = \int_{\mathcal{C}} (-\omega_\tau) d\tau. \tag{1.47}$$

But is this quantity well-defined? As discussed in Sec. 1.2.4, the connection coefficient $\omega_\tau$ depends on the parametrization of the path and on the basis choice in the fibers. Let us examine the behavior of $\Delta\alpha(\mathcal{C})$ under these transformations. First, let $\gamma$ be another parameter along the curve $\mathcal{C}$, specified by the function $\tau(\gamma)$. Two connection coefficients are related as in Eq. (1.36):

$$\omega_\gamma = \frac{d\tau}{d\gamma}\omega_\tau \quad \Rightarrow \quad \int_{\mathcal{C}} (-\omega_\gamma)d\gamma = \int_{\mathcal{C}} \frac{d\tau}{d\gamma}(-\omega_\tau)d\gamma = \int_{\mathcal{C}} (-\omega_\tau)d\tau, \tag{1.48}$$

where we assume that $\gamma$ increases in the positive direction. Thus, the value of $\Delta\alpha$ does not depend on the choice of the parametrization for $\mathcal{C}$.

Moreover, the quantity

$$\Delta\alpha \bmod 2\pi, \tag{1.49}$$

is independent of the choice of basis $\mathbf{1}$ along the curve $\mathcal{C}$. Indeed, let $\mathbf{1}' = e^{i\beta(\tau)}\mathbf{1}$ be another smooth basis section. Suppose that $\tau$ increases from 0 to 1 on $\mathcal{C}$. Then we have:

$$\Delta\alpha' = \int_{\mathcal{C}} (-\omega'_\tau)d\tau = \int_{\mathcal{C}} (-\omega_\tau - \partial_\tau\beta)d\tau = \Delta\alpha - \int_{\mathcal{C}} (\partial_\tau\beta)d\tau. \tag{1.50}$$

At first sight, the integral $\int_{\mathcal{C}} (\partial_\tau\beta)d\tau$ must vanish, since $\beta(0) = \beta(1)$ for any smooth real function on a closed contour. However, the smooth function in question is not $\beta(\tau)$ itself, but the exponential $e^{i\beta(\tau)}$. It is possible that

$$\beta(1) = \beta(0) + 2\pi n, \quad n \in \mathbb{Z}, \tag{1.51}$$

which corresponds to the situation when vectors of $\mathbf{1}'$ make $n$ full turns with respect to the vectors of $\mathbf{1}$ while going around the contour. Thus, the value of $\Delta\alpha$ given by (1.47) can be shifted by an integer multiple of $2\pi$ by the choice of basis section.

One can interpret the invariance of (1.49) geometrically by thinking of parallel transported vectors $\mathbf{v}_{PT}$ as having been engraved on the planes along the path $\mathcal{C}$. This vector field itself depends only on the geometry of embedding of the planes into the ambient space. Its shape is clearly unaffected by reparametrization of the path. The angle between initial and final vectors is determined up to $2\pi$, which is reflected in the ambiguity of $\Delta\alpha$ discussed above.

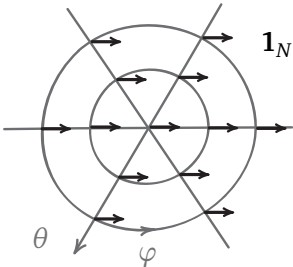

Figure 1.7: Basis section $\mathbf{1}_N$ of $TS^2$ near the north pole of the sphere.

**Exercise 1.3.** This exercise illustrates the $2\pi$ ambiguity of the parallel transport angle $\Delta\alpha$.

1. Introduce a new section $\mathbf{1}_N$, obtained by parallel transporting some vector from the north pole along all meridians (see Fig. 1.7). Express the section $\mathbf{1}_N$ in terms of $\mathbf{1} = e_\theta$.

2. Compute the connection coefficient $\omega_\varphi^N$ with respect to $\mathbf{1}_N$ using the transformation law.

3. Use these results to find the daily rotation angle $\Delta\alpha$ of the Foucault pendulum and compare with Eq. (1.46).

[2.3, §3.1.3, §4.1.2]

## 1.4 Connection on a vector bundle

### 1.4.1 Covariant derivative, parallel transport, and connection

The notions of parallel transport and covariant derivative are closely related to each other. To see this, consider a section $\mathbf{1}_{PT}$ of a tangent bundle $T\mathcal{B}$, which satisfies the equation of the parallel transport (1.38) along a curve $\mathcal{T}$. Let $\mathbf{1}$ be another basis section, such that along the curve we have

$$\mathbf{1} = e^{i\beta(\tau)}\mathbf{1}_{PT}. \tag{1.52}$$

Then the covariant derivative of $\mathbf{1}$ reads:

$$\nabla_\tau\mathbf{1} = i(\partial_\tau\beta)\mathbf{1}. \tag{1.53}$$

Comparing this with the definition of the complex connection coefficient (1.28), we conclude that $\omega_\tau$ is essentially the angular velocity (measured per unit of $\tau$) of rotation of the section $\mathbf{1}$ with respect to the section $\mathbf{1}_{PT}$. This explains the form of Eq. (1.47): observe that $\mathbf{1}_{PT}$ rotates with respect to the section $\mathbf{1}$ with velocity $-\omega_\tau$. In Eq. (1.47) we integrated this angular velocity and obtained the total angle of rotation of a parallel transported vector. One can also re-interpret the expression for the covariant derivative from this perspective. If we differentiate a section $\boldsymbol{v}$ of a constant modulus, the formula (1.31) simply describes the addition of the relative angular velocities. Indeed, the derivative $\nabla_\tau\boldsymbol{v}$ measures the angular velocity of $\boldsymbol{v}$ with respect to $\mathbf{1}_{PT}$, while the expression $(\partial_\tau + i\omega_\tau)\boldsymbol{v}$ corresponds to the sum

$$\text{(angular velocity of } \boldsymbol{v} \text{ w.r.t. } \mathbf{1}) \quad + \quad \text{(angular velocity of } \mathbf{1} \text{ w.r.t. } \mathbf{1}_{PT}). \tag{1.54}$$

The equation (1.38) defines the parallel transport along the curve from the covariant derivative. One can also go the other way round and define the covariant derivative from the parallel transport, as follows. Suppose that for each curve $\mathcal{T}$ on a surface $\mathcal{B}$, we know how to introduce a smooth section $\mathbf{1}_{PT}$ of $T\mathcal{B}$ over some neighborhood of the curve, which describes the parallel transport along the curve. From this data, we *define* the covariant derivative $\widetilde{\nabla}_\tau$ along $\mathcal{T}$ as

$$\widetilde{\nabla}_\tau\boldsymbol{v} = (\partial_\tau v)\mathbf{1}_{PT}, \tag{1.55}$$

where $\boldsymbol{v} = v\mathbf{1}_{PT}$ is some section of $T\mathcal{B}$. This definition implies that $\widetilde{\nabla}_\tau\mathbf{1}_{PT} = 0$, so the section $\mathbf{1}_{PT}$ indeed describes the parallel transport that corresponds to $\widetilde{\nabla}_\tau$ via Eq. (1.38). Moreover, the coordinate expression for $\widetilde{\nabla}_\tau\boldsymbol{v}$ has a familiar form:

**Exercise 1.4.** Let $\mathbf{1}$ be a basis section, such that $\mathbf{1} = e^{i\beta(\tau)}\mathbf{1}_{PT}$ on the curve $\mathcal{T}$. Find the coordinate expression of $\widetilde{\nabla}_\tau\boldsymbol{v}$ along the curve with respect to the basis section $\mathbf{1}$.

Covariant derivative and parallel transport are in fact two faces of a single structure called **connection on a vector bundle**, which can be defined by specifying either of them. Let us recapitulate the connections we have encountered thus far. For the tangent bundle $T\mathbb{R}^2$, we declared that the basis sections obtained from the Cartesian coordinates are constant (according to the identification given by Eq. (1.13)). In other words, these sections define a parallel transport along any curve on the plane. The corresponding covariant derivative is the ordinary component-wise differentiation (1.11). In a similar way, we specified a connection on $T\mathbb{R}^3$, which in turn allowed us to define a covariant derivative on the tangent bundle $T\mathcal{B}$ of a surface $\mathcal{B} \subset \mathbb{R}^3$. To this end, we used the orthogonal projection of the covariant derivative in $T\mathbb{R}^3$ to the fibers of $T\mathcal{B}$. To describe this relationship, we will call the resulting connection on $T\mathcal{B}$ a **projected connection**. All these connections are far from unique. While we will not discuss connections in full generality, note that one can use any set of smooth basis sections to define the parallel transport in $T\mathbb{R}^3$. This gives infinitely many connections on $T\mathbb{R}^3$, together with the corresponding projected connections on $T\mathcal{B}$. We will encounter a similar situation in Sec. 6.2.3.

### 1.4.2 Complex line bundle with connection

We have endowed the tangent bundle $T\mathcal{B}$ of a surface $\mathcal{B} \subset \mathbb{R}^3$ with several additional structures: an induced metric (1.7), a projected connection (1.15), and a complex structure introduced in Sec. 1.2.3. Note that none of these constructions relies on the condition that the fibers are tangent to the base space. Now we lift this condition and define these structures on a more general real plane bundle.

Consider a smooth, non-vanishing three-dimensional vector field $\boldsymbol{m}$ defined on a surface $\mathcal{B} \subset \mathbb{R}^3$. One can think of it as a section of the bundle $T\mathbb{R}^3|_{\mathcal{B}}$. At each point $p \in \mathcal{B}$, the vector $\boldsymbol{m}(p)$ determines the subspace $M_p \subset T_p\mathbb{R}^3$ such that $\boldsymbol{m}(p) \perp M_p$. This defines a vector bundle $M$ of real planes over $\mathcal{B}$, as shown in Fig. 1.8. In the particular case when the vectors $\boldsymbol{m}$ are surface normals, the bundle $M$ coincides with the tangent bundle $T\mathcal{B}$ of the surface $\mathcal{B}$. We define the bundle metric on $M$ as a metric induced from the bundle $T\mathbb{R}^3|_{\mathcal{B}}$.

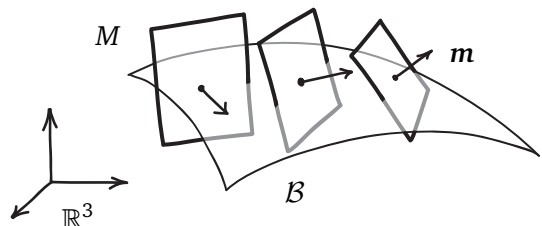

Figure 1.8: A real plane bundle $M$ defined by a vector field $\boldsymbol{m}$ over a two-dimensional surface $\mathcal{B} \subset \mathbb{R}^3$. The fibers of $M$ are the planes orthogonal to the vectors $\boldsymbol{m}$.

Each plane $M_p$ is oriented by its normal vector $\boldsymbol{m}$ and the right-hand rule (i.e., the orientation of the three-dimensional space $T_p\mathbb{R}^3$). Thus, it can be identified with a complex one-dimensional space, making $M$ into a complex line bundle over $\mathcal{B}$. Note that the switching of the direction of normals, $\boldsymbol{m} \to -\boldsymbol{m}$, leaves the real planes invariant, but changes their orientation. In complex terms, this amounts to the replacement $i \to -i$, or to complex conjugation. As we will see, it is important to distinguish between a complex line bundle and its conjugate version: this affects the sign of curvature (Sec. 2.2) and of the Chern number (Sec. 4.2.1).

As a final ingredient, we define the projected connection on $M$. This gives us the way to

perform the parallel transport of vectors and to find the covariant derivative of sections. Once a unit basis section $\mathbf{1}$ is chosen, the connection coefficient $\omega_\tau$ can be calculated along any curve $\mathcal{T}$ in $\mathcal{B}$. This allows one to compute the covariant derivative $\nabla_\tau$ of a section $\boldsymbol{v}$ of $M$ given by Eq. (1.31). In this way, we establish a correspondence:

$$\text{vector field } \boldsymbol{m} \text{ over } \mathcal{B} \subset \mathbb{R}^3 \quad \Rightarrow \quad \text{complex line bundle } M \text{ with connection over } \mathcal{B}. \qquad (1.56)$$

A crucial difference between the bundle $M$ and a tangent bundle is that sections of $M$ are not related to the base space. Recall that we constructed the basis sections $\{\boldsymbol{e}_\theta, \boldsymbol{e}_\varphi\}$ for $TS^2$ from velocities $\partial_\alpha \boldsymbol{r}$ of a point moving on the sphere along the coordinate curves (Sec. 1.1.2). In contrast, the fibers and sections of the bundle $M$ are independent of the geometry and coordinates of the base space $\mathcal{B}$. Finally, we note that the construction is easily generalized to base spaces of other dimension: for example, $\mathcal{B}$ can be a curve, or a three-dimensional region of $\mathbb{R}^3$.

## 1.5 Summary and outlook

Above, we introduced the language of vector bundles, developed machinery of covariant differentiation and discussed the closely related concept of the parallel transport. Vector bundles arise naturally as families of tangent spaces containing velocity vector fields of points moving on a surface. Further, they are generalized to the cases when such family of vector spaces is not related to the surface profile. In what follows, we will consider the local (geometric) and global (topological) characteristics of vector bundles, their mutual interplay and their manifestations in the models of physical phenomena.

One of the key tools in the study of geometry of vector bundles and their physical applications is the covariant derivative. Let us summarize our main findings related to the covariant derivative and parallel transport.

- Covariant derivative $\nabla_\tau$ gives a basis-independent way to differentiate sections of a vector bundle.

- We defined the covariant derivative by projection from the ambient space (which is not the most general form of this operator).

- Covariant derivative of a basis section $\mathbf{1}$ defines connection coefficients $\omega_\tau$, which enter in the coordinate expression for $\nabla_\tau$.

- Connection coefficients depend both on the coordinates used for the base space and on the choice of the basis sections, as described by the respective transformation laws.

- Parallel transport is a way to move a vector along a curve, such that the covariant derivative of the resulting vector field vanishes.

- After traversing a closed path, parallel transported vector need not return to its initial state. In our case, the change is described by the parallel transport angle $\Delta\alpha$.

- The parallel transport angle can be computed by integrating the connection coefficient. The result is independent of the parametrization of the path. The value of $\Delta\alpha \bmod 2\pi$ does not depend on the choice the basis section.

In our discussion, we used the ambient space $\mathbb{R}^3$ to describe geometric objects and additional structures on them. In this way, the embedding of a surface $\mathcal{B} \subset \mathbb{R}^3$ enabled us to define the induced metric and projected connection on the tangent bundle $T\mathcal{B} \subset T\mathbb{R}^3|_{\mathcal{B}}$. All these objects can be defined abstractly, without any reference to the ambient space. Here, we give a brief outline of these constructions. The reader can find further details in the textbooks on differential geometry, such as Refs. [3–5].

▷ **Smooth manifolds.** An abstract version of a two-dimensional surface is a smooth 2-manifold. Informally, a **smooth $n$-manifold** $\mathcal{B}^n$ is a space that can be locally identified with a region of the $n$-dimensional Euclidean space $\mathbb{R}^n$. Such an identification is made by a coordinate system, which associates an $n$-tuple of numbers $(x_1(p), \ldots, x_n(p)) \in \mathbb{R}^n$ to a point $p \in \mathcal{B}^n$. In general, $\mathcal{B}^n$ cannot be covered by a single coordinate system. Instead, one covers $\mathcal{B}^n$ with several overlapping coordinate charts. On the overlaps, the different coordinate systems must be related by smooth coordinate transformations. This allows one to differentiate and integrate functions on $\mathcal{B}^n$ expressed in terms of local coordinates. Strictly speaking, such formalism must be applied even for the two-dimensional sphere $S^2$, since the standard coordinates $(\theta, \varphi)$ have singularities. But in practice, one simply avoids computing derivatives $\partial_\varphi f$ and $\partial_\theta f$ at the poles. As for the integration, singularities are point-like and do not affect the values of surface integrals.

▷ **Riemannian geometry.** By considering velocities of points moving on a smooth manifold $\mathcal{B}^n$, one can define tangent vectors. As in our discussion above, this leads to the concept of tangent bundle $T\mathcal{B}^n$. A smooth choice of the inner product in the tangent spaces defines a metric and makes $\mathcal{B}^n$ a **Riemannian manifold**. A connection on $T\mathcal{B}^n$ is defined as a differential operator with certain properties, such as linearity and Leibniz rule. One connection on the tangent bundle of a Riemannian manifold, known as **Levi-Civita** connection, is especially important. In particular, it is compatible with the metric, which means that the parallel transport preserves lengths of vectors and angles between them. For example, the projected connection on $TS^2$ introduced above is the Levi-Civita connection compatible with the induced metric.

Riemannian geometry provides the mathematical language for General relativity. The gravitational field is described in terms of the metric and corresponding Levi-Civita connection on the tangent bundle of a 4-manifold, which represents the spacetime. The trajectory of the inertial motion of a test mass is a **geodesic**, which can be characterized by the property that the velocity vector is parallel transported along the curve.

▷ **Abstract vector bundles.** Finally, we comment on the precise meaning of the "collection of smoothly varying vector spaces", which are not necessarily tangent to the base space. An abstract **real vector bundle** $V$ over a manifold $\mathcal{B}^n$ is itself a smooth manifold with a special form of the coordinate transformations. If its fibers have dimension $k$, a vector $\boldsymbol{v}(p) \in V_p$ is described in coordinates as

$$(x_1(p), \ldots, x_n(p), v_1(p), \ldots, v_k(p)) \in \mathbb{R}^n \times \mathbb{R}^k. \tag{1.57}$$

The first $n$ numbers are the coordinates of the point $p \in \mathcal{B}^n$, while the remaining $k$ numbers are the fiber coordinates, or the components of $\boldsymbol{v}(p)$ with respect to the basis $\{\boldsymbol{e}_1(p), \ldots, \boldsymbol{e}_k(p)\}$ for $V_p$. For a coordinate chart $U \subset \mathcal{B}^n$, the fiber coordinates in $V|_U$ are fixed by choosing $k$ smooth basis sections over $U$. On the overlaps, there are several sets of basis sections, and corresponding fiber coordinates are related by smooth *linear* transformations.

The formalism of abstract vector bundles lies at the heart of the classical gauge theories, whose quantized versions describe the fundamental interactions in the Standard model of particle physics. We will touch upon the geometry of gauge fields in the next section. However, we will not need this abstract machinery for the theory of topological insulators. All relevant vector bundles will arise naturally as subbundles of other bundles and will be equipped with projected connections.

# 2 Electromagnetic field and curvature of connection

Connections on vector bundles provide the natural mathematical language for description of classical gauge fields. A characteristic feature of a gauge theory is a certain redundancy: there are many field configurations that correspond to a given physical situation. Such configurations are related by coordinate-dependent **gauge transformations**. Predictions of any physical model that includes gauge fields must be invariant under gauge transformations — in short, the model must have **gauge invariance**.

In this section, we consider electromagnetism as a simple example of gauge theory and discuss the geometric meaning of gauge invariance. Then we define the curvature of connection, which describes parallel transport locally and corresponds to the field strength tensor of a gauge field. As the name suggests, in some cases the curvature can be related to the "shape" of the bundle. Our main task will be to find such a relation between the curvature of projected connection on a real plane bundle (1.56) and the configuration of the vector field of plane normals. This will allow us to construct an analogy between the curvature of a two-dimensional surface and the strength of the magnetic field. Such intuitive picture will be helpful in later sections, when we will deal with the more abstract Berry curvature.

## 2.1 Electromagnetism as a gauge theory

### 2.1.1 Algebra of gauge invariance

Recall that the field strengths $E$ and $B$ can be expressed in terms of the scalar potential $\phi$ and the vector potential $A$ as

$$E = -\nabla\phi - \frac{\partial A}{\partial t}, \qquad B = [\nabla \times A]. \tag{2.1}$$

The choice of the potentials is not unique. Consider the following gauge transformations:

$$A \to A' = A - \nabla\chi, \qquad \phi \to \phi' = \phi + \frac{\partial \chi}{\partial t}, \tag{2.2}$$

where $\chi(x, y, z, t)$ is some smooth function. One checks that the new potentials describe the same values of the field strength. Thus, the observable quantities are invariant under these transformations.

The deeper significance of the potentials is revealed in the context of quantum mechanics. To see this, we derive the Schrödinger equation of a particle in the background electromagnetic field. The classical Hamiltonian of such particle reads

$$H = \frac{1}{2m}\sum_j (p_j - qA_j)^2 + q\phi, \tag{2.3}$$

where $q$ is the electric charge of the particle and $j = x, y, z$. If the particle is microscopic, quantum mechanics prescribes to replace the momentum with the differential operator $\hat{p}_j = -i\hbar\partial_j$, so that the quantum Hamiltonian becomes

$$\hat{H} = \sum_j \frac{1}{2m}(-i\hbar\partial_j - qA_j)^2 + q\phi. \tag{2.4}$$

Then the wave function $\psi$ describing the particle obeys the Schrödinger equation

$$i\hbar\partial_t\psi = \hat{H}\psi, \tag{2.5}$$

which can be rewritten as

$$i\hbar\left(\partial_t + i\frac{q}{\hbar}\phi\right)\psi = \sum_j \frac{(-i\hbar)^2}{2m}\left(\partial_j - i\frac{q}{\hbar}A_j\right)^2\psi. \tag{2.6}$$

Note that the electromagnetic field is represented in this equation by the potentials. For a fixed wave function $\psi$, the equation is not invariant under the gauge transformations (2.2). This is unacceptable, since in such case the behavior of the particle in the electromagnetic field would be affected by an arbitrary choice of the function $\chi$. The gauge invariance is restored if we demand that the gauge transformations (2.2) be accompanied by the following change of the wave function:

$$\psi \to \psi' = \psi \exp\left(-i\frac{q}{\hbar}\chi\right). \tag{2.7}$$

Thus, the electromagnetic potentials of the background field are closely related to the phase of the wave function of the particle.

The argument goes in the other direction, too. Note that the Schrödinger equation is invariant under the global change of the phase of the wave function. Indeed, if $\psi$ is a solution of Eq. (2.5), then $\psi e^{i\alpha}$ also satisfies it, for some constant $\alpha \in \mathbb{R}$. And what about *local* phase rotations? It follows from our discussion that one can change the phase of $\psi$ by an arbitrary function, as in Eq. (2.7), provided that the potentials in the Schrödinger equation (2.6) are changed according to Eqs. (2.2). In particular, this works in the case when the electromagnetic field is zero. The corresponding potentials are said to describe the **pure gauge**.

### 2.1.2 Geometry of gauge invariance

There is a concise way to explain the form of the gauge transformations (2.2), their relation to the phase rotations (2.7), and the invariance of the Schrödinger equation. Observe that the combinations in parentheses in Eq. (2.6) look like the coordinate expression for the covariant derivative in Eq. (1.31). Thus, we can "multiply" the Schrödinger equation (2.6) by the basis section **1** on the right and obtain

$$i\hbar\nabla_t\boldsymbol{\psi} = \sum_j \frac{1}{2m}(-i\hbar\nabla_j)^2\boldsymbol{\psi}, \tag{2.8}$$

which suggests the following geometric interpretation. Over spacetime $\mathbb{R}^4$, we have a complex line bundle with connection. A section $\boldsymbol{\psi}$ of this bundle describes the state of the quantum particle, and the connection represents the background electromagnetic field. Once a basis section **1** is chosen, the section $\boldsymbol{\psi}$ can be expressed as

$$\boldsymbol{\psi} = \psi\mathbf{1}, \tag{2.9}$$

where $\psi$ is the usual complex-valued wave function. Here, we assume that the section **1** exists; as we will see in Sec. 4, this is a safe assumption provided that the region of interest does not contain magnetic monopoles.

The connection determines four covariant derivatives $\nabla_\tau$ along the coordinate curves, $\tau = x, y, z, t$. The spatial covariant derivatives give the momentum operators:

$$\hat{p}_j\boldsymbol{\psi} = -i\hbar\nabla_j\boldsymbol{\psi}, \tag{2.10}$$

while the temporal derivative $\nabla_t$ enters the left-hand side of the Schrödinger equation. The connection coefficients with respect to the basis section **1** are proportional to the electromagnetic potentials:

$$\omega_t = \frac{q}{\hbar}\phi, \qquad \omega_j = -\frac{q}{\hbar}A_j. \tag{2.11}$$

Now let us see what happens when we choose another basis section $\mathbf{1}' = e^{i\beta}\mathbf{1}$, where $\beta(x, y, z, t)$ is a smooth real function. First, note that the section $\boldsymbol{\psi}$ is an invariant object:

$$\boldsymbol{\psi} = \psi\mathbf{1} = \psi'\mathbf{1}'. \tag{2.12}$$

Thus, the new wave function reads

$$\psi' = \psi e^{-i\beta}. \tag{2.13}$$

From the transformation law for the connection coefficients (1.34) we find that

$$\omega'_t = \frac{q}{\hbar}\phi + \partial_t\beta = \frac{q}{\hbar}\left(\phi + \frac{\hbar}{q}\partial_t\beta\right), \tag{2.14}$$

and likewise for $\omega_j$. The transformations in the last two equations are equivalent to those given by Eqs. (2.7) and (2.2) if we substitute $\beta = \frac{q}{\hbar}\chi$. In this picture, a gauge transformation manifestly does not affect the state of the particle and the configuration of the electromagnetic field. It merely changes the way of description of the same physical and geometrical situation.

The correspondence between the connection coefficients and gauge potentials can be used to interpret any complex line bundle $M$ with connection in terms of the electromagnetic field, as follows. First, choose a basis section $\mathbf{1}$ of $M$, which determines the connection coefficients along the coordinate curves. Interpret the coordinates as describing the physical spacetime, and associate the connection coefficients with the gauge potentials via Eq. (2.11). Now suppose that a section $\boldsymbol{v}$ of $M$ represents the wave function of a particle, which obeys the Schrödinger equation (2.8). Assume that the charge of the particle equals the positive elementary charge, $q = e$. Then the particle feels the background electromagnetic field, whose potentials are determined by the connection coefficients with respect to the basis section $\mathbf{1}$. One can thus associate this electromagnetic field with the given complex line bundle:

$$\text{complex line bundle } M \text{ with connection} \quad \Rightarrow \quad \text{configuration of the EM field.} \tag{2.15}$$

Below we consider such electromagnetic interpretation of the complex line bundle defined by (1.56). A solid understanding of this construction will prove useful in the context of topological insulators: they are often described in terms of the "magnetic field in the momentum space", which is introduced in a similar manner.

### 2.1.3 Magnetic flux and parallel transport angle

Consider a three-dimensional vector field $\boldsymbol{m}$ over $\mathbb{R}^3$, and let $M$ be a complex line bundle defined by $\boldsymbol{m}$ according to (1.56). We interpret $\mathbb{R}^3$ as a physical space with Cartesian coordinates $x_j = x, y, z$. The connection coefficients $\omega_j$ with respect to a smooth basis section $\mathbf{1}$ correspond under (2.15) to the components of the magnetic vector potential $A_j$. What is the geometric meaning of the magnetic field strength $\boldsymbol{B}$? It must be a local gauge-invariant quantity determined by the geometry of the bundle, that is, by the configuration of the field $\boldsymbol{m}$. Recall that the parallel transport angle $\Delta\alpha$, albeit non-local, has a certain degree of gauge invariance, as discussed in Sec. 1.3.3. Consider an oriented closed contour $\mathcal{C} \subset \mathbb{R}^3$ defined parametrically by three functions $x_j(\gamma)$. Then the connection coefficient with respect to the parameter $\gamma$ is given by the sum $\omega_\gamma = (\partial_\gamma x_j)\omega_j$, according to Eq. (1.37). Let us find the parallel transport angle for the contour $\mathcal{C}$:

$$\Delta\alpha(\mathcal{C}) = \int_{\mathcal{C}}(-\omega_\gamma)d\gamma = \int_{\mathcal{C}}(-\omega_j)dx_j = \frac{q}{\hbar}\int_{\mathcal{C}}\boldsymbol{A}\cdot d\boldsymbol{l} = 2\pi\frac{\Phi(\Sigma)}{\Phi_0}, \tag{2.16}$$

where $\Phi(\Sigma)$ is the magnetic flux threading some surface $\Sigma$ bounded by the contour, $q = e$ is the charge of the particle, and $\Phi_0 = \frac{h}{e}$ is the magnetic flux quantum. Thus, we can express the flux

through the surface $\Sigma$ in terms of the parallel transport angle associated with its boundary $\mathcal{C}$:

$$\Phi(\Sigma) = \frac{\Delta\alpha(\mathcal{C})}{2\pi}\Phi_0 \,. \tag{2.17}$$

The problem with the last equation is that the value of $\Delta\alpha(\mathcal{C})$ can be changed by $2\pi$ by the choice of the basis section along the contour (see Sec. 1.3.3). On the other hand, the flux $\Phi(\Sigma)$ is a well-defined observable quantity, which is determined by the magnetic field strength at the surface. The equality (2.17) suggests that the geometry of the bundle over the surface $\Sigma$ must somehow fix the unique value of $\Delta\alpha(\mathcal{C})$. We invite the reader to think about how this might happen, before we give the answer in Sec. 2.2.2. Our discussion will be based on the following simple calculation.

Consider a small oriented plaquette $\square$ lying on the $xy$ plane, with a contour $\partial\square$ as a boundary:

$$\partial\square = (x_0, y_0) \to (x_0 + \Delta x, y_0) \to (x_0 + \Delta x, y_0 + \Delta y) \to (x_0, y_0 + \Delta y) \to (x_0, y_0) \,. \tag{2.18}$$

Let us compute the magnetic flux through the plaquette $\square$ as a contour integral of the potential:

$$\Phi(\square) = \int_{\partial\square} \boldsymbol{A} \cdot d\boldsymbol{l} \,. \tag{2.19}$$

For the second edge, in linear approximation, one has:

$$\int_{y_0}^{y_0+\Delta y} A_y(x_0 + \Delta x, y) dy \approx A_y(x_0 + \Delta x, y_0)\Delta y \approx A_y(x_0, y_0)\Delta y + \frac{\partial A_y}{\partial x}\bigg|_{(x_0, y_0)} \Delta x \Delta y \,. \tag{2.20}$$

Adding together analogous expressions for the other edges of an infinitesimal contour, we obtain a familiar result:

$$\Phi(\square) = (\partial_x A_y - \partial_y A_x) dx dy = F_{xy} dx dy = B_z dS \,, \tag{2.21}$$

where $F_{xy}$ is a component of a field strength tensor, or equivalently the $z$ component of the magnetic field strength $\boldsymbol{B}$.

### 2.1.4 A note on orientation

One last technical note before we begin. To fix the signs of line and surface integrals, we need to define an **orientation** of spaces in a coherent way.

- One says that a contour is oriented, if there is a preferred direction. This can be fixed by specifying a tangent vector at some point.

- For a two-dimensional surface, orientation means a preferred sense of rotation. This is given by an ordered pair of tangent vectors.

- To orient a three-dimensional space, one chooses handedness, or an ordered triple of vectors.

These definitions are illustrated in Fig. 2.1. We will deal only with **orientable** spaces, meaning that they admit a consistent global choice of orientation in all tangent spaces.

Once a given space is oriented, there is a standard way to orient its subspaces. For example, let us orient $\mathbb{R}^3$ by the right-hand rule. To orient a surface $\mathcal{B} \subset \mathbb{R}^3$, we need to specify a normal $\boldsymbol{n}$ (if the surface is the boundary of some region, it is common to choose the outer normal). Then a pair $\{\boldsymbol{e}_1, \boldsymbol{e}_2\}$ of tangent vectors is positively-oriented if the triple $\{\boldsymbol{n}, \boldsymbol{e}_1, \boldsymbol{e}_2\}$

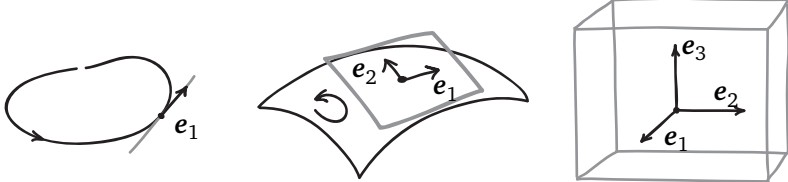

Figure 2.1: Orientation of spaces given by the ordered sets of tangent vectors.

is right-handed in $\mathbb{R}^3$. Note that the orientation of the sphere $S^2$ we used above follows this convention.

In a similar way, one can orient any contour $\mathcal{C}$ on the surface. Suppose the contour is a boundary of some region $\Sigma \subset \mathcal{B}$, which is denoted as $\mathcal{C} = \partial\Sigma$. Consider a pair of vectors tangent to the surface, $\{e_n, e_1\}$, where $e_1$ is tangent to the contour, and $e_n$ is an outer normal of the region $\Sigma$. Then $e_1$ defines positive direction in $\mathcal{C}$, if the pair is positively-oriented. As an example, consider a contour which is the boundary a small cap on the sphere $S^2$. The reader should check that the positive direction on the contour corresponds to the counterclockwise motion around the cap. On the other hand, the contour can also be considered as the boundary of the large region that covers the whole sphere except the cap. In this case, the orientation of the contour is reversed, since the outer normal $e_n$ now points inside the cap.

We will also need the notion of **positively-oriented coordinates**. To any coordinate curve one can associate a velocity vector of a point that moves in the positive direction. Thus, the orientation defined in terms of ordered sets of tangent vectors similarly applies to the ordered sets of coordinates.

## 2.2 Curvature of connection

In Sec. 2.1.3, we considered the electromagnetic interpretation (2.15) of the complex line bundle (1.56) defined by a vector field $m$ over a surface $\Sigma \subset \mathbb{R}^3$. This interpretation raised several questions:

- How to remove the $2\pi$ ambiguity from Eq. (2.17), which relates the flux $\Phi(\Sigma)$ with the parallel transport angle $\Delta\alpha$?

- How to compute $\Phi(\Sigma)$ using the vector field $m$ on $\Sigma$?

- What is, geometrically, the magnetic field strength $B$?

Below, we start by introducing the geometric analogue of the filed strength *tensor*, and consider Stokes' theorem, which answers the first question. The second question is addressed in Sec. 2.2.4. Finally, we discuss the third question and its applications in geometry of surfaces in Sec. 2.3.

### 2.2.1 Curvature components

We generalize the calculation that led to Eq. (2.21) as follows. Let $M$ be a complex line bundle with connection over an oriented surface $\Sigma \subset \mathbb{R}^3$ with positively-oriented coordinates $(x_1, x_2)$. Let us compute the parallel transport angle for the boundary of a coordinate plaquette $\square$:

$$\Delta\alpha(\partial\square) = \int_{\partial\square} (-\omega_\gamma) d\gamma. \tag{2.22}$$

SciPost Phys. Lect. Notes 67 (2023)

By the same token as above, we obtain

$$\Delta\alpha(\partial\Box) = \big[\partial_1(-\omega_2) - \partial_2(-\omega_1)\big]dx_1 dx_2\,. \tag{2.23}$$

The quantity

$$f_{12} = \partial_1(-\omega_2) - \partial_2(-\omega_1)\,, \tag{2.24}$$

is called the 12-component of the **curvature of connection**. Thus, the curvature can be thought of as a local characterization of the parallel transport. Importantly, the passage from (2.22) to (2.23) assumes that connection coefficients are smooth functions of coordinates, that is, the basis section **1** over $\Box$ is smooth. Note also that the sign of $\Delta\alpha$ depends both on the orientation of the base space and on the orientation of the fiber (the latter is given here by the complex structure, cf. Sec. 1.4.2). If the base space has dimension more than two, there is a curvature component for each pair of coordinate indices. Since the electromagnetic field strength is invariant under gauge transformations, one can expect that the curvature is independent of the basis choice.

> **Exercise 2.1.** Consider another basis section, $\mathbf{1}' = e^{i\beta(x_1,x_2)}\mathbf{1}$, where $\beta$ is some smooth real function. Show that the curvature component given by Eq. (2.24) is invariant under the corresponding transformation of connection coefficients.

> **Exercise 2.2.** One can define the curvature component in a manifestly gauge-invariant way through the commutator of covariant derivatives $\nabla_j$ of a section $\mathbf{v}$ along the coordinate curves $x_j$. Prove that
>
> $$[\nabla_1, \nabla_2]\mathbf{v} = \frac{1}{i}f_{12}\mathbf{v}\,. \tag{2.25}$$

[§3.4]

### 2.2.2 Curvature for $TS^2$ and Stokes' theorem

As an example, we calculate the curvature of the projected connection on the tangent bundle over the sphere $TS^2$. Recall that if we choose $\mathbf{1} = \mathbf{e}_\theta$ as a basis section, the only non-zero connection coefficient is $\omega_\varphi = \cos\theta$. Then

$$\Delta\alpha(\partial\Box) = (\partial_\theta(-\omega_\varphi) - \partial_\varphi(-\omega_\theta))d\theta d\varphi = \sin\theta\, d\theta d\varphi\,. \tag{2.26}$$

Thus, the $\theta\varphi$-component of the curvature reads

$$f_{\theta\varphi} = \sin\theta\,. \tag{2.27}$$

This implies that for any region $U$ on the sphere, it is impossible to define a **covariantly constant** vector field $\mathbf{v}$, which satisfies $\nabla_\tau\mathbf{v} = 0$ along any curve in the region $U$ (as we anticipated in Sec. 1.3.1). Indeed, otherwise one would be able to use $\mathbf{v}$ as a basis section, which would give vanishing connection coefficients and zero curvature, and we know that the curvature must be independent of the basis choice. One might be tempted to conclude that the converse is true: if the curvature vanishes in some region $U$ of a surface, then there exists a covariantly constant section over $U$. However, this is not the case, as we will see in Sec. 2.3.3.

Observe that for the contour $\partial\Box$, the parallel transport angle is given by the elementary solid angle: $\Delta\alpha(\partial\Box) = d\Omega$, as illustrated in Fig. 2.2. Consider now a region $\Sigma \subset S^2$ with a boundary $\partial\Sigma$ parameterized by $\gamma$. Assume that the section **1** is smooth over $\Sigma$. Let us break $\Sigma$

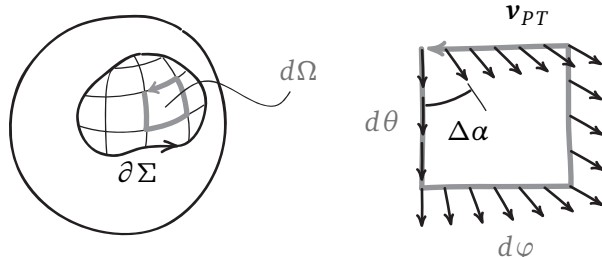

Figure 2.2: LEFT: A region $\Sigma$ of the sphere divided by the coordinate curves into small plaquettes. RIGHT: Parallel transport angle for an elementary coordinate contour $\Delta\alpha(\partial\boxdot)$.

into small plaquettes $\boxdot$ and add together all contour integrals over their boundaries $\partial\boxdot$. Since the contributions from the internal links cancel out, we have for the parallel transport angle

$$\int_{\partial\Sigma}(-\omega_\gamma)d\gamma = \sum_\boxdot \int_{\partial\boxdot}(-\omega_j)dx_j = \sum_\boxdot \Delta\alpha(\partial\boxdot) = \int_\Sigma d\Omega\,. \qquad (2.28)$$

It follows that the parallel transport angle for the boundary $\partial\Sigma$ is given by the solid angle spanned by the region $\Sigma$:

$$\Delta\alpha(\partial\Sigma) = \Omega(\Sigma)\,. \qquad (2.29)$$

This is a purely geometric quantity, which is manifestly independent of the basis choice. If the region $\Sigma$ is negatively-oriented, so are the constituent plaquettes, and the parallel transport angle changes its sign.

Now we reconcile this result with the $2\pi$ ambiguity of $\Delta\alpha$ related to the basis choice along the boundary $\partial\Sigma$. Recall that Eq. (2.23) requires that the basis section be smooth over the plaquette $\boxdot$. Thus, Eq. (2.28) holds only if the section is smooth over all plaquettes that constitute the region $\Sigma$. It turns out that there is no ambiguity in $\Delta\alpha$, once we demand that the basis section be smooth over the whole region $\Sigma$. Indeed, let **1** be such a section. Then the basis section changes along $\partial\Sigma$ that lead to $2\pi$ jumps of $\Delta\alpha(\partial\Sigma)$ inevitably create singularities of the section in the interior of $\Sigma$, and thus are not allowed. This smoothness condition removes the ambiguity from Eq. (2.17).

> **Exercise 2.3.** Consider a circle $\mathcal{C}$ of latitude $\theta$ as a boundary of the spherical cap covering the north pole of the sphere $S^2$. Compute the parallel transport angle $\Delta\alpha(\mathcal{C})$ as a surface integral of the curvature over the cap. Compare the answer with Eq. (1.46) and with the results of Exercise 1.3, and explain your observations.

The approach used in Eq. (2.28) is not restricted to $TS^2$ and can be applied to any complex line bundle $M$. For a two-dimensional region $\Sigma$ of the base space, the parallel transport angle over the boundary contour $\partial\Sigma$ is related to the curvature of the bundle over $\Sigma$ as

$$\Delta\alpha(\partial\Sigma) = \int_{\partial\Sigma}(-\omega_\gamma)d\gamma = \int_\Sigma f_{12}dx_1 dx_2\,, \qquad (2.30)$$

provided that the basis section **1** is smooth over $\Sigma$, so that there is no ambiguity in the first integral. In terms of electromagnetism and vector calculus, this is a version of the **Stokes'**

**theorem**, which relates circulation of the vector potential with the magnetic flux piercing the surface:

$$\int_{\partial\Sigma} \boldsymbol{A} \cdot d\boldsymbol{l} = \int_{\Sigma} [\boldsymbol{\nabla} \times \boldsymbol{A}] \cdot d\boldsymbol{S}. \tag{2.31}$$

Note that the last equation is a theorem from vector analysis applied to three-dimensional vector field $\boldsymbol{A}$ over a surface $\Sigma$ in $\mathbb{R}^3$. In contrast, Eq. (2.30) belongs to a very different context: it characterizes the geometry of a complex line bundle with connection defined over the surface $\Sigma$.

### 2.2.3  Curvature of real plane bundle

Consider the plane bundle $M$ over a surface $\Sigma$ defined by the vector field $\boldsymbol{m}$ of plane normals according to (1.56). We wish to relate the curvature of the projected connection on $M$ to the configuration of the field $\boldsymbol{m}$. To achieve this, we first reduce the problem to the known case of $TS^2$ and then transfer the results from the sphere back to the surface $\Sigma$.

Note that the nowhere-vanishing three-dimensional vector field $\boldsymbol{m}$ over $\Sigma$ gives rise to a map from $\Sigma$ to $S^2$. Indeed, at each point $p$, the vector $\boldsymbol{m}(p)$ specifies a ray in $\mathbb{R}^3$, which intersects the sphere $S^2 \subset \mathbb{R}^3$ in a point. We define the map $m : \Sigma \to S^2$ by sending $p$ to this point of intersection, as shown in Fig. 2.3. Thus, we have the correspondence

$$\text{non-vanishing vector field } \boldsymbol{m} \text{ over } \Sigma \quad \Rightarrow \quad \text{map } m : \Sigma \to S^2. \tag{2.32}$$

Let $(x_1, x_2)$ be the coordinates on the surface near the point $p \in \Sigma$. In coordinates, the map $m$ is described by two functions $\theta(x_1, x_2)$ and $\varphi(x_1, x_2)$. In a "good" situation (to be specified in a moment), one can consider these functions as a definition of the new coordinates on the sphere near $m(p)$, as discussed in Sec. 1.2.4. Let us express the curvature of $TS^2$ in these coordinates.

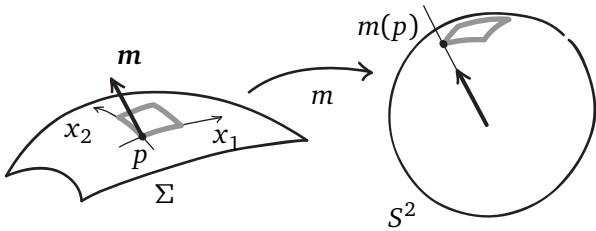

Figure 2.3: Vector field $\boldsymbol{m}$ over the surface $\Sigma$ gives rise to a map $m : \Sigma \to S^2$. A coordinate expression of the map $m$ allows one to define new coordinates on the sphere.

Using Eq. (1.37), we have:

$$f_{12} = \partial_1(-\omega_2) - \partial_2(-\omega_1) = \partial_1(-(\partial_2\alpha)\omega_\alpha) - [1 \leftrightarrow 2], \tag{2.33}$$

where we sum by $\alpha = \theta, \varphi$, thus extending the summation convention to the names of variables used as indices. By the chain rule, $\partial_1\omega_\alpha$ is given by the sum

$$\partial_1\omega_\alpha = (\partial_\beta\omega_\alpha)(\partial_1\beta). \tag{2.34}$$

**Exercise 2.4.**  Show that the curvature component with respect to the coordinates $(x_1, x_2)$ on the sphere is

$$f_{12} = J f_{\theta\varphi}, \tag{2.35}$$

where $J$ stands for the Jacobian determinant

$$J = \det \begin{pmatrix} \partial_1 \theta & \partial_2 \theta \\ \partial_1 \varphi & \partial_2 \varphi \end{pmatrix}. \tag{2.36}$$

[§3.3.4]

The matrix in the last equation is known as the **differential** $Dm$ of the map $m$, which is a linear transformation acting from the tangent space at $p$ to the tangent space at $m(p)$. The differential maps velocity vector of a point that moves through $p$ to the velocity vector of its image. Thus, if the images of velocity vectors associated with coordinate curves are linearly independent, the images of these curves can define coordinates near $m(p)$. This happens when $Dm$ is non-degenerate, that is, $J$ is non-zero. It is the "good" situation we referred to above.

Now we need to return from the sphere to the surface $\Sigma$. For each point $p \in \Sigma$ one can identify $M_p$ and $T_{m(p)}S^2$ as subspaces of $\mathbb{R}^3$. This allows us to transfer the section of $TS^2$ over some neighborhood of $m(p)$ to the section of $M$ near $p$. The curvature of $M$ over $\Sigma$ is determined by the parallel transport angle $\Delta\alpha(\partial\square)$ for the infinitesimal contour $\partial\square$ near the point $p$. Now observe that the planes over this contour, the basis section, and parametrization are by construction equivalent to those of the contour $m(\partial\square)$ on the sphere. Note also that in the degenerate case $J = 0$, the coordinate contour is mapped to a line or even a point, so the angle $\Delta\alpha(\partial\square) = 0$, and the curvature vanishes. We conclude that $f_{12}$ given by Eq. (2.35) is the desired curvature of the projected connection on the bundle $M$.

### 2.2.4 Flux as total solid angle

We are now in a position to compute the magnetic flux though the surface $\Sigma$ that corresponds under (2.15) to the complex line bundle defined by a vector field $\boldsymbol{m}$ over $\Sigma$ according to (1.56). To this end, let us use the Stokes' theorem to find the parallel transport angle $\Delta\alpha(\partial\Sigma)$ along the boundary contour of the surface $\Sigma$. The curvature of the plane bundle $M$ over $\Sigma$ is given by Eq. (2.35), and the curvature $f_{\theta\varphi}$ of $TS^2$ equals $\sin\theta$. We have:

$$\Delta\alpha(\partial\Sigma) = \int_\Sigma f_{12} dx_1 dx_2 = \int_\Sigma J \sin\theta \, dx_2 dx_2 = \int_{m(\Sigma)} d\Omega. \tag{2.37}$$

In the last equality we used change of variables formula for multi-dimensional integrals. The result is, in a sense, the **total solid angle** $\Omega(m(\Sigma))$ spanned by the image $m(\Sigma)$ on the sphere.[5] Note that integration takes the sign of $J$ into account. Next, we switch to the physical interpretation and upgrade Eq. (2.17) to the basis-independent version:

$$\Phi(\Sigma) = \Phi_0 \frac{\Omega(m(\Sigma))}{2\pi}. \tag{2.38}$$

We conclude that the magnetic flux through the surface $\Sigma$ described by the plane bundle $M$ with projected connection is determined by the total solid angle spanned by the plane normals $\boldsymbol{m}$.

To clarify the geometric meaning of this result, let us first consider an analogous situation in lower dimension. Let $\boldsymbol{m}$ be a planar vector field on a line segment $L$, as shown in Fig. 2.4. Similarly to the construction (2.32), the field defines a map $m : L \to S^1$ to the circle. Assume that $L$ and $S^1$ are oriented, and $\gamma$ and $\beta$ are their respective positively-oriented coordinates.

---

[5]In this section, we use the term "image" in the sense that differs from the standard mathematical notion (that is, a subset of the codomain of a function containing all points that have pre-images). By image $m(\Sigma)$, we mean the surface traced out by $m(p)$ as $p$ varies in $\Sigma$. In particular, this surface can have multiple layers.

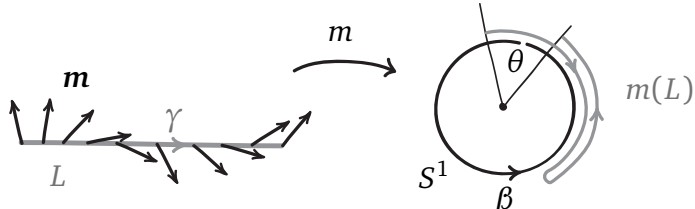

Figure 2.4: Planar vector field $\boldsymbol{m}$ on a line segment $L$ defines a map $m$ from $L$ to the circle $S^1$.

In these coordinates, the map $m$ is given by a function $\beta(\gamma)$. We are interested in the total angle $\theta$ spanned by the image $m(L)$:

$$\theta = \int_{m(L)} d\beta = \int_L \frac{d\beta}{d\gamma} d\gamma. \qquad (2.39)$$

Essentially, we are integrating the angular velocity and obtain the resulting angle of rotation. The image $m(L)$ has folding, but different layers have opposite signs of the derivative, and their contributions cancel each other out. Note that the sign of the derivative also tells us whether the map preserves the orientation at a given point. Finally, the value of $\theta$ is determined by the image $m(\partial L)$ of the boundary of the line segment, or its endpoints. If the image $m(L)$ covers the circle more that once, this value will be shifted by an integer multiple of $2\pi$.

These observations are readily generalized to the two-dimensional situation described by Eq. (2.37). Here, the Jacobian plays the role of the derivative above. Once the coordinates $(x_1, x_2)$ and $(\theta, \varphi)$ are positively-oriented, the sign of $J(p)$ indicates whether the map $m$ preserves orientation at $p$ (one can deduce this from the geometric meaning of the differential). If the image $m(\Sigma)$ develops folding, the overlapping layers with different signs of $J$ do not contribute to the integral. The parallel transport angle $\Delta\alpha(\partial\Sigma)$ is determined by the solid angle spanned by the image of the boundary $m(\partial\Sigma)$. This value can be shifted by $4\pi$ if the image covers the full sphere.

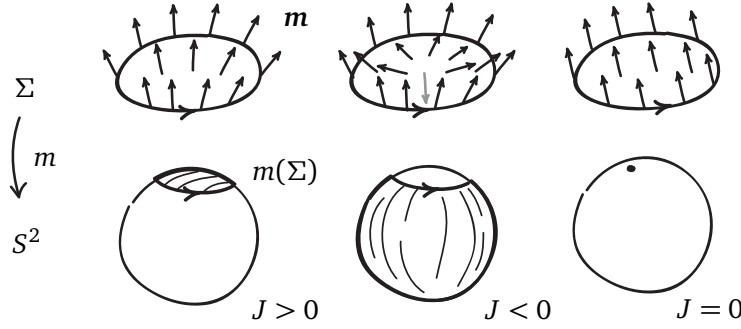

Figure 2.5: Three vector fields $\boldsymbol{m}$ over a surface $\Sigma$ and the images $m(\Sigma)$ of the corresponding maps $m : \Sigma \to S^2$ defined by $\boldsymbol{m}$ according to (2.32). LEFT: An orientation-preserving map. MIDDLE: An orientation-reversing map. The configuration of $\boldsymbol{m}$ at the boundary $\partial\Sigma$ is the same as in the left panel. RIGHT: A degenerate map sending $\Sigma$ to a single point.

For example, consider the vector fields $\boldsymbol{m}$ over a surface $\Sigma$ and corresponding images $m(\Sigma)$ on the sphere shown in Fig. 2.5 (we assume that no folding occurs here). The arrow on the boundary contour $\partial\Sigma$ indicates the orientation of the surface. On the left, the map $m$ preserves orientation, since the positively-oriented boundary $\partial\Sigma$ is mapped to a positively-oriented cap near the north pole. In the middle panel, the image of the boundary is the same, but the whole image is negatively-oriented on the sphere, hence the map reverses the orientation. On the right, we have a constant map whose image is collapsed into a single point. The map is degenerate, and its Jacobian vanishes.

Now we can determine the sign of the curvature of the projected connection on the plane bundle $M$ defined by the field $\boldsymbol{m}$ according to (1.56). We know that the curvature component reads $f_{12} = J\sin\theta$, where the indices refer to some positively-oriented coordinates $(x_1, x_2)$ on the surface $\Sigma$. Thus, the sign of the curvature is given by the sign of $J$. It follows that the three vector fields shown in the figure define vector bundles with positive, negative, and zero curvature. In terms of the electromagnetic interpretation (2.15), the last case corresponds to the vanishing magnetic field strength.

The first two cases also illustrate that the total solid angle $\Omega(m(\Sigma))$ is determined by the image of the boundary $m(\partial\Sigma)$, up to $4\pi$. Suppose that in the first case the solid angle spanned by the image is $\Omega_1$. Then the solid angle for the second case is $\Omega_2 = -(4\pi - \Omega_1)$, where the first minus sign is due to the orientation reversal. Thus, $\Omega_1 = \Omega_2 + 4\pi$, so the two images "differ by a full sphere".

## 2.3 Field strength and geometry of surfaces

Above, we considered the electromagnetic interpretation of the complex line bundle defined by the vector field of plane normals. Here, we focus on a particular case when such bundle is the tangent bundle of a two-dimensional surface in $\mathbb{R}^3$. Later, this will help to explain the part of the analogy described in the Preface that links magnetic monopoles and the Gauss–Bonnet theorem.

### 2.3.1 Electromagnetic tensor and field strength

Our discussion indicates that the component of the electromagnetic field strength tensor

$$F_{ij} = \partial_i A_j - \partial_j A_i, \tag{2.40}$$

is proportional to the curvature of connection on an (abstract) complex line bundle. This expression has the same form in any coordinate system. In contrast, the components of field strength vectors are related to the area elements. Consider the magnetic flux through the surface $\Sigma$:

$$\Phi(\Sigma) = \int_\Sigma F_{12}\, dx_1\, dx_2 = \int_\Sigma \boldsymbol{B} \cdot d\boldsymbol{S}. \tag{2.41}$$

In the Cartesian coordinates $x_j = x, y, z$, the area element $dS_{ij}$ is given simply by $dx_i dx_j$. Taking the surface $\Sigma$ to be an elementary coordinate contour, one finds

$$F_{xy} = B_z, \quad F_{xz} = -B_y, \quad F_{yz} = B_x. \tag{2.42}$$

Here, $B_y$ has the negative sign, for if one orients $xz$ plane by specifying the normal $\boldsymbol{n} = \boldsymbol{e}_y$, the pair $(\boldsymbol{e}_x, \boldsymbol{e}_z)$ has the negative orientation. Geometrically, these components are proportional to the three curvature components: $F_{ij} = \frac{\Phi_0}{2\pi} f_{ij}$. In a similar fashion, the components of the electric field $E_j$ are determined by curvatures over the elementary space-time contours $(x_j, t)$.

In a curvilinear coordinate system, the components of $\boldsymbol{B}$ can be found by "inverting" Eq. (2.41), as follows. Let $\mathcal{C}$ be the boundary of a small surface element near a point $p$ with the normal $\boldsymbol{n}$. Then the normal component of $\boldsymbol{B}$ is

$$B_n(p) = \lim_{\mathcal{C} \to p} \frac{\Phi(\mathcal{C})}{S(\mathcal{C})}, \tag{2.43}$$

where $\Phi(\mathcal{C})$ is the magnetic flux through the surface element and $S(\mathcal{C})$ is its area. In the limit, the contour shrinks to the point $p$.

### 2.3.2 Gaussian curvature

The notion of the field strength has its counterpart in the geometry of surfaces. Consider the tangent bundle $M = T\Sigma$ of a surface $\Sigma \subset \mathbb{R}^3$. It can be thought of as the plane bundle (1.56) defined by the vector field $\boldsymbol{n}$ of surface normals. According to (2.32), the field $\boldsymbol{n}$ determines the map $n : \Sigma \to S^2$ from the surface to the sphere $S^2$, called the **Gauss normal map**. Introducing a quantity similar to the field strength $B_n$, one obtains the **Gaussian curvature** of the surface:

$$\kappa(p) = \lim_{\mathcal{C} \to p} \frac{\Omega(n(\mathcal{C}))}{S(\mathcal{C})}, \tag{2.44}$$

where $\Omega(n(\mathcal{C}))$ is the solid angle enclosed by the image $n(\mathcal{C})$ on the sphere and $S(\mathcal{C})$ is the area of the region enclosed by $\mathcal{C}$ on the surface. Following the pattern of Eq. (2.41), the parallel transport angle for the boundary of the surface is given by

$$\Delta\alpha(\partial\Sigma) = \int_\Sigma f_{12} dx_1 dx_2 = \int_\Sigma \kappa dS. \tag{2.45}$$

It turns out that the Gaussian curvature can also be expressed as

$$\kappa(p) = \frac{1}{R_1(p)R_2(p)}, \tag{2.46}$$

where $R_j$ are **principal radii of curvature**, which are defined as follows. Consider a plane that contains the surface normal at $p$. The intersection of this plane with the surface gives some plane curve. The curve is characterized by the radius of curvature $R$ at $p$, which depends on the choice of the plane. Then $R_1$ and $R_2$ are minimal and maximal values of radius of curvature. For example, a sphere of radius $r$ has $R_1 = R_2 = r$, and $\kappa = \frac{1}{r^2}$. Then the parallel transport angle for some contour $\mathcal{C} = \partial\Sigma$ is, not surprisingly,

$$\Delta\alpha(\mathcal{C}) = \int_\Sigma \kappa dS = \int_\Sigma \frac{dS}{r^2} = \Omega(\Sigma). \tag{2.47}$$

### 2.3.3 Geometry of a conical surface

Finally, we look at the curvature and the parallel transport on a conical surface. These considerations will be important in the context of several physical effects to be discussed in Sec. 3. Let $\Sigma$ be a conical surface shown in Fig. 2.6. The tip is rounded, so the field $\boldsymbol{n}$ is smooth everywhere on $\Sigma$. We are interested in the parallel transport angles for the contours that lie away form the tip, that is, belong to the lateral surface. Under the normal map $n$, any point of the lateral surface is mapped to a point in a certain circle of constant latitude (the cone would touch the sphere along this circle). Let $\Omega_0$ be the solid angle enclosed by the circle. Then for any contour on the lateral surface of the cone one has $\Delta\alpha = N\Omega_0$, where $N \in \mathbb{Z}$ counts how many times the contour encircles the tip. In the figure, $N = 1$ for $\mathcal{C}_1$ and $N = 0$ for $\mathcal{C}_2$.

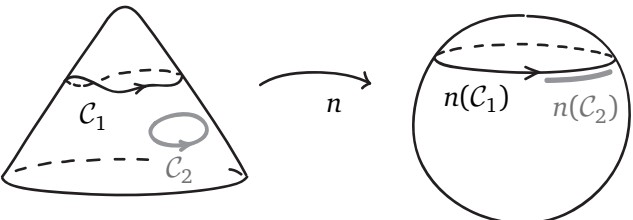

Figure 2.6: Images of contours on a surface of a cone with a rounded tip under the normal map. This map is associated under (2.32) to the vector field of surface normals.

The Gaussian curvature $\kappa$ of the lateral surface vanishes, since taking the limit $\mathcal{C} \to p$ requires $N = 0$. One can also deduce that $\kappa = 0$ from Eq. (2.46): since the generator of the cone is a straight line, the maximal radius of curvature is $R_2 \to \infty$. Whenever a contour lies in the zero-curvature region, the resulting $\Delta\alpha$ is independent of the shape of the contour.



Figure 2.7: Familiy of conical surfaces filling the space $\mathbb{R}^3$ and the corresponding vector field of normals $\boldsymbol{n}$. Only the vectors over the selected horizontal line are shown.

**Exercise 2.5.** Consider the space $\mathbb{R}^3$ filled with conical surfaces, as shown in Fig. 2.7. Taken together, vector fields of surface normals form a vector field $\boldsymbol{n}$ over $\mathbb{R}^3$. Consider the bundle $M$ over $\mathbb{R}^3$ defined by $\boldsymbol{n}$ according to (1.56). Which configuration of the magnetic field corresponds to $M$ under (2.15)? [§3.1.2]

Vanishing of the curvature of the cone almost everywhere gives an interesting way to find a parallel transport vector field $\boldsymbol{v}_{PT}$ along the circumference of the cone. This time, consider a cone with a point-like tip, which touches a sphere along a circle of latitude with azimuthal angle $\theta$, as shown in Fig. 2.8 on the left. The parallel transport on the cone along this circle is the same as in $TS^2$, since the tangent spaces coincide. Now we can cut the cone along its generating line and lay it flat on the plane. Note that this deformation preserves lengths of paths that do not cross the cut: the surface bends without stretching. There is no distortion of the coordinate system (in contrast with maps of the Earth). This deformation is an **isometry**, meaning that it preserves lengths of tangent vectors and angles between them.

Let us see what happens after such flattening to the vectors $\boldsymbol{v}_{PT}$ that were parallel transported along the circumference of the cone, as shown in Fig. 2.8 on the right. Recall from Sec. 1.3.3 that this vector field is expressed as $\boldsymbol{v}_{PT} = e^{i\alpha}\boldsymbol{1}$, where $\alpha(\varphi) = -\varphi \cos\theta$, and the

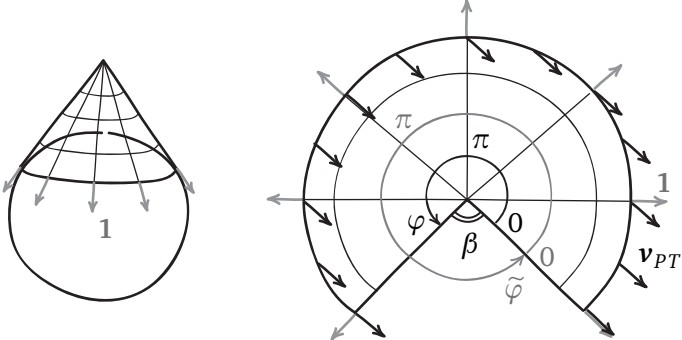

Figure 2.8: Parallel transport on a cone. LEFT: The cone touching the sphere along a circle of latitude. Also shown is the basis section $\mathbf{1} = \boldsymbol{e}_\theta$. RIGHT: The cone after cutting along a generator and flattening on the plane. Two different polar angles $\varphi$ and $\widetilde{\varphi}$ are defined on the cone and on the plane, respectively. The field $\boldsymbol{v}_{PT}$ shows parallel transported vectors.

basis section is $\mathbf{1} = \boldsymbol{e}_\theta$. On the plane, the shape of the cone is characterized by the opening angle $\beta$. From the elementary geometry one finds that

$$\cos\theta = 1 - \frac{\beta}{2\pi}\,. \tag{2.48}$$

Thus the parallel transport equation reads

$$\alpha = -\varphi\left(1 - \frac{\beta}{2\pi}\right) = -\widetilde{\varphi}\,, \tag{2.49}$$

where $\widetilde{\varphi}$ is the polar angle *on the plane*. We conclude that after flattening, the parallel transported vectors on the cone become parallel on the plane, in the usual sense. Folding the cone back makes $\boldsymbol{v}_{PT}$ discontinuous at the cut. The parallel transport angle equals the opening angle $\beta$, which therefore measures the amount of curvature concentrated in the tip of the cone. We encourage the reader to draw a constant vector field along the circumference of a flat paper circle and then to fold it into a cone. When put on a globe, the vectors will show the rotation of the oscillation plane of the Foucault pendulum.

Recall from the discussion below Eq. (2.27) that a non-zero value of the curvature at a point $p$ makes it impossible to define a covariantly constant vector field in a neighborhood $U$ of $p$. Consider the surface $\mathcal{B}$ obtained from the cone by removing the tip. Everywhere on $\mathcal{B}$, the curvature vanishes, which suggests that one can define a constant field near each point. Indeed, on the flattened surface shown in Fig. 2.8, such field corresponds to a field of parallel vectors over the region $U$ of the plane. Note, however, that there is no *global* smooth covariantly constant vector field on the surface $\mathcal{B}$. Once the region $U$ contains a path going around the tip, one cannot define a constant field over $U$. We will discuss a physical implication of this fact in Sec. 3.1.3.

## 2.4 Summary and outlook

In this section, we introduced curvature, a local characteristic of a vector bundle with connection. It is defined in terms of the parallel transport along the boundary of an infinitesimal coordinate plaquette. By definition, the curvature component (2.24) depends on the choice of coordinates; orientation of the surface affects the sign of the curvature. The integral of

curvature over a surface measures the parallel transport angle $\Delta\alpha$ over the surface boundary. According to the Stokes' theorem (2.30), the same value of $\Delta\alpha$ can also be found by means of the contour integral of connection coefficient $\omega_\gamma$, provided that the basis section used to find $\omega_\gamma$ is smooth over the whole surface. On a more abstract level, it is important to keep in mind that the curvature is not a property of the bundle itself; different connections introduced on a given bundle can have different curvatures.

In physics, the curvature of connection on a vector bundle plays the role of the strength of the gauge filed. Above, we considered such geometric picture of the electromagnetic field, with the following key results:

- One can give a geometric interpretation of the Schrödinger equation for a charged particle in the background electromagnetic field. The wave function becomes a section of a complex line bundle with connection. Gauge potentials are proportional to the connection coefficients, and field strength is described by the curvature.

- In the geometric picture, the Schrödinger equation includes covariant derivatives and is manifestly gauge-invariant. Gauge transformations correspond to the changes of the basis section.

- Any complex line bundle with connection can be interpreted as a description of the effective electromagnetic field.

- For a real plane bundle with projected connection, the effective magnetic flux is determined by the total solid angle covered by the vector field of plane normals.

- For a tangent bundle of a two-dimensional surface $\mathcal{B} \subset \mathbb{R}^3$ with projected connection, the strength of the effective magnetic field corresponds to the Gaussian curvature of the surface.

In the next section, we will continue studying the geometric nature of the electromagnetic field, and then will apply these ideas to an entirely different quantum system. The Stokes' theorem will help us to compute a *topological* invariant of a complex line bundle over a two-dimensional surface in Sec. 4.2.2. Below we make final remarks regarding mathematical and historical aspects of gauge theories.

▷ **Non-Abelian gauge fields.** The close relationship between the field strength and the curvature of connection on a vector bundle is a common feature of all gauge theories. The electromagnetism is the simplest gauge theory due to the fact that phase rotations (2.7) commute. This is not the case for other fundamental interactions, which are described by **non-Abelian** gauge fields. For example, the gravitational field strength is related to the Riemann curvature tensor. It describes the curvature of the Levi-Civita connection on a tangent bundle of a four-dimensional (pseudo-)Riemannian manifold. The parallel transport of a vector along a closed path results in a *linear transformation*. Accordingly, the curvature tensor has four indices: two of them correspond to the coordinates, as in Eq. (2.24), and the other two are matrix indices of the linear transformation. We will encounter a related expression for a non-Abelian curvature in Sec. 3.4.

▷ **Differential forms.** Maxwell equations were originally written in components, and only later they were transformed into the compact form using the vector analysis. In fact, the number of equations can be further reduced to just two, if one adopts a more abstract machinery of **differential forms**. This formalism also absorbs some of calculations we made above. An introduction to differential forms can be found in Ref. [3]. Here, we simply show how some of our equations will look like in this mathematical language.

An $n$-form is an object that can be naturally integrated over an $n$-dimensional surface. For example, the connection coefficients can be used to construct the connection one-form $\omega = \sum_i (-\omega_i) dx_i$. The differential $d$ is an operator transforming an $n$-form into an $(n+1)$-form. This single operator includes gradient, divergence, and curl as special cases. One defines the curvature two-form $f$ as the differential of $\omega$:

$$f = d\omega, \qquad (2.50)$$

which agrees with our definition given in components (2.24). The forms interact with maps between spaces via the pullback. In the context of Fig. 2.3, this construction allows one to define a two-form $m^* f$ over $\Sigma$ given the two-form $f$ over $S^2$. In components, these forms will be related by Eq. (2.35). Finally, the Stokes' theorem (2.30) assumes a concise form:

$$\int_{\partial \Sigma} \omega = \int_{\Sigma} d\omega. \qquad (2.51)$$

The machinery of differential forms incorporates the usual vector analysis and generalizes it to the higher-dimensional spaces with curvilinear coordinates.

▷ **Discovery of gauge invariance.** The principle of gauge invariance has a long and tortuous history, starting with the early works on electromagnetism and culminating in the Standard model of particle physics. A detailed historical overview of these developments is given in Ref. [24]. One of the milestones was the discovery of the relationship between the electromagnetic gauge transformations (2.2) and the phase rotation of the wave function (2.7). It was first formulated by Fock [25], who interpreted the complex phase as a "fifth coordinate" and derived the equation of motion of a quantum particle in the form of a geodesic equation in the five-dimensional space. However, the discovery is often attributed to Weyl, who coined the term "gauge invariance" and promoted it to a fundamental principle of physics. The term has a curious origin: initially, Weyl considered coordinate-dependent transformations of *scale* in an attempt to unify electromagnetism and General relativity. However, Einstein objected that this would imply the possibility of changing the size of an object by transporting it along a closed path. Later the scale transformation was replaced by the complex phase factor of the wave function, but the term remained unchanged.

It is interesting to consider the history of gauge theories in parallel with the mathematical development of abstract vector bundles and connections [26]. Weyl, who navigated both mathematical and physical worlds, played here a central role. He understood General relativity in the context of Riemannian geometry and the works of Levi-Civita on parallel transport. In these early studies, the parallel transport was defined using projection for the case of a Riemannian manifold embedded in the ambient space (we used a similar construction in Sec. 1). Weyl realized that a connection can be defined axiomatically on an abstract Riemannian manifold, without reference to the ambient space. The next step was to disengage the parallel transport from the metric. This naturally led to the possibility of changing the length of parallel transported vectors, or, equivalently, the freedom of choosing the scale locally. Then, Weyl tried to relate these changing scales with the electromagnetic field in a way Riemannian geometry is linked to gravitation. In modern terms, he introduced a connection on a real line bundle, whose curvature was to describe the electromagnetic field strength. From this perspective, finding the "correct" form of gauge invariance amounts to replacing a real line bundle with connection by a *complex* line bundle with connection. We conclude that Fock discovered the algebraic form of gauge invariance (Sec. 2.1.1), and Weyl first understood its geometric nature, discussed in Sec. 2.1.2.

# 3 Geometry of quantum states

In this section, we will consider several quantum effects whose description includes geometric properties of a relevant vector bundle. We start with two systems based on a solenoid containing magnetic flux: a particle on a ring and the double-slit experiment. These systems will be described in terms of the geometric picture of the electromagnetic field. In the first case, we will focus on the covariant derivative, and in the second one on the parallel transport.

Then we will define another type of vector bundle, which is associated with the collection of eigenstates of a quantum system. This bundle will be equipped with a connection related to the adiabatic evolution of a quantum state. Finally, we will see how the two pictures can combine in such a way, that the geometry of eigenspaces creates an effective electromagnetic field.

## 3.1 Quantum particle on a ring

Particle on a ring is a simple but important system, which can illustrate geometrical and topological aspects of quantum physics (see, for example, Ref. [27]). Here, we use it to discuss the relationship between physical observables and geometry of the complex line bundle that describes the electromagnetic interaction. We begin with the physical discussion and then interpret the results geometrically. After that, we describe a real plane bundle, which models the process of the flux insertion.

### 3.1.1 Spectrum and flux

Consider a charged particle, which moves freely on a ring $S^1$ surrounding a solenoid with the magnetic flux $\Phi$. In the case when $\Phi = 0$, the (angular) momentum operator is

$$\hat{p} = \frac{\hbar}{i}\partial_\varphi \,. \tag{3.1}$$

The eigenfunctions and the corresponding momentum eigenvalues are

$$\psi_n = e^{in\varphi}\,, \quad p_n = n\hbar\,. \tag{3.2}$$

Note that the eigenfunction uniquely determines the momentum of the particle. The number $n$ can be thought of as an "angular velocity" of the complex unit vector, which rotates as a function of $\varphi$. Since the wave function must be periodic,

$$\psi(\varphi + 2\pi) = \psi(\varphi) \quad \Rightarrow \quad n \in \mathbb{Z}\,, \tag{3.3}$$

and the momentum of the particle is quantized.

This familiar situation changes dramatically when the flux is non-zero. We assume that the magnetic field strength is concentrated inside the coil and vanishes outside. The vector potential along the ring must satisfy $\int A_\varphi d\varphi = \Phi$, and we choose it to be constant, $A_\varphi = \frac{\Phi}{2\pi}$. Let the particle have the positive elementary charge $q = e$. Then the momentum operator becomes

$$\hat{p} = \frac{\hbar}{i}\left(\partial_\varphi - i\frac{q}{\hbar}A_\varphi\right) = \hbar\left(-i\partial_\varphi - \frac{\Phi}{\Phi_0}\right), \tag{3.4}$$

where $\Phi_0 = \frac{h}{e}$ is the magnetic flux quantum. The eigenfunctions are the same as for $\Phi = 0$, while the momentum eigenvalues are shifted:

$$\psi_n = e^{in\varphi}\,, \quad \hat{p}\psi_n = \hbar\left(n - \frac{\Phi}{\Phi_0}\right)\psi_n\,. \tag{3.5}$$

The spectrum of the Hamiltonian $\hat{H} = \frac{\hat{p}^2}{2m}$ is

$$\varepsilon_n = \frac{p_n^2}{2m} = \frac{\hbar^2}{2m}\left(n - \frac{\Phi}{\Phi_0}\right)^2. \tag{3.6}$$

The spectrum as a function of $p$ consists of the set of points on the parabola determined by the condition $n \in \mathbb{Z}$, as shown in Fig. 3.1. The dependence of the momentum on the magnetic flux gives rise to the **persistent currents**. Note that if the ratio $\frac{\Phi}{\Phi_0}$ is not equal to $\frac{m}{2}$ for an integer $m$, it is impossible to populate the eigenstates with electrons in such a way that their total momentum vanishes. In other words, the ring must carry a perpetual flow of charge. Of course, this picture does not include many real-world effects, such as non-zero temperature, the presence of disorder, and finite resistance of the ring. Remarkably, the persistent currents were indeed observed in the metallic mesoscopic rings. The only way to detect this current is to measure its tiny magnetic moment, which requires ingenious experimental techniques [28]. See Ref. [29] for a comprehensive description of the step-by-step refinements of this simple model, which eventually lead to the predictions comparable with the experimental results.

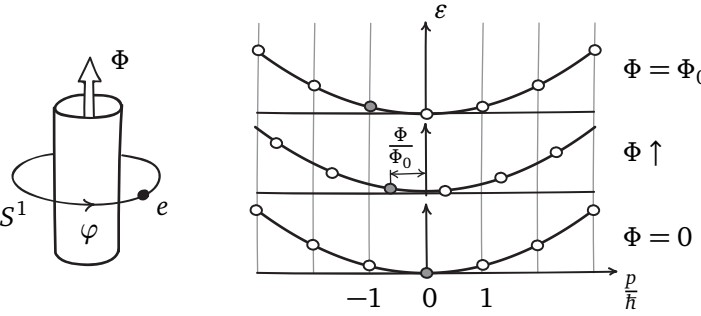

Figure 3.1: LEFT: Particle on a ring $S^1$ pierced by the magnetic flux $\Phi$. RIGHT: The spectra of the system for three different values of the flux. The flux insertion leads to the shift of the lattice of eigenvalues along the parabola. Shaded circle represents the state corresponding to the wave function $\psi_0$.

There are several interesting features of the momentum eigenstates, eigenvalues, and the spectrum:

- A particle described by the wave function $\psi_n$ can have any given value $\tilde{p} \in \mathbb{R}$ of the momentum. For example,

$$\hat{p}\psi_0 = \tilde{p}\psi_0, \quad \text{for} \quad \Phi = -\frac{\tilde{p}}{\hbar}\Phi_0. \tag{3.7}$$

  In particular, the momentum is not quantized in units of $\hbar$.

- For a fixed value of the flux $\Phi$, the possible values of the momentum form a lattice with the period equal to $\hbar$. If the flux $\Phi$ is not an integer multiple of the flux quantum $\Phi_0$, there is no eigenstate with $p = 0$.

- The variation of the field strength, or **flux insertion**, changes the momentum and energy eigenvalues smoothly while the eigenfunctions remain the same. After insertion of the flux quantum, $\Phi \to \Phi + \Phi_0$, the final spectrum is identical to the initial one, with each state replaced by its neighbor.

Note also that one can make a "large gauge transformation"

$$\psi'_n = e^{ik\varphi}\psi_n, \qquad A' = A + k\frac{\Phi_0}{2\pi}, \qquad \hat{p}'\psi'_n = \hat{p}\psi_n, \tag{3.8}$$

which simultaneously changes the wave function and shifts the flux by an integer number of flux quanta, while keeping the momentum eigenvalue invariant. According to the definition (2.2), this is not a gauge transformation in the entire space, since it changes the field strength inside the solenoid. On the other hand, in a situation when the flux is unknown, one can think of it as a mapping between various ways to describe a given state of the particle. Some authors tend to identify large gauge transformations with the flux insertion. However, this does not seem to be entirely correct, since the flux insertion is a physical process associated with the gradual shift of momentum eigenvalues, and the gauge transformation is an instantaneous formal mathematical operation. We will come back to this point in Sec. 7.1.2.

Now let us interpret these observations in terms of the geometric picture of the electromagnetic field detailed in Sec. 2.1.2. Recall that the state of the particle is described by a section $\boldsymbol{\psi} = \psi\mathbf{1}$ of a complex line bundle with connection, and the momentum operator is proportional to the covariant derivative. Note that the wave *function* $\psi$ does not have an intrinsic geometric meaning. To find the value of the momentum, we need to compute the covariant derivative of the section $\boldsymbol{\psi}$, which is expressed in terms of $\psi$ and the basis section $\mathbf{1}$. Since the connection coefficient is determined by the flux,

$$\omega_\varphi = -\frac{q}{\hbar}A_\varphi = -\frac{\Phi}{\Phi_0}, \tag{3.9}$$

the basis section $\mathbf{1}$ must change during the flux insertion. Thus, the section $\boldsymbol{\psi}$ and the momentum eigenvalue can vary even if $\psi$ remains unchanged.

As for the absence of the quantization, observe that the eigenstate equation

$$\frac{\hbar}{i}\nabla_\varphi\boldsymbol{\psi} = p\boldsymbol{\psi}, \tag{3.10}$$

has the same form as the definition (1.28) of the connection coefficient, $\nabla_\varphi\mathbf{1} = i\omega_\varphi\mathbf{1}$. It follows from the discussion in Sec. 1.4.1 that one can interpret $\frac{p}{\hbar}$ as the angular velocity of the complex vector $\boldsymbol{\psi}(\varphi)$ with respect to the parallel transport. Recall that the parallel transported vectors do not generally form a continuous vector field over a closed path. Hence, the angular velocity $\frac{p}{\hbar}$ need not be quantized, even though $\boldsymbol{\psi}$ is continuous on the ring $S^1$ (in a sharp contrast with the case of the ordinary complex functions).

Finally, a large gauge transformation is nothing else but the result of a basis change that shifts the value of the parallel transport angle by $2\pi n$ (see Sec. 1.3.3 and discussion below Eq. (2.17)).

### 3.1.2 Flux insertion as a bundle deformation

It is instructive to contemplate an explicit visualization of the vector bundle picture for the flux insertion. To this end, we employ the electromagnetic interpretation (2.15) of the real plane bundle (1.56).

Consider a plane bundle $M$ defined by a vector field $\boldsymbol{m}$ over the plane $\mathbb{R}^2$ that contains the ring $S^1$. We focus on two spatial dimensions, and the third coordinate will be used to describe the evolution of the bundle in time. Denote $D$ the disc formed by the intersection of the solenoid and the plane $\mathbb{R}^2$. The case of $\Phi = 0$ is described by a constant vector field, say, $\boldsymbol{m} = \boldsymbol{e}_z$. First, we need to find a bundle configuration that would represent a non-zero magnetic flux. That is, the curvature must be non-zero inside $D$ and vanish outside. Note that this condition is satisfied by the vector field shown in Fig. 2.7. According to Eq. (2.38), the

amount of flux is determined by the solid angle spanned by the vectors $\boldsymbol{m}$ over the disc $D$. Thus, the process in which the flux increases from 0 to $2\Phi_0$ as a function of time $t$ can be described by the bundle deformation shown in Fig. 3.2 on the left. Note that in contrast with the setting of Exercise 2.5, here we consider a general plane bundle (1.56) and not a tangent bundle to a surface. The bundle $M$ is formed by the planes normal to the vectors. The vector field over three-dimensional spacetime is obtained from the field shown in the figure by rotation around the central vertical axis.

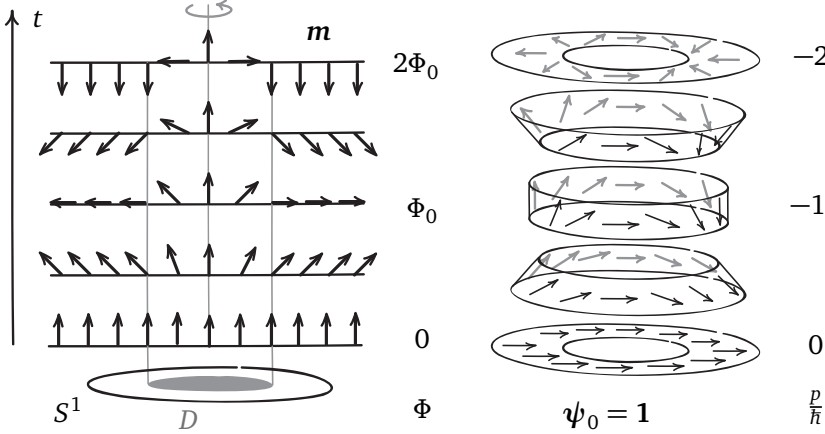

Figure 3.2: LEFT: Deformation of the plane bundle $M$, which describes the flux insertion. Arrows represent the plane normals $\boldsymbol{m}$. The field $\boldsymbol{m}$ has cylindrical symmetry about the central vertical axis. Each row of arrows shows the configuration of the bundle $M$ over the plane $\mathbb{R}^2$ containing the ring $S^1$, at the moment $t$. The value of the magnetic flux is shown for three configurations. RIGHT: Evolution of the section $\boldsymbol{\psi}_0$ of $M$ describing acceleration of the particle. Fibers of the plane bundle along the circle $S^1$ are merged into conical bands for clarity. The set of bands corresponds to the set of bundle configurations in the left panel (the vector field $\boldsymbol{m}$ is the normal field for the bands). Thin arrows inside the bands represent the section $\boldsymbol{\psi}_0$. The value of the momentum in units of $\hbar$ is shown for three sections.

In the process of the flux insertion, increasing amount of the curvature contained inside $D$ leads to the rotation of planes outside the solenoid, by continuity of $\boldsymbol{m}$. In this way, the field strength affects the geometry of the bundle in the zero-field region. Moreover, this picture incorporates the Faraday's law of induction. Consider the time-dependent bundle defined by $\boldsymbol{m}(r,\varphi,t)$ as a bundle over spacetime, where $r,\varphi$ are the polar coordinates on the plane. Then variation of the curvature in the plane inside the solenoid (magnetic field strength) leads to the non-zero curvature component $f_{\varphi t}$, which describes the strength of the circular electric field.

### 3.1.3 Wave function as a section

Now that we know the evolution of the bundle, we can consider the behavior of its section representing the state of the particle. To this end, we focus on the part of the bundle over the ring $S^1$, which is shown in the right panel of Fig. 3.2. For clarity, the individual fibers are merged into bands. Each band has the same shape as a restriction of the tangent bundle $TS^2$ to a circle of constant latitude. The deformation of the bundle for increasing $t$ corresponds to the "sliding" of the circle from the vicinity of the north pole (normals look upwards) through the equator to the south pole of the sphere (downward normals).

Consider a section $\psi_0$ shown in the figure. We claim that the evolution of this section describes the process of acceleration of a quantum particle from the zero momentum state to the state with $p = -2\hbar$. In the initial state, the fibers of the bundle are horizontal, so the covariant derivative coincides with the ordinary derivative on the plane. The section $\psi_0$ is a constant vector field, hence the particle has zero momentum. In the middle state, $\Phi = \Phi_0$, the bundle resembles the tangent bundle $TS^2$ restricted to the equator. Recall from Sec. 1.3.1 that the parallel transport along the equator is given, for example, by the basis section $e_\theta$. Here, the vectors clearly rotate relative to the parallel transport and make *one full turn* when going around the ring $S^1$. Assuming that the rotation is uniform, this section describes a particle with the momentum $p = -\hbar$. In the final state, the fibers are again horizontal, with the normals looking down. The vectors of $\psi_0$ make two complete turns, which corresponds to the momentum value $p = -2\hbar$. Thus, the absolute value of the momentum $p$ indeed increases during the process. In physical terms, this represents the acceleration of the particle under the action of the circular electric field. Recall that there is no eigenstate with zero momentum if $\frac{\Phi}{\Phi_0} \notin \mathbb{Z}$. Now, we can relate this to the absence of a covariantly constant field on a conical surface discussed in Sec. 2.3.3.

It is also interesting to note that the vector bundle picture allows one to restore the usual relationship between the momentum and the "winding number" $n$ in Eq. (3.2) (this works only for $\frac{\Phi}{\Phi_0} \in \mathbb{Z}$, when the parallel transport vector vector field is continuous). As discussed above, the vectors in the section $\psi_0$ make one full turn for $\Phi = \Phi_0$ and two full turns for $\Phi = 2\Phi_0$. Note that this number of turns changes during a *continuous* deformation that does not change the length of vectors! In contrast, one cannot continuously transform an ordinary complex function $\psi(\varphi) = 1$ into $\psi(\varphi) = e^{-2i\varphi}$ while preserving the modulus at each point.

Now let us describe the situation shown in the figure formally. First, we need to choose a basis section $\mathbf{1}$ for $M$, such that the corresponding connection coefficient gives the constant potential $A_\varphi = \frac{\Phi}{2\pi}$. To this end, identify the conical bands in the figure with the restrictions of the bundle $TS^2$ to the circles of constant latitude (the initial state corresponds to a small circle near the north pole). Then, using this identification, transfer the section $\mathbf{1}_N$ of $TS^2$ defined in Exercise 1.3 to the set of the conical bands. The result is shown in the right panel of Fig. 3.2. As a next step, we extend this section to the whole bundle $M$. In the initial state, the bundle over the plane is flat, with all fibers lying in the horizontal plane. The section along $S^1$ is a uniform vector field, so we extend it to the whole plane. Now consider what happens with this section as $t$ increases. Since the deformation of the bundle is smooth, this section will also deform smoothly. In this way, we obtain a smooth basis section $\mathbf{1}$ defined over the whole spacetime region in the left panel Fig. 3.2.

Owing to the smoothness of the section, the parallel transport angle computed as contour integral $\int_{S^1} (-\omega_\varphi) d\varphi$ equals the solid angle $\Omega$ spanned by the vectors $\mathbf{m}$ inside the area bounded by $S^1$. By the results of Exercise 1.3, the connection coefficient $\omega_\varphi$ does not depend on $\varphi$ (in the exercise, it is denoted $\omega_\varphi^N$). It follows that

$$\omega_\varphi = -\frac{\Omega}{2\pi}. \tag{3.11}$$

Geometrically, the momentum operator is given by the covariant derivative:

$$\hat{p} = \frac{\hbar}{i} \nabla_\varphi, \tag{3.12}$$

and we are looking for the eigenstate section $\psi_n$ such that

$$\nabla_\varphi \psi_n = i \frac{p_n}{\hbar} \psi_n. \tag{3.13}$$

For $n = 0$, we have:

$$\nabla_\varphi \boldsymbol{\psi}_0 = i\left(-\frac{\Phi}{\Phi_0}\right)\boldsymbol{\psi}_0. \tag{3.14}$$

Now observe that the basis section **1** defined above satisfies this equation:

$$\nabla_\varphi \mathbf{1} = i\omega_\varphi \mathbf{1} = i\left(-\frac{\Omega}{2\pi}\right)\mathbf{1} = i\left(-\frac{\Phi}{\Phi_0}\right)\mathbf{1}, \tag{3.15}$$

where we used Eq. (2.38). We conclude that for general $n$, the eigenstate section is given by

$$\boldsymbol{\psi}_n = e^{in\varphi}\mathbf{1}. \tag{3.16}$$

**Exercise 3.1.** Find a bundle deformation that starts from the same initial state $\boldsymbol{m} = \boldsymbol{e}_z$, but leads to the momentum value $p = \hbar$. [§4.3.2]

## 3.2 Aharonov–Bohm effect

Recall that the covariant derivative is closely related to the parallel transport. If we treat wave function as a section of a vector bundle, the parallel transport looks like a phase rotation of a state vector characterizing a point-like particle during its motion along some path. At first sight, this is not compatible with quantum physics, since quantum particles are delocalized and do not have definite trajectories. However, there is a way to formulate quantum mechanics that includes this picture of the parallel transport: it is the Feynman path integral approach [30]. Here we use this framework to discuss the effects of the gauge potential on the interference pattern of electrons in the double-slit experiment, following Ref. [31].

### 3.2.1 Path integral formulation

Suppose that an electron is emitted from the source at the point $a$ at the moment $t = 0$, as shown in Fig. 3.3. Let us find the probability of detecting the electron at the point $b$ of the screen at $t = T$. It is given by the modulus squared of the wave function, $|\psi(b, T)|^2$. This can be obtained from the initial state $|\psi(0)\rangle = |a\rangle$ by applying the unitary evolution operator:

$$\psi(b, T) = \langle b|\psi(T)\rangle = \langle b|\hat{U}(T)|a\rangle. \tag{3.17}$$

The last expression is called the **amplitude** of going from $a$ to $b$ in time $T$. According to the path integral method, the amplitude is proportional to the sum of the complex phases

$$\sum_{\mathcal{P}_i} \exp\left(\frac{i}{\hbar}S(\mathcal{P}_i)\right), \tag{3.18}$$

over all possible paths $\mathcal{P}_i$ connecting points $a$ and $b$. Here, $S(\mathcal{P}_i)$ is the classical action along the path $\mathcal{P}_i$. Each path can be thought of as a map from the time interval to the space:

$$\mathcal{P}_i : [0, T] \to \mathbb{R}^3, \qquad \mathcal{P}(0) = a, \qquad \mathcal{P}(T) = b. \tag{3.19}$$

It turns out that the interference of the complex phases associated with the paths leads to the formation of the fringes in the probability distribution $|\psi(b, T)|^2$ on the screen. Even if the source emits only one electron at a time, there are certain points on the screen that will never be hit by an electron. See Ref. [32] for a detailed computation of the resulting interference pattern using the path integral approach.

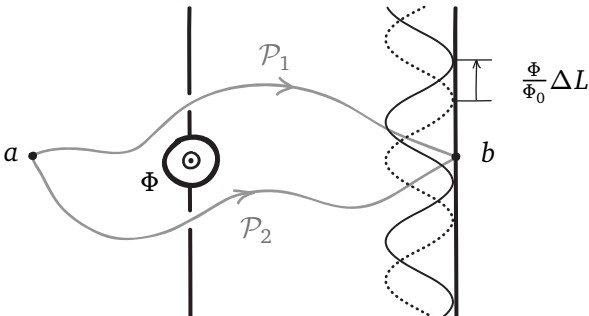

Figure 3.3: The Aharonov–Bohm effect. Electrons are emitted from the source $a$ and then go though the two slits. The probability of detecting an electron at a given point $b$ of the screen is determined by the phases associated with all possible trajectories, such as $\mathcal{P}_1$ and $\mathcal{P}_2$. The probability distribution on the screen has the form of an interference pattern (dotted line). Due to the magnetic flux $\Phi$ in the solenoid installed between the slits, the interference pattern is shifted by $\frac{\Phi}{\Phi_0}\Delta L$ (solid line), where $\Delta L$ is its spatial period.

### 3.2.2 Parallel transport

Now let us insert a solenoid with the magnetic flux in the space between the slits. In the presence of the magnetic field, one adds the term $L^B = q\dot{x}_j A_j$ to the Lagrangian, where $j = x, y, z$, and $A_j$ are the vector potential components. The corresponding change in the action reads

$$S^B = q\int \dot{x}_j A_j\, dt = q\int_{\mathcal{P}} \boldsymbol{A}\cdot d\boldsymbol{x}\,. \tag{3.20}$$

According to the relation (2.11) between the vector potential components and connection coefficients,

$$\exp\left(q\frac{i}{\hbar}\int_{\mathcal{P}} \boldsymbol{A}\cdot d\boldsymbol{x}\right) = \exp\left(i\int_{\mathcal{P}} (-\omega_j)dx_j\right) = \exp\left(i\int_{\mathcal{P}} (-\omega_\tau)d\tau\right) \equiv e^{i\alpha}\,, \tag{3.21}$$

where $\tau$ is the coordinate along the path. Note that the phase $\alpha$ is obtained by integrating the negative of the connection coefficient. It follows that the last expression describes a complex coordinate of the vector $e^{i\alpha}\boldsymbol{1}$ that is *parallel transported* along the path. Here, the connection coefficients are determined with respect to the basis section $\boldsymbol{1}$. In this way, the concept of parallel transport naturally appears in the path integral formalism.

Now consider any pair of paths going through the different slits.[6] The additional "magnetic" difference between the associated phases is

$$\Delta\phi_{12}^B = \int_{\mathcal{P}_2} (-\omega_j)dx_j - \int_{\mathcal{P}_1} (-\omega_j)dx_j = = \int_{\mathcal{P}_2-\mathcal{P}_1} (-\omega_j)dx_j = \int_{\mathcal{C}} (-\omega_j)dx_j\,, \tag{3.22}$$

where $\mathcal{C}$ is a closed contour formed by the difference of the two paths. It follows that

$$\Delta\phi_{12}^B = 2\pi\frac{\Phi}{\Phi_0}\,, \tag{3.23}$$

---

[6]We do not consider the paths that wind around the solenoid: according to the principle of stationary phase, the main contribution to the sum (3.18) comes from the trajectories that are close to the classical ones.

where $\Phi$ is the magnetic flux through the solenoid. By a similar argument, the phase difference between any two paths going through the same slit is zero. Effectively, the magnetic flux acts as a phase shifter installed into one of the slits. This leads to the shift of the interference pattern on the screen, a phenomenon known as the **Aharonov–Bohm effect** [33]. Note that when the flux is an integer multiple of $\Phi_0$, the phase difference equals $2\pi n$, and the interference pattern is the same as for $\Phi = 0$.

The Aharonov–Bohm effect was observed experimentally in several studies [34]. It was not easy for the physics community to accept that the static magnetic flux can affect the behavior of the electrons moving in the field-free region. The first experiments led to a controversy: it was argued that the effect resulted from the leakage of the magnetic field. The conclusive experiment was based on a toroidal ferromagnet fully coated by a superconductor. The stray magnetic field was expelled from the superconductor due to the Meissner effect, and the flux trapped inside was quantized in units of $\frac{\Phi_0}{2}$. The observed shifts of the interference pattern in different samples were either zero or half of the period of the pattern. This indicated that the superconductor behaved as expected, and confirmed the Aharonov–Bohm effect.

The relationship between the magnetic field and phases is used to create artificial gauge fields for *neutral* atoms trapped in an optical lattice [35]. It is possible to ensure that the atom acquires a phase during the "hopping" from one lattice site to another. Then the phase shifts are tuned in such a way that hopping around a closed loop is accompanied by a non-vanishing phase. As a result, the neutral atoms behave like charged particles in this synthetic magnetic field. In Sec. 3.3.5, we will consider a similar phenomenon, in which the phases appear as a result of an adiabatic evolution.

## 3.3 Geometric phase

As discussed above, the geometric picture of the electromagnetic field is based on an abstract complex line bundle over the spacetime. Now we are going to consider quantum mechanical vector bundles of a different kind. They arise as collections of eigenspaces of a quantum system over a **parameter space**. Each point of this space corresponds to a set of values of some parameters, which control the Hamiltonian of the system. The variation of the parameters can be thought of as traversing a path in the parameter space. The adiabatic evolution of an eigenstate over a cyclic path will result in the appearance of a phase factor, similar to the daily rotation angle of the Foucault pendulum. Moreover, both phenomena stem from the restriction of the evolution to some subspace. We will discuss this analogy in detail and provide a dictionary between the two settings.

### 3.3.1 Vector bundle over parameter space

Consider a Hamiltonian, which acts on a finite-dimensional space of states $\mathcal{H}$ and depends on the values of the control parameters:

$$\hat{H}(r_1, \ldots, r_m) \equiv \hat{H}(r), \quad r \in R. \tag{3.24}$$

Here, $R$ denotes the space of control parameters, formed by all possible combinations of their values. The parameter space can assume many shapes: for example, a single real parameter gives $R = \mathbb{R}$, a direction in three-dimensional space is described by a point on a sphere $R = S^2$ and two periodic variables give $R$ the form of a torus $T^2$.

We attach a copy $\mathcal{H}_r$ of $\mathcal{H}$ to each point $r \in R$, forming the **Hilbert space bundle** $\{\mathcal{H}_r\}$. Let $|n_r\rangle \in \mathcal{H}_r$ be an $n$-th Hamiltonian eigenstate separated by an energy gap from the other levels for all combinations of the control parameters. At each $r$, the eigenvector determines the eigenspace

$$V_r^n = \{a|n_r\rangle, a \in \mathbb{C}\}, \tag{3.25}$$

as a complex one-dimensional subspace of $\mathcal{H}_r$. Taken together, these subspaces form a complex line bundle over $R$, called an $n$-th **eigenspace bundle** $V^n$. Note the difference with the electromagnetic case, where the state of the particle is described by a *section* of a bundle.

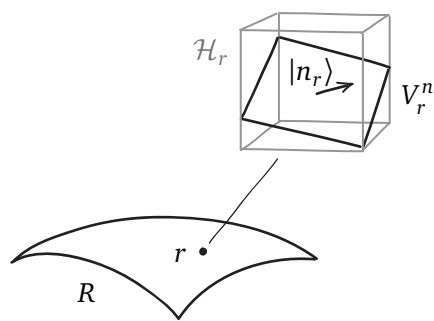

Figure 3.4: Eigenspace bundle $V^n$ over parameter space $R$. The bundle is formed by the eigenspaces $V^n_r$ associated with the energy level $\varepsilon_n$ of the Hamiltonian $\hat{H}(r)$. The point $r$ in the space $R$ is a combination of the control parameters of the Hamiltonian. The arrow represents the eigenstate $|n_r\rangle$, which belongs to the eigenspace $V^r_n$ lying in the space of states $\mathcal{H}_r$.

### 3.3.2 Adiabatic evolution and parallel transport

Assume that at the moment $t = 0$ the system is prepared in the eigenstate $|n_{r(0)}\rangle$ of the Hamiltonian $\hat{H}(r(0))$. Let $r(t)$ describe an adiabatically slow variation of control parameters, such that the system remains in the eigenspace $V^n_{r(t)}$ during the process (see Ref. [36] for a discussion of the adiabatic theorem in this context). Note that $|n_r\rangle$ as a complex vector is a specific element of the one-dimensional space $V^n_r$, and $e^{i\alpha}|n_r\rangle$ for some $\alpha \in \mathbb{R}$ describes the same physical state. We wish to consider, after Berry [37], the adiabatic evolution of the state in this general form.

Let $e^{i\alpha(t)}|n_{r(t)}\rangle$ be an eigenstate of $\hat{H}(r(t))$. Multiplying the Schrödinger equation

$$i\hbar\partial_t(e^{i\alpha}|n\rangle) = \hat{H}e^{i\alpha}|n\rangle, \tag{3.26}$$

by $\langle n|$ on the left, one finds

$$\partial_t\alpha = i\langle n|\partial_t n\rangle - \frac{\varepsilon_n}{\hbar}. \tag{3.27}$$

The last term describes the ordinary time evolution, or the dynamical phase, which can be eliminated by considering $\hat{H}_n = \hat{H} - \varepsilon_n\hat{\mathbb{1}}$ or equivalently by setting $\varepsilon_n = 0$.

The adiabatic evolution defines a way to transport the state vector $|n(0)\rangle$ through the set of eigenspaces over the path $r(t) \subset R$. Geometrically, we can use this as a definition of a parallel transport in the bundle $V^n$, and define in this way the **Berry connection**. Let us find the form of the covariant derivative associated with this parallel transport. To do this, we compare the present setting with that of the Foucault pendulum.

| Geometry | Foucault pendulum | Quantum state |
|---|---|---|
| Base space | surface of the Earth $S^2$ | parameter space $R$ |
| Ambient bundle | $T\mathbb{R}^3|_{S^2}$ (see Sec. 1.1.1) | Hilbert space bundle $\{\mathcal{H}_r\}$ |
| Subspace bundle | tangent planes $TS^2$ | eigenspaces $V^n$ |
| Basis section | $\mathbf{1}$ | $|n\rangle$ |
| Parallel transport | $\mathbf{v}_{PT} = e^{i\alpha}\mathbf{1}$ | $e^{i\alpha}|n\rangle$ |
| PT equation | $\partial_\varphi\alpha = -\omega_\varphi$ | $\partial_t\alpha = i\langle n|\partial_t n\rangle$ |

It follows that the connection coefficient is $\omega_t = -i\langle n|\partial_t n\rangle$. The negative of this,

$$A_t = i\langle n|\partial_t n\rangle, \tag{3.28}$$

is called the **Berry potential**, similarly to a component of the electromagnetic vector potential. The corresponding parallel transport angle for a closed contour $\mathcal{C}$,

$$\Delta\alpha(\mathcal{C}) = \int_{\mathcal{C}} A_t dt, \tag{3.29}$$

is known as the **Berry phase**. The properties of the parallel transport angle discussed in Sec. 1.3.3 apply to the Berry phase as well: it is invariant under reparametrizations of the path, and $\Delta\alpha$ mod $2\pi$ is independent of the choice of the basis section $|n\rangle$. Similarly to Eq. (3.28), the potential $A_\tau$ can be defined for any parameter $\tau$ describing a curve in the parameter space. The phase $\Delta\alpha(\mathcal{C})$ is determined by the geometry of the eigenspaces along the contour, and is also called the **geometric phase**.

We proceed with the analogy. The definition of $\omega_\tau$ translates as:

| Connection coefficient | $\nabla_\tau \mathbf{1} = i\omega_\tau \mathbf{1}$ | $\nabla_\tau|n\rangle = \langle n|\partial_\tau n\rangle|n\rangle$ |
|---|---|---|

The last expression can be rewritten as $|n\rangle\langle n|\partial_\tau n\rangle$. Introducing the projection operator

$$\text{Proj}^n = |n\rangle\langle n|, \tag{3.30}$$

on the subspace $V_r^n$ at each $r$, we find:

| Covariant derivative | $\nabla_\tau \mathbf{1} = \text{Proj}(\partial_\tau^a \mathbf{1})$ | $\nabla_\tau|n\rangle = \text{Proj}^n|\partial_\tau n\rangle$ |
|---|---|---|
| Derivative in the ambient bundle | $\partial_\tau^a \mathbf{v} = \mathbf{e}_i \partial_\tau v_i^a$ | $|\partial_\tau n\rangle = \sum_\alpha|\alpha\rangle\partial_\tau n_\alpha$ |
| Constant basis | $\{\mathbf{e}_x, \mathbf{e}_y, \mathbf{e}_z\}$ in $T\mathbb{R}^3$ | $\{|\alpha\rangle\}$ in $\mathcal{H}_r$ |

We conclude that the adiabatic evolution of the quantum state, subject to the combination of the Schrödinger equation and the adiabatic theorem, is mathematically equivalent to the rotation of the Foucault pendulum plane (up to the quantum dynamical phase). In both cases the parallel transport is defined by projection and depends only on geometry of certain subspaces.

There is, however, an important difference, which is contained in the last two lines of the table. Recall that for a projected connection, the covariant derivative relies on our ability to take the derivative $\partial^a$ in the ambient bundle. In the case of the Foucault pendulum, there is a standard Cartesian basis $\{\mathbf{e}_i\}$ in $T\mathbb{R}^3$, and we define $\partial^a$ by setting $\partial^a \mathbf{e}_i = 0$. The choice of $\{\mathbf{e}_i\}$ as a constant basis is justified by the linear structure of the base space $\mathbb{R}^3$ (see Eq. (1.13)) and by the identification between $\mathbb{R}^3$ and $T_0\mathbb{R}^3$. In contrast, there is no such relation between the parameter space $R$ and the fibers of the Hilbert space bundle $\{\mathcal{H}_r\}$. Therefore, we do not have a preferred basis section $|\alpha\rangle$ that can be declared to be constant. In Sec. 6.2.3, we will see how different choices of a constant basis in $\{\mathcal{H}_r\}$ can lead to different geometric phases for the same eigenspace bundle $V^n$.

### 3.3.3 Eigenspace bundle for two-level quantum system

Now we apply the general procedure described above to construct an eigenspace bundle, which will be extensively used in the next sections. Consider a quantum system with the two-dimensional Hilbert space of states $\mathcal{H}$. Once a basis $\{|\uparrow\rangle, |\downarrow\rangle\}$ is chosen, the space $\mathcal{H}$ can be identified with $\mathbb{C}^2$. A general form of the Hamiltonian matrix is

$$H = h_0\mathbb{I} + \sum_\alpha h_\alpha \sigma_\alpha = h_0\mathbb{I} + H_{\boldsymbol{h}}, \tag{3.31}$$

where $\mathbb{I}$ is the $2 \times 2$ identity matrix, and $\sigma_\alpha$ are Pauli matrices serving as the basis elements in the space of traceless Hermitian $2 \times 2$ matrices:

$$\sigma_x = \begin{pmatrix} 0 & 1 \\ 1 & 0 \end{pmatrix}, \quad \sigma_y = \begin{pmatrix} 0 & -i \\ i & 0 \end{pmatrix}, \quad \sigma_z = \begin{pmatrix} 1 & 0 \\ 0 & -1 \end{pmatrix}. \tag{3.32}$$

There is a natural parameter space associated with this Hamiltonian. Since $\hat{H}$ is determined by four real parameters, one may conclude that the parameter space must be four-dimensional. However, in the context of the geometric phase, we are interested only in those parameters whose changes affect the eigenstate. Suppose that $\psi$ is an eigenstate of the Hamiltonian (3.31):

$$H\psi = \varepsilon\psi. \tag{3.33}$$

Then it is also an eigenstate for $H_h$:

$$H_h\psi = (\varepsilon - h_0)\psi. \tag{3.34}$$

In other words, the eigenstate does not depend on the value of $h_0$. Moreover, $\psi$ is an eigenstate of the rescaled Hamiltonian $H_{\lambda h} = \lambda H_h$, where $\lambda \in \mathbb{R}$. It follows that $\psi$ depends only on the *direction* of the vector $h = (h_x, h_y, h_z)$. Thus, we can choose as a parameter space the unit sphere $S^2$ embedded in the three-dimensional space of Pauli matrices.[7]

The spectrum of the Hamiltonian (3.31) can be found by using the anti-commutation property of Pauli matrices: $\sigma_i \sigma_j = -\sigma_j \sigma_i$ for $i \neq j$. Together with $\sigma_i^2 = \mathbb{I}$, this implies that

$$H_h^2 = |h|^2 \mathbb{I}. \tag{3.35}$$

On the other hand, we have

$$H_h^2 \psi = (\varepsilon - h_0)^2 \psi, \tag{3.36}$$

for an eigenstate $\psi$. Hence, the eigenvalues of $H$ are

$$\varepsilon_\pm = h_0 \pm |h|. \tag{3.37}$$

To define the eigenstate bundle, we first attach a copy of $\mathcal{H}$ with a specified basis $\{|\uparrow\rangle, |\downarrow\rangle\}$ to each point $h \in S^2$. We further choose a high-energy state eigenvector $|\psi_{h+}\rangle \in \mathcal{H}$, which satisfies

$$H_h \psi_{h+} = \psi_{h+}. \tag{3.38}$$

Then we have a subspace

$$V_h^+ = \{a\psi_{h+} \mid a \in \mathbb{C}\} \subset \mathbb{C}^2, \tag{3.39}$$

attached to each point of the sphere. All such eigenspaces form a complex line bundle over $S^2$, which we will call the **monopole bundle** and denote $D^+$. As a base space for $D^+$, this sphere is known as the **Bloch sphere** in the context of spin-$\frac{1}{2}$ particle in the magnetic field $h$, or as the **Poincaré sphere** when it is used to describe the states of circularly polarized light.

To define a basis section, we need to find the eigenstates $\psi_{h+}$ on the sphere. To this end, we use the unitary matrix defined as

$$U(\theta, w) = \cos\frac{\theta}{2}\mathbb{I} + \sin\frac{\theta}{2}\sum_\alpha \frac{\sigma_\alpha}{i} w_\alpha, \tag{3.40}$$

---

[7]Here and throughout the notes, we will use the term "the space of Pauli matrices" as a shorthand for the "space of traceless Hermitian $2 \times 2$ matrices understood as a real three-dimensional vector space with Pauli matrices playing the role of basis vectors".

where $\boldsymbol{w}$ is a unit three-dimensional vector. This matrix has the property that under conjugation by $U(\theta, \boldsymbol{w})$, the Hamiltonian $H_{\boldsymbol{h}}$ is rotated in the space of Pauli matrices:

$$U(\theta, \boldsymbol{w})H_{\boldsymbol{h}}U(\theta, \boldsymbol{w})^{-1} = H_{\mathcal{R}(\theta,\boldsymbol{w})\boldsymbol{h}}, \tag{3.41}$$

where $\mathcal{R}(\theta, \boldsymbol{w})$ is the rotation through $\theta$ about $\boldsymbol{w}$ axis. If $\psi_{\boldsymbol{h}}$ is an eigenstate of $H_{\boldsymbol{h}}$, then the same-energy eigenstate of the rotated Hamiltonian reads

$$\psi_{\mathcal{R}(\theta,\boldsymbol{w})\boldsymbol{h}} = U(\theta, \boldsymbol{w})\psi_{\boldsymbol{h}}. \tag{3.42}$$

A physical discussion of this formalism can be found, for example, in Ref. [31]. In group theory, this is related to the representation of rotations using quaternions and to the two-to-one homomorphism between Lie groups $SU(2) \to SO(3)$. For a mathematical introduction to these topics, see the textbook [38]. A physicist-oriented presentation is given, for example, in Ref. [3].

Let us apply this unitary rotation to the eigenstate $\psi = (1,0)^T$ of $H_{\boldsymbol{h}} = \sigma_z$, so $\boldsymbol{h} = (0,0,1)^T$, and choose $\boldsymbol{w} = (-\sin\varphi, \cos\varphi, 0)^T$. Then the $\theta$ rotation of $\boldsymbol{h}$ about $\boldsymbol{w}$ axis gives the point with coordinates $(\theta, \varphi)$ on the sphere. The corresponding eigenstate is

$$\psi_+(\theta, \varphi) = \begin{pmatrix} \cos\frac{\theta}{2} \\ e^{i\varphi}\sin\frac{\theta}{2} \end{pmatrix}. \tag{3.43}$$

Note that this expression is conceptually similar to the column of components of $\boldsymbol{e}_\theta$ as a three-dimensional vector, Eq. (1.9). In both cases, we express a section of a bundle over $S^2$ in terms of the basis for a higher-dimensional ambient bundle.

Thus, we have defined the eigenspace bundle $D^+$ for a two-level quantum system and have specified its basis section $|\psi_+\rangle$.

### 3.3.4 Geometry of eigenspace bundles

Sometimes it is convenient to extend the base space of the monopole bundle $D^+$ to the whole space of Pauli matrices. Since the eigenstates are degenerate at $\boldsymbol{h} = 0$, we must exclude the origin. This gives the three-dimensional parameter space $\mathbb{R}^3 \setminus \{0\}$, which consists of all non-zero vectors $\boldsymbol{h}$. We denote the corresponding eigenstate bundle $D_3^+$. In this section, we compute connection coefficients and curvature components for the Berry connection on $D_3^+$. Then we use these results to find the curvature of a general eigenstate bundle defined by a two-level Hamiltonian, which varies over a parameter space $R$.

First, we introduce on $\mathbb{R}^3 \setminus \{0\}$ the spherical coordinate system $(\theta, \varphi, \rho)$, where $\rho = |\boldsymbol{h}|$. The eigenstates do not depend on $\rho$ and are given again by Eq. (3.43). Connection coefficients along the coordinate curves are

$$\omega_\varphi = -i\langle\psi_+|\partial_\varphi\psi_+\rangle = \sin^2\frac{\theta}{2}, \quad \omega_\theta = 0, \quad \omega_\rho = 0. \tag{3.44}$$

From the connection coefficients one finds the **Berry curvature** component of the bundle $D_3^+$ for each pair of coordinates:

$$f_{\theta\phi} = -\frac{1}{2}\sin\theta, \quad f_{\theta\rho} = f_{\varphi\rho} = 0. \tag{3.45}$$

Comparing this with the result for $TS^2$ discussed in Sec. 2.2.2, we conclude that the Berry phase for a boundary contour $\partial\Sigma$ of a surface $\Sigma \subset \mathbb{R}^3 \setminus \{0\}$ is

$$\Delta\alpha(\partial\Sigma) = -\frac{1}{2}\Omega(\Sigma), \tag{3.46}$$

where $\Omega$ is a solid angle enclosed by $\Sigma$ in the space of rays $\boldsymbol{h}$. The value given by this equation agrees with the line integral in Eq. (3.29), if the potential $A_\tau$ corresponds to a smooth basis section over $\Sigma$.

**Exercise 3.2.** Employ the unitary transformation $U(\theta, \boldsymbol{w})$ to find the negative-energy eigenstates $\psi_-$. Calculate connection coefficients and curvature components for the corresponding bundle $D_3^-$. [§4.3.2, 5.2, 5.3]

Now let us find the curvature of $D_3^+$ in terms of Cartesian coordinates $h_\alpha = h_x, h_y, h_z$. The solid angle spanned by an infinitesimal coordinate rectangle $dS_{\alpha\beta}$ located at the point with the position vector $\boldsymbol{h}$ is given by

$$d\Omega_{\alpha\beta} = \cos\nu \frac{dS_{\alpha\beta}}{|\boldsymbol{h}|^2}, \tag{3.47}$$

where $\nu$ is the angle between the normal to the rectangle and $\boldsymbol{h}$. Since the element $dS_{\alpha\beta}$ is orthogonal to the third coordinate $h_\gamma$, the cosine is given by $\frac{h_\gamma}{|\boldsymbol{h}|}$. Using Levi-Civita anti-symmetric symbol to take the orientation of the rectangle into account, we have

$$f_{\alpha\beta} = -\frac{1}{2} \frac{\varepsilon_{\alpha\beta\gamma} h_\gamma}{|\boldsymbol{h}|^3}. \tag{3.48}$$

In what follows, we will be interested in the eigenspace bundle of a two-level system defined in the general setting of Sec. 3.3.1. Let $\hat{H}$ be a two-level Hamiltonian, which depends on the control parameters $r = (r_1, \ldots, r_m)$ forming the parameter space $R$. The Hamiltonian $\hat{H}(r)$ determines the vector $\boldsymbol{h}(r)$ in the space of Pauli matrices. Globally, the this gives rise to the vector field $\boldsymbol{h}$ on the space $R$. For each point $r \in R$, we find the eigenspace corresponding to the high-energy eigenstate and thus define the eigenspace bundle $V^+$ over $R$:

$$\text{two-level Hamiltonians } \hat{H} \text{ over } R \;\Rightarrow\; \text{vector field } \boldsymbol{h} \;\Rightarrow\; \text{complex line bundle } V^+. \tag{3.49}$$

We wish to express the Berry curvature of $V^+$ in terms of the vector field $\boldsymbol{h}$. To do this, we adapt the solution of an analogous problem discussed in Sec. 2.2.3, where we computed the curvature of the real plane bundle $M$ defined by a vector field $\boldsymbol{m}$ of plane normals. Recall that the vector field $\boldsymbol{m}$ determined the map $m$ to the sphere, which allowed us to relate the curvature of $M$ with the curvature of $TS^2$. Here, the role of the standard bundle with the known curvature will be played by the monopole bundle $D_3^+$.

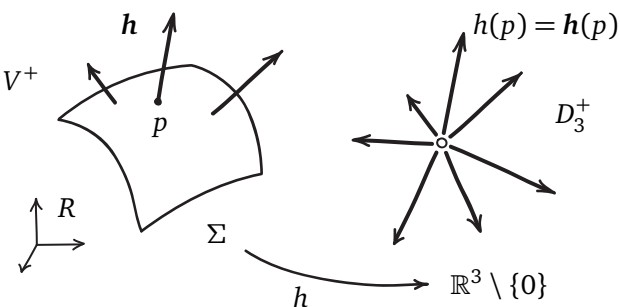

Figure 3.5: Eigenspace bundle $V^+$ over the parameter space $R$ with a specified surface $\Sigma \subset R$. The two-level Hamiltonian $\hat{H}$ over $R$ defines a vector field $\boldsymbol{h}$, which determines a map $h$ from $\Sigma$ to the base space $\mathbb{R}^3 \setminus \{0\}$ of the monopole bundle $D_3^+$. The map sends a point $p \in \Sigma$ to the vector $\boldsymbol{h}(p) \in \mathbb{R}^3 \setminus \{0\}$.

Consider a two-dimensional surface $\Sigma \subset R$ with coordinates $(x_1, x_2)$. The vector field $\boldsymbol{h}$ over $\Sigma$ defines the map $h : \Sigma \to \mathbb{R}^3 \setminus \{0\}$, which sends a point $p \in \Sigma$ to the corresponding vector $\boldsymbol{h}(p) \in \mathbb{R}^3 \setminus \{0\}$, as shown in Fig. 3.5. In coordinates, the map is described by three functions $h_\alpha(x_1, x_2)$. As we did in Sec. 2.2.3, we interpret these functions as a definition of new coordinates for $\mathbb{R}^3 \setminus \{0\}$. Note that Eq. (2.35) from the Exercise 2.4 can be written as

$$f_{12} = (\partial_1 \alpha)(\partial_2 \beta) f_{\alpha\beta}, \tag{3.50}$$

where we sum over the indices $\alpha, \beta$, which enumerate coordinates $\theta$ and $\varphi$ on the sphere. A similar result holds in the present context, with $h_\alpha$ playing the role of the coordinate $\alpha$ and $f_{\alpha\beta}$ describing the curvature components of $D_3^+$. Taking into account Eq. (3.48), we obtain

$$f_{12} = -\frac{1}{2} \frac{\varepsilon_{\alpha\beta\gamma} h_\gamma}{|\boldsymbol{h}|^3} (\partial_1 h_\alpha)(\partial_2 h_\beta) = -\frac{1}{2|\boldsymbol{h}|^3} \boldsymbol{h} \cdot [\partial_1 \boldsymbol{h} \times \partial_2 \boldsymbol{h}]. \tag{3.51}$$

The quantity $f_{12}$ describes the curvature of $D_3^+$ in the new coordinates $(x_1, x_2)$, and also gives the desired curvature over the surface $\Sigma$ in the parameter space $R$. The mixed product divided by $|\boldsymbol{h}|^3$ in the last expression is the "surface density" of the solid angle formed by the nearby vectors $\boldsymbol{h}$. Similarly to the case of the real plane bundle, the Berry phase, or the parallel transport angle, for the boundary of the surface $\Sigma$ is given by

$$\Delta\alpha(\partial\Sigma) = \int_\Sigma f_{12} dx_1 dx_2 = -\frac{1}{2}\Omega(h(\Sigma)), \tag{3.52}$$

where $\Omega(h(\Sigma))$ is the solid angle covered by the image of the surface $\Sigma$ under the map $h$ in the space $\mathbb{R}^3 \setminus \{0\}$.

### 3.3.5 Dirac monopole and emergent electrodynamics

Finally, let us interpret the monopole bundle $D_3^+$ over $\mathbb{R}^3 \setminus \{0\}$ with the Berry connection as a description of the magnetic field $\boldsymbol{B}$ in the three-dimensional physical space, according to (2.15). Note that the position vector $\boldsymbol{h}$ does *not* describe the magnetic field in question. Consider a region $\Sigma \subset S^2$ of the sphere of radius $r$ centered at the origin of $\mathbb{R}^3 \setminus \{0\}$. From Eqs. (2.17) and (3.46), the magnetic flux through $\Sigma$ is given by

$$\Phi(\Sigma) = \Phi_0 \frac{\Delta\alpha(\partial\Sigma)}{2\pi} = -\Phi_0 \frac{\Omega(\Sigma)}{4\pi}. \tag{3.53}$$

Thus, the total flux through the sphere is $\Phi(S^2) = -\Phi_0$, which means that the sphere contains a source of the magnetic field. We know that the only non-zero curvature component is $f_{\theta\varphi}$, which gives the radial component of the magnetic field strength. In the limit of region $\Sigma$ shrinking to a point $p$ on the sphere,

$$B_r = \lim_{\Sigma \to p} \frac{\Phi(\Sigma)}{S(\Sigma)} = -\frac{1}{r^2} \frac{\Phi_0}{4\pi} = -\frac{1}{r^2} \frac{\hbar}{2e}. \tag{3.54}$$

This is the field configuration of the **Dirac monopole** with the magnetic charge $-\frac{\hbar}{2e}$, placed at the origin.

Note that this electromagnetic interpretation mixes the two types of vector bundles mentioned in the introductory paragraph of Sec. 3.3. In the eigenstate bundle $V^+$, each fiber $V_r^+$ corresponds to the state of the system at a given point $r$ of the parameter space $R$. In the electromagnetic case, the particle is delocalized over a region of the physical space $\mathbb{R}^3$, and its state is described by a *section* $\boldsymbol{\psi}$ of a complex line bundle. Interestingly, there is a situation in which coexistence and interplay of these two settings give rise to observable physical effects.

Consider a **skyrmion**, a vortex-like distribution of magnetization $\boldsymbol{h}(x, y)$ on a plane, shown in Fig. 3.6 on the left. Such structures exist in non-centrosymmetric magnetic materials and are stabilized by the Dzyaloshinskii–Moriya intercation (see Ref. [39] for a recent overview). We assume that the magnetization has the constant modulus $|\boldsymbol{h}|$. The skyrmion is said to have a topological charge, since the image of the plane under the map $h : \mathbb{R}^2 \to S^2$ defined by the magnetization vector $\boldsymbol{h}$ covers the whole sphere. But for now we are interested in the local geometric effects of the spatial inhomogeneity of the magnetization.

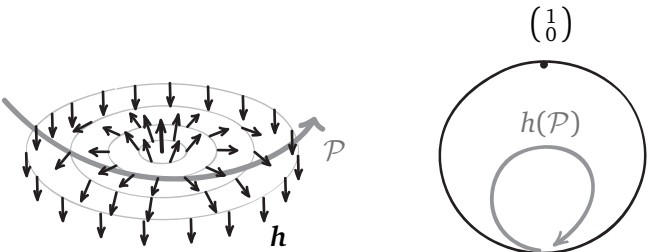

Figure 3.6: Since the electron's spin is aligned with the magnetization $\boldsymbol{h}$, the real-space trajectory $\mathcal{P}$ of an electron moving through the skyrmion (LEFT) is mapped to the trajectory on the Bloch sphere (RIGHT).

Suppose that the skyrmion exists in a magnetic metal, and conductance electrons are free to flow through the structure. We further assume that their motion is semi-classical and adiabatic, in the following sense. First, the electron is localized in a region with an approximately constant direction of $\boldsymbol{h}$. Second, electron's spin is always aligned with the magnetization, which acts as a background magnetic field. Under these assumptions, any trajectory traced out by the electron in the plane is mapped to the space of spin states, or the Bloch sphere. In this way, the evolution of the spin state becomes equivalent to the Berry phase problem for the spin in a slowly rotating field. Note that the plane containing the skyrmion acts both as a physical space for the charged particle and as a parameter space for the two-level Hamiltonian, which describes the coupling between the spin and the magnetization $\boldsymbol{h}$.

Now let us look at the motion of the electron from the path integral perspective. Recall that the sum of phases along all possible paths appears as a result of decomposition of the unitary evolution operator. Here, such evolution splits into the real-space part and the spin-space part. Thus, besides the phase (3.18), each trajectory will be weighted by the Berry phase. But regardless of the origin of the phase shift, it leads to the real Lorentz force felt by electrons. In a similar way, time-dependent magnetic structures can create effective electric field. These effects are manifestations of **emergent electrodynamics** [40].

> **Exercise 3.3.** Estimate the strength of the emergent magnetic field, given that the typical diameter of a skyrmion is 100 nm.

## 3.4 Summary and outlook

In this section, we discussed how geometric properties of vector bundles manifest themselves in various quantum systems. In Sec. 6.4, we will encounter phenomena related to the flux insertion (both for the magnetic flux and for the Berry flux). The shape of the eigenstate bundle encoded in the Berry connection will play a key role in our discussion of the topological band theory. Here is the summary of our results:

- The energy and momentum values of a particle on a ring are affected by the magnetic flux piercing the ring, even though the particle moves in the field-free region.

- Geometrically, the curvature (magnetic field strength) in the interior region changes the parallel transport along the ring and thus modifies the covariant derivative (the momentum operator). For a general value of the flux, there is no covariantly constant section of the bundle over the ring (no zero-momentum state).

- Aharonov–Bohm effect occurs in a two-slit diffraction experiment, which is modified by insertion of a solenoid between the slits. The magnetic flux inside the solenoid result in the shift of the interference pattern. In the path integral picture, this is a result of an additional phase (given by the parallel transport angle), which changes the phase factor associated with each path.

- For each isolated energy level of a Hamiltonian depending on the control parameters, one defines an eigenspace bundle over the parameter space $R$. This bundle can be equipped with Berry connection. The associated parallel transport along a path in $R$ corresponds to the adiabatic evolution of a state due to the slow changes of the control parameters.

- Berry phase $\Delta\alpha$ is the parallel transport angle associated with a closed path in the parameter space. Its value modulo $2\pi$ is gauge-invariant.

- Berry curvature is the curvature of the Berry connection. For a two-level Hamiltonian (3.31) depending on the control parameters, the curvature of the eigenstate bundle is related to the "solid angle density" of the vector field $\boldsymbol{h}$ defined over $R$.

- Berry phase acquired by an electron during its motion through a skyrmion leads to the emergent magnetic field, which acts on the electron with the Lorentz force.

Geometric phases arise in a variety of physical contexts, as discussed in a book [41] (the reader may also consult a recent overview [42]). Applications to the electronic structure theory are reviewed in an article [43] and are thoroughly discussed in a textbook [9]. Below, we make two remarks on the possible generalizations of the Berry phase.

▷ **Beyond adiabatic evolution.**  The geometric interpretation of the Berry phase in terms of connection on a complex line bundle is due to Simon [44]. He showed that the parallel transport defined by the adiabatic evolution (without the dynamical phase) corresponds to a standard connection on an embedded bundle. In this way, the physics of adiabatic evolution was linked with the geometry of complex subspaces. It turns out that the latter point of view is more fundamental, as the following examples show.

Aharonov and Anandan introduced the geometric phase associated with a cyclic, but not necessarily adiabatic, evolution of a quantum state [45]. They considered a solution $|\psi(t)\rangle$ of the Schrödinger equation

$$\hat{H}(t)|\psi(t)\rangle = i\hbar\frac{d|\psi(t)\rangle}{dt}, \tag{3.55}$$

which defines a path in the space of rays. Here, a **ray** associated with the state $|\psi(t)\rangle$ is the corresponding one-dimensional subspace[8] of $\mathcal{H}$ (cf. Eq. (3.25)). If the path returns to the starting ray, the state acquires a phase, which has a dynamical and a geometric parts. The key difference with the setting considered by Berry is that here the state $|\psi(t)\rangle$ need not be

---

[8]In mathematics, the space of rays is known as the projective space. We will briefly discuss such spaces in Sec. 4.5.

an eigenstate of the Hamiltonian $\hat{H}(t)$. A simple instance of the Aharonov–Anandan phase occurs in the case of spin-$\frac{1}{2}$ sate "precessing" around the static magnetic field described by the Hamiltonian $\hat{H}(t) = \sigma_z$. If the state moves along the line of latitude $\theta$, it acquires the phase $\pi(1 - \cos\theta)$, that is, half the solid angle subtended by its trajectory.

Another example of a geometric phase, which generalizes the Berry phase in several ways, is the Zak phase [46]. It is determined by the geometry of the eigenspace bundle associated with Bloch eigenstates $|\psi_k\rangle$ of some band in a one-dimensional crystal. The Bloch Hamiltonian $H_k$ depends on the crystal momentum $k$, which varies across the periodic Brillouin zone. Originally, Zak derived this phase by considering adiabatic "motion" of an eigenstate through the Brillouin zone under the applied electric field. He noted, however, that the phase can be "utilized for specifying entire bands". Today, the Zak phase plays a key role in the modern theory of electric polarization, and is important in topological band theory in general. In this context, the Zak phase is a static property of the band in question and is not related to the physical adiabatic evolution. We will discuss the geometric and physical meaning of the Zak phase in Sec. 6.2.

▷ **Non-Abelian geometric phase.** The procedure described in Sec. 3.3.2 relies on the condition that the level of interest does not become degenerate with other levels. In fact, this can be relaxed, giving rise to the non-Abelian phase factors associated with groups of energy levels, which were introduced by Wilczek and Zee in Ref. [47]. Consider a group of $N$ energy levels, which are possibly degenerate at some points of the parameter space, and denote $V$ the corresponding eigenstate bundle. Let $\{|\chi_n\rangle\}$ for $n = 1, \ldots, N$ be a set of smooth basis sections of $V$. In other words, at each point of the parameter space, this set of vectors span the same subspace as the eigenstates from the group (note that $|\chi_n\rangle$ itself need not be an eigenstate). One can define the covariant derivative in $V$ as

$$\nabla_\tau |\varphi\rangle = \text{Proj}\, |\partial_\tau \varphi\rangle\,, \tag{3.56}$$

where $\text{Proj} = \sum_n |\chi_n\rangle\langle\chi_n|$ is the projection onto the subspace spanned by the eigenstates from the group, and $|\varphi\rangle$ is a section of $V$. In components, we have

$$\nabla_\tau |\varphi\rangle = \nabla_\tau \sum_m \varphi_m |\chi_m\rangle = \sum_m \Big(\partial_\tau \varphi_m - i \sum_n \varphi_n A_\tau^{mn}\Big)|\chi_m\rangle\,, \tag{3.57}$$

where $A_\tau^{mn} = i\langle\chi_m|\partial_\tau \chi_n\rangle$ is an element of the connection *matrix*. The parallel transport around a closed loop results now in a unitary transformation, instead of a scalar phase factor, which appears in the single-state case. The curvature can be defined as in Exercise 2.2, via the commutator of covariant derivatives along the coordinate curves $x_1$ and $x_2$:

$$[\nabla_1, \nabla_2]|\varphi\rangle = \frac{1}{i}\Big(\partial_1 A_2 - \partial_2 A_1 - i[A_1, A_2]\Big)|\varphi\rangle \equiv \frac{1}{i}F_{12}|\varphi\rangle\,, \tag{3.58}$$

as the reader may check. Note the appearance of the commutator term due to the non-Abelian nature of connection.

# 4 Topology of vector bundles

Above, we considered two differential-geometric devices, which allow one to make measurements in vector bundles. One is bundle metric (Sec. 1.1.2), which measures length of vectors in a fiber and angles between them. The other is connection, which "connects" fibers at nearby points by providing a way to transport vectors from one fiber to another; at the same time,

it gives rise to the covariant derivative. Then, parallel transport around an infinitesimal loop defines the curvature of connection. Note that all these measurements are *local*, meaning that they happen near a given point. One can argue that the parallel transport along a curve is a non-local phenomenon, but in fact it is defined as a solution of the differential equation (1.38) at all points of the curve.

Vector bundles also have *global* properties, which do not depend on the results of local measurements. These properties are topological, meaning that they can be defined in terms of continuous maps, without any reference to metric or connection. In this section, we consider one such property of a complex line bundle $M$: an integer-valued invariant called Chern number $c(M)$. Surprisingly, while the Chern number does not depend on geometry, it can be computed from the local geometric data, as we discuss in Sec. 4.2. Then we will learn how to evaluate the Chern number in an important particular case, when the bundle is associated with a map (Sec. 4.3). This will help us to sketch a proof of the Gauss–Bonnet theorem. In later sections, we will see how the Chern number provides a basic classification of topological states of matter. We introduce the mathematical formalism underlying such classifications in Sec. 4.4.

## 4.1 Trivial and non-trivial bundles

### 4.1.1 What is topology about?

Topological space is a set equipped with an additional structure that allows one to determine whether a map between two such spaces is continuous. This structure is called **topology**, also giving the name to the branch of mathematics that studies the concept of continuity in this general setting. One of the central problems of topology is the classification of spaces by their topological properties, which can be defined in terms of the continuous maps.

For example, consider a map $f : \mathbb{R} \to \mathbb{R}$, or an ordinary real-valued function of real numbers. In this case, the topological definition of continuity coincides with the $\varepsilon$-$\delta$ definition from the real analysis. Now let us replace the codomain of $f$ by the integers $\mathbb{Z} \subset \mathbb{R}$. As a result, our choice of continuous functions is restricted:

$$\text{any continuous function } f : \mathbb{R} \to \mathbb{Z} \text{ is constant.} \tag{4.1}$$

This indicates that the domain and codomain of $f$ must have certain topological properties (namely, $\mathbb{R}$ is connected and $\mathbb{Z}$ is discrete).

Another essentially topological statement is the intermediate value theorem and its corollaries. Consider a continuous function $f : [0,1] \to I$ from the unit interval to another interval $I = [-1,1]$. If $f(0) = a < 0$ and $f(1) = b > 0$, then the function must vanish at some point. If we interpret the domain of $f$ as a time interval, the function defines a trajectory of a point moving from $a$ to $b$. The statement tells us that any such trajectory must go through the zero $0 \in I$. The situation is different for the continuous maps $f : [0,1] \to S^1$ from the interval to the circle. We can try to reproduce the previous setting: choose any three distinct points on the circle and denote them $a$, $b$ and $0$. As above, the "origin" $0$ lies between $a$ and $b$; however, there always exist a function such that $f(0) = a$, $f(1) = b$, but $f(t) \neq 0$ for all $t \in [0,1]$. We conclude that the circle $S^1$ and the interval $I$ are topologically distinct (both spaces are connected, but only the interval is *simply*-connected).

Topological properties of a space capture the general aspects of its shape, but are independent of the particular geometric form. Instead of the circle $S^1$, we could take an ellipse; it does not matter if the interval $I$ is a part of a straight line or a segment of a one-dimensional curve. So, intuitively, topological properties are invariant under continuous deformations, such as bending or stretching. Examples of operations that can change the topology of a space are cutting and gluing, which can turn $S^1$ into $I$ and vice versa.

### 4.1.2 Topology of line bundles

Recall that a section $\boldsymbol{v}$ of a vector bundle $V$ is a smooth function $\boldsymbol{v} : \mathcal{B} \rightarrow V$ from the base space $\mathcal{B}$ to the bundle, such that $\boldsymbol{v}(p) \in V_p$ for all $p \in \mathcal{B}$. Following the pattern of the examples discussed above, we will now use sections to define a topological property of a vector bundle. We begin with real line line bundles over the circle and then generalize this approach to the complex line bundles over closed two-dimensional surfaces.

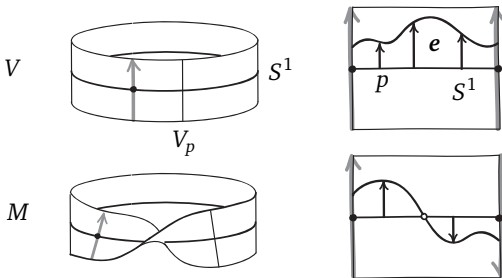

Figure 4.1: The cylinder $V$ and the Möbius band $M$ as examples of trivial and non-trivial real line bundles over the circle $S^1$. The fibers are isomorphic to $\mathbb{R}$, but are shown as intervals, for clarity. LEFT: Two bundles embedded into the three-dimensional space. RIGHT: Same bundles represented using identifications (shown by arrows). A global non-vanishing section of $V$ is denoted $\boldsymbol{e}$. A section of $M$ has a single zero, which is marked by an empty circle.

Let $V$ be a real line bundle over the circle $S^1$, constructed as follows. We attach a real one-dimensional vector space to each point of the circle in such a way that the spaces form the surface of the cylinder, as shown in Fig. 4.1. The choice of a basis vector in a fiber $V_p$ allows us to represent it as a real line:

$$V_p \xrightarrow[\text{basis } \boldsymbol{e}(p)]{\text{choice of}} \mathbb{R}, \tag{4.2}$$

since any vector $\boldsymbol{v}(p) = v\boldsymbol{e}(p)$ is described by its component $v \in \mathbb{R}$. Globally, this yields an identification of the bundle with a product space:

$$V \xrightarrow[\text{section } \boldsymbol{e}]{\text{choice of}} S^1 \times \mathbb{R}, \tag{4.3}$$

once a non-vanishing basis section $\boldsymbol{e}$ is chosen. Note also that in this case any other section $\boldsymbol{v} = v\boldsymbol{e}$ is described by a real function

$$v : S^1 \rightarrow \mathbb{R}. \tag{4.4}$$

Bundles that can be identified with a product space by a choice of a global non-vanishing section are called **trivial**. Note that the circle $S^1$ can be thought of as an interval $[0,1]$ with endpoints glued together. In this case, we need to specify how the fibers $V_0$ and $V_1$ are identified. This is indicated in Fig. 4.1 by gray arrows.

Now suppose that we reverse one of the fibers over the endpoints in the planar picture, and obtain a new bundle $M$. In three dimensions, this gives the bundle the shape of the Möbius band (of infinite width). Such twisted identification forces any smooth section to pass through zero, and makes the bundle **non-trivial**. This bundle cannot be represented as a product space, and its sections cannot be described as functions, in contrast with Eqs. (4.3) and (4.4). Note that *locally* bundles $V$ and $M$ look similar. By choosing a non-vanishing section along an interval $I \subset S^1$, one can identify the part of either bundle over $I$ with the product $I \times \mathbb{R}$. The

difference between bundles $V$ and $M$ can only be detected if we look at the whole circle $S^1$. We conclude that being (non-)trivial is a global topological property of these bundles. In fact, this property is defined in a similar way for any vector bundle.

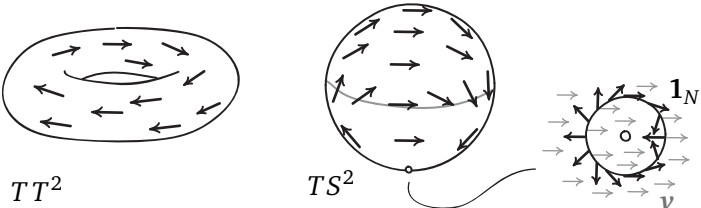

Figure 4.2: Tangent bundles of the torus and the sphere as examples of trivial and non-trivial complex line bundles. LEFT: There is a non-vanishing tangent vector field on the torus, which allows one to identify $TT^2$ with $T^2 \times \mathbb{C}$. RIGHT: The section $\mathbf{1}_N$ of $TS^2$ introduced in Exercise 1.3 has a singularity at the south pole. Inset shows the section $\mathbf{1}_N$ on a circle near the south pole and a smooth section $\boldsymbol{v}$.

A complex line bundle over the base space $\mathcal{B}$ can be identified with $\mathcal{B} \times \mathbb{C}$ if there exists a smooth non-vanishing section $\mathbf{1}$ over all of $\mathcal{B}$. If no such section exists, the bundle is non-trivial. More generally, a bundle with $n$-dimensional complex vector space as a fiber is said to be trivial if it can be identified with a product $\mathcal{B} \times \mathbb{C}^n$. To make such an identification, one needs $n$ global linearly independent sections forming together a basis in each fiber.

Consider the tangent bundle of the torus as a complex line bundle (Fig. 4.2). Since there is a smooth non-vanishing field of tangent vectors that goes along one of generating circles, the bundle is trivial. The tangent bundle of the sphere $TS^2$ has different nature. Its section $\mathbf{1} = \boldsymbol{e}_\theta$ has two singularities at the poles. One can try to construct a smooth section $\mathbf{1}_N$ by transporting some vector from the north pole along the meridians (see Exercise 1.3). The resulting section is smooth at the north pole, but there is a singularity at the south pole, which winds twice around it, as shown in Fig. 4.2. We will see that any section of $TS^2$ must have singularities, and the bundle is topologically non-trivial. This implies that $TS^2$ cannot be identified with the product space $S^2 \times \mathbb{C}$. It follows that a section $\boldsymbol{v}$ of $TS^2$ cannot be described by a single complex function $v : S^2 \to \mathbb{C}$.

Thus, the global twisting of the bundle space leads to the appearance of singularities in its sections. We wish to use these point-like singularities to characterize the non-trivial topology of complex line bundles. In this context, it is promising to consider bundles over two-dimensional closed surfaces, for the following reasons. First, a section of a bundle can have stable point-like singularities when the dimensionality of the base space coincides with that of the fiber (as a real vector space); we will discuss this principle in more detail in Sec. 10.1. For the real line bundles this gives $d = 1$ (hence the choice of the circle $S^1$ as a base space in the examples above), and for the complex line bundles $d = 2$. Second, the base space must be closed, since otherwise the singularities can be pushed away through the boundary, and the bundle is always trivial.

## 4.2  Chern number

Note that in contrast with real line bundles, a singularity in a complex line bundle can be labeled by an index, which shows how many times the vectors of the section rotate when one goes around a small contour near the singularity. As we will see below, the sum of these numbers is a topological invariant of a complex line bundle.

### 4.2.1 Index of a singularity

We start with a formal definition of the index. The basis sections $\mathbf{1}$ and $\mathbf{1}_N$ of $TS^2$ are examples of local unit sections: they are not defined over the whole base space and consist of vectors of unit length. In general, if a unit section $\mathbf{1}$ is not defined in an isolated point $p$ of the base space, we will say that $\mathbf{1}$ has a **singularity** at $p$. Let $\mathbf{v}$ be some smooth unit section in a neighborhood of $p$. Choose a positively-oriented contour $\mathcal{C}$ near $p$ such that both sections are defined on it. Then along the contour, the two sections are related by $\mathbf{1} = e^{i\beta}\mathbf{v}$, where $\beta$ is a smooth function of the coordinate $\gamma$ on $\mathcal{C}$. We define the **index of a singularity** of the section $\mathbf{1}$ at $p$ as

$$\text{ind}[\mathbf{1}(p)] = \frac{1}{2\pi}\int_{\mathcal{C}}(\partial_\gamma \beta)d\gamma\,. \tag{4.5}$$

For example, the singularity of $\mathbf{1}_N$ shown on the right in Fig. 4.2, has the index $+2$.

The index is an integer, and its value does not depend on the choice of the smooth section $\mathbf{v}$. Note that the sign of the index depends on the orientation of the surface, which determines that of the contour. There is also an implicit dependence on the orientation of the fiber, which is encoded in the complex structure (see Sec. 1.4.2).

### 4.2.2 Singularities and integral of curvature

Consider a complex line bundle $M$ over a closed two-dimensional surface $\mathcal{B}$. Let $\mathbf{1}$ be a basis section with singularities at points $p_i \in \mathcal{B}$. Let us try to find the integral of curvature $f_{12}$ over $\mathcal{B}$ using the Stokes' theorem (2.30). The theorem relates the integral of curvature with the integral of connection coefficient along the boundary, and also requires a smooth basis section. But in our case, $\mathcal{B}$ has no boundary and the section $\mathbf{1}$ is not smooth, which makes application of the theorem problematic. We can mend the situation by cutting a hole around each singularity.

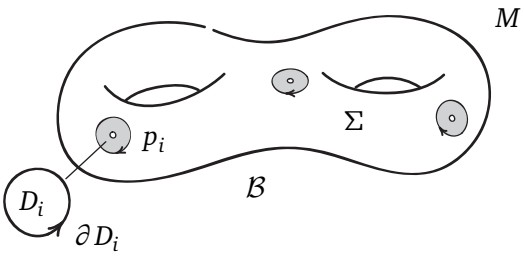

Figure 4.3: A complex line bundle $M$ over the closed surface $\mathcal{B}$ has a section $\mathbf{1}$ with singularities at the points $p_i$. The surface $\Sigma$ is obtained from $\mathcal{B}$ by removing small disks $D_i$ located near the points $p_i$.

Let $\{D_i\}$ be the set of small discs near the points $\{p_i\}$. Consider a region $\Sigma \subset \mathcal{B}$ that is obtained by removing these discs from $\mathcal{B}$:

$$\Sigma = \mathcal{B} \setminus \cup_i D_i\,. \tag{4.6}$$

The section $\mathbf{1}$ is smooth over $\Sigma$. The boundary of $\Sigma$ is given by

$$\partial \Sigma = \cup_i(-\partial D_i)\,, \tag{4.7}$$

since the boundary of each hole has the orientation opposite to that of the boundary of the disc (as discussed in Sec. 2.1.4). Now we apply the Stokes' theorem to the region $\Sigma$:

$$\frac{1}{2\pi}\int_\Sigma f_{12}dx_1dx_2 = \frac{1}{2\pi}\sum_i \int_{-\partial D_i}(-\omega_{\gamma_i})d\gamma_i\,, \tag{4.8}$$

where $\gamma_i$ parameterizes $\partial D_i$.

Consider one of the integrals from the sum. Let $\mathbf{1}'$ be a smooth section defined over the disc $D_i$. Then the singular section can be expressed as $\mathbf{1} = e^{i\beta(\gamma_i)}\mathbf{1}'$ along $\partial D_i$. Using the transformation law of connection coefficients (1.34), one finds:

$$\int\limits_{\partial D_i} \omega_{\gamma_i} d\gamma_i = \int\limits_{\partial D_i} \omega'_{\gamma_i} d\gamma_i + \int\limits_{\partial D_i} (\partial_{\gamma_i}\beta) d\gamma_i = -\int\limits_{D_i} f_{12} dx_1 dx_2 + 2\pi \operatorname{ind}[\mathbf{1}(p_i)]. \tag{4.9}$$

Finally, we take the limit $D_i \to p_i$, in which disks shrink to the points. Then the integrals of the curvature over the discs vanish $\int_D f \to 0$, while the region $\Sigma$ approaches the whole surface $\mathcal{B}$. Thus the limit of Eq. (4.8) is

$$\frac{1}{2\pi} \int_{\mathcal{B}} f_{12} dx_2 dx_2 = \sum_i \operatorname{ind}[\mathbf{1}(p_i)]. \tag{4.10}$$

This is a remarkable result: it provides a link between two seemingly unrelated quantities. On the left, we integrate the curvature $f_{12}$ of connection on $M$, which is a local geometric quantity. It does not depend on the choice of the basis section $\mathbf{1}$, but different connections can have different curvatures. The sum of the indices in the right-hand side is an integer corresponding to a particular section $\mathbf{1}$. It is a global characteristic of this section and does not depend on connection. The equality (4.10) implies that each side enjoys the properties of the other one:

1. The integral of the curvature $f_{12}$ of connection on $M$ is an integer multiple of $2\pi$ and does not depend on the choice of connection.

2. For any section of $M$ with point-like singularities, the sum of their indices is the same.

The sum of indices in Eq. (4.10) is an intrinsic topological property of any complex line bundle $M$ over a closed two-dimensional surface. We will call this integer the **Chern number** $c(M)$ of the bundle $M$. If there is a connection on $M$, one can compute $c(M)$ by integrating the curvature. This definition is suitable only for complex line bundles over two-dimensional closed surfaces. More generally, one defines a family of Chern numbers, which can characterize a complex bundle over a higher-dimensional base space. In this context, $c(M)$ is called the *first* Chern number.

### 4.2.3 Topological stability

What will happen with the Chern number if we deform the bundle $M$? We are interested in two types of complex line bundles: first, the plane bundles (1.56) defined by the vector field of plane normals $\boldsymbol{m}$; second, the eigenspace bundles (3.49) associated with eigenstates of a non-degenerate energy level of some Hamiltonian $\hat{H}$. By a "deformation of $M$", we will mean a smooth dependence of the corresponding vector field $\boldsymbol{m}$ (resp. the Hamiltonian $\hat{H}$) on an external parameter $t \in [0,1]$, such that the bundle remains well-defined during the process. In the first case, this means that the vector $\boldsymbol{m}_t(p) \neq 0$ for all points $p \in \mathcal{B}$ and all values of $t$. Here, $\boldsymbol{m}_t$ denotes the family of vector fields parameterized by $t$. For an eigenspace bundle, we require that the energy level of interest not become degenerate with the other levels.

Consider a deformation of a plane bundle $M_t$ for $t \in [0,1]$. Recall that the curvature component can be expressed in terms of derivatives of the vector field $\boldsymbol{m}$, and thus $f_{12}(p,t)$ is a smooth function of $t$ at each $p \in \mathcal{B}$. It follows that the Chern number $c(M_t)$ is also a smooth function of $t$, and by (4.1) it must be constant. We conclude that the Chern number is invariant under smooth deformations of the bundle: if two bundles $M_0$ and $M_1$ over $\mathcal{B}$ are related by a

smooth deformation $\boldsymbol{m}_t$ of of the corresponding vector fields, then $c(M_0) = c(M_1)$. One can also rephrase this in the spirit of the intermediate value theorem. If the Chern numbers of two bundles are different, then one cannot smoothly deform one into another. Still, we can consider a smooth interpolation $\boldsymbol{m}_t$ between their vector fields:

$$c(M_0) \neq c(M_1) \quad \Rightarrow \quad \text{there exist } p' \in \mathcal{B} \text{ and } t' \text{ such that } \boldsymbol{m}_{t'}(p') = 0. \tag{4.11}$$

The same argument applies to the eigenspace bundles and shows that changing of the Chern number $c(V^n)$ forces the energy level $\varepsilon_n$ to become degenerate at some point. We will consider an example of this situation in Sec. 8.2.4.

Here, we focused on the deformations of the vector field defining a bundle over a fixed base space. More generally, one can also allow smooth deformations of the base space. For example, this is relevant in the case of a tangent bundle, where the shape of the base space determines the configuration of the fibers.

### 4.2.4 Chern numbers of familiar bundles

▷ **Tangent bundle** $TS^2$. The curvature of the projected connection is $f_{\theta\phi} = \sin\theta$. The integral of the curvature gives

$$c(TS^2) = \frac{1}{2\pi} \int_{S^2} d\Omega = 2, \tag{4.12}$$

which implies that one cannot define a non-vanishing smooth tangent vector field on the sphere. This fact is colloquially known as the "hairy ball theorem", which says that one cannot comb hairs on a sphere without creating a cowlick. Indeed, the section $\mathbf{1} = \boldsymbol{e}_\theta$ has two singularities at the poles, each with index $+1$:

$$\text{ind}[\mathbf{1}(S)] + \text{ind}[\mathbf{1}(N)] = 2. \tag{4.13}$$

The section $\mathbf{1}_N$ has only one singularity at the south pole, but

$$\text{ind}[\mathbf{1}_N(S)] = 2. \tag{4.14}$$

Note that the different orientation of the sphere would change signs of the curvature, indices, and Chern number $c(TS^2)$. On the other hand, we could have defined $i\boldsymbol{e}_\theta$ to be $-\boldsymbol{e}_\phi$ instead of $\boldsymbol{e}_\phi$, resulting in the conjugation of the complex coordinate and reversing the sign of the Chern number.

More generally, consider a plane bundle $M$ over a closed surface $\mathcal{B}$. For any smooth vector field $\boldsymbol{m}$ over $\mathcal{B}$, the total solid angle $\Omega(m(\mathcal{B}))$ must be an integer multiple of $4\pi$, since the image of a closed surface $m(\mathcal{B})$ cannot cover a fraction of the sphere. It follows that the Chern number of a plane bundle (1.56) over a closed surface is even. For the case of the tangent bundle of a closed orientable surface, we will compute this number in Sec. 4.3.4.

▷ **Monopole bundle and topological quantization.** For the eigenstate bundle $D^+$ over the Bloch sphere described in Sec. 3.3.3, one has

$$c(D^+) = \frac{1}{2\pi} \int_{S^2} \left(-\frac{1}{2}\right) d\Omega = -1. \tag{4.15}$$

One can also obtain this in terms of singularities:

**Exercise 4.1.** The section $\psi_+$ defined by Eq. (3.43) has a single singularity at the south pole. Note that the section $e^{-i\varphi}\psi_+$ is smooth at this point. Find the index of the singularity of $\psi_+$.

Recall from Sec. 3.3.5 that in terms of the electromagnetic field, the bundle $D_3^+$ over $\mathbb{R}^3 \setminus \{0\}$ describes the magnetic field configuration of the Dirac monopole placed at the origin. For any closed surface $\mathcal{B} \subset \mathbb{R}^3$ that encloses the origin, the magnetic flux through $\mathcal{B}$ equals $-\Phi_0$, and we have $c(D_3^+|_{\mathcal{B}}) = -1$. Thus, a wave function $\psi$ of an electron in the vicinity of the monopole has a singularity on $\mathcal{B}$. By smoothness of $\psi$, these singularities form a line of zeroes of the wave function, which starts at the monopole. The same is true for the basis section **1**. This leads to the line of singularities of the vector potential, known as the **Dirac string**. Topology also puts constraints on the possible values of the monopole charge. In terms of the magnetic field, Eq. (4.10) tells that the magnetic flux through any closed surface is quantized in the units of $\Phi_0$ (see Sec. 3.3.5). In other words, any magnetic monopole must have magnetic charge $n \frac{\hbar}{2e}$, where $n \in \mathbb{Z}$. This is known as the **Dirac quantization condition**, which was derived in Ref. [48].

▷ **Gaussian curvature of the torus.** Now consider the tangent bundle of the torus $T^2$. We already know that this bundle is trivial, so its Chern number is zero. Thus, the integral of curvature of any connection must vanish:

$$\int_{T^2} f_{12} dx_2 dx_2 = 0 . \tag{4.16}$$

In particular, we have

$$\int_{T^2} \kappa dS = 0 , \tag{4.17}$$

for the Gaussian curvature (see Eq. (2.45)). Moreover, this will be true for the tangent bundle of any closed surface obtained from the torus by a smooth deformation. Thus, whenever a surface has a single hole, the integral of the curvature will vanish. Such surfaces are said to be of **genus** $g = 1$ (for the sphere $S^2$, $g = 0$). All closed orientable two-dimensional surfaces are classified by the genus, up to a smooth deformation. From the results for the sphere and the torus, we have:

$$\frac{1}{2\pi} \int_{\mathcal{B}_0} \kappa dS = 2 , \qquad \frac{1}{2\pi} \int_{\mathcal{B}_1} \kappa dS = 0 , \tag{4.18}$$

where $\mathcal{B}_g$ denotes a genus $g$ surface.

## 4.3 Pullback construction and topology

Consider the tangent bundle $T\mathcal{B}_g$ of a genus $g$ surface as a complex line bundle. In this section, we will derive a formula for the Chern number $c(T\mathcal{B}_g)$, generalizing results of Eq. (4.18). Here, finding the Chern number directly by application of either side of Eq. (4.10) would be a tedious task. Fortunately, there is a way to do this without any computation. Recall that the real plane bundle (1.56) and the eigenspace bundle for two-level quantum system (3.49) can be defined by a vector field. In both cases, the vector field over the base space $\mathcal{B}$ gives rise to the map from $\mathcal{B}$ to the sphere $S^2$, according to (2.32). Below, we consider a general way to define a vector bundle $V$ over $\mathcal{B}$ given a map from $\mathcal{B}$ to the base space of another bundle $E$. Then we will discuss how the properties of this map and the topology of $E$ determine the topology of the new bundle $V$.

### 4.3.1 Bundles induced by a map

Let $m : \mathcal{B} \to \mathcal{S}$ be a smooth map between surfaces, and let $E$ be a vector bundle over $\mathcal{S}$. We wish to define from this data a vector bundle over the surface $\mathcal{B}$. This can be done as follows:

at each point $p \in \mathcal{B}$, define the fiber of the new bundle $m^*E$ as

$$(m^*E)_p = E_{m(p)} \,. \tag{4.19}$$

In this way, a vector space is associated with every point of $\mathcal{B}$. The bundle $m^*E$ is called **pullback** bundle, or **induced** bundle. In a similar way one defines a **pullback section** $m^*\mathbf{1}$ of $m^*E$:

$$(m^*\mathbf{1})(p) = \mathbf{1}(m(p)) \,, \tag{4.20}$$

where $\mathbf{1}$ is some section of $E$. Fibers of $m^*E$ at different points are assumed to be independent,[9] so that there are sections of the pullback bundle that are not pullback sections.

As a physical example, consider an eigenspace bundle $V^+$ of a two-level Hamiltonian defined over the parameter space $R$. At a point $r \in R$, the direction of the vector $\boldsymbol{h}(r)$ associated with $\hat{H}(r)$ is described by two angles, $\theta(r)$ and $\varphi(r)$. It is then natural to define a basis section of $V^+$ as $\psi_+(\theta(r), \varphi(r))$, where $\psi_+$ is given by Eq. (3.43). This is nothing else but the pullback section $h^*\psi_+$, where $h : R \to S^2$ is the map associated with the Hamiltonian.

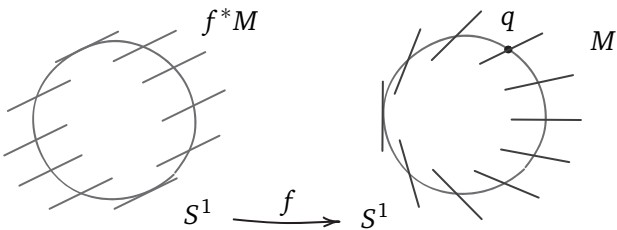

Figure 4.4: Pullback $f^*M$ of the Möbius bundle under the map $f : S^1 \to S^1$ defined by $f(p) = q$ for all $p \in S^1$.

As a mathematical example, consider the pullback of the Möbius bundle $M$ under a constant map from the circle $S^1$ to itself, shown in Fig. 4.4. The map sends all points of $S^1$ to the point $q$. According to the definition (4.19), the fibers of the pullback bundle $f^*M$ are the copies of the fiber $M_q$. If we choose a vector $\boldsymbol{v} \in M_q$, it will determine a global non-vanishing pullback section of $f^*M$, so this bundle is trivial. This example illustrates that a vector bundle can change its topological properties under pullback. Below, we will focus on such changes in topology of complex line bundles.

### 4.3.2 Curvature of pullback bundle and degree of a map

Recall that in Sec. 2.2.3 we computed the curvature of the real plane bundle $M$. The bundle was described by a vector field of the plane normals $\boldsymbol{m}$, or equivalently by the map $m : \mathcal{B} \to S^2$. To find the curvature, we used the identification of the vector spaces $M_p$ and $T_pS^2$ as subspaces of $\mathbb{R}^3$. The pullback bundle is a more abstract version of the same construction, with the map $m$ playing now a central role. The whole argument can be rephrased in terms of pullbacks, leading to the same expression for the curvature. Let $f_{\alpha\beta}$ be the curvature component of a bundle $E$ over the $\mathcal{S}$ in coordinates $(x_\alpha, x_\beta)$. Then the curvature of the pullback bundle over $\mathcal{B}$ is

$$(m^*f)_{12} = J f_{\alpha\beta} \,, \tag{4.21}$$

where $J$ is the Jacobian determinant for the map $f : \mathcal{B} \to \mathcal{S}$ expressed in coordinates.

---

[9]See Ref. [6] for the precise description of the pullback construction.

We are interested in topology of induced bundles, and from now on we assume that surfaces $\mathcal{B}$ and $\mathcal{S}$ are closed. Similarly to Eq. (2.37), we obtain for the Chern number of $m^*E$:

$$c(m^*E) = \frac{1}{2\pi} \int_{m(\mathcal{B})} f_{\alpha\beta} dx_\alpha dx_\beta = c(E)\deg(m), \tag{4.22}$$

where **degree of a map** $\deg(m)$ is an integer that shows how many times the image of $\mathcal{B}$ under $m$ covers the surface $\mathcal{S}$. The integrality of the degree follows from the fact that continuous maps preserve boundaries. Since $\partial\mathcal{B} = \varnothing$, the image $m(\mathcal{B})$ also does not have a boundary and covers all of $\mathcal{S}$, perhaps, several times. Let us consider some examples.

▷ **Pullbacks of $TS^2$ under reflection and inversion.** Let $\sigma_x : S^2 \to S^2$ be the reflection in the $yz$ plane (the sphere is centered at the origin). This map preserves areas of regions on the sphere but reverses orientation of contours. Thus the curvature of the tangent bundle $TS^2$ changes its sign under pullback by $\sigma_x$, and we have[10]

$$c(\sigma_x^*TS^2) = -2. \tag{4.23}$$

In a similar fashion, the inversion map $\mathcal{I}$ sending $\boldsymbol{r} \to -\boldsymbol{r}$ gives

$$c(\mathcal{I}^*TS^2) = -2. \tag{4.24}$$

▷ **Pullback of $D^+$ over the Bloch sphere under inversion.** By the same token as above, the Chern number is

$$c(\mathcal{I}^*D^+) = -c(D^+) = 1. \tag{4.25}$$

This bundle has a simple physical interpretation. According to Eq. (4.19), we attach a vector space defined by $|\psi_{\boldsymbol{h}+}\rangle$ to the point $-\boldsymbol{h}$. Note that

$$\hat{H}_{-\boldsymbol{h}}|\psi_{\boldsymbol{h}+}\rangle = -\hat{H}_{\boldsymbol{h}}|\psi_{\boldsymbol{h}+}\rangle = -|\psi_{\boldsymbol{h}+}\rangle, \tag{4.26}$$

that is, the positive-eigenvalue state of $\hat{H}_{\boldsymbol{h}}$ at the point $\boldsymbol{h}$ coincides with the negative-eigenvalue state of $\hat{H}_{-\boldsymbol{h}}$ at the opposite point of the Bloch sphere. It follows that

$$\mathcal{I}^*D^+ = D^-, \tag{4.27}$$

where $D^-$ is the bundle of low energy eigenspaces (compare with the results of Exercise 3.2.).

At each point $\boldsymbol{h}$ of the Bloch sphere, the eigenvectors $|\psi_{\boldsymbol{h}+}\rangle$ and $|\psi_{\boldsymbol{h}-}\rangle$ are orthogonal. The direct sum of the corresponding eigenspaces is the whole Hilbert space:

$$V_{\boldsymbol{h}}^+ \oplus V_{\boldsymbol{h}}^- = \mathcal{H}. \tag{4.28}$$

Globally, this gives the direct sum of vector bundles:

$$D^+ \oplus D^- = S^2 \times \mathcal{H}. \tag{4.29}$$

Thus, two eigenstate bundles form together a trivial bundle with fiber $\mathcal{H}$, which can be identified with $\mathbb{C}^2$ once the basis $\{|\uparrow\rangle, |\downarrow\rangle\}$ is chosen. For the Chern numbers, we have

$$c(D^+) + c(D^-) = 0. \tag{4.30}$$

---

[10]The bundle $\sigma_x^*TS^2$ may be useful in the context of Exercise 3.1.

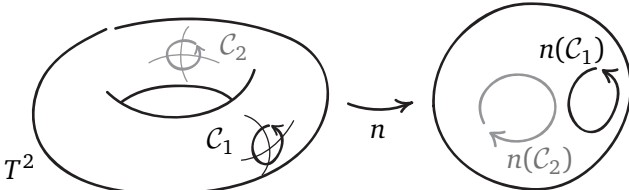

Figure 4.5: The transformation of orientation of contours under the normal map indicates that the tangent bundle of the torus has curvatures of opposite signs at the outer side (contour $\mathcal{C}_1$) and at the inner side (contour $\mathcal{C}_2$).

▷ **The tangent bundle of the torus.** This bundle can be thought of as a pullback of $TS^2$ under the normal map $n : T^2 \to S^2$. We already know that the integral of the curvature vanishes. Let us look at the curvature at some points of the torus. Choose two positively-oriented contours, as shown in Fig. 4.5. The contour $\mathcal{C}_1$ on the outer side of the torus is mapped to a positively-oriented contour on the sphere, while the orientation of $\mathcal{C}_2$ is reversed. Thus the curvature has opposite signs on the outer side and on the inner side of the hole. Heuristically, the image of the outer side covers the sphere once, while the image of the inner part is "turned inside out", so the degree of the map is zero.

If the torus has a perfect round shape, there are two circles that divide regions with opposite signs of curvature. At these circles, the Jacobian vanishes. This happens because the map is degenerate, since the circles are mapped to the north and south poles of the sphere (if the torus lies on the horizontal plane).

### 4.3.3 Indices of singularities in pullback section

Now let us see how the index of a singularity of a section of $E$ is transformed under the pullback. We return to the general setting of Sec. 4.3.1. Suppose that a section $s$ of $E$ has a singularity at $q \in \mathcal{S}$ with index ind$[s(q)]$. Let $p \in \mathcal{B}$ be some **pre-image** of $q$ under $m$, that is, a point that satisfies $m(p) = q$. We also demand that the map $m$ is invertible near $p$, so that Jacobian $J(p) \neq 0$. Then at $p$ we will have a singularity of the pullback section with index ind$[m^*s(p)]$.

Choose a contour $\mathcal{C}$ near $p$ and denote its image near $q$ by $m(\mathcal{C})$. We choose the orientation of $\mathcal{C}$ in such a way that $m(\mathcal{C})$ is positively-oriented in $\mathcal{S}$. Then, if $J(p) > 0$, the map preserves orientation of the surface at $p$, and the two indices will be the same. In the case of $J(p) < 0$, the contour $\mathcal{C}$ has negative orientation with respect to the surface, and the index of the singularity changes its sign. Thus

$$\text{ind}[m^*s(p)] = \text{sign}[J(p)]\,\text{ind}[s(q)]. \tag{4.31}$$

**Exercise 4.2.** To show this formally, define positively-oriented coordinates $\varphi$ for $\mathcal{C}$ and $\gamma$ for $m(\mathcal{C})$. The map $m$ at the contour is given by the function $\gamma(\varphi)$. Introduce a smooth section $v$ of $E$ near $q$, so that $s = e^{i\beta(\gamma)}v$ on $m(\mathcal{C})$. Then pull both sections back to $\mathcal{C}$ and find the index of $m^*s$ at $p$.

Now we can relate the Chern number of the pullback bundle $c(m^*E)$ with $c(E)$, as follows. Choose a point $q \in \mathcal{B}$ and consider the set of all of its pre-images $\{p_i\}$, which we denote by $m^{-1}(q)$. Suppose that the determinant $J \neq 0$ in some neighborhood of each pre-image $p_i$. It is a safe assumption, since generically a smooth real function of two variables takes zero value along some one-dimensional curves. If some of the pre-images lie on the curve where $J = 0$,

one can move the point $q$ or deform the map $m$ until $J \neq 0$ for all pre-images. Let $\mathbf{1}$ be a section of $E$ that has a single singularity at $q$ with index $\text{ind}[\mathbf{1}(q)] = c(E)$.

Then the pullback section $m^*\mathbf{1}$ will have singularities near $p_i$, and there will be no other singularities, since pullback preserves smoothness of a section by construction. Thus, for the sum of indices of the singularities, we have:

$$c(m^*E) = c(E) \sum_{p_i \in m^{-1}(q)} \text{sign}[J(p_i)]. \tag{4.32}$$

Besides the Chern number of the pullback bundle, this formula provides a useful expression for the degree of a map as a discrete sum (cf. Eq. (4.22)):

$$\deg(m) = \sum_{p_i \in m^{-1}(q)} \text{sign}[J(p_i)]. \tag{4.33}$$

We will use this result to compute the Chern number of an eigenspace bundle associated with a Bloch Hamiltonian in a crystal in Sec. 8.2.

**Exercise 4.3.** Plot the singularities of $\sigma_x^* \mathbf{1}_N$ and $\mathcal{I}^* \mathbf{1}_N$, where $\mathbf{1}_N$ is a section of $TS^2$ with a single singularity at the south pole. Convince yourself that in each case, the singularity of the pullback section has the index given by the Chern number.

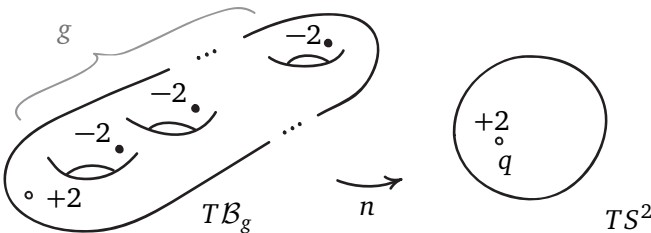

Figure 4.6: Counting singularities of pullback section of the tangent bundle $T\mathcal{B}_g$ of the genus $g$ surface $\mathcal{B}_g$. The section of $TS^2$ has a singularity with index $+2$ at the point $q$. The signs of singularities in the pullback section are determined by the signs of the curvature.

### 4.3.4 Gauss–Bonnet theorem

Consider the tangent bundle of the genus $g$ surface $T\mathcal{B}_g$ as a complex line bundle. This bundle can be thought of as a pullback of the tangent bundle of the sphere

$$T\mathcal{B}_g = n^* TS^2, \tag{4.34}$$

under the normal map $n : \mathcal{B}_g \to S^2$. Let us compute the Chern number $c(T\mathcal{B}_g)$ using the formula (4.32).

On the sphere, we introduce the section $\mathbf{1}_N$ (see Fig. 4.2) and move its sole singularity of index $+2$ to the point $q$, as shown in Fig. 4.6. In the figure, the point $q$ has $g+1$ pre-images on the surface $\mathcal{B}_g$. From discussion of the curvature of the torus, we know that the curvature of the inner side of a hole is negative, so the corresponding $g$ singularities of the pullback section have index $-2$. We also have one pre-image of $q$ in the region with the positive curvature.

Adding all indices together, we have $c(T\mathcal{B}_g) = 2(1-g)$. Taking into account Eqs. (2.45) and (4.10), we conclude that the integral of the Gaussian curvature of $\mathcal{B}_g$ is

$$\frac{1}{2\pi} \int_{\mathcal{B}_g} \kappa \, dS = 2(1-g), \tag{4.35}$$

which is the celebrated **Gauss–Bonnet theorem**. Note that the result does not depend on the particular shape of $\mathcal{B}_g$ used in the calculation because of the topological stability of the Chern number.

## 4.4 Topological classification of two-band Hamiltonians

As our final topological example, we consider the classification of non-degenerate two-band Hamiltonians $\hat{H}(x_1, x_2)$ which depend on two periodic parameters:

$$\hat{H}(x_1 + 2\pi, x_2) = \hat{H}(x_1, x_2), \quad \hat{H}(x_1, x_2 + 2\pi) = \hat{H}(x_1, x_2). \tag{4.36}$$

We identify the points in the parameter space that have the same Hamiltonians, which turns the parameter space $R$ into a torus $T^2$.

### 4.4.1 Classifications in general

First, we introduce the language of equivalence classes, which will help us to discuss classifications of topological matter in sections 7–9. Any classification is based on a certain equivalence relation, in the following sense. A **relation** $\sim$ on a set $A$ is a rule that tells us whether two elements $a, a' \in A$ are related. If this is the case, one writes $a \sim a'$. A relation $\sim$ is called an **equivalence relation**, if for all $a, a', a'' \in A$ it is true that

- $a \sim a$,

- $a \sim a' \quad \Rightarrow \quad a' \sim a$,

- $a \sim a'$ and $a' \sim a'' \quad \Rightarrow \quad a \sim a''$.

Any two elements related by an equivalence relation are said to be equivalent (with respect to the given relation). A set with an equivalence relation becomes a union of non-intersecting subsets called **equivalence classes**. The equivalence class containing an element $a$ is denoted $[a]$. Two elements belong to the same class if and only if they are equivalent:

$$[a] = [b] \quad \Longleftrightarrow \quad a \sim b. \tag{4.37}$$

Note that "classification" of elements of $A$ literally means dividing $A$ into classes; this is exactly what an equivalence relation does.

It is often desirable to describe equivalence classes by the values of a function defined on $A$. First, this function must respect the equivalence relation:

$$a \sim a' \quad \Rightarrow \quad f(a) = f(a'). \tag{4.38}$$

Such a function is constant on the equivalence classes and is an **invariant** associated with $\sim$. However, it is still possible that the values of $f$ do not fully reflect the structure of the classes of $A$, since $f$ can assume the same value at the elements of different classes. To avoid this, we further demand that

$$f(a) = f(a') \quad \Rightarrow \quad a \sim a'. \tag{4.39}$$

If both conditions are satisfied, the set of values of $f$ on the elements of $A$ is in one-to-one correspondence with the set of the equivalence classes. Hence, we can label each equivalence class by the value of $f$ on one of its elements. In this case, $f$ is a **complete invariant** of the classification.

### 4.4.2 Classes of Hamiltonians over a torus

Consider the set of non-degenerate two-band Hamiltonians defined over a torus:

$$\{\hat{H} \text{ over } T^2, \text{ such that } |\boldsymbol{h}(x)| \neq 0 \text{ for all points } x \in T^2\}. \tag{4.40}$$

We introduce the following relation on this set:

$$\hat{H}_0 \sim \hat{H}_1 \quad \Longleftrightarrow \quad \text{there is a smooth, nowhere-degenerate deformation } \hat{H}_0 \rightarrow \hat{H}_1. \tag{4.41}$$

In other words, there is a smooth family of Hamiltonians $\hat{H}_t$ for $t \in [0;1]$, such that each $\hat{H}_t$ is non-degenerate and for $t = 0, 1$, the Hamiltonian coincides with the given $\hat{H}_0$ and $\hat{H}_1$, respectively. One immediately checks that $\sim$ is an equivalence relation. Our goal is to describe the corresponding equivalence classes of the two-band Hamiltonians.

Recall from Sec. 3.3.4 that each non-degenerate Hamiltonian $\hat{H}(x_1, x_2)$ determines a complex line bundle of low-energy eigenstates $V^-$ over the parameter space. As discussed in Sec. 4.2.3, the Chern number of the bundle is invariant under smooth deformations. It follows that

$$\hat{H}_0 \sim \hat{H}_1 \quad \Rightarrow \quad c(V_0^-) = c(V_1^-), \tag{4.42}$$

so the Chern number of the eigenstate bundle is an invariant of the classification.

But is it a *complete* invariant? Consider two Hamiltonians $\hat{H}_0, \hat{H}_1$ such that $c(V_0^-) = c(V_1^-)$. We need to determine if there exists a smooth deformation connecting the two Hamiltonians. First, consider the terms proportional to the identity matrix. We deform these terms to zero in both Hamiltonians. We further deform the remaining terms in such a way that the vector fields associated to Hamiltonians by (3.49) become unit vector fields, $|\boldsymbol{h}| = 1$. According to (2.32), a unit vector field $\boldsymbol{h}$ on the torus $T^2$ carries the same information as a map $h : T^2 \rightarrow S^2$ to the unit sphere. The bundle $V^-$ can be thought of as the pullback of the monopole bundle under this map:

$$V^- = h^*(D^-). \tag{4.43}$$

Applying Eq. (4.22), we find that

$$c(V^-) = c(D^-)\deg(h) = \deg(h). \tag{4.44}$$

Thus, the maps associated with the two Hamiltonians have the same degree:

$$\deg(h_0) = \deg(h_1). \tag{4.45}$$

Our problem is reduced to the question whether two maps $h_i : T^2 \rightarrow S^2$ can be deformed into each other, given that they have the same degree. The answer is positive and is given by the **Hopf theorem** in homotopy theory [49]. We conclude that the Chern number $c(V^-) \in \mathbb{Z}$ of the eigenspace bundle defined by a Hamiltonian $\hat{H}$ is a complete invariant of the classification based on the relation (4.41). Thus, the equivalence classes of Hamiltonians can be labeled by integers. This result will be useful in the context of charge pumps (Sec. 7.3) and Chern insulators (Sec. 8.2).

Note the similarity of this situation with the Gauss-Bonnet theorem: there, we started from the classification of closed orientable two-dimensional surfaces based on their genus $g$. Then the theorem, Eq. (4.35), allowed us to compute the genus of a given surface by measuring and integrating a local geometric quantity, the Gaussian curvature $\kappa$:

$$g = 1 - \frac{1}{4\pi} \int_{\mathcal{B}_g} \kappa dS. \tag{4.46}$$

In the context of two-band Hamiltonians, we have shown that the classification of the Hamiltonians $\hat{H}$ coincides with the classification of eigenspace bundles $V^-$ (of course, the bundle $V^+$ would have worked equally well). The complete invariant here is the Chern number, which can also be found from the local geometry of the bundle:

$$c(V^-) = \frac{1}{2\pi} \int_{T^2} f_{12}^- dx_1 dx_2 \,, \tag{4.47}$$

where $f_{12}^-$ is the Berry curvature.

## 4.5 Summary and outlook

Above, we considered global topological properties of vector bundles, which do not depend on local details, but can be computed from the geometric data. As we will see in later sections, such properties of eigenspace bundles underlie the classification of phases in the topological band theory. Let us summarize our main findings:

- Topology studies continuous maps between topological spaces. One example of such map is a section of a vector bundle.

- An obstruction to finding a global non-vanishing section indicates that the bundle is topologically non-trivial.

- Any section of a non-trivial complex line bundle over a closed two-dimensional surface can have point-like singularities. Each singularity is characterized by an index.

- The sum of indices of singularities of a section is the Chern number $c(M)$ of the complex line bundle $M$. It does not depend on the choice of the section. The Chern number vanishes if and only if the complex line bundle is trivial.

- If there is a connection on the bundle, one can compute the Chern number by integrating the curvature of connection. The result does not depend on the choice of connection.

- Pullback construction allows one to define a bundle over $\mathcal{B}$ given a map $f : \mathcal{B} \to \mathcal{S}$ to the base space $\mathcal{S}$ of another bundle.

- Gauss-Bonnet theorem relates the genus of a closed orientable two-dimensional surface with the integral of the Gaussian curvature.

- Chern number of the eigenspace bundle is a complete invariant of classification of non-degenerate two-band Hamiltonians defined over a torus.

This section concludes our brief journey into the geometry and topology of vector bundles. We focused only on the simplest cases, which will be relevant to the future topics and can be discussed without invoking abstract machinery. Below, we give directions towards deeper mathematical discussion. Note that some of the sources require undergraduate background in algebra, topology, or differential geometry.

▷ **Flavors of topology.**   The definitions of topological space, continuous map, and of basic topological properties, such as connectedness and compactness, belong to the field of **point-set** topology. A motivated introduction to these ideas can be found in Ref. [50]. **Algebraic** topology aims to encode some information about topological spaces in the form of algebraic structures. In other words, one defines a "function", which takes as an input a topological space and produces, for example, a group. Importantly, this function preserves maps: a continuous

map between topological spaces turns into a group homomorphism. A physicist-oriented exposition of the homotopy and (co)homology groups is given in Refs. [3, 5]. A comprehensive mathematical introduction to the subject can be found in a textbook [51]. Another kind of topology relevant to our discussion is the **differential** topology, which relates global topological properties with the local ones studied by differential geometry [49, 52]. In fact, we use this latter setting, assuming smoothness of maps rather than just continuity.

▷ **Chern numbers in higher dimensions.** We were mainly concerned with the topology of complex line bundles over two-dimensional surfaces, which is the simplest case of interest. Topological properties can be defined for vector bundles, which have the base space or the fiber of higher dimension. For example, **Chern–Weil** construction produces a family of Chern numbers given the curvature of some connection on the bundle $M$. In particular, a complex bundle over a two-dimensional base space $\Sigma$ is characterized by the first Chern number

$$c_1 = \frac{1}{2\pi} \int_{\Sigma} \mathrm{tr}(F_{12}) dx_1 dx_2 , \tag{4.48}$$

where $F_{12}$ is the curvature matrix, such as one defined in Eq. (3.58) for an eigenstate bundle. In the case of complex *line* bundles, this reduces to the integral of curvature in Eq. (4.10). For base spaces of higher dimensions, there are higher Chern numbers $c_n$, which can be nonzero over $2n$-dimensional base space. For a discussion of the Chern–Weil construction, see Refs. [3, 5].

▷ **Classifying spaces.** Pullback construction plays an immensely important role in the classification problem of vector bundles. The classification is based on an appropriate notion of equivalence, known as isomorphism of vector bundles. There is an object, called **classifying space** $B$, which is the base space of a **universal bundle** $E$. The classifying space has the property that any vector bundle over the space $\Sigma$ is isomorphic to the pullback $f^*E$ for some map $f : \Sigma \rightarrow B$. Moreover, two vector bundles are isomorphic if and only if two corresponding maps are homotopic, that is, can be continuously deformed one into another. In this way, the study of isomorphism classes of vector bundles amounts to the study of homotopy classes of maps (if the structure of the classifying space is known). This is the key idea behind the theory of characteristic classes associated to vector bundles [7].

In the discussion above, we encountered elementary prototypes of this situation. For complex line bundles, the Bloch sphere $S^2$ with the monopole bundle played the role of such "classifying space". In Sec. 4.4.2, we replaced the equivalence of eigenstate bundles with the equivalence of maps to the sphere $S^2$. In a similar way, two equivalence classes of real line bundles over the circle can be described in terms of maps to the circle carrying the Möbius band bundle (see Fig. 4.4). Each map $f : S^1 \rightarrow S^1$ is characterized by the winding number; the parity of this number determines whether the pullback will be isomorphic to the trivial bundle or to the Möbius band.

▷ **Tautological line bundles.** It turns out that the monopole bundle over the Bloch sphere and the Möbius band bundle over a circle are closely related in a certain geometric sense. To see this, we need to make a detour and to introduce the idea of projective space. Consider a complex two-dimensional vector space $\mathbb{C}^2$, which consists of pairs $(z_1, z_2)$ of complex numbers. Each pair, except $(0, 0)$, defines a one-dimensional complex subspace

$$\{(\lambda z_1, \lambda z_2) \mid \lambda \in \mathbb{C}\} \subset \mathbb{C}^2 . \tag{4.49}$$

We wish to describe the set of all such subspaces of $\mathbb{C}^2$, which is called the **complex projective line** $\mathbb{C}P^1$. One can say that each point of $\mathbb{C}P^1$ represents a subspace of $\mathbb{C}^2$. Note that the

subspace (4.49) can be labeled by a single complex number

$$u = \frac{z_2}{z_1} = \frac{\lambda z_2}{\lambda z_1}, \tag{4.50}$$

provided that $z_1 \neq 0$. So, the number $u$ is a complex coordinate for $\mathbb{C}P^1$, which is defined everywhere except one point that represents the subspace (4.49) with $z_1 = 0$. In a similar way, one defines another complex coordinate $v = \frac{z_1}{z_2}$ for all subspaces with $z_2 \neq 0$. Thus, one can cover $\mathbb{C}P^1$ with two coordinate systems, which are related on the overlap by $u = \frac{1}{v}$.

These coordinates allow one to identify $\mathbb{C}P^1$ with a sphere $S^2$, which can be parameterized by complex numbers in a similar fashion by using the stereographic projection (here, we follow Ref. [3]). Consider a sphere $S^2$ of radius $r = \frac{1}{2}$ together with the tangent planes at the north and south poles. To each point $p \in S^2$, we associate points $u$ and $v$ in the tangent planes, as shown in Fig. 4.7 on the left. A point in each plane is obtained by a stereographic projection from the opposite pole of the sphere. Now let us interpret the points in the planes as complex numbers, $u = u_x + iu_y$ and $v = v_x + iv_y$, with axes shown in the figure. It follows from the elementary geometry that the complex numbers associated with the point $p$ are

$$u = e^{i\varphi} \tan \frac{\theta}{2}, \qquad v = e^{-i\varphi} \cot \frac{\theta}{2}, \tag{4.51}$$

where $(\theta, \varphi)$ are the coordinates of the point $p$. In this way, any point $p \in S^2$ away from the south pole (resp. north pole) is now described by a complex coordinate $u$ (resp. $v$), which are related by $u = \frac{1}{v}$ away from the poles. This makes the sphere a complex one-dimensional manifold called **Riemann sphere**. Because of the identical coordinate descriptions, we can visualize the complex projective line $\mathbb{C}P^1$ as the sphere $S^2$.

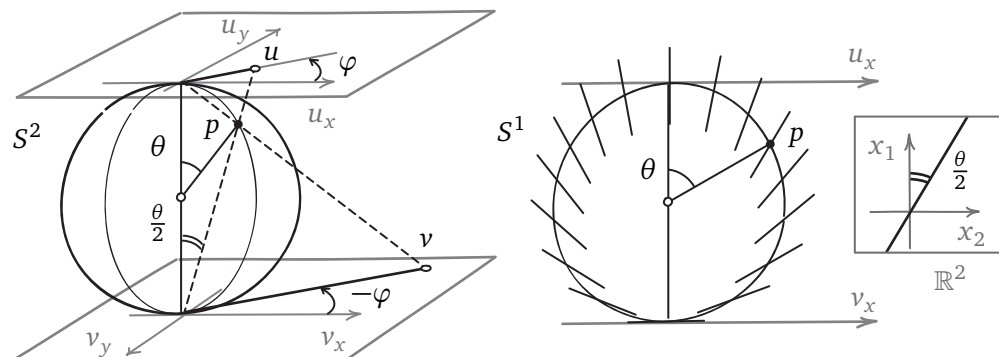

Figure 4.7: LEFT: Visualization of the complex projective line $\mathbb{C}P^1$ as a sphere $S^2$. Points $u$ and $v$ in the tangent planes are obtained from the point $p \in S^2$ by stereographic projection from the poles (dashed lines). RIGHT: Tautological line bundle over the real projective line $\mathbb{R}P^1$ visualized as a circle $S^1$. Inset shows the original vector space $\mathbb{R}^2$ with a subspace represented by the point $p \in S^1$.

Projective spaces naturally serve as base spaces for **tautological bundles**. Each point of a projective space represents a vector space; let us attach to the point this very space (hence the term "tautological"). Consider such tautological complex line bundle over $\mathbb{C}P^1$ visualized as a sphere $S^2$. Which subspace of $\mathbb{C}^2$ is associated to the point $p \in S^2$ with coordinates $(\theta, \varphi)$?

Its complex coordinate $u$ reads

$$u = \frac{e^{i\varphi} \sin \frac{\theta}{2}}{\cos \frac{\theta}{2}} = \frac{z_2}{z_1}. \tag{4.52}$$

It follows that the subspace in question contains the vector $(\cos \frac{\theta}{2}, e^{i\varphi} \sin \frac{\theta}{2})$, which is nothing else but the value of the section $\psi_+$ of the eigenstate bundle $D^+$, as given by Eq. (3.43)! We conclude that the bundle $D^+$ can be defined by a purely geometric construction, without any reference to the Hamiltonian of a two-state system.

Now let us consider a *real* projective line $\mathbb{R}P^1$, which is a collection of all one-dimensional subspaces of $\mathbb{R}^2$. To this end, we simply discard the imaginary part of all numbers in the discussion above. The pair $(z_1, z_2)$ becomes a pair of real numbers $(x_1, x_2)$. The space $\mathbb{R}P^1$ can be identified with a circle $S^1$. It is covered by two coordinate patches $u_x$ and $v_x$, which are defined by the stereographic projection to the tangent lines. In this case, the tautological line bundle can be readily visualized by attaching to each point of the circle the subspace of $\mathbb{R}^2$ that contains the vector $(\cos \frac{\theta}{2}, \sin \frac{\theta}{2})$. The bundle has the shape of the Möbius band, as shown in the right panel of Fig. 4.7. Thus, the Möbius band sits inside the monopole bundle $D^+$. Both bundles arise as tautological bundles over projective lines. In this sense, the bundle $D^+$ is a complex version of the Möbius band. For a mathematical discussion of the geometric phase theory understood as a result of such "informal complexification", see Ref. [53].

# 5 Tight-binding models and Bloch theory

Tight-binding approximation allows one to construct a simple quantum mechanical model of a crystal. One starts with a periodic array of atomic orbitals, which serve as a basis for the Hilbert space of states of an electron. The Hamiltonian operator is described by a matrix whose elements are interpreted as on-site potentials and amplitudes of hopping between the orbitals. Hamiltonian diagonalization gives the energy spectrum and the corresponding eigenstates. Then these states are populated according to the Pauli exclusion principle. Tight-binding models give a qualitative description of many properties of a crystal, including those related to the geometry and topology of the eigenspace bundle. These properties will be our main subject in later sections.

In this section, we introduce the formalism of tight-binding approximation, discuss the representation of crystal symmetries, and consider a model of graphene as an example. Here, we focus on technical details, which will provide the ground for further physical discussion. For general information about the tight-binding method and its limits of applicability, see, for example, Ref. [54].

## 5.1 Momentum space

### 5.1.1 Dimer as a two-level system

To begin with, we consider a finite system, a molecule with two orbitals, $a$ and $b$. We denote the corresponding electronic states as $|a\rangle$ and $|b\rangle$. The Hamiltonian takes the form

$$\hat{H} = U_a |a\rangle\langle a| + U_b |b\rangle\langle b| + t|b\rangle\langle a| + \bar{t}|a\rangle\langle b|, \tag{5.1}$$

where $U_\alpha$ with $\alpha = a, b$ are on-site potentials, and $t$ is the hopping amplitude from $a$ to $b$ site. Hermiticity requires that the hopping amplitude in the opposite direction be the complex conjugate of $t$. In the basis $\{|a\rangle, |b\rangle\}$, the Hamiltonian matrix reads

$$H = \begin{pmatrix} \langle a|\hat{H}|a\rangle & \langle a|\hat{H}|b\rangle \\ \langle b|\hat{H}|a\rangle & \langle b|\hat{H}|b\rangle \end{pmatrix} = \begin{pmatrix} U_a & \bar{t} \\ t & U_b \end{pmatrix}. \tag{5.2}$$

For simplicity, let us assume that there is no overall shift of energy levels, $U_a = -U_b = \Delta$, and that the hopping amplitude is real, $t \in \mathbb{R}$. Then the Hamiltonian matrix is expressed in terms of the Pauli matrices as

$$H = \sigma_x t + \sigma_z \Delta. \tag{5.3}$$

### 5.1.2 Diatomic chain in real space

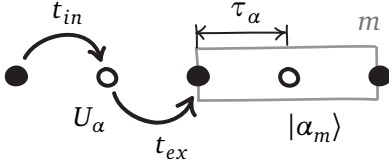

Figure 5.1: Tight-binding model of a diatomic chain. Here, $t_{in}$ and $t_{ex}$ are internal and external hopping amplitudes, the index $\alpha = a, b$ enumerates orbitals inside each unit cell, $U_\alpha$ stands for on-site potential, $m$ is the index of a unit cell, $\tau_\alpha$ is the distance between the origin of the unit cell and the orbital $\alpha$. Tight-binding orbitals are denoted $|\alpha_m\rangle$.

Now we construct an infinite periodic crystal from dimers as unit cells, as shown in Fig. 5.1. Besides the internal hopping $t_{in}$ inside each unit cell, we add external hopping $t_{ex}$ between neighboring atoms of two adjacent cells. The Hamiltonian operator is

$$\hat{H} = \sum_{\alpha m} U_\alpha |\alpha_m\rangle\langle\alpha_m| + \sum_m \left[ t_{in}|b_m\rangle\langle a_m| + t_{ex}|a_{m+1}\rangle\langle b_m| + h.c. \right], \tag{5.4}$$

where $m$ is the unit cell index and $\alpha = a, b$ enumerates orbitals. There are $N$ unit cells, and we impose periodic boundary conditions, $|\alpha_m\rangle = |\alpha_{m+N}\rangle$, so the system is invariant under translation by the lattice constant. For simplicity, we consider only nearest-neighbor hoppings; in general, the Hamiltonian can include longer hopping amplitudes. On the other hand, this basic model already has a lot of potential. Later, we will use it as a starting point in deriving model Hamiltonians for an adiabatic charge pump, a Chern insulator, and a Weyl semimetal.

In the basis $\{|\alpha_m\rangle\}$ the Hamiltonian is represented by $2N \times 2N$ matrix that describes $N$ coupled dimers. Generally, it has the form:

$$\hat{H} = \sum_{\substack{\alpha\beta \\ mn}} |\alpha_{m+n}\rangle H_n^{\alpha\beta} \langle\beta_m|. \tag{5.5}$$

Here, the matrix element $H_n^{\alpha\beta}$ gives the amplitude $|\beta_m\rangle \rightarrow |\alpha_{m+n}\rangle$. Note that this amplitude does not depend on $m$ because of the periodicity.

**Exercise 5.1.** Find the matrices $H_n$ defined by Eq. (5.5) that describe the Hamiltonian (5.4) for $n = -1, 0, 1$. [§5.1.3]

### 5.1.3 Bloch waves and momentum space

The problem of diagonalizing the Hamiltonian can be greatly simplified by introducing a new basis that respects the discrete translational symmetry of the crystal lattice. It is the **Bloch wave basis**:

$$|\alpha_k\rangle = \frac{1}{\sqrt{N}} \sum_m e^{imk} |\alpha_m\rangle, \tag{5.6}$$

where we set the lattice constant to unity. One checks that $|\alpha_k\rangle$ is indeed an eigenstate of the translation operator acting on the orbitals as $\hat{T}_1|\alpha_m\rangle = |\alpha_{m+1}\rangle$. The periodicity of the Hamiltonian (5.5) implies that $[\hat{H}, \hat{T}_1] = 0$. Thus, we can find the simultaneous eigenstates of both operators and label them by the **crystal momentum** $k$. The inverse Fourier transform gives the expression of the real-space orbitals in terms of Bloch waves:

$$|\alpha_m\rangle = \frac{1}{\sqrt{N}} \sum_m e^{-imk} |\alpha_k\rangle. \tag{5.7}$$

Periodic boundary conditions demand that $e^{ikN} = 1$, so the crystal momentum $k$ takes discrete values in the interval $[0, 2\pi)$, called the **Brillouin zone**, with an increment $\Delta k = \frac{2\pi}{N}$.

The states $|\alpha_k\rangle$ form an orthonormal basis, which follows from $\langle \alpha_m | \beta_n \rangle = \delta_{\alpha\beta}\delta_{mn}$ and the delta function identity:

$$\frac{1}{N} \sum_m e^{imk} = \delta_{k,0}, \tag{5.8}$$

where

$$\delta_{p,q} = \begin{cases} 1, & p = q \\ 0, & p \neq q \end{cases}, \tag{5.9}$$

is the Kronecker delta.

We substitute the expression for $|\alpha_m\rangle$ as an inverse transform of $|\alpha_k\rangle$ into Eq. (5.5) and obtain

$$\hat{H} = \sum_{\alpha\beta k} |\alpha_k\rangle \left[ \sum_n e^{-ink} H_n^{\alpha\beta} \right] \langle \beta_k | \equiv \sum_{\alpha\beta k} |\alpha_k\rangle H_k^{\alpha\beta} \langle \beta_k | = \sum_k \hat{H}_k, \tag{5.10}$$

where $\hat{H}_k$ is the **Bloch Hamiltonian**. Note that, in contrast with Eq. (5.5), here we do not have any coupling between states with different $k$. Thus, in the Bloch wave basis the Hamiltonian of the diatomic chain describes $N$ *independent* two-level systems parameterized by the crystal momentum $k$. Instead of diagonalizing $2N \times 2N$ matrix, we need to diagonalize $N$ matrices, each of dimension $2 \times 2$. In general, the rank of the matrix $H_k$ is determined by the number of orbitals in the unit cell.

According to Eq. (5.10), the matrix $H_k$ is given by the Fourier transform of $H_n$. For our diatomic chain, the Bloch Hamiltonian matrix reads

$$H_k = \begin{pmatrix} U_a & \overline{t_{in}} + t_{ex} e^{-ik} \\ t_{in} + \overline{t_{ex}} e^{ik} & U_b \end{pmatrix}. \tag{5.11}$$

The reader should check the results of Exercise 5.1 by using the inverse transform

$$H_n = \frac{1}{N} \sum_k e^{ink} H_k. \tag{5.12}$$

Again we assume that the hopping amplitudes are real and there is no overall shift of the on-site potentials, $U_a = -U_b = \Delta$, so the Hamiltonian matrix in the $\{|a_k\rangle, |b_k\rangle\}$ basis is

$$H_k = \sigma_x (t_{in} + t_{ex} \cos k) + \sigma_y t_{ex} \sin k + \sigma_z \Delta = \sum_i h_i \sigma_i. \tag{5.13}$$

The eigenstates are solutions of $\hat{H}_k |\psi_k\rangle = \varepsilon_k |\psi_k\rangle$ and can be thought of as the states of spin-$\frac{1}{2}$ particle in the magnetic field $\boldsymbol{h} = (h_x, h_y, h_z)$. According to Eq. (3.37), the energies of the two bands are

$$\varepsilon_{k\pm} = \pm\sqrt{(t_{in} + t_{ex}\cos k)^2 + (t_{ex}\sin k)^2 + \Delta^2}. \tag{5.14}$$

We will be mostly concerned with low-energy, or valence band eigenstates.

**Exercise 5.2.** Consider two Bloch Hamiltonians (5.13) for the diatomic chain, defined in terms of parameters $(\Delta, t_{in}, t_{ex})$ as

$$H_k^0 : (0, 1, 0) \qquad H_k^1 : (0, 0, 1). \tag{5.15}$$

Find the corresponding eigenstates $|\psi_k\rangle$ for the valence band (use the results of Exercise 3.2). [6.1, 6.2, §9.1.1, §9.1.2, §9.1.3]

### 5.1.4 Another momentum-space basis

So far, Bloch theory has not included the real-space positions of the orbitals. Denote $\tau_\alpha$ the position of the orbital of the type $\alpha$ inside a unit cell. Then the position operator acts on $|\alpha_m\rangle$ as follows:

$$\hat{x}|\alpha_m\rangle = (m + \tau_\alpha)|\alpha_m\rangle. \tag{5.16}$$

With the lattice constant set to unity, the cell coordinate is an integer $m \in \mathbb{Z}$ and the orbital coordinate takes values in the unit interval $\tau_\alpha \in [0, 1)$. Later we will need the basis in the momentum space that is aware of positions of the orbitals:

$$|\widetilde{\alpha_k}\rangle = e^{ik\tau_\alpha}|\alpha_k\rangle. \tag{5.17}$$

The Bloch Hamiltonian eigenstates can be expressed in terms of either basis:

$$|\psi_k\rangle = \sum_\alpha \psi_{\alpha k}|\alpha_k\rangle = \sum_\alpha u_{\alpha k}|\widetilde{\alpha_k}\rangle. \tag{5.18}$$

Note that while the states $|\alpha_k\rangle$ obey

$$|\alpha_{k+2\pi}\rangle = |\alpha_k\rangle, \tag{5.19}$$

the new basis vectors $|\widetilde{\alpha_k}\rangle$ are not periodic in the momentum space because of the additional phase factor $e^{ik\tau_\alpha}$. For our diatomic chain, we place the origin at the orbital of type $a$. The coordinates of the orbitals are $\tau_a = 0$ and $\tau_b = \frac{1}{2}$, which implies that $|\widetilde{b_{k+2\pi}}\rangle = -|\widetilde{b_k}\rangle$. Since the eigenstates $|\psi_k\rangle$ are $k$-periodic, the components $u_{\alpha k}$ are not. We will use the new basis in Sec. 6.2.2.

## 5.2 Symmetry of tight-binding models

In this section, we consider several examples of how symmetry of the crystal affects the form of the tight-binding Hamiltonian both in the real space and in the momentum space.

### 5.2.1 Inversion symmetry

Suppose that the diatomic chain introduced in Sec. 5.1.2 is invariant under inversion symmetry, with the inversion center lying at the center of a unit cell with index $m = 0$. Then the inversion is represented by a linear operator $\hat{\mathcal{I}}$ that maps orbitals in the $m$-th unit cell to those in the cell with index $-m$. The matrix of $\hat{\mathcal{I}}$ is defined as follows:

$$\hat{\mathcal{I}}|\alpha_m\rangle = \sum_\beta |\beta_{-m}\rangle\langle\beta_{-m}|\hat{\mathcal{I}}|\alpha_m\rangle \equiv \sum_\beta |\beta_{-m}\rangle\mathcal{I}^{\beta\alpha}. \tag{5.20}$$

In the sum, the index $\beta$ runs over the orbitals inside a unit cell. The action of $\hat{\mathcal{I}}$ on the Bloch basis states (5.6) reads

$$\hat{\mathcal{I}}|\alpha_k\rangle = \frac{1}{\sqrt{N}}\sum_{m\beta} e^{ikm}|\beta_{-m}\rangle\mathcal{I}^{\beta\alpha} = \sum_\beta |\beta_{-k}\rangle\mathcal{I}^{\beta\alpha}. \tag{5.21}$$

The crystal is inversion-symmetric if the symmetry maps hopping amplitudes and on-site potentials to those of equal strength. In such case, the symmetry operator commutes with the Hamiltonian:

$$\hat{\mathcal{I}}\hat{H} = \hat{H}\hat{\mathcal{I}}. \tag{5.22}$$

To translate this into momentum space, we evaluate each product on a Bloch basis state:

$$\hat{\mathcal{I}}\hat{H}|\alpha_k\rangle = \hat{\mathcal{I}} \sum_\beta |\beta_k\rangle H_k^{\beta\alpha} = \sum_\gamma |\gamma_{-k}\rangle \mathcal{I}^{\gamma\beta} H_k^{\beta\alpha}, \tag{5.23}$$

$$\hat{H}\hat{\mathcal{I}}|\alpha_k\rangle = \hat{H} \sum_\beta |\beta_k\rangle \mathcal{I}^{\beta\alpha} = \sum_\gamma |\gamma_{-k}\rangle H_{-k}^{\gamma\beta} \mathcal{I}^{\beta\alpha}. \tag{5.24}$$

It follows that the symmetry condition in the momentum space reads:

$$\mathcal{I}H_k = H_{-k}\mathcal{I} \quad \Rightarrow \quad \mathcal{I}H_k\mathcal{I}^{-1} = H_{-k}. \tag{5.25}$$

Similar results hold for inversion-symmetric crystals in two and three spatial dimensions.

### 5.2.2 Time-reversal: spinless particles

One can guess the form of the time reversal operator $T$ from the action on the plane wave $\psi_p = e^{ipx}$. We demand that the coordinate be invariant under $T$, while the momentum must be reversed. Then

$$T\psi_p = \psi_{-p} = e^{-ipx} = \overline{\psi_p}, \tag{5.26}$$

which suggests that time reversal acts as complex conjugation, $T = K$. Indeed, one checks that for any solution $\psi(x,t)$ of the Schrödinger equation, the conjugate wave function $\overline{\psi}(x,t)$ gives the solution for the time-reversed problem. A detailed discussion of the time reversal operation in quantum mechanics can be found in Ref. [31]. Below, we give a brief summary of the results we will use in what follows.

The time reversal operator is **anti-linear**, as it satisfies

$$K(|\psi\rangle + |\chi\rangle) = K|\psi\rangle + K|\chi\rangle, \qquad K(a|\psi\rangle) = \overline{a}K|\psi\rangle, \tag{5.27}$$

where $|\psi\rangle$, $|\chi\rangle$ are state vectors and $a$ is a complex scalar. Such an operator cannot be represented by a matrix and depends on the basis choice:

$$K|\alpha\rangle = |\alpha\rangle \quad \Rightarrow \quad K|\psi\rangle = \sum_\alpha \overline{\psi_\alpha}|\alpha\rangle. \tag{5.28}$$

In the plane wave example above it is natural to assume that $T|x\rangle = |x\rangle$. Thus $T = K$ holds in the basis of position operator eigenstates.

Let us see how $T$ interacts with the inner product. Since $\langle K\psi| = \sum_\alpha \langle\alpha|\psi_\alpha|$, we have

$$\langle K\psi|K\varphi\rangle = \overline{\langle\psi|\varphi\rangle} = \langle\varphi|\psi\rangle. \tag{5.29}$$

The operators with this property are called **anti-unitary**.

Consider now Bloch waves in a crystal. From $T|\alpha_m\rangle = |\alpha_m\rangle$, one finds that $T|\alpha_k\rangle = |\alpha_{-k}\rangle$. Thus, $T$-invariant Bloch Hamiltonian must satisfy:

$$TH_kT^{-1} = H_{-k} \quad \Rightarrow \quad \overline{H_k} = H_{-k}. \tag{5.30}$$

### 5.2.3  Time reversal: spinful particles

When acting on a spinful particle, time reversal must also reverse spin. For spin-$\frac{1}{2}$ particles, this is realized by

$$T = \exp\left(\frac{\pi}{2}\frac{\sigma_y}{i}\right)K = -i\sigma_y K = \begin{pmatrix} 0 & -1 \\ 1 & 0 \end{pmatrix} K. \tag{5.31}$$

Now $T$ is a product of a unitary and anti-unitary operators, and is again anti-unitary.

Consider the action of $T$ on a state $\psi_s$ on the Bloch sphere. For the parametrization given by Eq. (3.43), the complex conjugation $K : \varphi \mapsto -\varphi$ acts as a reflection in the plane $\varphi = 0$, and the exponential acts as a $\pi$ rotation about $y$ axis. The resulting state is proportional to the $\psi_{-s}$ state for eigenstates both of high and low energy.

> **Exercise 5.3.**  Act with $T$ on the spin eigenstates $\psi_\pm(\theta, \varphi)$ defined in Eq. (3.43) and in Ex. 3.2. Check that the time reversal operator flips the spin direction.

A crucial property of $T$ for spin-$\frac{1}{2}$ particles is that

$$T^2 = \begin{pmatrix} 0 & -1 \\ 1 & 0 \end{pmatrix}^2 = -\mathbb{I}. \tag{5.32}$$

This leads to the following **Kramers theorem**. Suppose that the Hamiltonian $\hat{H}$ commutes with some operator $T$, which is anti-unitary and squares to $-\mathbb{I}$. Then each energy level is at least two-fold degenerate. To prove this, first note that

$$\hat{H}|\psi\rangle = \varepsilon|\psi\rangle \quad \Rightarrow \quad \hat{H}T|\psi\rangle = T\hat{H}|\psi\rangle = T\varepsilon|\psi\rangle = \varepsilon T|\psi\rangle, \tag{5.33}$$

so $T|\psi\rangle$ is an eigenstate with the same eigenvalue as $|\psi\rangle$. This does not necessarily mean that the two states are degenerate, since they can be linearly dependent. However, the properties of $T$ imply that

$$\langle\psi|T\psi\rangle = \langle T^2\psi|T\psi\rangle = -\langle\psi|T\psi\rangle \quad \Rightarrow \quad |\psi\rangle \perp |T\psi\rangle, \tag{5.34}$$

so the states $|\psi\rangle$ and $T|\psi\rangle$ are distinct and thus degenerate.

In order to describe spinful electrons in a crystal, one replaces each real-space orbital $|\alpha_m\rangle$ with a basis of a two-level spin system $|\alpha_{m\sigma}\rangle$, where $\sigma = \uparrow, \downarrow$. In this way, Bloch eigenstates become $|\psi_k\rangle = \sum_{\alpha\sigma} \psi_{\alpha k\sigma} |\alpha_{k\sigma}\rangle$. Time reversal sends spin eigenstate $\psi_s$ at crystal momentum $k$ to the state proportional to $\psi_{-s}$ at $-k$.

## 5.3  Application: Graphene

The simplest tight-binding model of graphene includes two orbitals per unit cell, as shown in Fig. 5.2. In Sec. 8.3, we will see that this Hamiltonian, in a sense, describes a critical phase between a trivial and a topological insulator. We will also encounter it the context of topological semimetals in Sec. 10. A detailed discussion of the physical origin of the model can be found, for example, in the first section of Ref. [55]. For a general overview of graphene, see Ref. [56].

### 5.3.1  Brillouin zone and band structure

The Bravais lattice of graphene is hexagonal, while the atomic sites form the honeycomb lattice.[11] We label unit cells by vectors $\boldsymbol{m} = m_1\boldsymbol{a}_1 + m_2\boldsymbol{a}_2$, where $\{\boldsymbol{a}_i\}$ is a real space basis for

---

[11]Here, we use the language of crystallography. For a brief introduction, see Ref. [57].

the Bravais lattice. In a similar way, any vector in the reciprocal space is decomposed in terms of the basis $\{b_i\}$ as $k = k_1 b_1 + k_2 b_2$. The Bloch states are defined as

$$|\alpha_k\rangle = \frac{1}{N} \sum_m e^{ik\cdot m} |\alpha_m\rangle, \tag{5.35}$$

where the dot product means

$$k \cdot m = \sum_i k_i m_i. \tag{5.36}$$

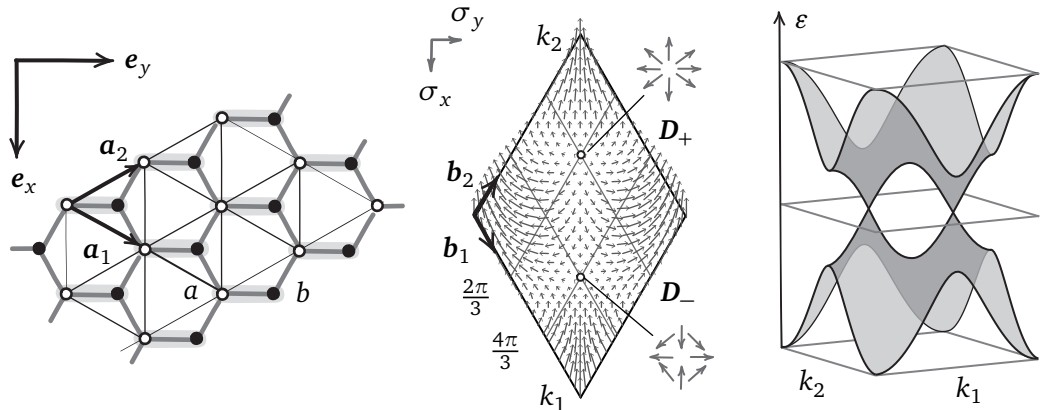

Figure 5.2: LEFT: Crystal lattice of graphene. Empty (filled) circles show atomic orbitals of $a$ ($b$) sublattice. Black lines show hexagonal Bravais lattice with basis vectors $a_1, a_2$. Also shown is the Cartesian basis $e_x, e_y$. Heavy gray lines form the honeycomb lattice and indicate hopping amplitudes $t$. MIDDLE: Brillouin zone of graphene with Dirac points $D_\pm$. Gray arrows represent the vector field $h_k$ describing the Hamiltonian. Insets show vector fields for linearized Bloch Hamiltonian near Dirac points. RIGHT: Energy spectrum of graphene $\varepsilon_\pm = \pm|h_k|$ with the conical band intersections at the Dirac points.

Let us express $a_i$ in terms of the orthonormal Cartesian basis $\{e_x, e_y\}$:

$$a_{1,2} = \begin{pmatrix} \pm\frac{1}{2} \\ \frac{\sqrt{3}}{2} \end{pmatrix}. \tag{5.37}$$

One can also associate $b_i$ with real-space vectors, and interpret the dot product above as the standard inner product on the plane. For example, let us find the direction of $b_1$. The corresponding Bloch wave $e^{ib_1\cdot m} = e^{ik_1 m_1}$ has the wave front along $a_2$. Since $b_1$ plays the role of wave vector, we find that $b_1 \perp a_2$, and similarly $b_2 \perp a_1$. This can also be understood algebraically: Eq. (5.36) holds only if $b_i \cdot a_j = \delta_{ij}$. We have

$$b_{1,2} = \begin{pmatrix} \pm 1 \\ \frac{1}{\sqrt{3}} \end{pmatrix}, \tag{5.38}$$

and the Brillouin zone assumes the shape shown in the middle panel of Fig. 5.2.

Now we construct the tight-binding model. Suppose that only nearest-neighbor hopping is present, with the hopping amplitude $t \in \mathbb{R}$. For example, hopping from $b$ site of a cell with coordinate $m$ to the neighboring $a$ sites is described by the following three terms:

$$t\left(|a_m\rangle\langle b_m| + |a_{m+a_1}\rangle\langle b_m| + |a_{m+a_2}\rangle\langle b_m|\right). \tag{5.39}$$

In the momentum space, this becomes

$$H_k^{ab} = t\left(1 + e^{-ik \cdot a_1} + e^{-ik \cdot a_2}\right) = t\left(1 + e^{-ik_1} + e^{-ik_2}\right) \equiv f_k. \tag{5.40}$$

Thus, the Bloch Hamiltonian in $\{|a_k\rangle, |b_k\rangle\}$ basis reads:

$$H_k = \begin{pmatrix} 0 & f_k \\ \overline{f_k} & 0 \end{pmatrix}. \tag{5.41}$$

The corresponding vector field $h_k$ is shown in Fig. 5.2 in the middle panel.

There are two special points $D_\pm$ in the Brillouin zone, in which the Hamiltonian is gapless, $H_{D_\pm} = 0$. Their coordinates are easily found to be

$$D_\pm = \pm\frac{2\pi}{3}\begin{pmatrix} 1 \\ -1 \end{pmatrix}. \tag{5.42}$$

These points are known as **Dirac points** because of the conical form of the band touchings, reminiscent of the linear dispersion relation of massless relativistic particles. The analogy, however, is not exact: the Dirac equation operates with four-component wave functions of a spinful particle, and we have only two energy levels originating from the sublattice degree of freedom. The spectrum of the Hamiltonian is given by

$$\varepsilon_\pm = \pm|h_k|, \tag{5.43}$$

and is shown in Fig. 5.2 on the right.

Let us find the expansion of the Hamiltonian around Dirac points $D_\pm$ in terms of a long-wavelength parameter $|q| \ll 2\pi$. We start with the expression

$$f_{D_\pm + q} = t\left(1 + \sum_j e^{-i(D_\pm + q) \cdot a_j}\right). \tag{5.44}$$

**Exercise 5.4.** Show that expansion of $f_{D_\pm + q}$ to linear order in $q$ reads

$$f_{D_\pm + q} \approx -\frac{t\sqrt{3}}{2}\left(\pm q \cdot e_x - iq \cdot e_y\right). \tag{5.45}$$

Denoting dot products in the last expression by $q_x$ and $q_y$, we obtain the Hamiltonian

$$H_{D_\pm + q} \approx -\frac{t\sqrt{3}}{2}\left(\pm q_x \sigma_x + q_y \sigma_y\right). \tag{5.46}$$

The Hamiltonian near $D_\pm$ is linear in terms of Pauli matrices, which agrees with the conical shape of band touchings at the Dirac points. Note that the vector $h_k$ has opposite sense of rotation when going around $D_+$ and $D_-$, as shown in the middle panel of Fig. 5.2.

### 5.3.2 Symmetry considerations

Conical band intersections at the Dirac points are the hallmark of the graphene spectrum. They are protected by symmetries of graphene, in the sense that one cannot open the gap by adding small symmetry-preserving perturbations to the Hamiltonian. Here, we consider two such symmetries (the full symmetry group contains more elements: see, for example, Ref. [58]).

First, note that $f_k = \overline{f_{-k}}$, so the Hamiltonian satisfies Eq. (5.30) and has time-reversal symmetry $T$. In terms of the vector field $h_k$, this means that the vectors $h_k$ and $h_{-k}$ are

related by the reflection in the $xz$ plane, since complex conjugation of the Hamiltonian matrix reverses the sign of $h_y$.

The Hamiltonian is also symmetric under inversion with respect to the middle point between the two orbitals. We select the inversion center lying in the unit cell with $\boldsymbol{m} = 0$, so the symmetry is represented by

$$\hat{\mathcal{I}}|a_{\boldsymbol{m}}\rangle = |b_{-\boldsymbol{m}}\rangle\,, \qquad \hat{\mathcal{I}}|b_{\boldsymbol{m}}\rangle = |a_{-\boldsymbol{m}}\rangle\,. \tag{5.47}$$

Thus, the matrix of the inversion operator is $\sigma_x$. In the momentum space, the Hamiltonian satisfies

$$\sigma_x H_{\boldsymbol{k}} \sigma_x = H_{-\boldsymbol{k}}\,. \tag{5.48}$$

The conjugation by $\sigma_x$ amounts to the $\pi$ rotation around $\sigma_x$ axis in the space of Pauli matrices. It follows that the vectors $\boldsymbol{h_k}$ and $\boldsymbol{h_{-k}}$ must have opposite signs of $h_y$ and $h_z$ components.

If the Hamiltonian respects both $\mathcal{I}$ and $T$, then it is invariant under their combination $\mathcal{I} \circ T$. The converse, however, is not true, and one should consider this combined symmetry separately. Due to the double reversal of the sign of $\boldsymbol{k}$, this symmetry acts in the momentum space *locally*:

$$\sigma_x \overline{H_{\boldsymbol{k}}} \sigma_x = H_{\boldsymbol{k}}\,. \tag{5.49}$$

In the space of Pauli matrices, the transformation on the left is the combination of reflection in the $xz$ plane with $\pi$ rotation around $x$ axis. The result is the reflection in the $xy$ plane. Thus, the $\mathcal{I} \circ T$ symmetry forces the vector $\boldsymbol{h_k}$ to lie in the $xy$ plane, and does not put any other constraints. Note that any vector field $\boldsymbol{h_k}$ with vanishing $h_z$ describes a Hamiltonian with $\mathcal{I} \circ T$ symmetry; at the same time, $\mathcal{I}$ and $T$ may be broken individually, if $h_y(\boldsymbol{k}) \neq -h_y(-\boldsymbol{k})$.

This symmetry constraint protects the Dirac points from small symmetry-preserving perturbations. As we will discuss in Sec.10.1, such perturbations can only change the position of an individual Dirac point, but cannot destroy it (however, under a large enough perturbation, two points can merge and annihilate). Thus, in order to open the gap at the Dirac point by a small perturbation, one has to break $\mathcal{I} \circ T$, which in turn requires breaking either $\mathcal{I}$ or $T$. We will come back to this point in Sec. 8.3.1.

# 6 Modern theory of electric polarization

Electric polarization is a basic property of crystals, which is commonly associated with the spatial distribution of the charge density $\rho(x)$. If the charge distribution of a whole crystal is known, it is straightforward to compute its dipole moment. Then, dividing by the volume of the crystal, one obtains the value of the electric polarization as the dipole moment per unit volume. However, this approach becomes problematic when we focus on a single unit cell with periodic boundary conditions, which is the setting of numerical studies of electronic structure. In Sec. 6.1, we discuss the origin of this problem and its solution on the classical level. In what follows, we develop the corresponding quantum theory using the tight-binding formalism and show that the polarization is determined by a certain geometric phase.

## 6.1 Difficulties with polarization

### 6.1.1 From charge density to currents

Recall that a system of point charges $q_i$ with coordinates $\boldsymbol{r}_i$ is characterized by the dipole moment $\boldsymbol{d} = \sum_i q_i \boldsymbol{r}_i$. If the system is neutral, $\sum_i q_i = 0$, the dipole moment does not depend on the choice of the origin. This is readily generalized to the case of the continuous charge density.

Consider a one-dimensional crystal of length $L$ consisting of the ionic cores and the electronic cloud shown in Fig. 6.1 on the left. The dipole moment has two respective contributions:

$$d = \sum_i q_i x_i + \int_L x\rho(x)dx\,, \tag{6.1}$$

where $q_i$ and $x_i$ describe ionic cores, and $\rho(x)$ is the electronic charge density. The polarization of the crystal is the dipole moment per unit volume, which becomes $P = \frac{d}{L}$ in one dimension.

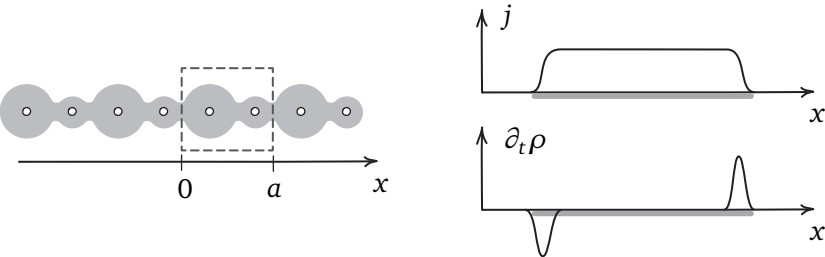

Figure 6.1: LEFT: Charge distribution of a finite crystal consists of the point-like positive ionic cores and continuous electronic charge density. One unit cell is selected, $a$ denotes the lattice constant. RIGHT: Spatial distribution of the current $j$ and the corresponding evolution of the charge density $\partial_t \rho$. Heavy gray line represents the crystal.

In this way, the polarization of the crystal is determined by the static charge distribution. This is not, however, how the polarization is measured in experiments. For example, consider the piezoelectric effect. Suppose that we wish to measure the polarization of a cubic sample, which results from squeezing it in the $x$ direction. This is done by placing the electrodes on the two faces normal to the $x$ axis and connecting them by the shorting circuit. The deformation of the crystal leads to the redistribution of the charge in the bulk, which results in the appearance of the surface charges. These charges are used as a measure for the bulk polarization. They can be found from the current that flows between the electrodes in the external circuit during the process of deformation.

Thus, the classical definition of the polarization uses the static charge distribution, while in experiments one measures currents caused by the changes of the polarization. Two approaches are related by the continuity equation:

$$\partial_x j(x,t) = -\partial_t \rho(x,t)\,, \tag{6.2}$$

where $j(x,t)$ is the current density (or simply the current, since the model is one-dimensional) and $\rho(x,t)$ is the charge density distribution. Fig. 6.1, right, shows how the bulk current gives rise to the charge accumulation on the right end of the crystal, according to the continuity equation.

The harmony between theory and experiment was disturbed by the first-principles numerical methods, which gave accurate predictions of the charge distribution in real materials. The quantum simulation of the whole crystal is inaccessible, so the model consists of a single unit cell with the periodic boundary conditions. The electronic contribution to the electric polarization is defined as the dipole moment of a unit cell:

$$P_{dip} = \frac{1}{a} \int_0^a x\rho(x)dx\,, \tag{6.3}$$

where $a$ is the lattice constant.[12] But it was found that the computed value of the polarization does not agree with the experimental data. This was a surprising result for a well-established field: what could be wrong with the century-old textbook formulas? The answer can be deduced from the right panel of Fig.6.1. Note that the charge redistribution follows a (highly hypothetical) scenario, in which the current $j(x,t)$ is *spatially uniform* inside the crystal: $\partial_x j = 0$, so that the bulk charge density remains unchanged. Thus, the experiment would show the charge accumulation at the right end, but it would remain completely invisible for the bulk theory based on the charge density $\rho$.

This difficulty is resolved by the **modern theory of electric polarization** [59], which redefines the polarization in terms of the bulk currents and thus aligns the cell-periodic theory with the experimental methods. Moreover, the theory provides the corresponding quantum-mechanical expression for $P$, which has now become a standard computational tool implemented in the *ab initio* software packages. To motivate the new definition, consider the time derivative of the dipole moment of the unit cell. Using the continuity equation, we obtain:

$$\partial_t P_{dip} = \frac{1}{a}\int_0^a x(\partial_t \rho)dx = -\frac{1}{a}\int_0^a x(\partial_x j)dx = -\frac{1}{a}(xj)\Big|_0^a + \frac{1}{a}\int_0^a j dx\,. \qquad (6.4)$$

The last term describes the average current flowing through the unit cell. We rewrite this as

$$\frac{1}{a}\int_0^a j dx = j(a) + \partial_t P_{dip}\,. \qquad (6.5)$$

In the situation shown in the figure, $\partial_t P_{dip}$ vanishes, and the equation tells us that the current flowing through the unit cell equals the current $j(a)$ through the cell boundary. In the modern theory, the electric polarization is *defined* as a quantity whose rate of change is given by the average current through the unit cell:

$$\partial_t P = \frac{1}{a}\int_0^a j dx\,. \qquad (6.6)$$

Note that, in contrast with Eq. (6.3), this gives an accurate description of the process. In particular, one can find the changes of the charge accumulated at the end of the finite crystal. In the same way as the experiments, this formula does not give an absolute value of the polarization, but only the difference between the values in the initial and final states.

### 6.1.2 Quantum systems

Our main goal is to obtain the expression for the electric polarization of a crystal from its Bloch Hamiltonian. Here, we discuss a tight-binding example showing that even a perfect knowledge of the cell-periodic charge density is not enough to determine the adiabatic current.

First, consider the model of a diatomic molecule described in Sec. 5.1.1:

$$H = \Delta(p)\sigma_z + t(p)\sigma_x\,, \qquad (6.7)$$

where on-site potentials and hopping amplitudes are now functions of the control parameter $p \in [0,\pi]$. Let the Hamiltonian parameters vary as shown in Fig. 6.2. In the initial state, $p = 0$, there is no hopping, and the ground state $|a\rangle$ has energy $-\Delta_0$. Then we turn the hopping on and gradually reverse on-site potentials. At the point $p = \frac{\pi}{2}$, the Hamiltonian matrix is proportional to $\sigma_x$, and the ground state is given by $\frac{1}{\sqrt{2}}(|a\rangle - |b\rangle)$. Finally, we arrive in another state in which the orbitals are decoupled, with the ground state $|b\rangle$. If the process

---

[12]In our units, $a = 1$, but we will keep the name of this variable in some formulas.



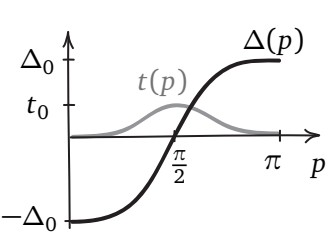

| $p$ | $H$ | $|\psi\rangle$ | $e|\psi|^2$ |
|---|---|---|---|
| $0$ | $-\Delta_0\sigma_z$ | $|a\rangle$ | |
| $\frac{\pi}{2}$ | $t_0\sigma_x$ | $\frac{1}{\sqrt{2}}(|a\rangle - |b\rangle)$ | |
| $\pi$ | $\Delta_0\sigma_z$ | $|b\rangle$ | |

Figure 6.2: Charge pumping in a single dimer. LEFT: Variation of the Hamiltonian parameters $t$ and $\Delta$ as functions of the control parameter $p$. RIGHT: Hamiltonian matrices $H$, ground state wave functions $|\psi\rangle$, and electric charge densities $e|\psi^2|$ for the three values of $p$. Shading of circles indicates on-site potentials (the darker, the lower).

is slow enough, the adiabatic theorem asserts that the system will remain in the ground state at each stage. Note that the charge density $\rho = e|\psi|^2$ gets shifted from $a$ site to $b$ site.

Now consider a one-dimensional crystal made of such dimers and set $t_{in}(p) = t(p)$ and $t_{ex} = 0$ for $p \in [0, \pi]$, as shown in the left panel of Fig. 6.3. For $p \in [\pi, 2\pi]$, let the charge shift *between* two unit cells in a similar process: the on-site potential difference $\Delta$ goes back to the negative value, while the external hopping $t_{ex}$ is turned on and $t_{in}$ is zero. As a result, the electrons shift to the right by one lattice constant, and for $p = 2\pi$ we arrive in the initial state.

But what if the roles of internal and external hopping amplitudes are interchanged? Suppose that $t_{in} = 0$ and $t_{ex} \neq 0$ in the first half of the cycle, while $t_{in} \neq 0$ and $t_{ex} = 0$ for the second half. Clearly, in this case the charge flows from the right to the left (Fig. 6.3, right panel). Note that for both protocols, the evolution of the charge density $e|\psi|^2$ is exactly the same. It is the periodic nature of the crystal that makes it possible to go in opposite directions while moving from $a$ to $b$ sublattice in both cases.

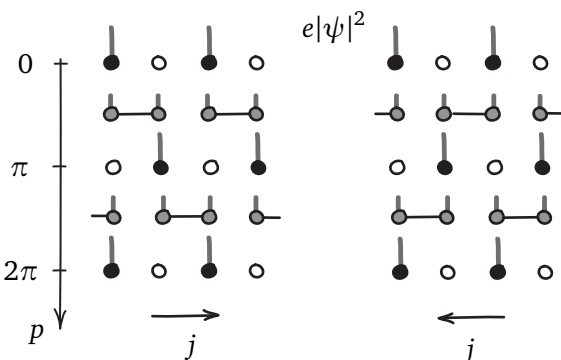

Figure 6.3: Two charge-pumping protocols with identical charge density evolution and opposite directions of current. Shading of circles indicates on-site potentials (the darker, the lower). Vertical bars represent the charge density $\rho = e|\psi|^2$.

This example illustrates that one cannot determine the electric polarization (6.6) from the charge density alone. Classically, the difference stems from the term that describes the current through the unit cell boundary $j(a)$. In the first case, $j(a) = 0$ for $p \in [0, \pi]$ and $j(a) \neq 0$ for $p \in (\pi, 2\pi)$, while in the second case the situation is reversed. Note that the Hamiltonian

$\hat{H}_k$ and thus its eigenstates $|\psi_k\rangle$ do contain information about the current $j(a)$, but it is lost when we take the modulus squared of the state vector. This should not be surprising, since in general the probability current is sensitive to the phase of the wave function.

In such extreme limit of decoupled dimers, we are able to track positions of electrons simply by inspection. In a more realistic situation, both hopping amplitudes are non-zero, and electrons are delocalized. We still can compute the cell-periodic charge density and $P_{dip}$, but finding the current through the boundary $j(a)$ becomes a non-trivial problem.

## 6.2 Wannier functions and geometry

Due to the periodic and delocalized character of the Bloch eigenstate $|\psi_k\rangle$, it does not make much sense to act on it with the position operator. On the other hand, we know that the combination of plane waves can result in a well-localized wave packet. In the context of Bloch theory, this motivates the definition of the Wannier functions. We will see that the center-of-mass coordinate of the Wannier function is determined by the geometry of the eigenstate bundle. Later, these functions will help us to compute the elusive current $j(a)$.

### 6.2.1 Definition and basic properties

For simplicity, we consider a single occupied band with eigenstates $|\psi_k\rangle$, which is the case for the diatomic chain at the half-filling (the chain is defined in Sec. 5.1.2). We define $n$-th **Wannier function** as the inverse Fourier image of the Bloch eigenstates:

$$|w^n\rangle = \frac{1}{\sqrt{N}} \sum_k e^{-ikn} |\psi_k\rangle. \tag{6.8}$$

These functions are not Hamiltonian eigenstates, but they are orthonormal and span the same Hilbert space as Bloch functions. The real space components of Wannier function $|w^n\rangle$ depend only on the difference $m - n$,

$$w^n_{\alpha m} = \frac{1}{N} \sum_k e^{ik(m-n)} \psi_{\alpha k}, \tag{6.9}$$

so that Wannier functions with different $n$ are related by a lattice translation. We will also need the inverse transform relating the components of $|\psi_k\rangle$ with the components of zeroth Wannier function:

$$\psi_{\alpha k} = \sum_m e^{-ikm} w^0_{\alpha m}. \tag{6.10}$$

Since the Bloch states are periodic in the real space, one can expect that Wannier functions are localized and thus well-suited to be acted on by the position operator. The expectation value

$$x_n = \langle w^n|\hat{x}|w^n\rangle, \tag{6.11}$$

is called $n$-th **Wannier center**. Note that the Fourier transform is sensitive to the phases, so the Wannier states (6.8) are not gauge-invariant. For example, a new set of Bloch states

$$|\psi_k\rangle' = e^{-ikm}|\psi_k\rangle \quad \Rightarrow \quad |w^n\rangle' = |w^{n+m}\rangle, \tag{6.12}$$

results in the relabeling of the Wannier functions. A more general gauge transformation can also change the shape of the Wannier functions. However, despite these ambiguities, the coordinate $x_n$ turns out to be gauge-invariant modulo lattice constant, as we will see in Sec. 6.2.3.

Let us find the cell-periodic charge density. For the diatomic chain at the half-filling, each unit cell contributes one electron, and charge density must satisfy

$$\sum_{\alpha m} \rho_{\alpha m} = N e. \tag{6.13}$$

From the orthonormality conditions, we have

$$\sum_{n} \langle w^n | w^n \rangle = \sum_{k} \langle \psi_k | \psi_k \rangle = N. \tag{6.14}$$

It follows that the charge density can be expressed in terms of either set of functions:

$$\rho_{\alpha m} = \frac{e}{N} \sum_{k} |\psi_{\alpha k}|^2 = e \sum_{n} |w^n_{\alpha m}|^2. \tag{6.15}$$

**Exercise 6.1.** Consider two Bloch Hamiltonians defined in Exercise 5.2. Find Wannier functions $|w^n\rangle$ as the inverse Fourier transforms of $|\psi_k\rangle$. Check that two expressions for charge density given by Eq. (6.15) agree in both cases. Compute positions of the Wannier centers and note that they coincide with the centers of charge of the diatomic "molecules". [6.2]

### 6.2.2 Zak phase

Let us calculate the coordinate of the zeroth Wannier center $x_0$, following Ref. [9]. Taking into account equations from sections 5.1.3 and 5.1.4, we obtain

$$\hat{x} |w^0\rangle = \frac{1}{N} \sum_{\alpha m k} \psi_{\alpha k} e^{imk} \hat{x} |\alpha_m\rangle = \frac{1}{N} \sum_{\alpha m k} \psi_{\alpha k} e^{imk} (m + \tau_\alpha) |\alpha_m\rangle, \tag{6.16}$$

where $m$ is the cell index, $\tau_\alpha \in [0, 1)$ is the coordinate of the orbital $|\alpha_m\rangle$ inside the unit cell, and the lattice constant is $a = 1$. Recall from Sec. 5.1.4 that the Bloch eigenstate component in the basis $|\widetilde{\alpha_k}\rangle$ is $u_{\alpha_k} = \psi_{\alpha k} e^{-ik\tau_\alpha}$. We rewrite the last equation in terms of $u_{\alpha k}$, as follows:

$$\hat{x} |w^0\rangle = \frac{1}{N} \sum_{\alpha m k} u_{\alpha k} \frac{1}{i} \frac{\partial}{\partial k} \left( e^{i(m + \tau_\alpha)k} \right) |\alpha_m\rangle. \tag{6.17}$$

If $N$ is large, one can consider the momentum space sum as an integral over the Brillouin zone: $\frac{1}{N} \sum_{k} \to \frac{1}{2\pi} \int_{BZ} dk$, which gives

$$\hat{x} |w^0\rangle = \frac{1}{2\pi i} \sum_{\alpha m} \int_{BZ} u_{\alpha k} \partial_k \left( e^{i(m + \tau_\alpha)k} \right) |\alpha_m\rangle dk. \tag{6.18}$$

Then, integration by parts yields:

$$\hat{x} |w^0\rangle = \sum_{\alpha m} |\alpha_m\rangle \frac{1}{2\pi i} \left[ u_{\alpha k} e^{i(m + \tau_\alpha)k} \Big|_0^{2\pi} - \int_{BZ} \left( \partial_k u_{\alpha k} \right) e^{i(m + \tau_\alpha)k} dk \right]. \tag{6.19}$$

We perform the summation over $m$, which turns $e^{imk} |\alpha_m\rangle$ into the Bloch wave $\sqrt{N} |\alpha_k\rangle$, and we have

$$\hat{x} |w^0\rangle = \frac{\sqrt{N}}{2\pi i} \sum_{\alpha} \psi_{\alpha k} \Big|_0^{2\pi} |\alpha_k\rangle + \frac{\sqrt{N}}{2\pi} \sum_{\alpha} \int_{BZ} i \left( \partial_k u_{\alpha k} \right) |\widetilde{\alpha_k}\rangle dk. \tag{6.20}$$

Note that the first term vanishes because of the periodicity of $\psi_{\alpha k}$. Finally, we multiply on the left by

$$\langle w^0 | = \frac{1}{\sqrt{N}} \sum_{\beta k'} \overline{u_{\beta k'}} \langle \widetilde{\beta_{k'}} |, \tag{6.21}$$

and obtain

$$x_0 = \langle w^0 | \hat{x} | w^0 \rangle = \frac{1}{2\pi} \int_{BZ} i \sum_\alpha \overline{u_{\alpha k}} \, \partial_k u_{\alpha k} dk \equiv \frac{\gamma}{2\pi}. \qquad (6.22)$$

The quantity $\gamma$ is called the **Zak phase** [46]. The form of the integrand resembles the Berry potential, which suggests the geometric origin. However, the phase $\gamma$ is not equivalent to *the* Berry phase for the Bloch eigenstates $|\psi_k\rangle$, as we will see in the next section.

> **Exercise 6.2.** For the two Hamiltonians defined in Exercise 5.2, compute the Zak phase for the valence band. Compare with the results of Exercise 6.1.

### 6.2.3 Geometric interpretation

To interpret the Zak phase geometrically, we first need to identify the relevant vector bundles. Interestingly, the Hamiltonian of the crystal $\hat{H}$ itself defines the parameter space. In the Bloch wave basis, the matrix of the Hamiltonian becomes block-diagonal, with blocks indexed by the crystal momentum $k$. Then we interpret the Brillouin zone circle, $k \in [0, 2\pi)$, as the parameter space for the Bloch Hamiltonian. For each $k$, the Bloch Hamiltonian $H_k$ acts on the Hilbert space $\mathcal{H}_k$. The dimension of $\mathcal{H}_k$ is determined by the number of degrees of freedom in each unit cell (for our diatomic chain, $\dim \mathcal{H}_k = 2$). Taken together, these spaces form the vector bundle $\{\mathcal{H}_k\}$ over the Brillouin zone. The lower-energy, or valence band eigenstate $|\psi_k\rangle$ of $H_k$ defines the valence-band eigenspace $V_k^v \subset \mathcal{H}_k$. We are interested in the geometry of the valence band eigenspace bundle $V^v$.

Let us find the Berry phase acquired by the eigenstate $|\psi_k\rangle$ after going around the Brillouin zone (note that this means taking the contour integral of the potential rather than a physical process of the adiabatic evolution). At first sight, the Berry potential is given by

$$A(k) = i \langle \psi_k | \partial_k \psi_k \rangle, \qquad (6.23)$$

but this expression can be misleading. In general, we know that the value of the inner product $\langle \psi | \chi \rangle$ does not depend on the basis choice; here, the situation is different. As discussed in Sec. 3.3.2, the Berry potential depends on the choice of the "constant basis" in $\{\mathcal{H}_k\}$ that one uses to write $|\psi_k\rangle$ in components. Here, we have two standard bases of Bloch waves over the Brillouin zone, $|\alpha_k\rangle$ and $|\widetilde{\alpha_k}\rangle$. As they are related to each other by the $k$-dependent transformation, the conditions $\partial_k |\alpha_k\rangle = 0$ and $\partial_k |\widetilde{\alpha_k}\rangle = 0$ clearly cannot be satisfied at the same time. Which basis is to be declared constant? Perhaps, one should choose the periodic one, and assume that $\partial_k |\alpha_k\rangle = 0$. In this case, Eq. (6.23) will invariably give $i \sum_\alpha \overline{\psi_{\alpha k}} \partial_k \psi_{\alpha k}$. How can we obtain the expression with $u_{\alpha k}$, as in Eq. (6.22)?

Formally, this issue can be resolved as follows. Introduce two differential operators in the bundle of Hilbert spaces $\{\mathcal{H}_k\}$:

$$D_k |\alpha_k\rangle = 0, \qquad \widetilde{D_k} |\widetilde{\alpha_k}\rangle = 0. \qquad (6.24)$$

Both operators act on scalar functions as an ordinary derivative $\partial_k$. These equations are to be understood as definitions of the operators rather than statements about the basis vectors. Note that $\widetilde{D_k}$ is well-defined despite the discontinuity of $|\widetilde{\alpha_k}\rangle$, since $|\widetilde{\alpha_k}\rangle$ and $|\widetilde{\alpha_{k+2\pi}}\rangle$ differ by a $k$-independent factor.

Each of the operators, when projected onto the eigenspace of $|\psi_k\rangle$, gives rise to the corresponding Berry potential:

$$A(k) = i \langle \psi_k | D_k \psi_k \rangle = i \sum_\alpha \overline{\psi_{\alpha k}} \partial_k \psi_{\alpha k}, \qquad (6.25)$$

and

$$\widetilde{A}(k) = i \langle \psi_k | \widetilde{D_k} \psi_k \rangle = i \sum_\alpha \overline{u_{\alpha k}} \, \partial_k u_{\alpha k} \,. \tag{6.26}$$

Note that in both cases we have the same ambient bundle $\{\mathcal{H}_k\}$ and the same set of eigenspaces as fibers of the subbundle $V^\nu$. However, the operators $D_k$ and $\widetilde{D_k}$ determine different connections on $V^\nu$, which result in different geometric phases for $|\psi_k\rangle$. Further discussion of this subtle form of gauge dependence can be found in Refs. [60, 61]. In what follows, we will use tilde to indicate the geometric quantities computed using the differential operator $\widetilde{D_k}$.

Geometrically, the operators $D_k$ and $\widetilde{D_k}$ can be thought of as covariant derivatives in the bundle of Hilbert spaces, defined in the spirit of Eq. (1.55). Each basis defines its own parallel transport in $\{\mathcal{H}_k\}$, according to Eq. (6.24). The fact that the basis $|\widetilde{\alpha_k}\rangle$ is not periodic in the Brillouin zone does not harm this geometric picture. Indeed, the parallel transported vector need not return to its initial state after going around a closed loop.

We conclude that the Zak phase is given by

$$\gamma = \int_{BZ} \widetilde{A}(k) dk \,, \tag{6.27}$$

where $\widetilde{A}(k)$ is the Berry potential obtained by the projection of the derivative $\widetilde{D_k}$ in $\{\mathcal{H}_k\}$ defined by the condition $\widetilde{D_k} |\widetilde{\alpha_k}\rangle = 0$. Now that we have described the Zak phase as a geometric phase, we can apply the general properties of the gauge invariance (1.49). Under any transformations of the basis section $|\psi_k\rangle' = e^{i\beta(k)} |\psi_k\rangle$, the Zak phase can change only by an integer multiple of $2\pi$. In this case, the position of the Wannier center (6.22) will change respectively by an integer number of lattice constants, as exemplified by Eq. (6.12).

## 6.3 Polarization as a geometric phase

We are now in a position to find the quantum expression for the electric polarization $P$, as defined by Eq. (6.6). To this end, we will follow Ref. [62] and analyze the physical meaning of the geometric phases obtained by integration of the potentials $A(k)$ and $\widetilde{A}(k)$. For the original derivation based on the linear-response calculation of the adiabatic current, see textbook [9] and lecture notes [63], both written by the founders of the modern theory of electric polarization.

### 6.3.1 Current through the boundary of a unit cell

Since the components of $|\psi_k\rangle$ in the two bases satisfy $\psi_{\alpha k} = e^{ik\tau_\alpha} u_{\alpha k}$, the Berry potentials are related as

$$i \sum_\alpha \overline{u_{\alpha k}} \partial_k u_{\alpha k} = i \sum_\alpha \overline{\psi_{\alpha k}} \partial_k \psi_{\alpha k} + \sum_\alpha \tau_\alpha |\psi_{\alpha k}|^2 \,. \tag{6.28}$$

We are interested in the corresponding decomposition of the Zak phase and its physical meaning. Taking into account Eq. (6.10), one checks that

$$\Delta Q \equiv \frac{e}{2\pi} \int_{BZ} A(k) dk = \frac{ie}{2\pi} \int_{BZ} \sum_\alpha \overline{\psi_{\alpha k}} \partial_k \psi_{\alpha k} dk = e \sum_{\alpha m} m |w_{\alpha m}^0|^2 \,. \tag{6.29}$$

Thus, multiplying both sides of Eq.(6.28) by $\frac{e}{2\pi}$ and integrating each term over the Brillouin zone, we obtain

$$\frac{e\gamma}{2\pi} = \Delta Q + \sum_\alpha \tau_\alpha \rho_{\alpha m} \,. \tag{6.30}$$

The term on the left is proportional to the Zak phase. The last term on the right has the meaning of the average dipole moment of a unit cell $P_{dip}$ (with the lattice constant set to unity).

What is the physical meaning of the term denoted by $\Delta Q$? Recall that the charge density can be expressed as a sum of contributions $\rho_{\alpha m}^n$ from the individual Wannier functions:

$$\rho_{\alpha m} = \sum_n e |w_{\alpha m}^n|^2 \equiv \sum_n \rho_{\alpha m}^n. \tag{6.31}$$

In each contribution, let us further sum over the orbitals to obtain its coarse-grained version. We obtain the quantity

$$\rho_m^n = \sum_\alpha \rho_{\alpha m}^n, \tag{6.32}$$

which measures the amount of charge carried by the $n$-th Wannier function in the $m$-th unit cell. With this notation, $\Delta Q$ can be expressed as

$$\Delta Q = \sum_m m \rho_m^0. \tag{6.33}$$

To understand the meaning of this sum, we plot the part of the graph of $\rho_m^n$ near the origin of

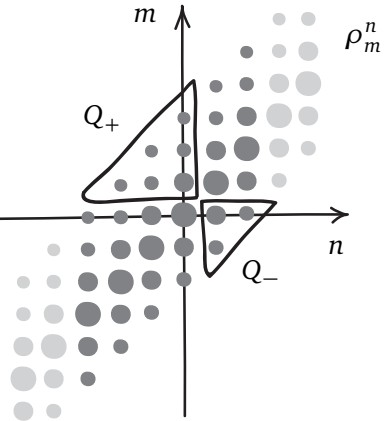

Figure 6.4: Contributions $\rho_m^n$ to the charge density. Circles represent electric charge carried by the $n$-th Wannier function in the $m$-th unit cell. Darker circles represent Wannier functions that have non-zero components in the unit cell with $m = 0$.

the $(m, n)$ torus, as shown in Fig.6.4. Each column of circles represents the charge distribution for some Wannier function. It follows from the figure that the quantity $\Delta Q$ measures the difference between total charges contained in the selected triangles,

$$\Delta Q = Q_+ - Q_-. \tag{6.34}$$

To see this, note that Wannier functions are related by lattice translations, and thus the values of $\rho_m^n$ are constant along diagonals.

The significance of the quantity $\Delta Q$ becomes clear if we consider the adiabatic change of the Hamiltonian. Suppose that we perform the experiment described in Sec. 6.1.1, and change the state of the crystal in a way that results in redistribution of charge. In this case Wannier functions will change their shape and shift along vertical lines. Observe that the

increase rate $\dot{Q}_+$ and decrease rate $-\dot{Q}_-$, taken together, measure exactly the current that flows through the boundary of the zeroth unit cell:

$$\Delta\dot{Q} = j(a). \tag{6.35}$$

Recall that one cannot find the current $j(a)$ from the charge density $\rho_{\alpha m}$, as illustrated by Fig. 6.3. Remarkably, the decomposition (6.31) of the same charge density in terms of Wannier functions allows us to extract this important piece of information.

### 6.3.2 Polarization and Wannier charge center

Consider the time derivative of Eq. (6.30):

$$\partial_t\left(\frac{e\gamma}{2\pi}\right) = \Delta\dot{Q} + \partial_t\left(\sum_\alpha \tau_\alpha \rho_{\alpha m}\right). \tag{6.36}$$

Note that the terms on the right correspond to the current through the unit cell boundary $j(a)$ and the time derivative of the dipole moment of a unit cell, $\partial_t P_{dip}$. Thus we have just obtained the quantum versions of the terms on the right in Eq. (6.5), which defines the electric polarization classically. We conclude that the changes in the electronic contribution to the polarization are determined by the Zak phase:

$$\Delta P = \frac{e}{2\pi}\int_{t_i}^{t_f} \dot{\gamma}\,dt. \tag{6.37}$$

With some caution in mind, this can be rewritten as

$$\Delta P = \frac{e}{2\pi}\Big(\gamma(t_f) - \gamma(t_i)\Big). \tag{6.38}$$

From the last equation, one defines the electronic contribution to the polarization as

$$P = \frac{e\gamma}{2\pi}, \qquad \gamma = \int_{BZ} \widetilde{A}(k)\,dk. \tag{6.39}$$

In other words, polarization is defined as a dipole moment of zeroth Wannier function charge distribution:

$$P = e\langle w^0|\hat{x}|w^0\rangle = e\sum_{\alpha m}(m+\tau_\alpha)|w^0_{\alpha m}|^2 = \sum_{\alpha m}(m+\tau_\alpha)\rho^0_{\alpha m}, \tag{6.40}$$

and equals the dipole moment of an elementary charge placed in the zeroth Wannier center. Geometric phase formula (6.39), along with the current-based definition (6.6), constitute the core of the modern theory of electric polarization. For two- and three-dimensional systems, the polarization is computed by averaging the value of the Zak phase in a given direction over the surface Brillouin zone.

    To illustrate the roles of the terms in Eq. (6.30), we consider the adiabatic current that flows through the dimerized chain shown in Fig. 6.3 on the left. The values of the terms of Eq. (6.30) as functions of the pumping parameter $p$ are shown in the first row in Fig. 6.5. The second row shows similar graphs for the charge-pumping protocol shown in Fig. 6.3 on the right. In both cases, the dipole moment $P_{dip}$ is determined solely by the charge density distribution $\rho_{\alpha m}$, and returns to its initial value $P_{dip} = 0$ after the complete cycle. The current through the boundary, $j(a) = \Delta\dot{Q}$, makes the whole difference. It determines the direction of the shift of Wannier centers, and thus the total current.

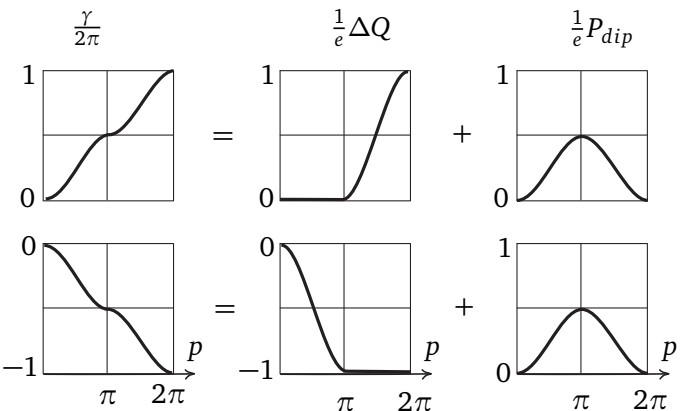

Figure 6.5: Two components of the Zak phase in Eq. (6.30) as functions of the pumping parameter $p$. The TOP (BOTTOM) row corresponds to the process shown in Fig. 6.3 on the left (right). For calculations, see Supplementary material of Ref. [62].

## 6.4 Magnetic flux and polarization quantum

Finally, we need to discuss why it takes caution to switch from Eq. (6.37) to the two-point formula (6.38). To see this, we compare changes in polarization with insertion of magnetic flux into a circular crystal.

### 6.4.1 Magnetic flux and Peierls substitution

Consider a periodic diatomic chain as a ring, and let it be threaded by the magnetic flux $\Phi$, as shown in Fig. 6.6. We choose the vector potential to be constant along the ring, $\Phi = \int A dl$. We set the coordinate of site $a$ to zero, $\tau_a = 0$ and that of $b$ site to $\tau_b$. Since the integral of $A$ over a unit cell is $\frac{\Phi}{N}$, we have

$$\int_0^{\tau_b} A dl = \tau_b \frac{\Phi}{N}, \qquad \int_{\tau_b}^1 A dl = (1 - \tau_b)\frac{\Phi}{N}, \tag{6.41}$$

where the lattice constant is set to unity. The first integration is performed from the $a$ site to the $b$ site inside a unit cell, and the second one from $b$ site to the $a$ site of the next unit cell.

In the tight-binding approximation, one includes magnetic field by altering hopping amplitudes, procedure known as **Peierls substitution**:

$$t_{ab} \quad \rightarrow \quad t_{ab} \exp\left(i\frac{q}{\hbar}\int_{\tau_a}^{\tau_b} A dl\right). \tag{6.42}$$

For example, the internal hopping amplitude is changed as

$$t_{in} \quad \rightarrow \quad t_{in} \exp\left(-i\Delta k \frac{\Phi}{\Phi_0}\tau\right), \tag{6.43}$$

where $\Delta k = \frac{2\pi}{N}$.

Since we are concerned with the real-space positions of the orbitals, we should use the basis $|\widetilde{\alpha_k}\rangle$. The corresponding Bloch Hamiltonian matrix $\widetilde{H}_k$ transforms as

$$\widetilde{H}_k = \begin{pmatrix} U_a & \overline{t_{in}}e^{ik\tau_b} + t_{ex}e^{-ik(1-\tau_b)} \\ t_{in}e^{-ik\tau_b} + \overline{t_{ex}}e^{ik(1-\tau_b)} & U_b \end{pmatrix} \quad \rightarrow \quad \widetilde{H}_{k+\Delta k \frac{\Phi}{\Phi_0}}. \tag{6.44}$$

SciPost Phys. Lect. Notes 67 (2023)

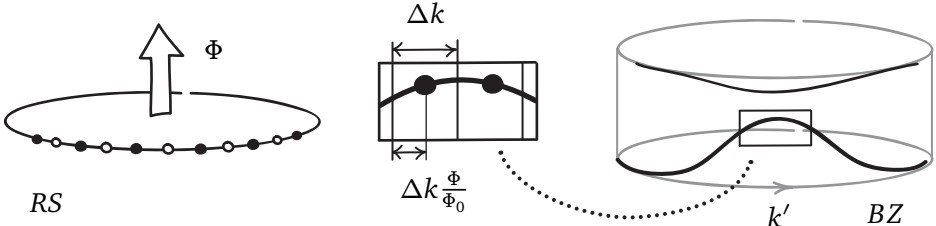

Figure 6.6: Magnetic flux insertion in the real space ($RS$) leads to the shift of momentum eigenstates along the graph of the spectrum in the Brillouin zone ($BZ$), which is parameterized by $k' = k + \Delta k \frac{\Phi}{\Phi_0}$.

The effect of the magnetic field here is similar to that in the case of the particle on a ring (see Sec. 3.1.1). Points that represent momentum eigenvalues are shifted along the curve of the spectrum by the amount proportional to the flux. The difference with the particle on a ring is that here, the periodicity of the crystal in the real space makes the momentum space compact, so the spectrum is also periodic. At the half-filling, all the states in the valence band are occupied. Thus, after the insertion of the flux quantum $\Phi_0$, the spectrum coincides with the initial one.

### 6.4.2 Berry flux and polarization quantum

Consider the Brillouin zone circle as a boundary of some fictitious disc (see Fig. 6.7). Here, the Zak phase can be thought as a Berry flux "threading" this disc: $\gamma = \int_{BZ} \widetilde{A} dk = \Phi_B$. The position of the Wannier charge center inside unit cell is given by $\frac{\Phi_B}{2\pi}$. Thus, we have an interesting symmetry between the real space and the momentum space: the geometric phase acquired in one space leads to the shift of "particles" in the reciprocal space.

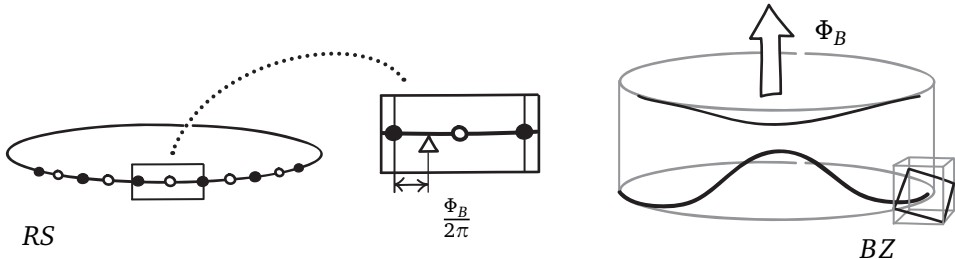

Figure 6.7: The shift of the Wannier charge center (triangle) in the real space ($RS$) is described by the Berry flux threading a fictitious disc bounded by the Brillouin zone ($BZ$).

The difference between real-space magnetic flux and momentum-space Berry flux $\Phi_B$ is that the latter is defined here only modulo $2\pi$. Indeed, since the points of the disc bounded by the Brillouin zone do not physically exist, one can always make on the boundary a large gauge transformation (6.12) with a non-zero winding number (see also Sec. 2.1.3 and Eq. (3.8)). This will result in the new values

$$\gamma' = \gamma + 2\pi m, \quad x_0' = x_0 + m, \quad P' = P + me, \qquad m \in \mathbb{Z}, \tag{6.45}$$

for the Zak phase, the position of the zeroth Wannier center, and the polarization, respectively. So, the value of $P$ is defined modulo the **polarization quantum**, which coincides with the elementary charge $e$ for a one-dimensional system. Formally, all gauge choices for $|\psi_k\rangle$ are equally appropriate, so there is no way to tell which of the possibilities describes the true value of $P$.

One way to deal with this ambiguity is to consider the whole lattice of the Wannier centers, instead of focusing on the particular coordinate $x_0$. Alternatively, this lattice can be thought of as the set of values taken by $x_0$ after all possible gauge transformations. This makes the polarization a *lattice-valued* quantity and allows for interesting behavior that cannot be described by an ordinary vector or scalar. For example, the value of $P$ can change after a cyclic evolution of the Hamiltonian, a situation we will consider in Sec. 7. In Sec. 9.2, we will find a non-trivial solution of the equation $P = -P$, which exists only because of the inherent ambiguity of $P$.

On the other hand, recall that in experiments, one always measures the *difference* between the two values of the polarization. Note that the shift of the lattice of Wannier centers can be described unambiguously. Indeed, one can compute the coordinate $x_0$ in the initial state and then track its value as a function of time until arriving in the final state. The total change in the polarization will be given by Eq. (6.37). The computation requires that the gauge $|\psi_k(t)\rangle$ be smooth for all $k$ and $t$. Thus, if one makes a gauge transformation in the initial state, a similar transformation must be applied for all $t$, so the difference $\Delta P$ remains gauge-invariant. The two-point formula (6.38) will give the same result as Eq. (6.37), provided that the basis sections $|\psi_k(t_i)\rangle$ and $|\psi_k(t_f)\rangle$ can be smoothly connected over the whole time interval.

Finally, note that the polarization quantum is present even in the classical picture described in Sec. 6.1.1, once we assume that the electric charge comes in portions of $e$. Indeed, for a given state of the crystal, imagine taking a single electron from one surface and moving it to the opposite side. As a result, polarization of the crystal is changed by the quantum, while the bulk remained intact.

## 6.5 Summary and outlook

For a finite crystal, the polarization can be computed from the dipole moment of the whole crystal. However, this approach fails when one needs to find the polarization based on the data from the single unit cell with periodic boundary conditions (as in the case of *ab initio* quantum simulations). This problem is solved by the modern theory of electric polarization both on the classical and on the quantum levels. The main takeaways of our discussion are:

- One cannot describe electric polarization in terms of the bulk, cell-periodic charge density.

- The polarization is defined by Eq. (6.6) as a quantity whose rate of change is given by the average current through the unit cell.

- This current-based definition agrees with the experimental methods of measuring electric polarization.

- The electronic contribution to the electric polarization is given by the Zak phase, according to Eq. (6.39).

- The Zak phase measures the coordinate of the zeroth Wannier center and is gauge-invariant modulo $2\pi$.

- The ambiguity of the Zak phase implies that the electric polarization is defined modulo polarization quantum. The changes of the polarization (6.37) are defined unambiguously, provided that the gauge $|\psi_k(t)\rangle$ is smooth for all $k$ and $t$.

The interplay between bulk and boundary physics of polarization, as well as its geometric interpretation, lies at the heart of the theory of topological insulators. In Sec. 7, we will see that a periodic change of polarization is characterized by a topological invariant. Many topological phases can be described by such a process, which is "frozen" in the momentum space. We will consider an example of such phase in Sec. 8.2. Below, we briefly discuss two practical applications of gauge dependence of Wannier functions and comment on possible generalizations.

▷ **Gauge freedom.** As discussed in Sec. 6.2.1, Wannier functions depend on the gauge choice of Bloch eigenstates. A general gauge transformation affects both the shape and position of the Wannier function; however, the position of the zeroth Wannier center associated with an isolated band remains gauge-invariant modulo the lattice constant. This gauge freedom can be used to obtain Wannier functions with desired properties. By choosing the gauge that minimizes their spread, one obtains **maximally localized** Wannier functions [64]. Another option is to make Wannier functions **symmetry-adapted** (see [65] and references therein). For each Wannier function, one considers the subgroup of the full symmetry group that leaves the Wannier center fixed. The symmetry-adapted Wannier functions transform under an irreducible representation of this subgroup, which simplifies symmetry analysis. They also play a key role in the topological quantum chemistry, which will be discussed in Sec. 9.3.

▷ **Multiple bands and Wilson loops.** The Zak phase generalizes to the multi-band case along the lines of the non-Abelian Berry phase introduced in Sec. 3.4, with the derivative $\partial_k$ replaced by the operator $\widetilde{D}_k$. Consider a group of $N$ bands and denote $V$ the eigenstate bundle formed by the vector spaces spanned by the eigenstates from the group. Choose a set of orthonormal state vectors at $k = 0$, and perform the parallel transport of this "frame" through the Brillouin zone. The parallel transported frame will be related to the initial one by a unitary transformation $U$ with eigenvalues of the form $e^{i\theta_n}$, for $n = 1, \ldots, N$. Importantly, the individual "Wannier centers" $\frac{\theta_n}{2\pi}$ become gauge-dependent in this setting (imagine a gauge transformation mixing different states). Still, their sum remains gauge-invariant, and the polarization is defined by

$$P = \frac{e}{2\pi} \sum_n \theta_n \,. \tag{6.46}$$

The gauge choice for the group of bands means choosing a set $\{|\chi_k^n\rangle\}$ of basis sections for the bundle $V$. Suppose that this set corresponds to the maximally localized Wannier functions; then it has an interesting geometric property. It turns out that this set of states is preserved under the parallel transport, in the sense that each state acquires an individual phase factor $e^{i\lambda_n}$ and does not get mixed with other states. The connection matrix with respect to this basis is diagonal and $k$-independent:

$$\widetilde{A}_k^{mn} = \frac{\lambda_n}{2\pi} \delta_{mn} \,. \tag{6.47}$$

The set of phases $\{\lambda^n\}$ is also known as the **Wilson loop spectrum**. The name comes from high-energy physics, where Wilson loop operator is also related to the non-Abelian parallel transport. A detailed discussion of multi-band formalism in the context of electric polarization can be found in the textbook [9]. The lecture notes [66] give an introduction to the machinery of discrete Wilson loops built as chains of projection operators.

▷ **Multipole moments.** The geometric phase formula for the electric polarization has recently been generalized to the case of quadrupole and octupole electric moments [67,68]. For example, a two-dimensional finite crystal with bulk quadrupole moment is characterized by the presence of the corner charges and "edge polarization". The latter can be described as a

polarization of the effective one-dimensional chain directed along an edge; it turns out that its electric polarization decays exponentially, when the effective chain moves into the bulk. The quadrupole moment may be quantized due to the spatial symmetries, in which case its value can be computed using machinery of *nested* Wilson loops. This formalism is based on the geometric phases associated with Wannier functions rather than with Bloch eigenstates. In the absence of the quantization, the values of the bulk quadrupole moment and edge polarization become gauge-dependent, while the corner charges remain gauge-invariant, as shown in Ref. [69] for inversion-symmetric crystals (note that inversion does not put any constraints on the quadrupole moment).

# 7 Charge pumping and topology

Suppose that the Bloch Hamiltonian $\hat{H}_k(p)$ of a one-dimensional two-band crystal undergoes an adiabatic evolution, which is controlled by the parameter $p$. Then the Zak phase (6.22) associated with the valence band eigenstates is also a function of $p$. In this section, we discuss an important special case when the evolution is *periodic*, that is,

$$\hat{H}(p + 2\pi) = \hat{H}(p). \tag{7.1}$$

Consider the trajectory of some Wannier center during one period of the evolution. Since the process is periodic in $p$, the final position of the Wannier center must coincide with the initial one. For an ordinary $\mathbb{R}$-valued scalar, such as a coordinate of a charged particle, this would mean that the total shift of the particle is zero. However, the coordinate of the Wannier center has a different character: due to the gauge ambiguity, it is natural to consider the lattice of all Wannier centers (see Sec. 6.4.2). Now, such periodic lattice can shift by several lattice constants in one direction and yet return to its initial state. If this is the case, the variation $\hat{H}(p)$ is said to describe an **adiabatic charge pump**, also known as the **Thouless pump** [70]. In this section, we will characterize charge pumps by a topological invariant. Then, we will consider what happens at the ends of a finite crystal, which works as a charge pump in the bulk.

## 7.1 Topological invariant of a charge pump

First, we consider charge pumping in a crystal with periodic boundary conditions. This allows us to use the Bloch theory in the momentum space and Wannier functions in the real space. It should be noted, however, that the periodic charge pumping process is not realistic in the context of the electric polarization (but is possible in other systems, as we will discuss in Sec. 7.4). Still, these considerations are of great conceptual importance, as they bridge together the geometric theory of the electric polarization and the realm of topological phases of matter.

### 7.1.1 Examples of charge pumps

We start by defining two charge pumping protocols, which lead to the shift of the zeroth Wannier center by one lattice constant.

▷ **Dimerized charge pump.** One can easily construct a pump out of charge-pumping dimers shown in Fig. 6.2, as described in Sec. 6.1.2. In this protocol, all electrons are always localized, either inside a unit cell or between two unit cells.

▷ **Continuous charge pump.** A more realistic model would include non-zero values of both hopping amplitudes for all $p$. The dimerized protocol suggests that the process of pumping is governed by the combination of the relative strength of the two hopping amplitudes and the sign of the difference $\Delta$ between the on-site potentials. Consider the Hamiltonian (5.13) with the following variation of the parameters:

$$\Delta(p) = -\Delta_0 \cos p \,, \qquad t_{in}(p) = t_0 + \delta \sin p \,, \qquad t_{ex} = t_0 \,. \tag{7.2}$$

The graphs of these functions and the corresponding evolution of the crystal in the real space are shown in Fig. 7.1. Note that since Wannier centers represent an observable quantity, they

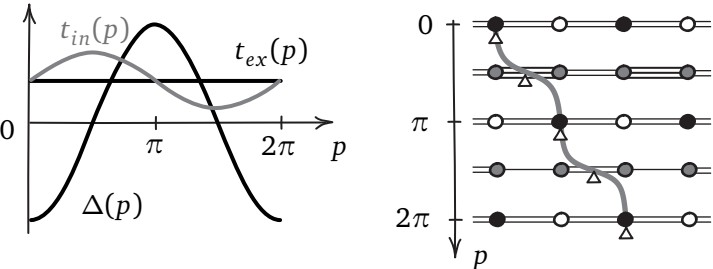

Figure 7.1: LEFT: Evolution of the parameters describing the continuous charge pump defined by Eq. (7.2). RIGHT: Gray curve shows the trajectory of the Wannier center (triangle) during the cycle. Number of the lines between atoms indicates the strength of the bonds. Shading of the circles shows the on-site potentials (the darker, the lower).

must respect symmetries of the crystal. For certain values of the pumping parameter $p$, this allows us to guess the position of the Wannier centers (we will discuss such symmetry constraints in Sec. 9.2.1). For $p = 0$, the chain is inversion-symmetric, with the symmetry center on an atomic site. Thus, the Wannier center can also be placed only on the atom, either of $a$ or $b$ type. Since the electron will be present mostly on the atom with the lower on-site potential, we find that at $p = 0$ the Wannier center has coordinate 0. At $p = \frac{\pi}{2}$, the inversion center lies between atoms, and one can expect that the Wannier center lies at the middle of the stronger bond, that is, inside the unit cell. Similar reasoning gives the positions of the Wannier center for other symmetric states: $p = \pi, \frac{3\pi}{2}, 2\pi$, shown in the right panel of Fig. 7.1 by triangles. The gray curve in shows the actual trajectory of the Wannier center for the protocol (7.2) computed using the PYTHTB package [71], thus confirming our guesses.

### 7.1.2 Quantization of charge transport

Now let us calculate the shift of the charge center during one cycle for a general charge pump with a given two-band Hamiltonian $\hat{H}(k, p)$. Since the Hamiltonian is periodic in both variables, the parameter space has the form of the torus $T^2$. The collection of valence band eigenspaces is a smooth vector bundle $V^v$ over $T^2$. We orient the torus in such a way that the coordinates $(k, p)$ are positively-oriented. Below, we compute certain geometric characteristics of the bundle $V^v$ using the connection defined by $\widetilde{D}_k$, and then discuss the possibility of using another connection (see Sec. 6.2.3).

Let us find the change in the Zak phase as a function of the pumping parameter $p$:

$$\Delta\gamma(p) = \gamma(p) - \gamma(0) = \int_{\mathcal{C}_p} \widetilde{A} dk - \int_{\mathcal{C}_0} \widetilde{A} dk \,, \tag{7.3}$$

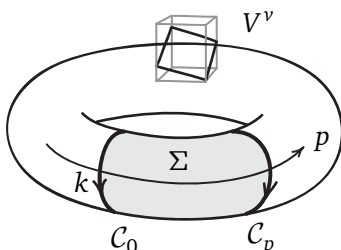

Figure 7.2: Vector bundle $V^\nu$ of valence band eigenstates over the parameter space of a charge-pumping Hamiltonian $\hat{H}(k, p)$. Brillouin zone circle at $p$ is denoted $\mathcal{C}_p$. The surface $\Sigma$ is bounded by two such contours.

where $\mathcal{C}_p$ is the Brillouin zone circle at $p$. Two circles form the boundary of the cylindrical segment $\Sigma$ of the torus. Taking orientation into account, we have $\partial\Sigma = \mathcal{C}_0 - \mathcal{C}_p$. Suppose that the states $|\psi_k(p)\rangle$, or the basis section, are smooth on $\Sigma$. Then, according to the Stokes' theorem (2.30),

$$\Delta\gamma(p) = -\int_{\partial\Sigma} \widetilde{A} dk = -\int_\Sigma \widetilde{f}_{kp} dk dp\,. \tag{7.4}$$

Thus, the shift of the Wannier center during the full cycle is given by

$$\Delta x_0 = \left.\frac{\gamma(p)}{2\pi}\right|_0^{2\pi} = -\frac{1}{2\pi}\int_{T^2} \widetilde{f}_{kp} dk dp = -c(V^\nu)\,, \tag{7.5}$$

and is determined by the topology of $V^\nu$. This gives a "physical" way to understand why Chern number takes integer values: after a periodic evolution, Wannier center can shift only by an integer number of lattice constants. Also note that the variation of the Hamiltonian that leads to a shift of Wannier centers by one lattice constant can be thought of as the insertion of the Berry flux quantum (see Sec. 6.4.2). In contrast, the large gauge transformation, such as one given by Eq. (6.12), corresponds to a relabeling of Wannier functions and is not related to any changes in the Hamiltonian. This illustrates the difference between flux insertion and large gauge transformations mentioned in Sec. 3.1.1.

The key observation of Thouless [70] is that the amount of electric charge pumped in one cycle is quantized in units of $e$. To see this, let us interpret the shift of Wannier centers in terms of polarization. Replacing the pumping parameter $p$ with the time $t$, we obtain from Eqs. (6.5), (6.6), and (6.39):

$$\left.e\frac{\Delta\gamma}{2\pi}\right|_0^{2\pi} = \left.\Delta P\right|_0^{2\pi} = \int_0^{2\pi} \dot{P} dt = \int_0^{2\pi} j(a) dt + \left.P_{dip}\right|_0^{2\pi} = q\,. \tag{7.6}$$

Since the dipole moment $P_{dip}$ is determined by the charge density, we have $P_{dip}(0) = P_{dip}(2\pi)$ after a cyclic evolution. The integral of the current through the boundary $j(a)$ gives the total charge $q$ pumped during the cycle, which is indeed an integer multiple of $e$.

Note that we could have computed the integral of the current $\int j(a) dt$ by Stokes theorem, since it is related to the geometric phase $\phi = \int_{BZ} A(k) dk$, as we discussed in Sec. 6.3.1. This illustrates the fact that the integral of curvature over a *closed* surface does not depend on the choice of connection (see Sec. 4.2.2). On the other hand, if we consider only a part of the pumping cycle, so the surface $\Sigma$ in Eq. (7.4) does not cover the whole torus, then $\Delta\gamma(p) \neq \Delta\phi(p)$. In this case, the integral of curvature is not quantized and does depend on the choice of connection. In physical terms, such discrepancy originates from the possible changes in $P_{dip}$. For yet another look at this situation, note that the distinction between

two geometric phases stems from the difference between two standard bases introduced in Sec. 5.1.4. Only one of the bases, $|\widetilde{\alpha_k}\rangle$, takes into account the spatial positions of the orbitals, or lattice geometry. This distinction is crucial in the context of the electric polarization, while it becomes irrelevant for the topological invariant of a charge pump. A detailed analysis of the role of lattice geometry in the Berry phase-related properties is given in Ref. [72].

## 7.2 End states of finite pumps

Until now, we have focused on the Bloch theory, which is based on the translation invariance and requires periodic boundary conditions. Here, we consider charge-pumping chains with open boundary conditions. We will call such systems **finite charge pumps**, as opposed to effectively infinite crystals with periodic boundary conditions. Starting from the real-space counterpart of the Bloch Hamiltonian $\hat{H}_k(p)$, we obtain its finite version $\hat{H}_f(p)$ by setting to zero the values of all hopping amplitudes between the first and the last unit cells:

$$\hat{H}_k(p) \quad \Rightarrow \quad \hat{H}_f(p). \tag{7.7}$$

Deep in the bulk, the physics described by $\hat{H}_f(p)$ must be similar to that of the periodic crystal: there must be an adiabatic current due to the variation of $p$. But what will happen at the ends, where the bulk current meets the open boundary? In this section, we examine two charge pumping protocols introduced in Sec. 7.1.1 from this point of view.

### 7.2.1 Finite dimerized pump

In the finite dimerized pump, electrons are always localized, and the spectrum can be found without any computations. We begin with a remainder on a general quantum mechanical phenomenon known as the **avoided level crossing**. Suppose that the energies of two eigenstates of a Hamiltonian $\hat{H}(p)$ cross for some value of the parameter $p$, as shown in Fig. 7.3 on the left. One can write

$$\hat{H} = \varepsilon_a |a\rangle\langle a| + \varepsilon_b |b\rangle\langle b|, \tag{7.8}$$

where all terms depend on $p$. In the basis $\{|a\rangle, |b\rangle\}$, the Hamiltonian matrix reads

$$H = \frac{\varepsilon_a - \varepsilon_b}{2}\sigma_z + \frac{\varepsilon_a + \varepsilon_b}{2}\mathbb{I} = h_z\sigma_z + h_0\mathbb{I}. \tag{7.9}$$

We ignore the overall shift of the levels and assume that $\varepsilon_a = -\varepsilon_b$. Then the spectrum is given by

$$\varepsilon_\pm = \pm|h_z|. \tag{7.10}$$

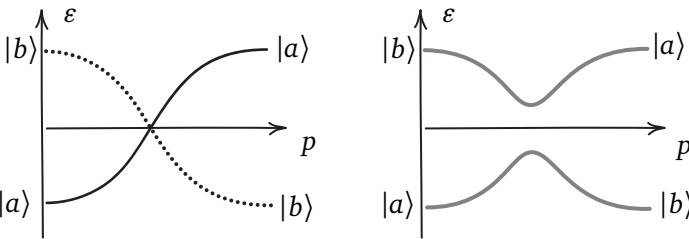

Figure 7.3: LEFT: Two energy levels of a Hamiltonian cross. RIGHT: The presence of the coupling terms in the Hamiltonian leads to the avoided level crossing.

However, this is not a typical situation. In general, the Hamiltonian matrix also contains a coupling term of the form $h_x \sigma_x$. Even if $h_x \ll h_z$, this term opens the gap in the spectrum, since now

$$\varepsilon_\pm = \pm\sqrt{h_z^2 + h_x^2}\,, \tag{7.11}$$

as shown in Fig. 7.3 on the right. So, generically, the levels "avoid" crossing (but can cross in special cases, for example, when the coupling terms are prohibited by symmetry). Here, we consider the evolution of the Hamiltonian controlled by a single parameter; in this case, it enough to have two independent components $h_x, h_z$ of the Hamiltonian to make the degeneracy point unstable. More generally, the stability depends on the combination of the dimension of the parameter space and that of the space of Hamiltonians, as we will discuss in Sec. 10.1.

A familiar example of a system with the spectrum given by Eq. (7.11) is the charge-pumping dimer shown in Fig. 6.2, where the states $|a\rangle$ and $|b\rangle$ are spatially separated, and $h_x$ is non-zero only in the middle of the process. In the avoided crossing scenario, the $|a\rangle$ state turns into $|b\rangle$ state, and vice versa. The electron occupies the ground state and shifts between the orbitals. On the contrary, if there is no coupling, the electron that started at $|a\rangle$ will remain at this orbital and will end up in the high-energy state. So, depending on the presence of the coupling, the electron either shifts in the real space or in the energy.

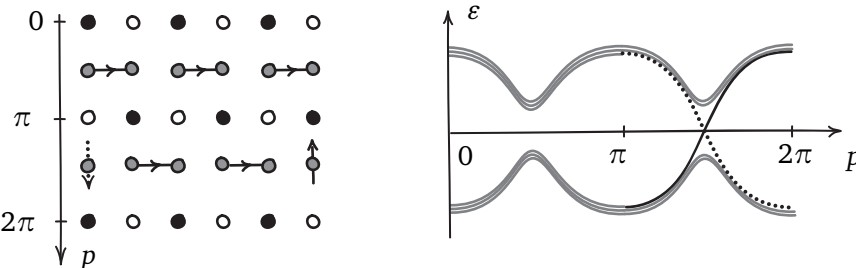

Figure 7.4: Dimerized charge pump of finite length. LEFT: The process of charge pumping in the real space. Shading of circles shows the on-site potentials (the darker, the lower). Horizontal arrows indicate the shift of charge in the real space. Vertical arrows show the shift of the energy of the state. RIGHT: The spectrum of the finite chain as a function of the pumping parameter.

Now consider a finite version of the dimerized charge pump consisting of $N$ unit cells. Such pump for $N = 3$ is shown in the left panel of Fig. 7.4. In the first half of the cycle, we have $N$ charge-pumping dimers. Their spectra have the shape of the avoided level crossing (Fig. 7.3, right). However, in the second half, there are only $N - 1$ dimers that pump charge between unit cells, and a single dimer with zero hopping due to the broken periodic boundary conditions. Its spectrum has the form shown in Fig. 7.3 on the left. It follows that the spectrum of the whole system is highly degenerate and takes the form shown in Fig. 7.4 on the right. In the second half of the cycle, there are two branches of the spectrum that connect lower and higher bands. Importantly, the states that correspond to these branches are localized at the opposite ends of the chain.

What will happen to the electron that occupies the rightmost orbital, in the next cycle? The electron starts in the high-energy state. Recall that in the discussion of a single charge-pumping dimer in Sec. 6.1.2, we focused on the occupied state with low energy. Now note that the high-energy state moves in the opposite direction during the process. It follows that Wannier centers of the high-energy band of the dimerized charge pump move backwards. This

can be expressed in terms of the Chern numbers for valence and conduction bands as

$$c(V^v) + c(V^c) = 0, \tag{7.12}$$

similar Eq. (4.30). Thus, after $N$ cycles, all electrons will be in the high-energy state, and will start to return to the low-energy state via the end state on the left.

### 7.2.2 Finite continuous pump

The end states in the spectrum of the dimerized pump might seem to be an artifact of this particular protocol, where electrons are locked in the orbitals at the ends. What if they will be able to escape once both hopping amplitudes are non-zero? We will find the answer in the spectrum of the continuous charge pump.

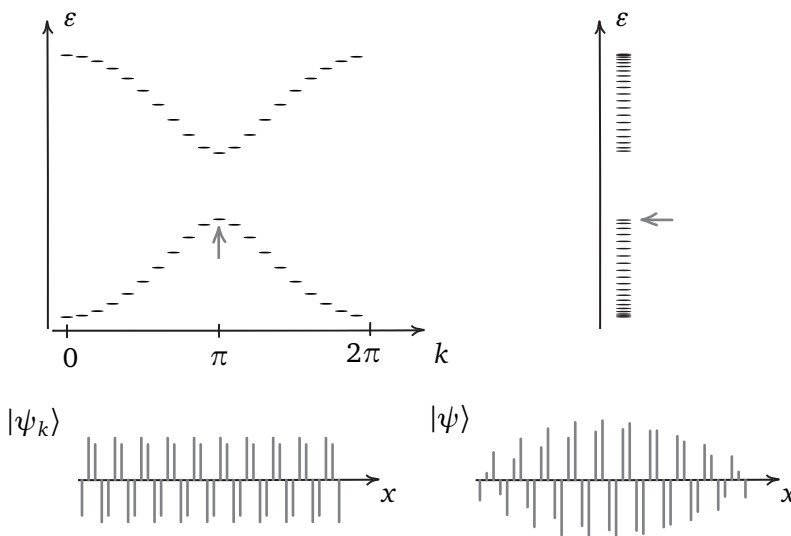

Figure 7.5: LEFT: A typical spectrum of the Bloch Hamiltonian $\hat{H}_k$ of a diatomic chain and the graph of the eigenstate $|\psi_k\rangle$ for $k = \pi$ in the real space. RIGHT: Energy levels for the corresponding finite Hamiltonian $\hat{H}_f$ and the graph of the eigenstate $|\psi\rangle$ whose energy is marked by an arrow. Both eigenstates happen to be purely real in the basis $\{|\alpha_m\rangle\}$, which allows to plot their components.

To begin with, let us see what happens with the eigenstates and the spectrum of the Bloch Hamiltonian once we switch to the open boundary conditions. A finite crystal is essentially a "long molecule", which is not translation-invariant as a whole, so the Bloch theory does not apply. To find eigenstates and their energies, we diagonalize $2N \times 2N$ real-space Hamiltonian matrix numerically, using the PYTHTB package [71]. The crystal momentum $k$ loses its meaning, and bands transform into groups of energy levels, as illustrated in Fig. 7.5. However, away from the boundaries, the Hamiltonian retains the local translation invariance, so we can expect that the eigenstates will resemble those of the Bloch Hamiltonian. The right panel of Fig. 7.5 indicates that this is indeed the case. In the spectrum, there are two groups of energy levels spanning the "projections" of the energy bands of the Bloch Hamiltonian. Individual levels are slightly shifted, as can be seen from the lifted degeneracy between the Bloch states at $\pm k$. Still, we can speak about the bulk gap, which is approximately the same as for the corresponding periodic crystal. The eigenstates with energies lying inside the bands have delocalized wave-like character, with their overall shape altered by the boundary conditions. But, as we will see in a moment, this is not the only type of eigenstates of the finite Hamiltonian.

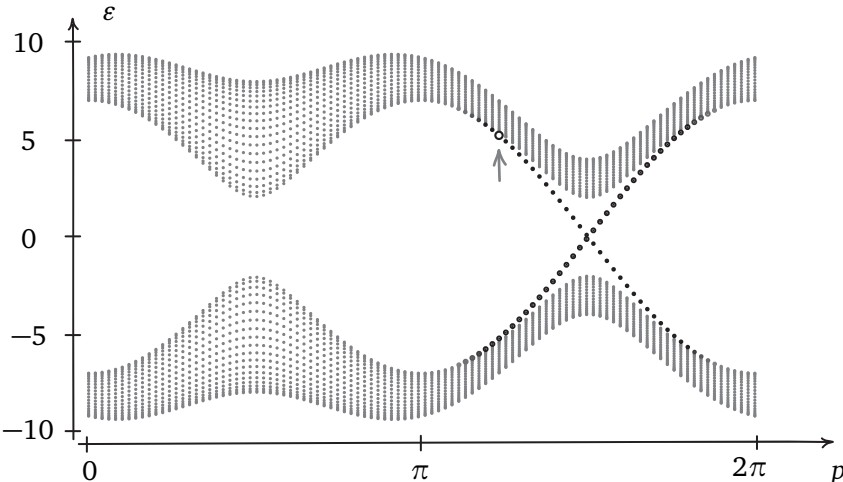

Figure 7.6: The energy spectrum of the finite continuous pump. The parameters of the protocol (7.2) are $\Delta_0 = 7$, $t_0 = 3$ and $\delta = 2$. The chain consists of $N = 20$ unit cells. The arrow marks the state that will be considered below.

Figure 7.6 shows the spectrum of the finite version of the delocalized pump with the protocol (7.2). In the first half of the cycle, the spectrum consists of the bulk bands. In the second half, two mid-gap branches appear. Since the energies of these states lie inside the bulk gap, they cannot be extended Bloch-like waves, and must be localized. The Hamiltonian is translation-invariant in the bulk, so the only inhomogeneity that can support these states is the boundary. Consider the state $|\psi\rangle$ marked by an arrow in Fig. 7.6. The shape of this state in the real space is shown in Fig. 7.7 on the left. The state is indeed localized at the left end of the crystal. The components of the state decay exponentially as it spreads into the bulk. Note also that the state is supported only on the sublattice of $a$ orbitals.

This state can be approximated by the following ansatz [73]:

$$|\psi_a\rangle = \sum_n r^n |a_n\rangle, \tag{7.13}$$

where $r \in \mathbb{C}$ is a fixed complex number. Consider the part of the Hamiltonian (5.4) that consists only of the on-site terms, $\hat{H}_{site} = \sum_{am} U_a |\alpha_m\rangle\langle\alpha_m|$. Then $|\psi_a\rangle$ is its eigenstate:

$$\hat{H}_{site}|\psi_a\rangle = U_a|\psi_a\rangle. \tag{7.14}$$

To make $|\psi_a\rangle$ an eigenstate of the full Hamiltonian, demand that hopping part acts on it as $\hat{H}_{hop}|\psi_a\rangle = 0$. One finds that

$$r = -\frac{t_{in}}{t_{ex}}. \tag{7.15}$$

By construction, the state is supported only on the orbitals of type $a$. One can say that the components of $|\psi_a\rangle$ vanish on $b$ sites as a result of the destructive interference of the hopping amplitudes from the adjacent $a$ sites. Note, however, that the solution is not exact, since the last $b$ site has a single neighbor of the $a$ type. Still, the solution is quite accurate even for the chain with $N = 20$ unit cells, as shown in the right panel of Fig. 7.7.

**Exercise 7.1.** Using an ansatz similar to Eq. (7.13), find an approximate eigenstate localized at the right end of the chain.

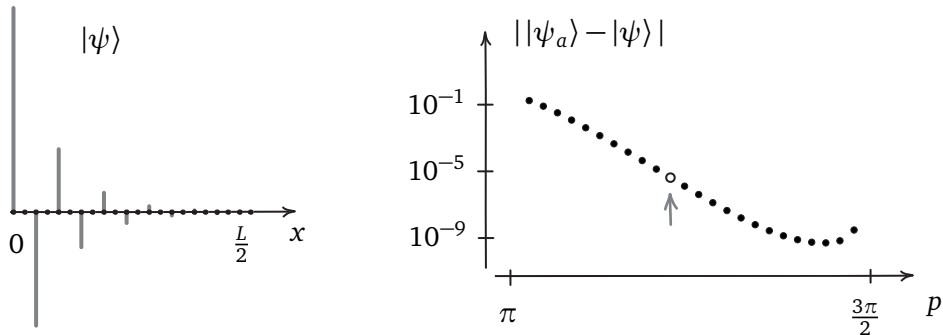

Figure 7.7: LEFT: The real-space representation of the eigenstate $|\psi\rangle$ decaying into the bulk. Dots on the axis represent atomic sites. $L$ stands for the length of the chain. The energy of the state is marked by an arrow in Fig. 7.6. RIGHT: The norm of the difference between the approximate analytical eigenstate $|\psi_a\rangle$ defined by Eq. (7.13) and the exact numerical eigenstate $|\psi\rangle$ for the states in the first half of the decreasing mid-gap branch of the spectrum. Both states are normalized. The marked point corresponds to the state shown in the left panel.

To analyze the localization behavior of the eigenstates, we plot the average position

$$\langle x \rangle = \langle \psi | \hat{x} | \psi \rangle \,, \tag{7.16}$$

for each eigenstate of the charge-pumping Hamiltonian, as shown in Fig. 7.8. For most of the states, the "center-of-mass" $\langle x \rangle$ is located near the middle of the chain; these are extended wave-like states. In the second half of the cycle we can see two branches of states, which are localized at the ends of the chain. They correspond to the mid-gap branches of the spectrum in Fig. 7.6. Note that the shift of the centers towards the ends is pronounced already for $p = \pi$, while the energies still belong to the bulk bands.

## 7.3 Two-band charge pumps: general case

In spite of the differences between the two charge-pumping protocols discussed above, we observe the striking similarities between the spectra of the corresponding finite pumps. Each spectrum has a pair of the peculiar branches with corresponding states localized at one of the ends. For the right end of the chain, these **chiral branches** have the following key features:

- The branch goes across the bulk gap from the valence band to the conduction band.

- The corresponding eigenstates are exponentially localized at the right end of the chain.

In this section, we argue that such chiral branches will appear in any finite chain that pumps charge in the bulk.

### 7.3.1 Stability of number of chiral branches

First, we introduce a formal way to count the number $n_c$ of the chiral branches in the spectrum of a finite pump. Then we will show that $n_c$ enjoys the same sort of topological stability as the Chern number associated with a periodic system.

Recall from Sec. 4.4.2 that the Chern number classifies all non-degenerate two-band Hamiltonians defined over a torus. In particular, this applies to the Hamiltonians of charge pumps

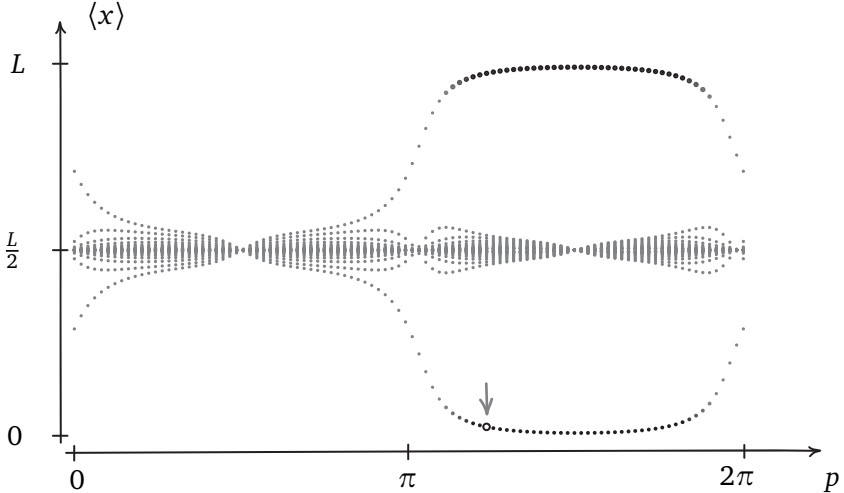

Figure 7.8: The average position $\langle x \rangle$ for the eigenstates of the continuous pump of finite length $L$. In the second half of the cycle, there are states localized near $x = 0$ and near $x = L$. The arrow shows the point corresponding to the state marked in Figs. 7.6 and 7.7.

$\hat{H}_k(p)$, where the Chern number measures the shift of a Wannier center, as discussed in Sec. 7.1.2. Here, we restrict our attention to a smaller set of all **gapped** Hamiltonians, which have a non-zero bulk gap

$$\Delta \varepsilon = \varepsilon^c_{min} - \varepsilon^v_{max}, \qquad (7.17)$$

defined as the difference between the bottom of the conduction band and the top of the valence band for all values of the pumping parameter $p$. If the Hamiltonian is gapped, then it is non-degenerate; the converse, however, is not true: a non-degenerate Hamiltonian can have $k$-dependent term proportional to the identity matrix, which shifts the energy levels such a way that the gap $\Delta \varepsilon$ closes. In other words, a non-degenerate Hamiltonian has only a direct gap, while Eq. (7.17) describes an indirect gap.

Let $\varepsilon_0$ be the value of a constant energy level, which lies inside the bulk gap of a finite charge-pumping Hamiltonian. Consider the chiral branches $\varepsilon(p)$ of the spectrum localized at the *right* end of the chain. Suppose that $\frac{d\varepsilon}{dp}\big|_{p=p_i} \neq 0$ for all points $p_i$ in which $\varepsilon(p_i) = \varepsilon_0$ (if this is not the case, one can slightly shift the constant level). We define the **number of chiral branches** of the spectrum as

$$n_c = \sum_{p_i} \text{sign}\left[ \frac{d\varepsilon}{dp}\bigg|_{p=p_i} \right], \qquad (7.18)$$

where we sum over all intersection points. For example, we have $n_c = 1$ for the two protocols discussed in the previous section.

Now consider the stability: is it possible to change $n_c$ by a smooth deformation of the parameters of the model? At first sight, the chiral branches in Fig. 7.6 are perfect candidates for the avoided level crossing described in Sec. 7.2.1. We only need to introduce the coupling term $\langle \psi_1 | \hat{H} | \psi_2 \rangle$ between the eigenstates corresponding to the chiral branches. However, the two states are exponentially localized at the opposite ends of the crystal. In order to couple these states, we must add long-range hopping amplitudes to the Hamiltonian, which is not realistic for the thermodynamic limit $N \to \infty$.

Thus, we can treat the two end states separately. Note that we used a very specific way to obtain a finite Hamiltonian from a periodic one (7.7). The boundaries of a real crystal



Figure 7.9: Topological stability of the number of intersections $n_c$ between a chiral branch and a constant level $\varepsilon_0$. Panels show the spectrum of states localized at the right end of a finite charge-pumping chain in a narrow energy window around the level $\varepsilon_0$ in the bulk gap. Empty (filled) circles indicate intersections between the spectral branches of end states and the level $\varepsilon_0$ with positive (negative) derivatives $\frac{d\varepsilon}{dp}$. From left to right: a chiral branch; pulling a trivial branch out of the valence band; opening the gap at the crossing point; deforming the chiral branch.

are subject to the **surface reconstruction**, or structural changes of the atomic lattice. This happens because the environment of the atoms at the surface differs from that of the atoms in the bulk. In the language of tight-binding models, this means that we can change the values of the hopping amplitudes and on-site potentials near the surface. These changes will not significantly affect the energies of the bulk states, but can deform the surface spectrum. As illustrated in Fig. 7.9, such deformations cannot change the key features of the chiral branch. Indeed, by continuity, the states obtained by a deformation will be localized at the same end as the original states. The most dramatic transformation that this branch can undergo is the avoided level crossing with a non-chiral branch pulled out one of the bulk bands. However, this only results in the appearance of a new chiral branch. Finally, note that the intersection points $p_i$ can be created or annihilated in pairs with opposite signs of the derivative $\frac{d\varepsilon}{dp}\big|_{p=p_i}$, so the total number $n_c$ remains unchanged. A similar reasoning shows that the number of chiral branches $n_c$ is invariant under the smooth deformations of the *bulk* Hamiltonian, as long as the gap (7.17) is preserved. One says that the chiral branches of the spectrum are protected by the bulk gap.

In fact, the value of $n_c$ is stable against even more radical changes of the surface than those discussed above:

> **Exercise 7.2.** Consider a finite charge-pumping diatomic chain, in which a single $b$ orbital is removed from the right end. In this case, the state given by Eqs. (7.13) and (7.15) becomes an exact eigenstate of the Hamiltonian, as argued in Ref. [73]. For the protocol (7.2), plot the corresponding branch of the spectrum and find $n_c$.

### 7.3.2 Classification of two-band charge pumps

Consider a general two-band charge pump. We can characterize it in two ways: under the periodic boundary conditions, we compute the Chern number $c(V^v)$ of the valence band bundle; for a finite version, there is a number $n_c$ of chiral branches in the spectrum. How are the two integers related? To find the answer, we will analyze the classifications of periodic and finite systems.

Figure 7.10 shows a bird's-eye view of all two-band gapped charge-pumping Hamiltonians. Consider first the periodic systems described by Bloch Hamiltonians. Each point in a white region represents the Bloch Hamiltonian $\hat{H}_k(p)$ of a charge pump. Hamiltonians in the two nearby points have only a slight difference, so that a path in this space corresponds to a

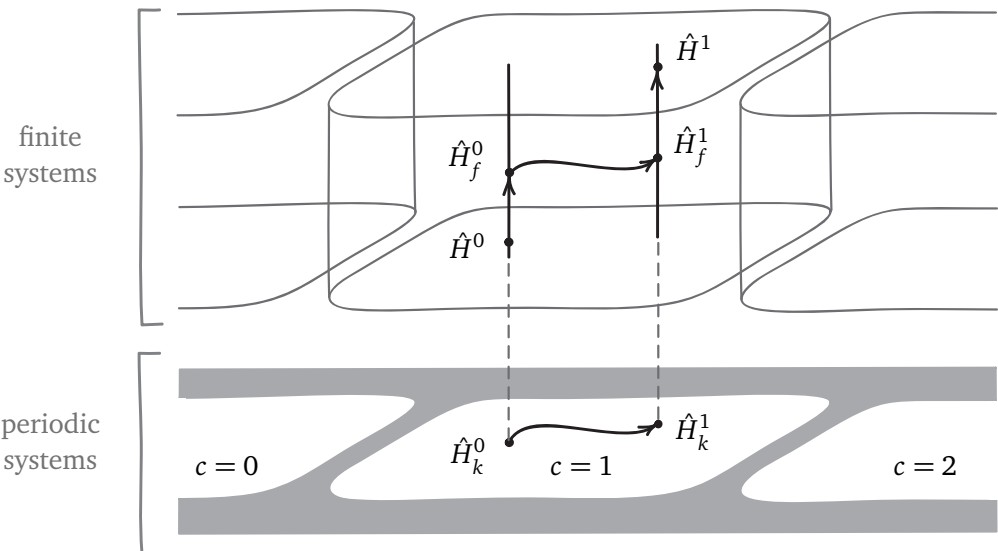

Figure 7.10: Phase diagram of periodic and finite Hamiltonians of charge-pumping chains. Each point represents a charge-pumping Hamiltonian (dependence on $p$ is omitted for brevity). A curve corresponds to a smooth, gap-preserving deformation between the Hamiltonians at the endpoints. For periodic systems, equivalence classes of gapped Hamiltonians (white regions) are labeled by Chern numbers and are separated by gapless Hamiltonians (gray area). Dashed lines connect periodic and finite Hamiltonians. Projecting down along a dashed line corresponds to imposing periodic boundary conditions on a finite Hamiltonian. The truncation of the hopping terms (7.7) describes a map that goes along a dashed line upwards.

smooth deformation between the Hamiltonians at its endpoints. We introduce the following equivalence relation:

$$\hat{H}_k^0(p) \sim \hat{H}_k^1(p) \Longleftrightarrow \text{there is a smooth gap-preserving deformation } \hat{H}_k^0(p) \to \hat{H}_k^1(p). \qquad (7.19)$$

Essentially, this is the relation (4.41) from Sec. 4.4.2 for non-degenerate Hamiltonians adapted to the smaller set of gapped Hamiltonians. Again, the classification has a complete invariant — the Chern number $c(V^v)$ of the bundle of valence band eigenspaces. In particular, any two gapped Hamiltonians with the same associated Chern numbers are connected by a smooth deformation that does not close the bulk gap. At the same time, any smooth deformation connecting two Hamiltonians from the different classes must include closing the gap for some $k$ and $p$. The corresponding path must go through the set of gapless Hamiltonians, represented as a gray area in the figure (we will consider an explicit example of such process in Sec. 8.2.4). Below, we omit the explicit dependence of the Hamiltonian on $p$, for brevity.

Now consider the set of finite systems. The vertical interval above each $\hat{H}_k$ represents all finite charge pumps corresponding to the given periodic Hamiltonian. This set includes the Hamiltonian $\hat{H}_f$ introduced by Eq. (7.7) and other Hamiltonians that differ from it by the terms at the ends of the chain. These additional boundary terms are described by a set of functions (hopping amplitudes and on-site potentials depending on $p$). By deforming them to zero functions, we obtain the Hamiltonian $\hat{H}_f$. Thus, any two finite Hamiltonians corresponding to a given periodic $\hat{H}_k$ can be deformed one into another. This corresponds to a path in the vertical direction in the diagram. One can also follow a path in the horizontal direction, which describes a variation of the bulk, translation-invariant parameters of the model.

We use the same equivalence relation (7.19) as for the periodic systems. Importantly, "the bulk gap" has identical meaning in both settings, since its value $\Delta\varepsilon$ is not affected by what happens at the boundaries in the thermodynamic limit. Note that Fig. 7.10 suggests a very specific relationship between the two classifications: namely, there is exactly one class of finite systems "above" each class of periodic systems. Here, we give a formal argument why this is in fact the case. First, introduce a "projection" $\pi$ from the set of finite Hamiltonians to the set of periodic ones. This function takes a finite Hamiltonian and imposes periodic boundary conditions, discarding any boundary terms. Then, for any two finite Hamiltonians $\hat{H}^0$ and $\hat{H}^1$, we have

$$\hat{H}^0 \sim \hat{H}^1 \quad \Leftrightarrow \quad \pi(\hat{H}^0) \sim \pi(\hat{H}^1). \tag{7.20}$$

Indeed, if two finite Hamiltonians are equivalent, $\hat{H}^0 \sim \hat{H}^1$, they are connected by a path in the diagram. "Projecting" this path onto the horizontal plane gives the deformation between two corresponding periodic Hamiltonians. Conversely, let $\hat{H}^0$ and $\hat{H}^1$ be finite Hamiltonians that correspond to a pair of equivalent periodic Hamiltonians. Note that one can connect them by the following series of deformations shown in Fig. 7.10:

$$\hat{H}^0 \to \hat{H}^0_f \to \hat{H}^1_f \to \hat{H}^1. \tag{7.21}$$

Here, the middle arrow comes from the deformation of the bulk parameters, and two other arrows correspond to the deformations of the boundary terms.

Now note that the map $\pi$ induces a map $\tilde{\pi}$ between the sets of equivalence classes (with notation from Sec. 4.4.1):

$$\tilde{\pi}([\hat{H}]) = [\pi(\hat{H})], \tag{7.22}$$

which is well-defined[13] by virtue of "$\Rightarrow$" part of Eq. (7.20). Since any periodic Hamiltonian $\hat{H}_k$ has the corresponding finite Hamiltonian $\hat{H}_f$ given by Eq. (7.7), the map $\tilde{\pi}$ is surjective:

$$[\hat{H}_k] = [\pi(\hat{H}_f)] = \tilde{\pi}([\hat{H}_f]). \tag{7.23}$$

This map is also injective:

$$\tilde{\pi}([\hat{H}^0]) = \tilde{\pi}([\hat{H}^1]) \quad \Rightarrow \quad [\hat{H}^0] = [\hat{H}^1], \tag{7.24}$$

as the reader may verify. We conclude that the map $\tilde{\pi}$ is a bijection between the sets of equivalence classes (note that the original map $\pi$ is *not* a bijection). In other words, equivalence classes of periodic Hamiltonians are in one-to-one correspondence with those of finite Hamiltonians.

Finally, we introduce a topological invariant of the classification of finite Hamiltonians. It follows from the discussion in Sec. 7.3.1 that the number of chiral edge modes is constant on the equivalence classes of finite Hamiltonians:

$$\hat{H}^0 \sim \hat{H}^1 \quad \Rightarrow \quad n_c^0 = n_c^1. \tag{7.25}$$

To determine the values of $n_c$, we introduce the representative models in each class, as follows. Let $\hat{H}_k(p)$ be the Bloch Hamiltonian of the continuous charge pump (7.2). Define a new Hamiltonian by

$$\hat{H}'_k(p) = \hat{H}_k(mp), \quad m \in \mathbb{Z}. \tag{7.26}$$

---

[13]A brief reminder on mathematical terminology: a map is well-defined, if its value does not depend on the choice made in its definition (here, we use a specific representative $\hat{H}$ of the class $[\hat{H}]$). For a map $f : M \to N$, a pre-image of $n \in N$ is an element $m \in M$ such that $f(m) = n$. The map $f$ is surjective, if each $n \in N$ has at least one pre-image. The map $f$ is injective, if each $n \in N$ has at most one pre-image. A bijective map is both injective and surjective.

If $m = -1$, the charge is pumped in the opposite direction. For $m = 0$, there is no pumping. The case $|m| > 1$ describes the insertion of several pumping cycles into the interval $p \in [0, 2\pi]$. The spectrum of the corresponding finite pump is obtained by juxtaposition of several copies of Fig. 7.6. Note that for each representative, the Chern number associated with $\hat{H}_k$ determines the number of chiral branches in the spectrum of the finite Hamiltonian:

$$-c(V^v) = n_c. \tag{7.27}$$

Recall that both the number of chiral branches $n_c$ and the Chern number $c(V^v)$ are topological invariants, which remain constant under smooth deformations preserving the bulk gap. Thus, either number is constant inside the respective equivalence class of models. It follows that the equality (7.27), while established only for representative models, holds for *any* finite two-band Hamiltonian $\hat{H}$ of a charge pump and for its periodic version $\pi(\hat{H})$. Since $c(V^v)$ is a complete invariant of classification of periodic systems, so is the number $n_c$ in the finite case.

## 7.4  Summary and outlook

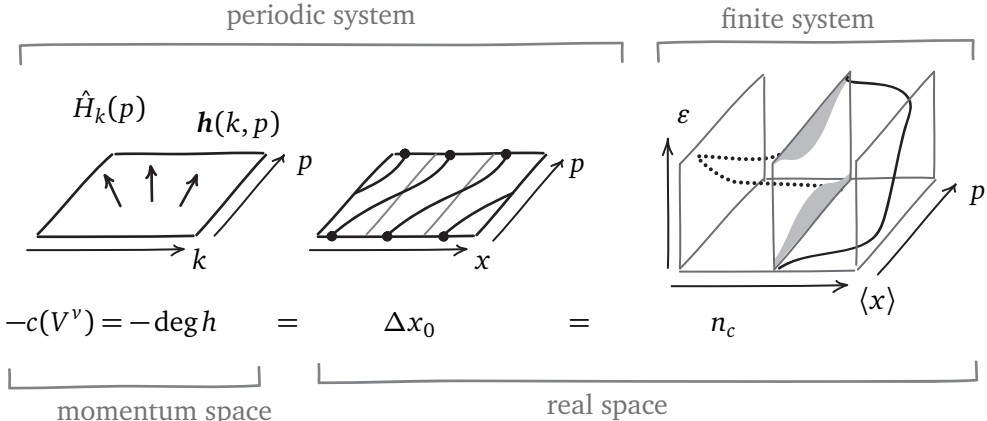

Figure 7.11: LEFT: The Bloch Hamiltonian $\hat{H}_k(p)$ of a charge pump gives rise to the vector field $\boldsymbol{h}(k, p)$. MIDDLE: The shift of Wannier centers $\Delta x_0$ during one pumping cycle is an integer (with lattice constant set to unity). RIGHT: The spectrum of a finite pump $\varepsilon(\langle x \rangle, p)$ as a function of the pumping parameter $p$ and of the center-of-mass coordinate $\langle x \rangle$ of a given eigenstate.

Figure 7.11 summarizes our results for the characteristics of two-band charge pumps defined in various settings. The Bloch Hamiltonian $\hat{H}_k(p)$ depending on the pumping parameter $p$ describes a periodic charge-pumping crystal in the momentum space. It is characterized by the Chern number of the valence band eigenspace bundle $c(V^v)$, which can be computed as a degree of the map $h : T^2 \to S^2$ from the momentum-parameter space to the Bloch sphere. This map arises from the vector field $\boldsymbol{h}(k, p)$ associated with the Bloch Hamiltonian. If we switch to the real space, while keeping the periodic boundary conditions, we will observe the shift $\Delta x_0$ of the zeroth Wannier center in one pumping cycle. Right panel of Fig. 7.11 schematically combines the graphs in Figs. 7.6 and 7.8 for a finite charge-pumping chain. The spectrum has chiral branches, which correspond to the states localized the ends of the chain. Their total number $n_c$ is defined by Eq. (7.18) for the states localized at the right edge.

The key results of the present section link these three pictures together:

- The shift of the Wannier center $\Delta x_0$ during one pumping cycle equals the negative of the Chern number $c(V^v)$ (the sign depends on the orientation conventions).

- The shift of the Wannier center $\Delta x_0$ for a periodic Hamiltonian $\hat{H}_k(p)$ equals the total number $n_c$ of chiral branches in the spectrum of any finite Hamiltonian with the same bulk parameters.

The localized surface states with mid-gap energies were known in the early days of quantum mechanics and band theory [74, 75]. The chiral spectral branches in the charge pumps have a conceptually new property: when we focus on a single end of the crystal, the end state branch *is not periodic* in $p$, despite the periodicity of the Hamiltonian (of course, the complete spectrum is periodic). This combination of localization and chiral nature makes the end states robust against variations of the bulk Hamiltonian that do not close the bulk gap. Under the periodic boundary conditions, such stability is reflected in the topological invariant associated with the Bloch eigenstates. In the next section, we will see how these phenomena can be realized in the momentum space of a two-dimensional crystal.

Below, we discuss the multi-band generalization of the topological invariant, the relationship between charge pumps and driven systems, and experimental realizations of charge pumps.

▷ **Multiple bands.** In the case of multiple occupied bands, a charge pump is characterized by the Chern number (4.48), which can be computed as

$$c = \frac{1}{2\pi} \sum_n \int_0^{2\pi} (\partial_p \lambda_n) dp \,, \tag{7.28}$$

where $\lambda_n$ are the Wilson loop eigenvalues (or centers of maximally localized Wannier functions) introduced in Sec. 6.5. This can be deduced using the generalization of the Stokes theorem (2.30) for the trace of the curvature matrix:

$$\int_\Sigma \text{tr}(F_{12}) dx_1 dx_2 = \int_{\partial\Sigma} \text{tr}(A_\gamma) d\gamma \,. \tag{7.29}$$

Note that the upon taking the trace, the commutator term in Eq. (3.58) vanishes.

▷ **Floquet theory.** Adiabatic topological charge pump is a particular case of a **driven** system: one can interpret the change of the Hamiltonian parameters as a result of an external influence. **Floquet theory** provides a general framework for description of such systems in the case when the driving is periodic. The theory focuses on the unitary time evolution operator. If the evolution is periodic, so $\hat{H}(t + T) = \hat{H}(t)$ for a period $T$, then the evolution operator $\hat{U}(T)$ plays a role analogous to that of the discrete lattice translation in Bloch theory. Accordingly, one can find the states that are transformed by $\hat{U}(T)$ into themselves, up to a phase factor:

$$\hat{U}(T)|\psi_\epsilon\rangle = e^{-i\epsilon T}|\psi_\epsilon\rangle \,. \tag{7.30}$$

The number $\epsilon$ is called **quasienergy**, because of its similarity with the energy of an eigenstate of a time-independent Hamiltonian. The distinction between the energy and quasienergy is topological in the basic sense discussed in Sec. 4.1.1. Like the crystal momentum in the Bloch theory, the quasienergy in the Floquet theory is periodic, so its range is a circle $S^1$ and not the real line $\mathbb{R}$, as in the case of the ordinary spectrum. Owing to this periodicity, each quasienergy band can be characterized by a winding number showing how many times it traverses the "Brillouin zone" of quasienergies. This picture provides an alternative view on the adiabatic charge pump: it turns out that the topological invariant is directly related to the winding of the quasienergy bands. For more details, see Ref. [76].

▷ **Cold atoms.** Ultracold atoms allow one to simulate and study a large variety of quantum phenomena [77]. When put in a periodic optical lattice, the atoms can model behavior of electrons in a crystal, in particular, their geometric and topological properties [78]. The Thouless charge pump was realized experimentally using ultracold atoms in a dynamical optical lattice [79,80]. Essentially, such lattices are standing waves formed by the counter-propagating laser beams. The maxima of intensity of the standing wave serve as potential minima for atoms. The atoms populate the lattice and behave as quantum particles, simulating the physics of a periodic crystal. In the experiments, two such lattices were superimposed, which allowed to control the shape of the potential wells by changing the phase of one of the lattices. Each atom was either localized in a single well or delocalized over a double well, and the pumping was realized in series of the tunneling events, similarly to the dimerized protocol shown in Fig. 7.4. The topological nature of charge pumping was confirmed by the quantized shifts of the center of mass of the atomic cloud. The authors of Ref. [80] also observed the absence of pumping in a protocol with zero Chern number. In the work [79], the authors experimentally verified that the states in the upper band move in the opposite direction to that of the states in the lower band (in agreement with Eq. (7.12)).

# 8 Chern insulators

Band theory of solids originates from solutions of the Schrödinger equation for a single electron in a periodic potential. Despite this oversimplified approach, the band theory gives a qualitative description of certain physical properties of crystals. In particular, it predicts the existence of the energy gap and explains the basic types of electrical conductivity. Metallic conductivity requires the presence of the Fermi surface, which arises as an intersection of the Fermi level with the energy bands. On the other hand, if the Fermi level lies in the gap, the valence bands are completely filled, and the crystal is insulating. But is it possible to further qualitatively distinguish two insulators in the framework of band theory — that is, without taking into account interactions, finite temperature, various types of order, non-equilibrium phenomena, etc.? For around eight decades of band theory, it has been believed that the answer is negative: while two insulators can differ in details of the band energies, they are essentially similar. This can be formalized by the following equivalence relation on the set of gapped Bloch Hamiltonians:

$$\hat{H}^0_{\boldsymbol{k}} \sim_\varepsilon \hat{H}^1_{\boldsymbol{k}} \iff \text{there is a smooth, gap-preserving deformation } \{\varepsilon^0_n(\boldsymbol{k})\} \to \{\varepsilon^1_n(\boldsymbol{k})\}, \quad (8.1)$$

where $\{\varepsilon_n(\boldsymbol{k})\}$ denotes the spectrum of the Hamiltonian. Note that the spectrum is simply a set of functions defined over the Brillouin zone. Any such function can be deformed to another one. So, if two insulators have the same number of occupied bands (and the same total number of bands), they are equivalent.

However, it turns out that one can define another notion of equivalence, which takes into account the whole *matrices*, and not just their eigenvalues. It leads to a non-trivial classification of gapped Hamiltonians, such that two inequivalent Hamiltonians describe crystals with distinct macroscopic physical properties. Below, we will introduce this equivalence relation and discuss the corresponding classification of the two-band Bloch Hamiltonians in one, two, and three spatial dimensions.

## 8.1 Topological equivalence of Bloch Hamiltonians

Consider a $d$-dimensional insulating crystal described by a tight-binding model and let $\hat{H}_{\boldsymbol{k}}$ be the corresponding gapped Bloch Hamiltonian. We assume that the total number of bands is fixed, as well as the number of occupied bands lying below the bulk gap. Define the following

equivalence relation on the set of such Hamiltonians:

$$\hat{H}^0_{\boldsymbol{k}} \sim \hat{H}^1_{\boldsymbol{k}} \quad \Longleftrightarrow \quad \text{there is a smooth, gap-preserving deformation } \hat{H}^0_{\boldsymbol{k}} \to \hat{H}^1_{\boldsymbol{k}}. \qquad (8.2)$$

The relation should look familiar: we discussed similar constructions in Sec. 4.4.2 and in Sec. 7.3.2. The key difference is that now the parameter space is the $d$-dimensional torus $T^d$ formed by the possible values of the crystal momentum $\boldsymbol{k}$.

Form now on, we restrict ourselves to the case of two bands and a single occupied band. Not only is it more simple, but also more rich than the general case, as we will discuss in Sec. 8.5. The matrix of the Bloch Hamiltonian has the form:

$$H_{\boldsymbol{k}} = h_0(\boldsymbol{k})\mathbb{I} + \sum_i h_i(\boldsymbol{k})\sigma_i. \qquad (8.3)$$

The terms proportional to the identity matrix restrict the set of interest (by the possible indirect gap closing, see Sec. 7.3.1), but do not affect the classification. So, we can focus on the vector field $\boldsymbol{h}_{\boldsymbol{k}} \equiv \boldsymbol{h}(\boldsymbol{k})$. Furthermore, the size of the direct gap $|\boldsymbol{h}_{\boldsymbol{k}}|$ at any point $\boldsymbol{k}$ can be tuned to the same value for any two gapped Hamiltonians. Thus, only the direction of the vector $\boldsymbol{h}_{\boldsymbol{k}}$ matters, and we can consider the flat-band Hamiltonians with $|\boldsymbol{h}_{\boldsymbol{k}}|$ set to unity. Then, the equivalence classes correspond to the homotopy classes of smooth maps $h : T^d \to S^2$ from the $d$-dimensional torus of the Brillouin zone to the two-dimensional sphere $S^2$ (cf. Sec. 4.4.2).

As a warm-up, consider the one-dimensional case, $d = 1$. The Brillouin zone is the circle $T^1 = S^1$, and we are interested in the classes of maps

$$h : S^1 \to S^2. \qquad (8.4)$$

Note that the image $h(S^1) \subset S^2$ is a closed loop on the surface of the sphere. Intuitively, any such loop can be shrunk to a point by a smooth deformation; formally, one says that the fundamental group of the sphere $S^2$ is trivial, $\pi_1(S^2) = 0$. In terms of vector fields, this means that any three-dimensional vector field on a circle can be deformed into a constant vector filed. Or, returning to the setting of Bloch Hamiltonians: we can deform any Hamiltonian $H_k$ of a one-dimensional crystal to the atomic limit $H_k = \sigma_z$ with zero hopping amplitudes. Thus, all such Hamiltonians fall into the single class under the relation $\sim$, and we obtain the same trivial classification as with $\sim_\varepsilon$.

## 8.2 From charge pumps to Chern insulators

Now we switch to the case of two spatial dimensions, $d = 2$. What are the equivalence classes of two-band Bloch Hamiltonians $\hat{H}_{\boldsymbol{k}}$ under the relation (8.2)? The good news is that the answer easily follows from our study of the charge pumps. Indeed, note that we can interpret the two-dimensional Hamiltonian

$$\hat{H}_{\boldsymbol{k}} = \hat{H}_{k_x, k_y} = \hat{H}_{k_x}(k_y), \qquad (8.5)$$

as a one-dimensional Hamiltonian $\hat{H}_{k_x}$ depending periodically on the parameter $k_y \in [0, 2\pi)$. Conversely, for any charge pump with Hamiltonian $\hat{H}_k(p)$, we can define a two-dimensional Bloch Hamiltonian by replacing $k \to k_x$ and $p \to k_y$. Below, we re-interpret our findings for charge pumps in this new physical context.

### 8.2.1 Chern number

We know that the Chern number $c(V^v)$ of the valence band eigenspace bundle is a complete invariant of the classification of charge pumps based on the equivalence (7.19). Thus, the

Chern number also classifies the two-dimensional Bloch Hamiltonians under (8.2). The insulators characterized by a non-zero Chern number are called **Chern insulators**. According to Eq. (4.44), the Chern number can be conveniently computed as the degree $\deg(h)$ of the map

$$h : T^2 \to S^2, \tag{8.6}$$

associated with the vector field $\boldsymbol{h}_k$.

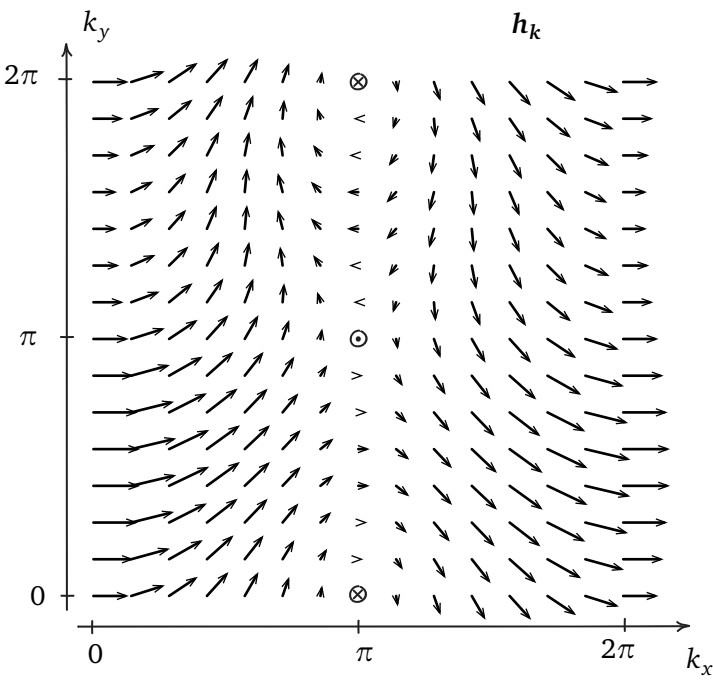

Figure 8.1: Vector field $\boldsymbol{h}_k$ for the Chern insulator obtained from the continuous charge pump (7.2) with parameters $\Delta_0 = 7$, $t_0 = 3$ and $\delta = 2$. The arrows show the projection of $\boldsymbol{h}_k$ to the $\sigma_x$-$\sigma_y$ plane. At the points $(k_x, k_y) = (\pi, 0)$ and $(\pi, \pi)$, this projection vanishes, and the direction of $\boldsymbol{h}_k$ is indicated by $\odot$ for $h_z > 0$ and by $\otimes$ for $h_z < 0$.

For example, take the pumping protocol (7.2) in the diatomic chain with the Hamiltonian Eq.(5.13). After replacing the pumping parameter $p$ with the crystal momentum $k_y$, we obtain the following two-dimensional Bloch Hamiltonian of a Chern insulator:

$$H_k^{CI} = \sigma_x(t_0(1 + \cos k_x) + \delta \sin k_y) + \sigma_y t_0 \sin k_x - \sigma_z \Delta_0 \cos k_y. \tag{8.7}$$

Let us check that the Chern number is indeed non-zero. The vector field $\boldsymbol{h}_k$ of the Hamiltonian $H_k^{CI}$ is shown in Fig. 8.1. We compute the degree of the corresponding map by counting the pre-images, as described by Eq. (4.33). Consider the north pole $NP$ of the sphere $S^2$. Under the map (8.6), it has a single pre-image in the Brillouin zone $T^2$:

$$h^{-1}(NP) = (\pi, \pi). \tag{8.8}$$

Hence, $|\deg(h)| = 1$ and we only need to find the sign. It is determined by whether the map $h$ preservers the orientation near the point in question. We orient the torus $T^2$ by declaring $(k_x, k_y)$ to be positively-oriented coordinates. The sphere $S^2 \subset \mathbb{R}^3 \setminus \{0\}$ sits in the three-dimensional space of Pauli matrices. The latter is oriented by specifying a positively-oriented

basis $\{\sigma_x, \sigma_y, \sigma_z\}$, which determines the orientation of the sphere in the standard way described in Sec. 2.1.4.

> **Exercise 8.1.** Determine the sign of the degree of the map $h : T^2 \to S^2$. Consider also the pre-image of the south pole of the sphere and check that the answer is the same.

> **Exercise 8.2.** Find the parameters of the real-space two-dimensional tight-binding model of the Chern insulator with Bloch Hamiltonian (8.7).

### 8.2.2 Wannier states

In the context of charge pumps, the Chern number measures the shift of the Wannier center during one pumping cycle. Similarly, the flow of the Wannier centers as a function of the crystal momentum is an essential feature of Chern insulators. Moreover, now we can treat both directions in the parameter space on an equal footing.

Let us perform the **partial Fourier transform** $\mathcal{F}_x^{-1}$ of the Bloch states

$$\mathcal{F}_x^{-1}|\alpha_{k_x k_y}\rangle = \frac{1}{\sqrt{N}} \sum_{k_x} e^{-i k_x m_x} |\alpha_{k_x k_y}\rangle \equiv |\alpha_{m_x k_y}\rangle. \tag{8.9}$$

In this way, we obtain the mixed real-momentum space with coordinates $(m_x, k_y)$, where $m_x$ is the unit cell index in the $x$ direction and $k_y$ is the crystal momentum in the $y$ direction. In the mixed basis, the Bloch Hamiltonian becomes a one-dimensional real-space Hamiltonian $H_{m_x}(k_y)$ of the charge-pumping chain. The partial Fourier transform of Bloch eigenstates gives us the **hybrid** Wannier functions

$$|w_{m_x}(k_y)\rangle = \mathcal{F}_x^{-1}|\psi_{k_x k_y}\rangle. \tag{8.10}$$

As discussed in Sec. 7.1.2, the corresponding Wannier charge centers shift by $-c(V^v)$ unit cells along the $x$ direction when $k_y$ increases from 0 to $2\pi$.

Alternatively, we can make the partial Fourier transform in the $y$ direction. How can we characterize the flow of Wannier center in this case? In the momentum space, the Hamiltonian $\hat{H}_{\boldsymbol{k}}$ is described by a vector field $\boldsymbol{h}(k_x, k_y)$. Let us introduce new coordinates $(k_y, -k_x)$. Then the same vector field can be expressed as a new function: $\boldsymbol{h}'(k_y, -k_x) = \boldsymbol{h}(k_x, k_y)$. Since the pairs of coordinates have the same orientation, the degrees of the corresponding maps coincide: $\deg h' = \deg h$. It follows that the chain described by $H_{m_y}(-k_x)$ pumps charge in the positive $y$ direction for *decreasing* $k_x$. In other words, if $k_x$ increases from 0 to $2\pi$, the hybrid Wannier centers shift by $c(V^v)$ unit cells in the $y$ direction.

Another important consequence of the non-trivial topology arises in the real space with the periodic boundary conditions. Define two-dimensional Wannier functions $|w_{m_x m_y}\rangle$ as the inverse Fourier images of the Bloch eigenstates $|\psi_{k_x k_y}\rangle$. The localization properties of the Wannier functions depend on the smoothness of the Bloch eigenstates in the momentum space (for details, see Ref. [81]). Recall that the Chern number $c(V^v)$ determines the number of singularities in any section of the complex line bundle of valence band eigenspaces $V^v$. This means that one cannot choose the global smooth phase for the valence band eigenstates $|\psi_{\boldsymbol{k}}\rangle$ over the whole Brillouin zone. Because of the singularities in $|\psi_{k_x k_y}\rangle$, any Wannier function $|w_{m_x m_y}\rangle$ has the power-law tails. Thus, in a Chern insulator, it is impossible to construct a Wannier state, which would be *exponentially* localized in all spatial directions [82]. This property is used as a basis for a definition of a topological phase with symmetry, as we will discuss in Sec. 9.

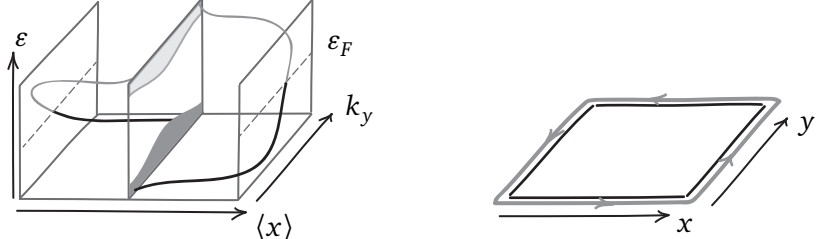

Figure 8.2: From chiral branches of the spectrum to edge modes. LEFT: A typical spectrum of a Chern insulator with $c(V^\nu) = -1$ as a function of $\langle x \rangle$ and $k_y$, where $\langle x \rangle$ is the average position of the eigenstates in the $x$ direction. All states below the Fermi level $\varepsilon_F$ are filled. The intersections of $\varepsilon_F$ and chiral branches describe edge modes. RIGHT: In the real space, the crystal has a unidirectional edge mode running along the boundary.

### 8.2.3 Edge modes and bulk-boundary correspondence

Perhaps, the most dramatic changes in the physical interpretation of our results for charge pumps occur when the crystal has open boundaries. Consider a Chern insulator, which has a finite length in the $x$ direction, but maintains periodicity in the $y$ direction. It is described by a family of one-dimensional real-space Hamiltonians $H_{m_x}(k_y)$ parameterized by $k_y$, which remains a good quantum number. The left panel of Fig. 8.2 shows a typical spectrum of such Hamiltonian for $c(V^\nu) = -1$ (cf. the right panel of Fig. 7.11). The states are additionally resolved by their average position $\langle x \rangle$. The spectrum contains two chiral branches, whose respective eigenstates are localized at the opposite edges of the crystal.

Recall that in Sec. 7.3.1, we artificially introduced a "constant energy level" $\varepsilon_0$ to compute the number of chiral branches. Here, we have a natural choice for the mid-gap energy level: the **Fermi level** $\varepsilon_F$, which divides the energies of the occupied states and the empty ones. Thus, the Fermi level intersects the edge branches of the spectrum. The points of intersection form a (discrete) Fermi surface, and the edges of the crystal become *metallic*. Each intersection point describes an **edge mode**, or an electron wave packet moving along the edge with the **Fermi velocity**

$$v_F = \frac{d\varepsilon(k_y)}{dk_y}, \tag{8.11}$$

where $\varepsilon(k_y)$ is the given chiral branch of the spectrum. Equation (7.18) translates as

$$n_c = \sum_i \text{sign}[v_F^i]. \tag{8.12}$$

In words, $n_c$ measures the difference between the numbers of right- and left-moving modes. In this light, Fig. 7.9 illustrates that during the deformations of the Hamiltonian, the modes appear and annihilate in pairs of opposite moving modes; the total number $n_c$ remains invariant. The sign of $n_c$ depends on the chosen edge and coordinate system. For definiteness, we characterize the crystal by the number

$$n_c: \quad \text{number of modes moving in the positive } y \text{ direction along the edge with } x > 0. \tag{8.13}$$

Now consider a crystal, which is finite in the $y$ direction and is periodic along $x$. Above, we found the direction of the flow of Wannier centers as function of $k_x$. The corresponding

edge modes in the finite geometry have the following velocities: $v_F(k_x) > 0$ for the edge with $y < 0$, and $v_F(k_x) < 0$ for the edge with $y > 0$.

Finally, we break the periodicity in both spatial directions. If the corresponding Bloch Hamiltonian is characterized by $c(V^v) = -1$, the finite crystal has an edge mode on each edge. Together, they form a single chiral edge mode running around the whole boundary, as shown in Fig. 8.2, right. Thus, we have a two-dimensional insulating crystal, whose edge supports a topologically stable metallic mode carrying a dissipationless unidirectional current! It follows that the Chern insulator necessarily breaks the time reversal symmetry, since the symmetry operation reverses the direction of the edge current. In general, the number of such modes is determined by the Chern number:

$$n_c = -c(V^v), \tag{8.14}$$

as we know from Eq. (7.27). Note that the two integers appearing in Eq. (8.14) have entirely different mathematical nature. The Chern number characterizes the eigenspace bundle of Bloch eigenstates in the momentum space, which is defined under the periodic boundary conditions. This is a global topological invariant associated with the bulk Hamiltonian. On the other hand, the edge modes appear in a finite crystal, for which the momentum space does not exist. But somehow the electrons at each point of the boundary of the crystal manage to "know" about the bulk topological invariant.[14] This results in a robustness of the boundary physics: if we cut from the crystal a piece with a complex shape, the chiral mode will cling to the edge, repeating its shape. Eq. (8.14) is a particular instance of the **bulk-boundary correspondence**, a general principle relating a bulk topological property with the physics on the boundary. This principle has numerous applications well beyond the present context of the two-band tight-binding Bloch Hamiltonians. Some of these generalizations will be discussed in Sec. 8.5.

The bulk-boundary correspondence for Chern insulators can also be justified by the following argument based on the charge conservation [84]. Suppose that the band structure has a single chiral edge mode, as in Fig. 8.2, right panel. Only the lower part of the chiral branch of the spectrum is occupied. Now consider the local charge density at the right end of the effective one-dimensional crystal along the $x$ direction as a function of $k_y$. At some value of $k_y$, the chiral branch crosses the Fermi level, and the charge density drops down. But the charge density must be a periodic function of $k_y$, so this change must be compensated by the charge inflow from the bulk. This is described by the shift of the Wannier charge centers, or the Chern number of the bundle $V^v$. The presence of $n_c$ chiral edge modes require that Wannier centers shift by $n_c$ unit cells for $k_y \in [0, 2\pi)$, which gives Eq. (8.14).

> **Exercise 8.3.** Equation (8.14) applies to the boundary between the Chern insulator and the vacuum. More generally, consider two rectangular Chern insulators brought into contact along one edge. Argue that the number of chiral modes at this interface is determined by the difference between the Chern numbers of two insulators.

### 8.2.4 Topological transition

The valence band bundle $V^v$ is an eigenspace bundle for a two-level quantum system defined over a closed two-dimensional parameter space. As noted in Sec. 4.2.3, changing of the Chern number of such bundle requires that the two levels become degenerate at some point. For a two-band insulator, this means closing the bulk gap. Here, we consider an example of such process in some detail, which will be helpful in the context of Weyl semimetals, Sec. 10.3.1.

---

[14]In other words, a global momentum-space property affects the local physics in the real space. Conversely, one can define a local real-space quantity, called **local Chern marker**, which measures $c(V^v)$ once averaged over the interior of the crystal [83].

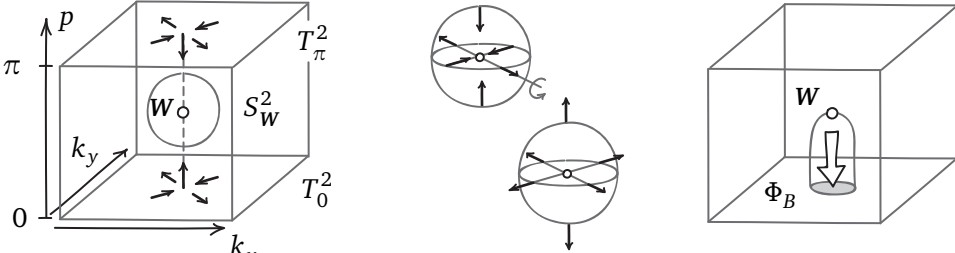

Figure 8.3: LEFT: Topological transition between a Chern insulator and a trivial in-
sulator controlled by the parameter $p$. The degeneracy point is denoted $W$. Bottom
(top) face represents the Brillouin zone torus $T_0^2$ ($T_\pi^2$) of a topological (trivial) insu-
lator. The point $W$ is surrounded by the sphere $S_W^2$. MIDDLE: The $\pi$ rotation of the
vector $\boldsymbol{h_k}(p)$ at each point around the indicated axis turns the vector field near $W$
into an outward-pointing field. RIGHT: The point $W$ acts as a source of the Berry flux.
The net flux $\Phi_B$ pierces all horizontal planes below $W$. This is shown schematically
as a "flux tube" starting at $W$.

We start with the Bloch Hamiltonian of the Chern insulator (8.7). Note that the center of
the Brillouin zone $(k_x, k_y) = (\pi, \pi)$ is the only point where the vector $\boldsymbol{h}$ has the components
$(0, 0, h_z)$ with $h_z > 0$. This allowed us to compute the Chern number by considering this
point as the pre-image of the north pole of the Bloch sphere $S^2$. Now imagine adding to the
Hamiltonian a $\boldsymbol{k}$-independent term $(0, 0, h_z')$ with negative $h_z'$. Physically, this means increasing
the difference between the on-site potentials. This will not affect the in-plane distribution
shown in Fig. 8.1, but will turn $\odot$ in the center into $\otimes$, for a strong enough perturbation.

Specifically, consider the following transformation of the Hamiltonian (8.7)

$$H_{\boldsymbol{k}}(p) = H_{\boldsymbol{k}}^{CI} + (\cos p - 1)\Delta_0 \sigma_z, \tag{8.15}$$

controlled by the parameter $p \in [0, \pi]$. In the center of the Brillouin zone, we have:

$$\boldsymbol{h}_{(\pi, \pi)}(p) = (0, 0, \Delta_0 \cos p), \tag{8.16}$$

so the bulk gap closes at $p = \frac{\pi}{2}$. Note that we have a two-band Hamiltonian $H_{\boldsymbol{k}}(p)$ defined over
each point of the three-dimensional space with coordinates $(k_x, k_y, p)$. It is non-degenerate
everywhere except at the point $W = (\pi, \pi, \frac{\pi}{2})$. The corresponding vector field is shown in
Fig. 8.3. We define the valence band bundle $V^v$ over this mixed momentum-parameter space
excluding the point $W$. For any closed two-dimensional surface $\mathcal{B}$ inside this space, we can
compute the Chern number of the restricted bundle $c(V^v|_{\mathcal{B}})$. We consider two kinds of such
surfaces: the Brillouin zone torus $T_p^2$ for a given $p$ and the sphere $S_W^2$ near the point $W$.

For $p = 0$, we have $c(V^v|_{T_0^2}) = -1$, since this is the valence bundle of the Chern insula-
tor (8.7). The insulator described by $H_{\boldsymbol{k}}(p = \pi)$ is trivial:

**Exercise 8.4.** Consider the vector field $\boldsymbol{h_k}(p = \pi)$ and the corresponding map
$h : T^2 \to S^2$. Deduce that $\deg h = 0$ by counting the pre-images of some point on
the Bloch sphere $S^2$.

Now consider the sphere $S_W^2$. The middle panel of Fig. 8.3 illustrates how the vectors on
the sphere can be deformed into an outward-pointing vector field. Thus, the original field

defines a map $h : S^2_W \to S^2$ with degree $\deg h = 1$. According to Eq. (4.44), the Chern number is

$$c(V^\nu|_{S^2_W}) = 1. \tag{8.17}$$

One can also compute this algebraically. To this end, consider the linearization of the Hamiltonian (8.7) near the degeneracy point in terms of a small vector $\mathbf{q} = (q_x, q_y, q_p)$:

$$h_{\mathbf{W}+\mathbf{q}} \approx (-\delta q_y, -t_0 q_x, -\Delta_0 q_p). \tag{8.18}$$

This can be rewritten in the matrix form

$$H_{\mathbf{W}+\mathbf{q}} \approx \sum_{ij} \sigma_i A_{ij} q_j = \begin{pmatrix} \sigma_x & \sigma_y & \sigma_z \end{pmatrix} \begin{pmatrix} 0 & -\delta & 0 \\ -t_0 & 0 & 0 \\ 0 & 0 & -\Delta_0 \end{pmatrix} \begin{pmatrix} q_x \\ q_y \\ q_p \end{pmatrix}. \tag{8.19}$$

Essentially, the matrix $A$ is the differential of the map between the parameter space and the space of Pauli matrices (cf. Sec. 2.2.3). The sign of the Jacobian $\det A$ tells us whether the map preserves or reverses the orientation of the three-dimensional space (and, consequently, that of the two-sphere). In our case, $\det A = \delta t_0 \Delta_0$ is positive, which confirms the result obtained by the first method (the values of the parameters are listed in the caption of Fig. 8.1).

Since the Hamiltonian $H_{\mathbf{k}}(p)$ for any $p$ above the point $\mathbf{W}$ is connected to $H_{\mathbf{k}}(\pi)$ by a smooth deformation, the average Berry flux through $T^2_p$ vanishes for $p \in (\frac{\pi}{2}, \pi]$. Similarly, the Berry flux through any plane below $\mathbf{W}$ equals $-2\pi$. Thus, the point of degeneracy acts as a source of the Berry flux in the three-dimensional parameter space, as shown schematically in the right panel of Fig. 8.3. Now let us interpret the complex line bundle $V^\nu$ with the Berry connection in terms of the magnetic field, according to (2.15). Then the figure shows a flux tube terminating with the magnetic monopole. Any horizontal plane pierced by the flux represents the Brillouin zone of a Chern insulator (we will come back to this point in Sec. 10.3.3 discussing surface states of topological semimetals). In this sense, a Chern insulator can be described in terms of the "magnetic field in the momentum space", which is created by the magnetic monopoles in the extendend parameter space. We encourage the reader to revisit at this point the analogy discussed in the Preface and interpret it in terms of vector bundles, connections, and sections with singularities.

As a side note, the magnetic structures with the shape schematically shown in the right panel of Fig. 8.3 were recently observed in experiments [85]. Such structures consist of the skyrmion tube ending with the Bloch point, which plays the role of magnetic monopole in magnetically ordered media.

## 8.3 Haldane model

In this section, we consider a modification of the graphene proposed by Haldane in 1988 [86], which is the first model of a Chern insulator. Remarkably, the term "Chern insulator", as well as the general classification problem of topological states of matter appeared nearly two decades later. The motivation behind the model was to reproduce the quantum Hall physics with zero net magnetic field. A detailed account of this physical and historical context can be found in Haldane's Nobel lecture [87].

### 8.3.1 Turning graphene into Chern insulator

Recall that graphene Hamiltonian (5.41) is gapless and contains only terms proportional to $\sigma_x$ and $\sigma_y$. Now let us see what happens when we add $\sigma_z$ terms, which open the gap at the Dirac points $\mathbf{D}_\pm$ and turn graphene into an insulator.

One way to do so is to add staggered on-site potentials $\pm\Delta$ with the opposite signs at the two sublattices. Physically this is realized in the hexagonal boron nitride, which has the same honeycomb lattice as graphene. The Hamiltonian becomes

$$H_{\boldsymbol{k}}^{BN} = \begin{pmatrix} \Delta & f_{\boldsymbol{k}} \\ \overline{f_{\boldsymbol{k}}} & -\Delta \end{pmatrix}. \tag{8.20}$$

To determine the Chern number, consider the Hamiltonian as a map $h : T^2 \to S^2$ from the Brillouin zone torus to the Bloch sphere. For graphene, the map is defined everywhere except at the Dirac points, and the image coincides with the equator. Once the constant term $\Delta\sigma_z$ is added, the image becomes a cap above or below the equator, depending on the sign of $\Delta$. Clearly, the image does not cover the full sphere, so the Chern number must be zero. Or, to put it differently, either the north pole or the south pole of the sphere $S^2$ does not have a pre-image in the Brillouin zone $T^2$. To interpret the triviality in physical terms, note that in the limit $\Delta \gg t$ the system becomes an atomic insulator that consists of decoupled orbitals and thus cannot have any properties of a Chern insulator.

> **Exercise 8.5.** For the case $\Delta \ll t$, consider the images of small contours in the Brillouin zone under the map $h : T^2 \to S^2$. Argue that the Berry curvature is concentrated near the Dirac points and has opposite signs at $\boldsymbol{D}_\pm$. [§8.5]

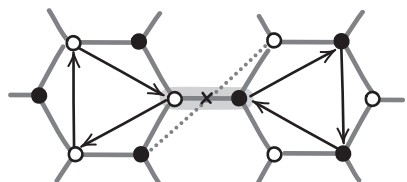

Figure 8.4: Haldane model. Black arrows indicate purely imaginary hopping amplitudes $it_2$ (for clarity, hoppings inside each sublattice are shown separately in two hexagons). Heavy gray lines show the real hopping amplitude $t_1$. Inversion center is marked by the cross. Dashed line connects two orbitals related by the inversion.

To open the gap in a topologically non-trivial way, we follow Haldane [86] and add the imaginary hopping amplitude $it_2$ between the next-nearest neighbors of the same type, as shown in Fig. 8.4. The phase accumulation after hopping around each triangular loop indicates the presence of the magnetic field (see Sec. 6.4.1). However, there is no such phase associated with going around the full hexagon of the honeycomb lattice, since the nearest-neighbor hopping amplitudes are real. Thus, the average magnetic flux through each hexagon must be zero.

The hopping elements between $a$ sites are:

$$it_2\left( |a_{\boldsymbol{m}+\boldsymbol{a}_1}\rangle\langle a_{\boldsymbol{m}}| + |a_{\boldsymbol{m}-\boldsymbol{a}_2}\rangle\langle a_{\boldsymbol{m}}| + |a_{\boldsymbol{m}+\boldsymbol{a}_2-\boldsymbol{a}_1}\rangle\langle a_{\boldsymbol{m}}| \right) + h.c., \tag{8.21}$$

where $\boldsymbol{a}_1$ and $\boldsymbol{a}_2$ are Bravais lattice vectors introduced in Sec. 5.3.1. In the momentum space, one has

$$H_{\boldsymbol{k}}^{aa} = 2t_2(\sin k_1 - \sin k_2 + \sin(k_1 - k_2)) \equiv \Delta_{\boldsymbol{k}}, \tag{8.22}$$

so the Hamiltonian reads

$$H_{\boldsymbol{k}}^{HM} = \begin{pmatrix} \Delta_{\boldsymbol{k}} & f_{\boldsymbol{k}} \\ \overline{f_{\boldsymbol{k}}} & -\Delta_{\boldsymbol{k}} \end{pmatrix}. \tag{8.23}$$

Crucially, the $\sigma_z$ component is now $k$-dependent. It is an anti-symmetric function of momentum, and in particular $\Delta_{D_+} = -\Delta_{D_-}$. Thus, the north pole of the Bloch sphere has now a single pre-image in the Brillouin zone, and $H_k^{HM}$ describes a Chern insulator.

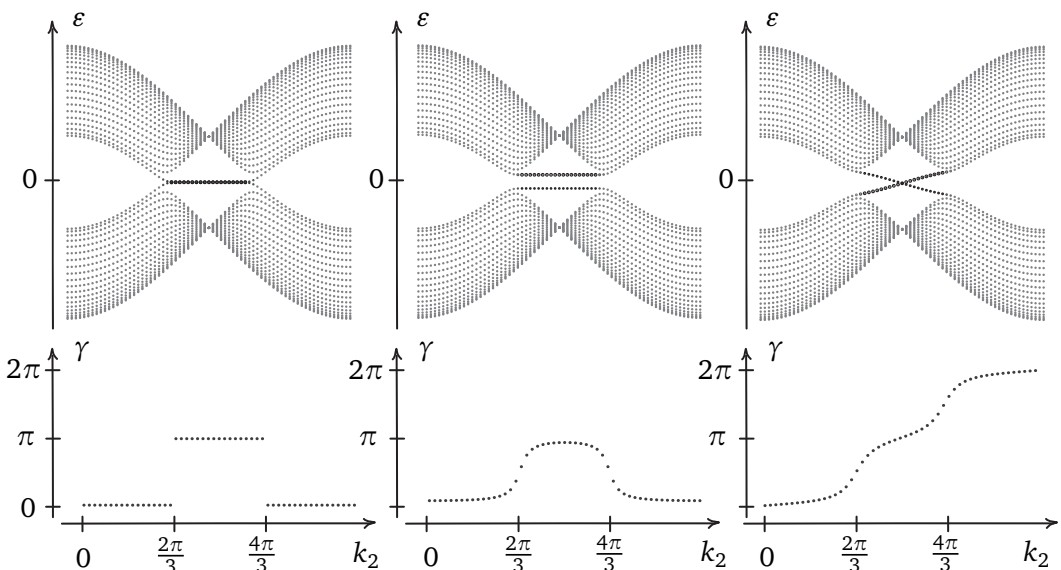

Figure 8.5: TOP ROW: Spectra of honeycomb lattice ribbons with zigzag edges as functions of the crystal momentum $k_2$. Ribbons have finite length ($N = 20$ unit cells) along the lattice vector $\boldsymbol{a}_1$. Heavy dots show the energies of the surface states localized at the right edge. BOTTOM ROW: Zak phase $\gamma$ as a function of the crystal momentum $k_2$. LEFT: Graphene, nearest-neighbor hopping $t = 1$. MIDDLE: Boron nitride, nearest-neighbor hopping $t = 1$, staggered on-site potential $\Delta = -0.15$. RIGHT: Haldane model, nearest-neighbor hopping $t = 1$, next-nearest-neighbor hopping $it_2 = -0.05i$ (the direction of the hopping is shown in Fig. 8.4). Numerical simulations were performed using PYTHTB package [71].

Next, we examine the topological properties of the three models in the mixed position-momentum space. The top row in Fig. 8.5 shows the spectra of ribbons, which are finite in the direction of the lattice vector $\boldsymbol{a}_1$ and are periodic along the other direction. This type of boundary is called **zigzag** for the characteristic shape of the edges. Such ribbon, of width $N = 3$ unit cells, is shown in Fig. 5.2 on the left. In the spectrum of graphene (left panel), the projections of the Dirac cones at $k_2 = \frac{2\pi}{3}, \frac{4\pi}{3}$ are connected by a pair of flat degenerate bands lying in the bulk gap. The corresponding eigenstates are localized at the opposite edges of the graphene ribbon. The middle panel shows the spectrum of the boron nitride, where the bulk gap is opened and the degeneracy of the edge states is lifted. Such surface states can be pushed into the bulk bands by an appropriate surface potential. Finally, the right panel shows the spectrum of the Haldane model. Here, we have a pair of chiral branches connecting the valence band and conduction band, which indicates the presence of non-trivial topology.

The corresponding graphs of the Zak phase along $k_1$ as a function of $k_2$ are shown in the bottom row of Fig. 8.5. Alternatively, these are trajectories of the zeroth Wannier center of a one-dimensional diatomic chain, whose Hamiltonian depends on the external parameter $k_2$. For the purpose of this computation, the origin of the unit cell is shifted to the middle of the horizontal bond connecting $a$ and $b$ atoms. With this choice of the origin, the Zak phase of graphene is quantized in units of $\pi$. In the one-dimensional chain, this corresponds to

the Wannier centers sitting either at the middle of the unit cells or between the two cells. Such precise quantization happens because of symmetry, as we will discuss in Sec. 9.2.1. For a detailed analytical study of the relationship between the Zak phase and edge states in graphene ribbons of general geometry, see Ref. [88].

The Zak phase plot for the boron nitride looks like that for graphene, with the sharp jumps smoothed out by the small perturbation $\Delta$. At first sight, the Zak phase for the Haldane model does not appear to be connected to that of graphene; note, however, that the Zak phase is defined modulo $2\pi$. In the case of graphene, we can choose the different branch for $k_2 \in [\frac{4\pi}{3}, 2\pi]$, replacing the value $\gamma = 0$ with $\gamma = 2\pi$. Then the dependence $\gamma(k_2)$ for the Haldane model can be thought of as the smoothed version of this graph. The flow of the Wannier center for Haldane model corresponds to that of a charge-pumping chain, showing again that we have a Chern insulator.

Finally, let us analyze the symmetry of the models introduced above. Recall from Sec. 5.3.2, that the Dirac points in graphene are protected by the combination of inversion and time-reversal symmetries, so that opening the gap requires breaking of at least one of them. The Hamiltonian of boron nitride $H_k^{BN}$ retains the time-reversal symmetry, but inversion symmetry is broken by the staggered on-site potential. The vectors at $\boldsymbol{h}_k$ and $\boldsymbol{h}_{-k}$ are related by reflection in the $xz$ plane, so the $h_z$ components at the Dirac points must be the same. In contrast, the Haldane model $H_k^{HM}$ has the inversion symmetry, which can be seen from Fig. 8.4: inversion preserves direction of arrows, or the sign of complex hopping amplitudes. However, it breaks time reversal since $\Delta_k \neq \overline{\Delta_{-k}}$. The vectors $\boldsymbol{h}_k$ at opposite momenta are related by $\pi$ rotation about $x$ axis, and $h_z$ components at Dirac points have opposite signs. We conclude that for these two models, the crucial sign choice in $h_z(\boldsymbol{D}_+) = \pm h_z(\boldsymbol{D}_-)$ is determined by symmetry constraints. Note that neither version of the last equality need to hold in the general case; the only symmetry requirement is that Chern insulator must break time-reversal symmetry.

### 8.3.2 Quantum anomalous Hall effect

Part of the title of the paper introducing the Haldane model reads "Model for a Quantum Hall Effect without Landau Levels". Indeed, the physics of Chern insulators is closely related to the quantum Hall effect, which is also characterized by the topological quantization and by the presence of edge modes (for details, see lecture notes [16] and references therein). For this reason, Chern insulators are sometimes referred to as **quantum anomalous Hall** insulators (the Hall effect is called anomalous if it exists without the external magnetic field). Below, we consider one aspect of this relationship: the response to the in-plane electric field.

We start with a simple one-dimensional crystal. Since the Bloch theory requires periodic boundary conditions, it is natural to consider such crystal as a ring. Then the electric field can be applied by varying the magnetic flux that threads the ring. As discussed in Sec. 6.4.1, the insertion of one flux quantum $\Phi_0$ leads to the shift of the momentum values by $\Delta k = \frac{2\pi}{N}$. If the crystal is an insulator, the Fermi level lies inside the gap, so the valence band is fully occupied. After the shift of the states by $\Delta k$, nothing changes: the insulator does not react to the electric field. The metallic crystal shown in Fig. 8.6 has a Fermi surface that consists of two points. They correspond to the modes with the group velocities $v_F$ of opposite signs. The shift of momentum states due to the electric field increases the occupation near one of the points, and reduces the number of electrons at the other. Thus, the electric field induces the current in the metallic crystal.

Consider now the edge state at the top of the cylindrical Chern insulator. The effective one-dimensional crystal at the edge is a metal, but the Fermi surface consists only of a single point. The shift of the momentum values increases the number of occupied states, which seems to violate the charge conservation. This suggests that one should take both edges into account. Since the other edge mode has the opposite chirality, its number of occupied states decreases.

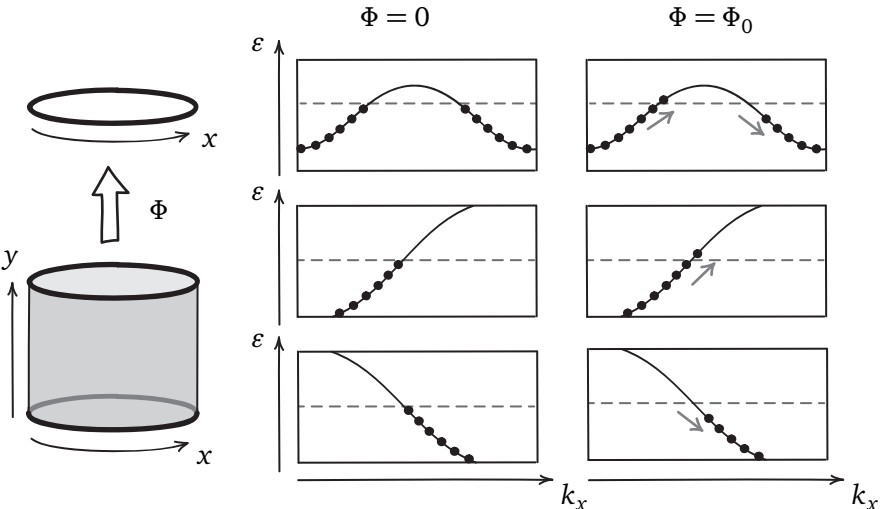

Figure 8.6: Spectra of metallic one-dimensional crystals before and after insertion of the magnetic flux quantum $\Phi_0$. Top row: Spectrum of a one-dimensional metallic circular crystal. Middle (bottom) rows: Spectra of edge modes localized at the top (bottom) edge of a cylindrical Chern insulator. Filled circles indicate occupied states. Arrows show the shift of the occupied states due to the flux insertion.

Thus, the insertion of the flux quantum leads to the transfer of a single electron between the edges. In other words, the electric field in the $x$ direction results in the difference in potential along the $y$ direction, akin to the Hall effect. Note, however, that there is no magnetic field perpendicular to the crystal surface.

A similar observation lies at the core of the Laughlin argument for the quantization of the conductivity in the quantum Hall effect [89]. There, the flux insertion leads to the shift of the real-space position of the bulk wave functions. If the flux is an integer multiple of the flux quantum $\Phi_0$, the amount of shifted charge is quantized in units of $e$.

## 8.4 Three dimensions: Hopf–Chern insulators

The classification of gapped two-band Hamiltonians under the relation (8.2) in the case $d = 3$ corresponds to the classification of maps

$$h : T^3 \to S^2 \,. \tag{8.24}$$

Here, $T^3$ is the three-dimensional torus, which can be thought of as a cube with opposite faces identified. The classification of such maps is due to Pontryagin, who showed that each map is uniquely characterized by four integers:

$$(l, c_1, c_2, c_3), \quad c_i \in \mathbb{Z}, \quad l \in \mathbb{Z}_n, \quad n = 2 \cdot \gcd(c_1, c_2, c_3) \,. \tag{8.25}$$

Here, $l$ takes values in $\mathbb{Z}_n$, or integers modulo $n$, which is the set of equivalence classes defined on $\mathbb{Z}$ by the relation

$$a \sim b \quad \Longleftrightarrow \quad a - b \text{ is a multiple of } n \,. \tag{8.26}$$

In other words, elements of one class have the same remainder after division by $n$. For example, if $(c_1, c_2, c_3) = (2, 0, 1)$, we have $l \in \{[0], [1], [2], [3]\}$; if all $c_i$ are zero, then $l \in \mathbb{Z}$. Below, we discuss the meaning of these integers, following Ref. [90].

### 8.4.1 Three Chern numbers $(c_1, c_2, c_3)$

First, we set $l = [0]$ and interpret the other three numbers $(c_1, c_2, c_3)$ associated with a given map $h : T^3 \to S^2$. To this end, consider the pre-image $h^{-1}(p) \subset T^3$ of some point $p \in S^2$. Recall that in the two-dimensional case, such pre-images were point-like; here, we can expect that they will be one-dimensional curves inside the torus $T^3$. Consider the latter as a cube with periodic boundary conditions. Then these curves can intersect the three faces of the cube, and the numbers $(c_1, c_2, c_3)$ are exactly the numbers of such intersections. To understand these numbers in physical terms, note that each torus $T^2$ inside $T^3$ is itself a Brillouin zone of a two-dimensional insulator. For example, consider the torus $T^2$ given by $k_z = 0$. If the Hamiltonian $\hat{H}(k_x, k_y, 0)$ describes a Chern insulator, the torus $T^2$ contains some point-like pre-images of the point $p \in S^2$. The number of these pre-images determines the degree of the map $h : T^2 \to S^2$ and the Chern number of the corresponding valence band bundle. On the other hand, this number equals $c_3$ as defined above. Thus, the triple $(c_1, c_2, c_3)$ is simply the Chern numbers of three two-dimensional insulators sitting in the planes given by $k_x = 0$, $k_y = 0$, and $k_z = 0$, respectively.

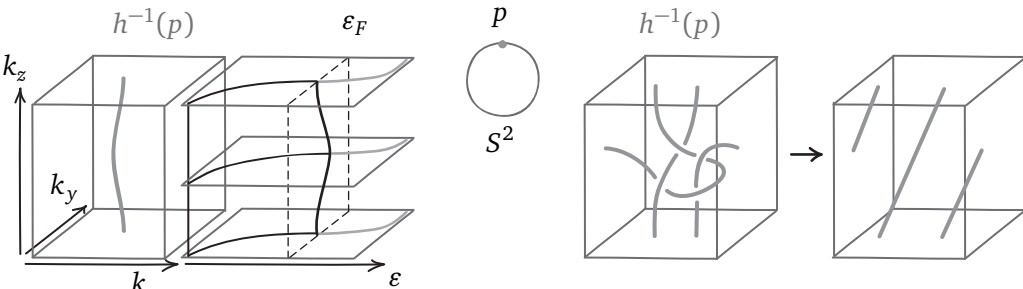

Figure 8.7: LEFT: 3D Chern insulator obtained by stacking 2D Chern insulators along the $z$ direction. The Brillouin zone contains a pre-image $h^{-1}(p)$ of the north pole $p$ of the Bloch sphere $S^2$. A crystal, which is finite in the $x$ direction has surface branches in the spectrum $\varepsilon(k_y, k_z)$ formed by the chiral branches of individual 2D Chern insulators. RIGHT: Any two Hamiltonians with the same numbers $(l, c_1, c_2, c_3)$ are topologically equivalent.

Left panel of Fig. 8.7 shows the simple case of $(c_1, c_2, c_3) = (0, 0, -1)$, which corresponds to stacking of Chern insulators with $c(V^v) = -1$ along the $z$ direction. Let us analyze the boundary states of this three-dimensional crystal. Suppose that the crystal is finite in the $x$ direction, and $k_y, k_z$ remain good quantum numbers. By fixing a certain value of $k_z$, we obtain a two-dimensional Brillouin zone $T^2$. The insulator over the torus $T^2$ has the Chern number $c(V^v) = -1$, so there must be an edge mode. On the right edge, $x > 0$, its energy $\varepsilon(k_y)$ goes up when $k_y$ increases. Imagine plotting this graph for all values of $k_z$. This gives a surface $\varepsilon(k_y, k_z)$ in the spectrum, which intersects the Fermi level plane $\varepsilon_F$ along a curve forming a one-dimensional Fermi surface (we will use a similar argument in discussion of Fermi arcs on the surface of Weyl semimetals in Sec. 10.3.3). In this way, the surface spectrum of the three-dimensional crystal extends that of the Chern insulator. One important difference is that Chern insulators have edge modes on the entire boundary; this need not be the case in three dimensions. For example, the top and bottom surfaces of our crystal do not support boundary modes, despite the crystal is topologically non-trivial.

One can devise a more complex pattern of the pre-images, such as that shown in the right panel of 8.7. However, by virtue of the classification theorem, we know that it can be smoothly deformed into a simple pattern shown on the right. The picture suggests that any three-

dimensional crystal characterized by $(c_1, c_2, c_3)$ and $l = [0]$ can be obtained by stacking Chern insulators along some crystallographic direction, which is indeed the case.

### 8.4.2 Linking number $l$

It turns out that two maps with identical triples $(c_1, c_2, c_3)$ still can be inequivalent. To see this, we will need a *pair* of pre-images of points $p, q \in S^2$. Specifically, consider a stack of Chern insulators with $c(V^\nu) = 1$ in the $z$ direction. If the layers are not coupled, the field $\boldsymbol{h_k}$ does not depend on $k_z$. Then the pre-images of the south pole $p$ and of the point $q$ at the equator are two straight lines, as shown in the left panel of 8.8. Now imagine another crystal with the following dependence of $\boldsymbol{h_k}$ on $k_z$: each vector $\boldsymbol{h}(k_x, k_y)$ rotates around the vertical axis so that the angle of rotation equals $k_z \in [0, 2\pi]$. Then the two pre-images will intertwine (Fig. 8.8, middle panel). This is measured by the **linking number** $l$, the first of the four integers in (8.25). For simplicity, we do not consider the orientation of the pre-images, which determines the sign of $l$. This number for a pair of intertwined loops in three-dimensional space was introduced by Gauss. Moreover, he found a way to express it as an integral, inspired by the early discoveries in electromagnetism. Later, the integral formula was independently derived by Maxwell in the context of knot theory (see Ref. [91] for a historical perspective).

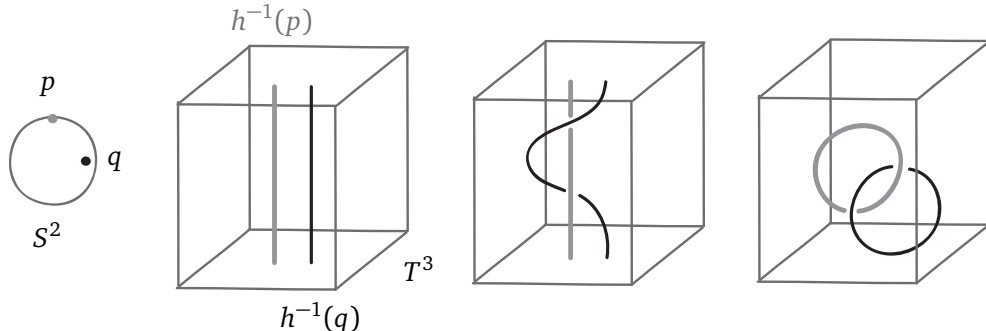

Figure 8.8: LEFT: A pair of pre-images of the points $p, q \in S^2$ in the Brillouin zone $T^3$ of a three-dimensional Chern insulator. MIDDLE: A pair of pre-images with a linking number $l = [1]$. RIGHT: Linking of the two pre-images in the Brillouin zone of the Hopf insulator.

A point $\boldsymbol{k}$ of the Brillouin zone cannot have more than one image $p = h(\boldsymbol{k}) \in S^2$, hence the curves $h^{-1}(p)$ and $h^{-1}(q)$ cannot intersect for $p \neq q$. Thus, the maps characterized by the integers $([1], 0, 0, 1)$ and $([0], 0, 0, 1)$ with pre-images shown in the figure indeed cannot be deformed one into another. However, one should be careful with such intuitive arguments: if the pre-images intertwine *twice*, which corresponds to the case $([2], 0, 0, 1)$, the map can be deformed to the map with no linking $([0], 0, 0, 1)$. See Ref. [90] for a pictorial proof of this equivalence. Formally, for the given values of $(c_1, c_2, c_3)$, the linking number takes values in $\mathbb{Z}_2$, where $[2] = [0]$.

One special case occurs when $(c_1, c_2, c_2) = (0, 0, 0)$. Then the linking number $l$ can be any integer. In this case, the map $h : T^3 \to S^2$ with a non-zero value of $l$ describes the Hamiltonian of a **Hopf insulator** [92]. The name comes from a famous map $S^3 \to S^2$ discovered by Hopf, which also appears in several other physical contexts [93]. In particular, it is closely related to the eigenspace bundle over the Bloch sphere introduced in Sec. 3.3.3. Hopf insulators have a special form of the bulk-boundary correspondence, which is discussed in Ref. [94].

## 8.5 Summary and outlook

One can define Chern insulator using simple ingredients: the momentum space, the Bloch Hamiltonian $\hat{H}_{\boldsymbol{k}}$, its eigenstates $|\psi_{\boldsymbol{k}}\rangle$, and their phase ambiguity. All these concepts were known already in 1930's. However, the Chern insulators were introduced only much later, motivated by the experimental discovery of the precise quantization of conductivity in the quantum Hall effect. Indeed, it would be difficult to invent a concept of Chern insulator from scratch. The phase of an individual eigenstate does not have physical significance; why should our ability to choose the phase *globally* have any? Any Hamiltonian matrix is unitary equivalent to its diagonal form; how can a smooth deformation of matrices be more restricted than a deformation of their spectra? In a sense, the topological band theory is a treasure, which remained hidden on the surface all along.

Let us recapitulate the main results discussed this section:

- In two dimensions, Chern number of the valence band bundle $c(V^v)$ is a complete invariant of classification of two-band insulators under the equivalence relation (8.2) based on the smooth gap-preserving deformations of Bloch Hamiltonians.

- Two-band Chern insulators are crystals described by Hamiltonians with $c(V^v) \neq 0$. They have the following properties:

  - The Hamiltonian of a Chern insulator breaks time-reversal symmetry.
  - In the momentum space, one cannot choose a smooth global phase of Bloch eigenstates. Two-dimensional Wannier states lack exponential localization.
  - In the mixed position-momentum space, there is a flow of hybrid Wannier centers as function of the momentum.
  - A crystal with open boundaries has chiral edge modes, whose number $n_c$ is determined by $c(V^v)$ via the bulk-boundary correspondence (8.14) (when the crystal is surrounded by a trivial insulator).

- Changing the Chern number of a Hamiltonian, which depends on an external parameter, requires closing of the bulk gap at some point. Such degeneracy points in the momentum-parameter space act as sources of the Berry flux.

- Haldane model is a first model of Chern insulator. It consists of the graphene Hamiltonian with an additional complex next-nearest-neighbor hopping amplitude, which breaks the time-reversal symmetry. The corresponding $h_z(\boldsymbol{k})$ term opens the gap at the Dirac points in a topologically non-trivial way.

- Insertion of the magnetic flux quantum into a cylindrical Chern insulator with $|c(V^v)| = 1$ leads to the transfer of a single electron between two boundary circles of the cylinder. This is a manifestation of the quantum anomalous Hall effect.

- In three dimensions, the equivalence relation (8.2) leads to a rich classification of two-band models, which include three-dimensional Chern insulators and Hopf insulators. In the latter case, the topological invariant has the meaning of a certain linking number.

In this section, we focused on the case of two-band tight-binding models. It turns out that our central result — the bulk-boundary correspondence for Chern insulators — holds in much more general settings. In condensed matter physics, it has been established using various techniques, including complex analysis on Riemann surfaces [95], initially developed by Hatsugai in the context of the quantum Hall effect [96]; topology of Green functions [97] based on the approach pioneered by Volovik [98]; dimensional reduction and scattering matrices [99];

advanced mathematical treatment using non-commutative geometry and $K$-theory of operator algebras [100]. The general discussion of these developments lies well beyond the scope of the present notes. Below, we highlight several specific generalizations of our findings and discuss relevant experimental results.

▷ **Multiple occupied bands.**   The bulk-boundary correspondence for two-band Chern insulators generalizes to the multi-band case. Recall that the electric polarization is determined by the average position of multi-band Wannier centers. The Chern number (7.29) gives the net charge transferred along the $x$ coordinate during one "pumping cycle" $k_y \in [0, 2\pi)$. The charge-conservation argument from Sec. 8.2.3 holds verbatim, and the Chern number equals the number of the chiral edge modes.

In terms of the topological classification, two insulators with different total numbers of bands (or different numbers of occupied bands) belong to different classes. In the case of Chern insulators, the distinction turns out to be physically irrelevant. One can thus include a possibility of adding trivial bands into the equivalence relation (8.2). Interestingly, there is a parallel situation in mathematics. Recall from Sec. 4.5 that classification of vector bundles over a base space $\mathcal{B}$ is equivalent to finding homotopy classes of maps from $\mathcal{B}$ to an appropriate classifying space. In such general form, this problem can be very difficult. Things become a bit simpler if one considers equivalence up to addition of trivial bundles. The resulting set of equivalence classes (which has the algebraic structure of a ring) is studied by the $K$-theory of vector bundles [6]. However, the simplification comes at the cost of "resolving power": some distinct homotopy classes can merge into a single class in the framework of $K$-theory. A physical example of this phenomenon is the two-band Hopf insulator: if one adds a trivial valence or conduction band, the Hamiltonian can be deformed to that of a trivial insulator. This is one of the factors underlying the difficulty of realizing the Hopf insulator as a condensed-matter system (which may in principle be overcome in a system of ultra-cold polar molecules in an optical lattice [101]).

▷ **Continuum limit.**   Despite their global nature, topological invariants admit a local characterization, which links the theory of topological matter with high-energy physics. One of important techniques of particle physics is lattice regularization, which replaces the spacetime continuum with a discrete lattice. Any crystal Hamiltonian in condensed matter can be thought of as a result of lattice regularization of some field theory. Conversely, consider an expansion of the Bloch Hamiltonian $\hat{H}_{k+q}$ near the point $k$ in terms of a small wave vector $q$. This gives an effective field theory of particles whose wavelength is much larger than the lattice constant, which is called a continuum limit of the lattice Hamiltonian.

For example, consider the Bloch Hamiltonian (8.15) at $p = \frac{\pi}{2}$. Near the degeneracy point in the center of the Brillouin zone, the dispersion is linear. In the language of high-energy physics, it describes a massless fermion. If we slightly change the value of $p$, the fermion will acquire a mass, encoded in the term proportional to $\sigma_z$. For the values of $p$ just above and below $\frac{\pi}{2}$, the mass term has opposite signs. One can also interpret the changing of $p$ as the motion of the monopole piercing the Brillouin zone of the insulator. Once the monopole crosses the Brillouin zone, the Chern number must change by 1, provided that no other degeneracies appear at other points. This allows one to deduce something about topology of an eigenspace bundle by looking at a single point.

Moreover, this analogy is helpful in the context of the bulk-boundary correspondence. Imagine that such transition occurs in the real space: divide an infinite two-dimensional crystal into two half-planes and set the opposite signs of the mass term in the two regions. Then the boundary will host a massless particle, known as a domain wall fermion in the particle physics (see [1] and references therein). Its spectrum is nothing else but the linearized dis-

person relation of the chiral boundary mode of a Chern insulator. This gives a powerful tool for studying the boundary physics of topological states of matter. Basic examples can be found in many introductory texts on the topic, such as Refs. [10] and [102]. Note that this mixed real-momentum space approach relies on the envelope function approximation, which is described in detail in Chapter 7 of Ref. [8]. For an introduction to the topological matter from the perspective of high-energy physics, see Refs. [13–15].

▷ **Classical waves.** The local description of Chern insulators suggests that the bulk topological invariants and associated boundary modes do not rely on the discrete lattice, and can in principle be present in any wave system.

The oceanic thermocline is an upper layer of relatively warm water in the world ocean. Its dynamics is approximately described by the shallow water model. One of the variables in the model is the Coriolis parameter, which accounts for the coupling between the two in-plane components of the velocity of water due to the Coriolis force. The model predicts two special branches in the spectrum of the planetary waves. They correspond to the Kelvin and Yanai waves, which are trapped near the equator and can propagate only eastwards. Recently it was found that the nature of these waves is in fact topological [103]. The Coriolis parameter has opposite signs in the northern and southern hemispheres and plays the role of the mass parameter, which opens the spectral gap. The topological analysis shows that the equator must support exactly two gapless unidirectional boundary modes. In this way, topological physics plays an important role in the equatorial climate dynamics, including the El Niño phenomenon.

Even without the compact momentum space, the presence of unidirectional boundary modes can be attributed to the topology of the eigenstate bundle of a differential operator. Lectures [104] provide a detailed discussion of this correspondence with an eye towards a deep mathematical result, the Atiyah-Singer index theorem, which relates topological and spectral properties of certain differential operators.

▷ **Amorphous systems.** One can also generalize the case of a periodic crystal by making the arrangement of the lattice sites irregular on a large scale. Such amorphous systems can have non-trivial topology, which manifests itself by the chiral boundary modes and is diagnosed by the local topological markers (see Ref. [105] for a recent overview of this topic).

Here, we briefly discuss the results of an experimental and numerical study of an amorphous gyroscopic metamaterial [106]. For the experiments, the authors construct an irregular array of physical pendulums, each of which contains a gyroscope with the angular momentum pointing along the rod. The neighboring gyroscopes interact with each other by repulsive magnetic forces between the permanent magnets installed on them. Due to this coupling, the precession of one gyroscope around the equilibrium point is transferred to the others, giving rise to the collective precession modes. The spectrum of these modes is found to be gapped. Exciting a single gyroscope with the mid-gap frequency results in a chiral precession mode. The precession propagates along the boundary only in one direction and does not enter the bulk region, indicating that the system is topological.

Like a Chern insulator, the gyroscopic metamaterial breaks the time-reversal symmetry. Intuitively, the sense of rotation of the gyroscopes should determine the chirality of the boundary mode; however, this intuition turns out to be wrong. Numerical simulations of networks of gyroscopes connected by springs show that the topological properties strongly depend on the geometry of the lattice. Here, "geometry" means the characteristic shape of the local environment of a given lattice site. For a particular lattice type, the system has two bulk gaps, which contain boundary modes of opposite chiralities.

▷ **Interactions, disorder, and experiments.**   Finally, we return to the field of condensed matter, now from the experimental viewpoint. A major physical simplification used in the tight-binding models is that they do not include effects of interactions and disorder. Recall that the bulk-boundary correspondence for Chern insulators can be interpreted in terms of the Wannier center flow. Intuitively, if we start from a non-trivial insulator characterized by such flow of charge, and then adiabatically deform it into an interacting or disordered system, the flow must survive. Thus, the topological properties are robust against moderate interactions and disorder, provided that they do not close the bulk gap. Ref. [107] gives a review of interacting topological phases. For an introduction to the effects of disorder, see Week 9 of the course [11].

This heuristic understanding is supported by the experimental evidence of topological quantization in real crystals, which necessarily include lattice imperfections and interactions. The Chern insulators are characterized by the quantization of the Hall conductivity in the absence of the external magnetic field. Since Chern insulators break time-reversal symmetry, the system must be magnetic. In the pioneering work [108], the authors used the doping by magnetic Cr atoms to open the gap in the topological surface states of a time-reversal invariant topological insulator $(Bi, Sb)_2 Te_3$ (we will briefly discuss such systems in Sec. 9.4). When the magnetic moments of dopants on the top and bottom surfaces are aligned due to ferromagnetic ordering, the system becomes a Chern insulator. Recently, the quantum anomalous Hall effect was observed in the intrinsic magnetic material $MnBi_2 Te_4$, which has A-type antiferromagnetic order: the crystal structure consists of layers with alternating direction of magnetic moments, while inside each layer the magnetization is uniform [109]. A moderate external magnetic field is used to induce ferromagnetic order in the whole sample. Nevertheless, the authors demonstrate that the nature of the topological quantization is intrinsic.

Perhaps, the most surprising system hosting the Chern insulator phase is the twisted bilayer graphene on the hexagonal boron nitride substrate [110], which does not contain any magnetic atoms. Moreover, its topological properties are, in a sense, induced by interactions, which can be understood as follows.

1. Recall from Sec. 8.3.1 that the on-site potential $\Delta$ added to the graphene Hamiltonian opens the gap. By results of Exercise 8.5, the Berry curvature is concentrated near the Dirac points. In this context, a neighborhood of a Dirac point is called a valley. In the case of the Hamiltonian (8.20), the curvature has opposite signs in the two valleys, so the Chern number vanishes.

2. In a twisted bilayer graphene, two sheets of graphene form a moiré superlattice. In the resulting Brillouin zone, the bands at the two valleys become independent. There are four degenerate Dirac cones.

3. The hexagonal boron nitride has the same lattice type as graphene, but a different lattice constant. When the graphene bilayer is aligned with the boron nitride substrate, this breaks the inversion symmetry of graphene and opens the gap at the Dirac points.

4. The bands near the Fermi level in the twisted bilayer graphene are nearly flat, so the physics is dominated by the interactions rather than by the kinetic energy. Interactions lift the degeneracy between bands and determine their order in energy.

5. By applying the gate voltage, one can control the filling of the bands. If the Chern numbers of the filled bands do not sum to zero, we have a Chern insulator. From the magnetic perspective, this state is an example of unusual orbital ferromagnetism, or the ordering of current loops rather than spin magnetic moments [111].

The details of these and other experimental realizations of Chern insulators are systematically reviewed in Ref. [112]. Although the Haldane model remains a thought experiment in the

context of condensed matter, it was brought to reality by use of the ultra-cold atoms [113]. The crucial breaking of the time-reversal symmetry was achieved by the circular "shaking" of the whole optical lattice.

# 9 Role of symmetry

Symmetry often determines properties of a physical system and also changes the way we think about the problem. The theory of topological phases is no exception. In this context, we focus on properties that are invariant under smooth, gap-preserving deformations of the Hamiltonian. If the system is symmetric, it is natural to restrict the possible deformations, allowing only those that respect the symmetry. This is formalized by modifying the topological equivalence of Bloch Hamiltonians (8.2), as follows. Two gapped Hamiltonians with a symmetry $S$ are equivalent, if

$$\hat{H}_1 \sim_S \hat{H}_2 \quad \Longleftrightarrow \quad \text{there is a deformation } \hat{H}_1 \to \hat{H}_2 \text{ that preserves } S \text{ and the bulk gap.} \tag{9.1}$$

Recall that in a Chern insulator, one cannot construct exponentially localized Wannier functions. In a crystal with additional symmetries, this transforms into the following definition: a system with symmetry $S$ is **topological**, if there is no exponentially localized Wannier functions respecting the symmetry $S$. The classification of topological matter with symmetries has two goals:

- Find the equivalence classes under the relation $\sim_S$.
- Determine which of the classes are topological.

$$\tag{9.2}$$

The strategy to tackle this problem depends on the type of symmetry. In the case of internal symmetries, the problem is solved by the mathematical methods of homotopy theory. The result is the periodic table of topological insulators and superconductors [114, 115], in which the phases are organized according to their dimensionality and the symmetry class. In Sec. 9.4, we consider one of the entries of the table, the spinful time-reversal symmetry in two dimensions, and provide a heuristic argument for the corresponding topological classification. The case of spatial crystalline symmetries requires different methods. A recent breakthrough in the problem (9.2) is the development of the topological quantum chemistry [116, 117], a theory based on the interplay between symmetry properties of the Bloch eigenfunctions in the momentum space and the localized Wannier orbitals in the real space. We apply a simplified version of this method to inversion-symmetric models in Sec. 9.3. In some cases, the goals of (9.2) can be achieved directly. One example is the inversion symmetry in one dimension, which we discuss in Sec 9.1. The implications of the inversion symmetry on the polarization of the one-dimensional chain will be considered in Sec 9.2.

## 9.1 Three looks at the inversion-symmetric chain

As our first example, we consider the diatomic chain shown in Fig 5.1 with additional inversion symmetry. We choose the inversion center lying in the middle of the unit cell with index $m = 0$, so the inversion acts on the real space orbitals as

$$\hat{\mathcal{I}}|a_m\rangle = |b_{-m}\rangle, \quad \hat{\mathcal{I}}|b_m\rangle = |a_{-m}\rangle. \tag{9.3}$$

This model includes the well-known SSH chain as a special case (as we discuss in Sec. 9.5). Below, we describe the equivalence classes of this system under the relation (9.1) in three complementary pictures.

It should be noted that the inversion symmetry of a diatomic chain can be realized differently: inversion can map each sublattice into itself. This is realized in the continuous charge

pumping protocol shown in Fig. 7.1 on the right, for the values of the pumping parameter $p = \pi n$, where $n \in \mathbb{Z}$. However, in this case inversion does not preserve unit cells, which makes the symmetry operator $k$-dependent and renders analysis cumbersome. We will come back to this setting in Exercise 9.3.

### 9.1.1 Symmetry of Bloch Hamiltonian

To begin with, we examine the constraints put on the form of the Bloch Hamiltonian by the inversion symmetry. We will use the Bloch basis $|\alpha_k\rangle$, which is periodic in the momentum space (as we did in Sec. 5.2). The matrix of the inversion operator is $\sigma_x$, and Eq. (5.25) gives

$$\sigma_x H_k \sigma_x = H_{-k}. \tag{9.4}$$

Recall that the conjugation by $\sigma_x$ acts as the $\pi$ rotation about the $\sigma_x$ axis in the space of Pauli matrices. Thus, the components of the field $\boldsymbol{h}_k$ describing the Hamiltonian must satisfy

$$h_x(k) = h_x(-k), \qquad h_y(k) = -h_y(-k), \qquad h_z(k) = -h_z(-k). \tag{9.5}$$

There are two special points $k^\star$ in the momentum space, which are fixed under inversion. These are solutions of the equation $k = -k$. Since the Brillouin zone is a circle, we have two such points:

$$k^\star = 0, \pi. \tag{9.6}$$

Then $h_y(k^\star)$ and $h_z(k^\star)$ are forced to vanish, so the matrix $H_{k^\star}$ must be proportional to $\sigma_x$. There is more freedom at other points, but the values in the two halves of the Brillouin zone are related by the conditions (9.5). Thus, we can focus only on the interval $k \in [0, \pi]$.

Note that we cannot change sign of the $h_x(k^\star)$ without closing the bulk gap. On the other hand, if two symmetric Hamiltonians have the same signs of $h_x(k^\star)$, they can be deformed one into another without breaking the symmetry and without closing the bulk gap. It follows that a pair of two-band Hamiltonians of the inversion-symmetric chain are equivalent under (9.1) if and only if they have the same signs of $h_x(k^\star)$:

$$\hat{H}^1 \sim_{\mathcal{I}} \hat{H}^2 \quad \Longleftrightarrow \quad \text{sign}[h_x^1(k^\star)] = \text{sign}[h_x^2(k^\star)], \quad k^\star = 0, \pi. \tag{9.7}$$

In total, we have four classes of Hamiltonians: there are two fixed points $k^\star$ and two possible signs of $h_x$ at each $k^\star$. Recall from Sec. 8.1 that without the symmetry, all one-dimensional two-band Hamiltonians are topologically equivalent. In this way, constraints imposed by symmetry refine the topological classification.

To visualize the conditions (9.5), we plot the images of the momentum space under the map $h : S^1 \to S^2$ from the Brillouin zone to the Bloch sphere, defined by the vector field $\boldsymbol{h}_k$. We consider Hamiltonians from only two of four classes, as the other two differ from these by the overall sign change. First, suppose that $h_x(0) > 0$ and $h_x(\pi) > 0$. The simplest Hamiltonian in this class is given by

$$H_k^0 = \sigma_x, \tag{9.8}$$

which we encountered in Exercise 5.2. One can deform it away from $\sigma_x$ at the intermediate points $k \neq k^\star$. In this case, the condition (9.5) requires that the images of the two halves of the Brillouin zone on the Bloch sphere be two symmetric loops with opposite orientation, as shown in Fig. 9.1 on the left.

For the second case, $h_x(0) > 0$ and $h_x(\pi) < 0$, we choose the Hamiltonian $H_k^1$ from Exercise 5.2,

$$H_k^1 = \sigma_x \cos k + \sigma_y \sin k, \tag{9.9}$$

as a representative. Here, the vector $\boldsymbol{h}_k$ undergoes a uniform rotation, making the full turn in the $xy$ plane in the space of Pauli matrices. The symmetry constraints 9.5 imply that any possible deformation is symmetric, so the loop always encloses exactly one half of the Bloch sphere (Fig. 9.1, right).

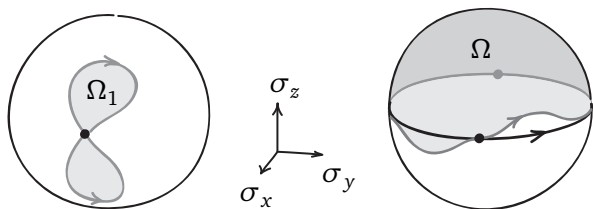

Figure 9.1: Images $h(S^1)$ of the Brillouin zone on the Bloch sphere defined by the two inversion-symmetric Hamiltonians. The black point shows the image of $k = 0$. the LEFT: A typical image describing the Hamiltonian with $h_x(0) > 0$ and $h_x(\pi) > 0$. RIGHT: Two typical images for the case $h_x(0) > 0$ and $h_x(\pi) < 0$.

### 9.1.2 Inversion eigenvalues

Now let us see how the inversion operator acts on a Bloch eigenstate $|\psi_k\rangle$ that corresponds to the valence band. Using Eq. (5.25), we obtain

$$\hat{H}_{-k}(\hat{\mathcal{I}}|\psi_k\rangle) = \hat{\mathcal{I}}\hat{H}_k|\psi_k\rangle = \varepsilon_k(\hat{\mathcal{I}}|\psi_k\rangle). \tag{9.10}$$

Thus, the state $\hat{\mathcal{I}}|\psi_k\rangle$ is an eigenstate of $\hat{H}_{-k}$ with the eigenvalue $\varepsilon_k$. In our simple two-band Hamiltonian, the bands are non-degenerate (that is, there are no level intersections at any given $k$). It follows that the state $\hat{\mathcal{I}}|\psi_k\rangle$ must be proportional to the state at $-k$ in the same band. As the state vectors are normalized, they can only differ by a phase factor:

$$\hat{\mathcal{I}}|\psi_k\rangle = e^{i\phi(k)}|\psi_{-k}\rangle. \tag{9.11}$$

Let us determine the most general form of the function $\phi(k)$. First, note that since $\mathcal{I}^2 = \mathbb{I}$, we have

$$e^{i(\phi(k)+\phi(-k))} = 1 \quad \Rightarrow \quad \phi(k) = -\phi(-k) \mod 2\pi. \tag{9.12}$$

So, $\phi(k)$ is an odd function up to $2\pi$. One example of such function is $\phi(k) = \pi$, which would be an *even* function without the "mod $2\pi$" condition. Since $k^\star = -k^\star$, we have

$$\phi(k^\star) = 0, \pi. \tag{9.13}$$

Now, suppose that we make the gauge transformation

$$|\psi_k'\rangle = e^{i\beta(k)}|\psi_k\rangle. \tag{9.14}$$

Then

$$\hat{\mathcal{I}}|\psi_k'\rangle = e^{i\beta(k)}e^{i\phi(k)}|\psi_{-k}\rangle = e^{i(\phi(k)+\beta(k)-\beta(-k))}|\psi_{-k}'\rangle. \tag{9.15}$$

Thus, the function $\phi(k)$ transforms into

$$\phi'(k) = \phi(k) + \beta(k) - \beta(-k). \tag{9.16}$$

Note that the additional term $\beta(k) - \beta(-k)$ is itself an odd function, which vanishes at $k^\star$. It follows that the values of $\phi(k^\star)$ are gauge-invariant. Away from the points $k^\star$, the function $\phi(k)$ can assume any shape allowed by the condition (9.12): one checks that the gauge transformation with $\beta = \frac{1}{2}(\bar{\phi} - \phi)$ turns $\phi$ into $\bar{\phi}$.

One particular gauge transformation that we will use in a moment is related to the choice of the inversion center. Denote $\hat{\mathcal{I}}_n$ the inversion with respect to the center of the unit cell

with the index $n$. In this notation, our standard inversion operator is $\hat{\mathcal{I}} = \hat{\mathcal{I}}_0$. Then the two operations are related by

$$\hat{\mathcal{I}}_n = \hat{T}_{2n} \circ \hat{\mathcal{I}}_0, \tag{9.17}$$

where $\hat{T}_{2n}$ is the translation by $2n$ lattice constants. Since the Bloch wave $|\psi_k\rangle$ is an eigenstate of the lattice translation, we obtain

$$\hat{\mathcal{I}}_n|\psi_k\rangle = e^{i\phi(k)-2ink}|\psi_{-k}\rangle. \tag{9.18}$$

Thus, if we choose the inversion center in the $n$-th unit cell, this amounts to the gauge transformation $\phi(k) \to \phi(k) - 2nk$, where $n \in \mathbb{Z}$.

At the fixed points $k^\star$ in the Brillouin zone, the Hamiltonian commutes with the inversion operator:

$$\hat{H}_{k^\star}\hat{\mathcal{I}} = \hat{\mathcal{I}}\hat{H}_{k^\star}. \tag{9.19}$$

Hence, one can choose eigenstates of $\hat{H}_{k^\star}$ such that they are also eigenstates of $\hat{\mathcal{I}}$:

$$\hat{\mathcal{I}}|\psi_{k^\star}\rangle = \lambda_{k^\star}|\psi_{k^\star}\rangle. \tag{9.20}$$

From Eq. (9.11) we conclude that the inversion eigenvalues are

$$\lambda_{k^\star} = e^{i\phi(k^\star)} = \pm 1, \tag{9.21}$$

which explains why $\phi(k^\star)$ do not depend on the gauge choice. As in Sec. 9.1.1, we have four cases: two possible eigenvalues at the two points $k^\star$. Since in our chain $\mathcal{I} = \sigma_x$ and $H_{k^\star} \sim \sigma_x$, the four combinations of inversion eigenvalues of $|\psi_k\rangle$ are in one-to-one correspondence with the four classes of Hamiltonians described above.

> **Exercise 9.1.** The equivalence relation 9.1 implies that any smooth deformation between inequivalent systems requires breaking the symmetry or closing the gap. Construct a symmetry-preserving smooth interpolation between the Hamiltonians $\hat{H}^0$ and $\hat{H}^1$ defined in Exercise 5.2. Observe that at some point, the bulk gap closes. Track the inversion eigenvalues of the valence and conduction bands during the process. [§9.3.1, §10]

### 9.1.3 Wannier functions

Finally, we examine how the symmetry affects the Wannier functions (6.8) of the valence band $|\psi_k\rangle$. We will focus on the zeroth Wannier function $|w^0\rangle = \frac{1}{\sqrt{N}}\sum_k|\psi_k\rangle$. The inversion operator acts as follows:

$$\hat{\mathcal{I}}|w^0\rangle = \frac{1}{\sqrt{N}}\sum_k e^{i\phi(k)}|\psi_{-k}\rangle. \tag{9.22}$$

In general, $\hat{\mathcal{I}}|w^0\rangle$ is not a Wannier function, since their shape is largely arbitrary and depends on the gauge choice for $|\psi_k\rangle$. On the other hand, we can benefit from this ambiguity by choosing $\phi(k)$ that is best suited for our needs. For example, suppose that the state $|\psi_k\rangle$ has inversion eigenvalues $\lambda_0 = -1$ and $\lambda_\pi = 1$. Accordingly, we have $\phi(0) = \pi$ and $\phi(\pi) = 0$. A simple linear function passing through these points is $\phi(k) = \pi - k$, which gives

$$\hat{\mathcal{I}}|w^0\rangle = \frac{1}{\sqrt{N}}\sum_k e^{i(\pi-k)}|\psi_{-k}\rangle = \frac{1}{\sqrt{N}}\sum_{k'}(-e^{ik'})|\psi_{k'}\rangle = -|w^{-1}\rangle. \tag{9.23}$$

In a similar way, one finds the appropriate functions $\phi(k)$ for the other three cases, which results in a set of **symmetry-adapted** Wannier functions, shown in Fig. 9.2 in the bottom rows.

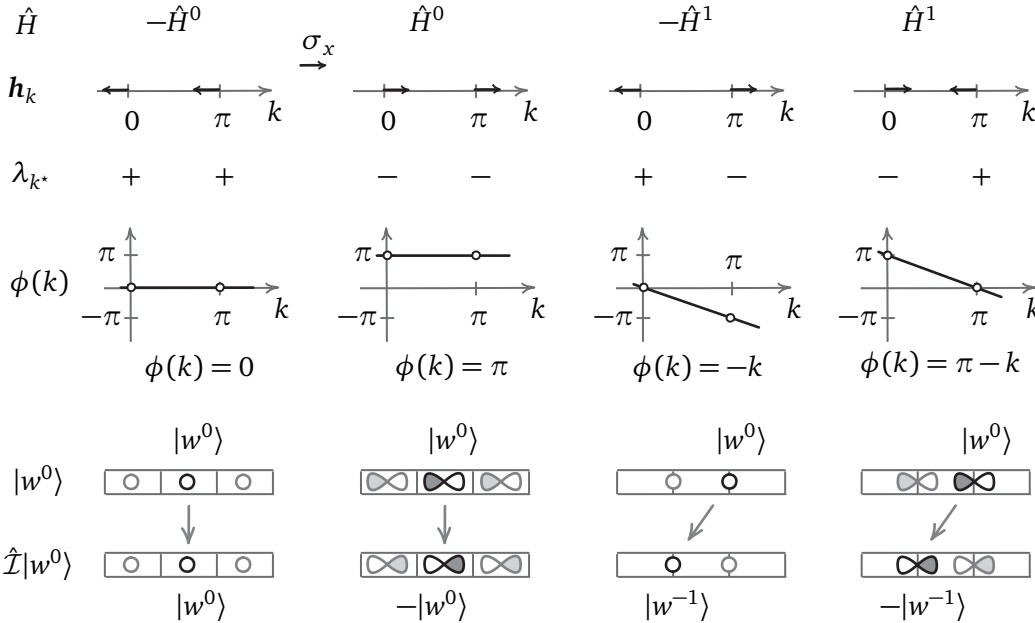

Figure 9.2: COLUMNS: Four classes of two-band inversion-symmetric Bloch Hamiltonians. ROWS: Representative Hamiltonians $\hat{H}$; directions of the vector $\boldsymbol{h}_k$ at $k^\star$; inversion eigenvalues $\lambda_{k^\star}$ of the valence band eigenstates $|\psi_k\rangle$; representative functions $\phi(k)$ defined by Eq. (9.11); action of the inversion operator on the zeroth symmetry-adapted Wannier function $|w^0\rangle$ as a part of the lattice.

Note that in each case, the *lattice* of Wannier functions is mapped by $\mathcal{I}$ to itself, with a possible change of sign. In the first two cases, one of the Wannier functions is placed at the inversion center and thus is transformed into itself. In the third and the fourth cases, all Wannier functions change their positions under inversion. So, the lattice of symmetry-adapted Wannier functions is characterized by the following two properties:

- Whether the Wannier functions change sign under inversion.

- Whether the lattice of Wannier centers contains the inversion center.

As above, in total there are four possible combinations. The first property is represented in the figure by the shapes of the functions resembling *s* and *p* orbitals. Note that these shapes only reflect the symmetry of the functions and do not refer to the atomic orbitals.

The assignment of the functions $\phi(k)$ is not unique, but this does not affect the essential features of Wannier functions. Indeed, if we demand that $\hat{\mathcal{I}}$ maps a Wannier function into another Wannier function, $\phi(k)$ must be linear in $k$ with an integer coefficient $n$. The value $\phi(0)$ determines if there is an additional constant term $\pi$. The parity of $n$ is fixed by whether $\phi(0)$ equals $\phi(\pi)$. Thus, the only freedom left is to change $n$ by an even integer, and we know from Eq. (9.18) that this corresponds to a different choice of the inversion center.

Fig. 9.2 summarizes three equivalent ways to describe the classes of two-band inversion-symmetric Hamiltonians in one dimension. In the language of the vector field $\boldsymbol{h}_k$ associated with the Bloch Hamiltonian, there are four choices of $\text{sign}[h_x(k^\star)]$. They correspond to the four combinations of the inversion eigenvalues $\lambda_{k^\star}$ of the valence band. Finally, there are four types of lattices formed by symmetry-adapted Wannier functions, which are linked to the inversion eigenvalues by the function $\phi(k)$. The symmetry constraints on the Hamiltonian are model-dependent; in contrast, the pictures based on inversion eigenvalues and Wannier

functions are universal, and are widely used in the study of topological phases with crystalline symmetries, as we will see in Sec. 9.3.

To look at a specific example, consider the Hamiltonian (9.9). As a function of $k$, the vector $\boldsymbol{h}_k$ traverses the equator of the Bloch sphere. By the result of Exercise 5.2, the valence band eigenstate is

$$|\psi_k\rangle = \frac{1}{\sqrt{2}}\left(-e^{-ik}|a_k\rangle + |b_k\rangle\right). \tag{9.24}$$

The corresponding zeroth Wannier function reads

$$|w^0\rangle = \frac{1}{\sqrt{2}}\left(-|a_1\rangle + |b_0\rangle\right). \tag{9.25}$$

**Exercise 9.2.**

1. Act with inversion operator on the state (9.24) and find the phase factor $\phi(k)$.

2. Find $\hat{\mathcal{I}}|w^0\rangle$ and express it in terms of other Wannier functions.

3. Use $|w^n\rangle$ to construct the states $|\psi_0\rangle$ and $|\psi_\pi\rangle$ in the real space and check their symmetry properties.

**Exercise 9.3.** Now consider the case when the inversion maps each lattice into itself. Find the $k$-dependent matrix of the inversion operator $\mathcal{I}_k$. Show that only two symmetry types of Wannier functions are realized in this case (the remaining two types can be obtained by using $p$-type atomic orbitals in the chain). [§9.1, §9.2.1]

## 9.2 Polarization and inversion symmetry

Our next goal is to find how the inversion symmetry affects the electric polarization of the diatomic chain. We will show that there are two possible values of the Zak phase, which correspond to the two allowed values of the bulk polarization. Then we apply these results to the chain of finite length. At first sight, the polarization must vanish in this case, since any finite inversion-symmetric charge distribution has zero dipole moment. However, we will see that the symmetry of the Hamiltonian does not necessarily determine the symmetry of the charge distribution, which in fact depends on how the eigenstates are filled with electrons.

**Warning:** The value of the Zak phase depends on the choice of the origin in the unit cell. In Sec. 6, we considered the diatomic chain with the origin at the orbital of $a$ type, so the coordinates of the orbitals were $\tau_a = 0$ and $\tau_b = \frac{1}{2}$ (see Sec. 5.1.4). In this section, we use another convention, which reflects the inversion symmetry: we set the origin at the inversion center in the middle of the unit cell, and the orbitals have coordinates $\tau_a = -\frac{1}{4}$ and $\tau_b = \frac{1}{4}$.

### 9.2.1 Quantization of Zak phase

The presence of inversion symmetry restricts possible values of the Zak phase. We will demonstrate this in two ways, one more visual and the other more formal.

First, we note that the charge density $\rho_{am}$ is symmetric under inversion. Indeed, note that the *real-space* Hamiltonian $\hat{H}$ commutes with the inversion $\hat{\mathcal{I}}$, which acts on the orbitals according to (9.3). Thus, one can find the simultaneous eigenstates of both operators. The

Bloch states $|\psi_k\rangle$ are eigenstates of $\hat{H}$, but are not eigenstates of $\hat{\mathcal{I}}$ unless $k = k^\star$. To fix this, we define the new states for $k \neq k^\star$ by

$$|\psi_{k\pm}\rangle = \frac{1}{\sqrt{2}}\left(|\psi_k\rangle \pm \hat{\mathcal{I}}|\psi_k\rangle\right).$$ (9.26)

One checks that the states $|\psi_{k\pm}\rangle$, together with $|\psi_{k^\star}\rangle$, are in fact eigenstates both for $\hat{H}$ and for $\hat{\mathcal{I}}$. Since each state makes an inversion-symmetric contribution to the charge density (6.15), the distribution $\rho_{am}$ is inversion-symmetric.

The symmetry of the charge density implies that the dipole moment of the unit cell $P_{dip}$ vanishes (recall that the inversion center lies in the middle of the unit cell). It follows from Eq. (6.28) that two Berry connections coincide $\widetilde{A}_k = A_k$, and the Zak phase equals the Berry phase of a two-level system. It thus can be easily found from the solid angle spanned by the image of the Brillouin zone on the Bloch sphere. Two typical situations are shown in Fig. 9.1. In the first case, if a single loop encloses a solid angle $\Omega_1$, we have for the Zak phase:

$$\gamma = \frac{\Omega_1}{2} - \frac{\Omega_1}{2} = 0.$$ (9.27)

In the second case the solid angle always equals $2\pi$, so that

$$\gamma = \frac{\Omega}{2} = \pi.$$ (9.28)

We conclude that under the symmetry constraints given by Eq. (9.5), the Zak phase is quantized and can be either 0 or $\pi$.

Next, we derive the quantization algebraically, following Ref. [118]. Note that the inversion operator does not depend on $k$ and satisfies $\hat{\mathcal{I}}^2 = \mathbb{I}$. Using Eq. (9.11), we obtain:

$$\widetilde{A}_k = i\langle\psi_k|\widetilde{D}_k|\psi_k\rangle = i\langle\psi_k|\hat{\mathcal{I}}\widetilde{D}_k\hat{\mathcal{I}}|\psi_k\rangle = i\langle\psi_{-k}|e^{-i\phi}\widetilde{D}_k e^{i\phi}|\psi_{-k}\rangle = -\widetilde{A}_{-k} - \partial_k\phi.$$ (9.29)

Then the Zak phase can be computed by dividing the Brillouin zone into two halves related by the inversion:

$$\gamma = \int_{-\pi}^{\pi}\widetilde{A}_k dk = \int_{-\pi}^{0}\widetilde{A}_k dk + \int_{0}^{\pi}\widetilde{A}_k dk = \int_{0}^{\pi}\widetilde{A}_{-k}dk + \int_{0}^{\pi}\widetilde{A}_k dk = -\int_{0}^{\pi}(\partial_k\phi)dk.$$ (9.30)

Thus, the Zak phase is given by

$$\gamma = \phi(0) - \phi(\pi) = \begin{cases} 0, & \lambda_0 = \lambda_\pi \\ \pi, & \lambda_0 \neq \lambda_\pi \end{cases},$$ (9.31)

where the phases are understood modulo $2\pi$ and $\lambda_{k^\star}$ are the inversion eigenvalues at the fixed points. For the case discussed in Exercise 9.3, the Zak phase quantization is proved in Ref. [119].

One can use the geometric argument to understand the quantization of the Zak phase in graphene, which we encountered in Sec. 8.3.1. Recall that we considered the Zak phase associated with the effective one-dimensional Hamiltonian $H_{k_1}(k_2)$, which depends on the crystal momentum $k_2$ as an external parameter. In Sec. 5.3.2 we discussed the combined action of inversion and time reversal, $\mathcal{I} \circ T$, which is local in the momentum space and forces the vector $\boldsymbol{h_k}$ to lie in the $xy$ plane in the space of Pauli matrices. Note that like inversion, $\mathcal{I} \circ T$ makes the charge distribution inversion-symmetric, so the Zak phase is again related to the solid angle. Thus, the value of $\gamma(k_2)$ is determined by the number of revolutions made by the vector $\boldsymbol{h_{k_1}}(k_2)$ in the $xy$ plane as it evolves for $k_1 \in [0, 2\pi)$. It follows that the Zak phase for graphene shown in Fig. 8.5 can be found simply by looking at the vector field $\boldsymbol{h_k}$ shown in the middle panel of Fig. 5.2, as the reader should verify.

SciPost Phys. Lect. Notes 67 (2023)

### 9.2.2 Polarization of periodic and finite chains

Now let us interpret the quantization of the Zak phase physically, in terms of the electric polarization. Recall that the Zak phase $\gamma$ measures the coordinate of the zeroth Wannier center, and the polarization $P$ is given by its dipole moment. Since the Wannier functions are related by lattice translations, the value of $\gamma$ determines the position of the whole lattice of Wannier centers. The quantization of the Zak phase can be understood classically, as a consequence of the symmetry constraint put on this lattice. Suppose that we start with a crystal formed by ionic cores. Due to the discrete translation symmetry, there is a lattice of inversion centers which lie, in our convention, in the middle of unit cells. How can we add the lattice of charge centers in an inversion-symmetric way? There are two possibilities: the charge centers must coincide with inversion centers or must be shifted by one half of the lattice constant. In the first case, $\gamma = 0$, and in the second case we have $\gamma = \pi$. According to Fig. 9.2, the position of the charge centers is determined by whether the inversion eigenvalues $\lambda_{k^\star}$ at $k^\star = 0, \pi$ coincide, in agreement with Eq. (9.31).

The picture of the lattice formed by the charge centers gives a visual interpretation of the fact that the bulk polarization is naturally defined as a phase rather than a vector. Indeed, the polarization changes sign under the inversion; and any reflection-odd vector must vanish in a symmetric system:

$$\boldsymbol{v} = -\boldsymbol{v} \quad \Rightarrow \quad \boldsymbol{v} = 0 \, .$$

In contrast, the phase is defined modulo $2\pi$, and

$$\gamma = -\gamma \quad \Rightarrow \quad \gamma = 0, \pi \, .$$

This corresponds to the two possible values

$$P = 0 \quad \mathrm{mod} \; e \, , \qquad P = \frac{e}{2} \quad \mathrm{mod} \; e \, , \tag{9.32}$$

defined modulo the polarization quantum. In both cases, the inversion maps the lattice of charge centers into itself.

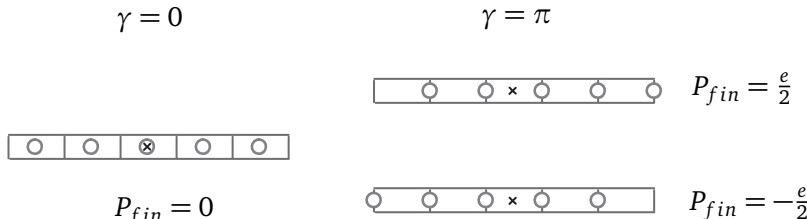

Figure 9.3: Charge center distributions in a finite inversion-symmetric chain. LEFT: When $\gamma = 0$, the charge distribution is symmetric, which gives a zero dipole moment. RIGHT: If $\gamma = \pi$, no inversion-symmetric charge distribution exists at half-filling. Two possible values of the polarization differ by the polarization quantum.

Now consider a finite inversion-symmetric crystal of length $L$ shown in Fig. 9.3. As we discussed in Sec. 6.1.1, a finite system has a well-defined dipole moment and thus a fixed value of the polarization. Let us interpret the predictions of the bulk theory in this context. In this discussion, we assume that the crystal consists of an integer number $N$ of unit cells, so that the ends of the crystal coincide with the cell boundaries.

A finite crystal has a single inversion center in the middle of the chain, which we use as the origin. Suppose that the positive charge distribution of the ionic cores is symmetric and does not contribute to the dipole moment. If the Zak phase vanishes, $\gamma = 0$, the electronic charge density is also symmetric under inversion, which gives the zero total dipole moment.

On the other hand, if $\gamma = \pi$, we cannot put $N$ charge centers into $N$ unit cells while preserving the symmetry. After filling $N - 1$ states in the bulk, we are left with two states at the ends and only a single electron. Thus, at the charge neutrality the lattice of the occupied charge centers must break the inversion symmetry, which is known as the **filling anomaly** (see Ref. [120] and references therein). Depending on which of the end states is filled, we have the dipole moment $d = \pm \frac{eL}{2}$ and the polarization $P_{fin} = \pm \frac{e}{2}$. Observe that both of the values agree with the bulk theory, which gives $P = \frac{e}{2} \mod e$. We conclude that in the finite case, the polarization restores its vectorial nature, but in a way compatible with the bulk theory based on the geometric phase.

Finally, we note that the non-zero value of the polarization may or may not be accompanied by the edge states. In general, the boundary itself breaks the inversion symmetry, so the end states can be pushed to the bulk bands by an appropriate surface potential. However, one can see such states in several models discussed above, for example, in graphene for $k_2 \in [\frac{2\pi}{3}, \frac{4\pi}{3}]$, as shown in Fig. 8.5. A detailed discussion of the symmetry conditions for the surface states can be found in Ref. [121].

## 9.3 Idea of topological quantum chemistry

Above we showed that there are four distinct classes of Hamiltonians of two-band inversion-symmetric chains. However, finding the classes is only the first part of the classification problem (9.2). For this system, the second — topological — part is easy: the representative Hamiltonians $\pm \hat{H}^0$ and $\pm \hat{H}^1$ have localized Wannier functions, so none of the classes is topological, according to our definition.

Inversion symmetry in one dimension is probably the simplest case; for other symmetries and dimensionalities, both parts of (9.2) can lead to much more difficult mathematical problems. Recall that in a Chern insulator, the absence of localization of Wannier functions stems from the singularities in the Bloch eigenstate $|\psi_k\rangle$ due to the non-trivial topology of the corresponding complex line bundle. One can adapt this to the present context and ask: is it possible to find a smooth section of the valence band bundle that satisfies symmetry constraints similar to Eq. (9.11)? For the internal symmetries, such as time reversal, one can define topological invariants, which answer this question directly [10]. There are also topological invariants tailored for some spatial symmetries, for example, the mirror Chern number for crystals with mirror symmetry [122]. The topological phases existing due to spatial symmetries are called **crystalline topological insulators** (see Ref. [123] for an introduction).

However, for a general crystalline symmetry, this approach is hopeless due to the large variety of possible symmetry groups and the intricate structure of the corresponding symmetry constraints. Another technique is based on the analysis of the Wannier center flow, or Wilson loop spectra, along certain high-symmetry paths in the Brillouin zone. But this is computationally demanding, as it requires finding the Bloch eigenstates over a dense mesh of points in the momentum space and often involves manual choice of the paths of integration. Examples of such case-studies can be found in Refs. [124] and [125].

Fortunately, there are efficient methods that address both parts of the classification problem (9.2) by providing not only the symmetry-based invariants of the classification, but also the criteria of topological phases. The methods were developed independently in two forms, known as topological quantum chemistry [116] and symmetry indicators [117]. The two approaches differ in details of mathematical formalism, but are essentially equivalent. We will use "topological quantum chemistry", or TQC, to refer to both.

The corresponding topological invariants are not complete, and give only partial solution to the classification problem. On the other hand, the method has relatively low computational costs, since it is based solely on symmetry of the eigenstates and not on their geometry. This makes possible an automated, high-throughput search of candidate topological materials using as an input the crystallographic databases, which contain tens of thousands of entries. The astonishing results show that approximately 30% of all non-magnetic crystalline materials are topologically non-trivial [126].

Below we give a brief outline of the method, which requires some familiarity with the language of representation theory of groups. Then we apply the algorithm to the familiar case of the inversion symmetry, partially following the lecture [127]. First, we re-derive our results for a one-dimensional system and then upgrade them to the two-dimensional case. A proper group-theoretic introduction to the topological quantum chemistry can be found in Ref. [128].

### 9.3.1 General algorithm

The algorithm of the topological quantum chemistry consists of two parts. The first part is the band structure combinatorics, a method introduced in Ref. [129], which produces a list of all symmetry-compatible band structures.

Consider a crystal with a spatial symmetry group $G$. The symmetry also acts in the momentum space. There are special subspaces of the Brillouin zone — such as points, lines, and planes — that are invariant under the action of the symmetry group or one of its subgroups (generalizing the fixed points $k^\star$ from the example above). Over these subspaces, the Bloch Hamiltonian commutes with the symmetry operators. Consequently, Hamiltonian eigenstates over these subspaces must transform according to a representation of the corresponding (sub)group (generalizing the relation (9.20)). In this way, one associates a representation, or a "symmetry label" with each eigenstate at each high-symmetry subspace.

Now turn the problem the other way round: choose randomly some sets of the symmetry labels corresponding to the representation of the group $G$ and place them on the high-symmetry points. Is it possible to find a Bloch Hamiltonian whose eigenstates will transform under the given representations at these points? One constraint comes from the dimensions of the representations: a each point, their sum must be equal to the number of the bands. Once this is satisfied, the next question is whether the different points can be connected by the bands. Recall that if an eigenstate transforms under an irreducible representation of dimension $n$, the state must be $n$-fold degenerate. So, the symmetry labels correspond to the band crossings of certain symmetry type. The possible ways to connect the crossings are restricted by the symmetry. For example, consider a pair of high-symmetry points connected by a high-symmetry line. Then the representations at the first point must split into some other representations along the line and then combine at the other point. All these transformations happen in a controlled way determined by the representation theory. This results in the set of compatibility rules between the possible sets of symmetry labels at the high-symmetry points. Thus, we have a combinatorial problem of assigning irreducible representations to the special points in a way compatible with the rules. The solution gives the list of all possible sets of symmetry labels of the Hamiltonian eigenstates. In other words, any Hamiltonian with the symmetry group $G$ must have a set of symmetry labels from this list. Moreover, the set of labels of valence bands turns out to be invariant under the relation (9.1), since a given set cannot be changed without closing the bulk gap (which would allow the exchange of symmetry labels between the valence and conduction bands, cf. Exercise 9.1).

The second part of the algorithm addresses the question of which systems are topological. Since we have already listed all possibilities, this is equivalent to asking which of them are trivial. The answer comes from the real-space orbitals, which give rise to the concept of band representation, introduced by Zak [130]. The idea is to start from a lattice of localized

symmetric orbitals and to find the corresponding symmetry labels in the momentum space.

The action of the symmetry on an orbital is two-fold: first, the symmetry changes the position of the orbital, and second, it transforms the orbital itself. The latter transformation must correspond to some representation of the symmetry group; this determines the type of the orbital. By choosing different types of the orbitals and different positions, one can obtain all possible lattices of localized orbitals compatible with the given symmetry. Next, one interprets these orbitals as the Wannier functions of some Hamiltonian. By the Fourier transform, one obtains the corresponding Bloch eigenstates and then determines their symmetry labels in the momentum space. As a result, one has a list of all possible sets of symmetry labels of Bloch states corresponding to the *localized* Wannier functions. It remains to compare it with the first list obtained from the band structure combinatorics. All the band structures that are in the first list and do not appear in the second one, must represent topological phases compatible with the symmetry.

### 9.3.2 Inversion symmetry in one dimension

Consider the inversion-symmetric chain with a single occupied band. Following the procedure described above, we first list the irreducible representations of the symmetry group, then describe all possible combinations of the corresponding symmetry labels in the momentum space, and finally determine the sets of symmetry labels originating from the localized symmetric real-space orbitals.

The symmetry group consists of two elements, identity and inversion: $G = \{e, \mathcal{I}\}$. This group has two irreducible representations:

$$
\begin{array}{c|cc}
 & e & \mathcal{I} \\
\hline
\chi_1 & 1 & 1 \\
\chi_s & 1 & -1
\end{array}
$$

If a state $|\psi\rangle$ transforms under the trivial representation $\chi_1$, then $\mathcal{I}|\psi\rangle = |\psi\rangle$. If it transforms according to the sign representation $\chi_s$, then $\mathcal{I}|\psi\rangle = -|\psi\rangle$. In other words, symmetry-compatible functions must be either even or odd under inversion.

Consider a Bloch eigenstate $|\psi_k\rangle$ of the valence band. At the fixed points $k^\star$, it must transform under one of the irreducible representations of the symmetry group, since the Hamiltonian commutes with $\mathcal{I}$ at these points. Here, the "symmetry labels" are simply the eigenvalues $\lambda_{k^\star}$. Thus, we have four combinations that correspond to the choice of the representation at the two fixed points. There are no other constraints, so the band structure combinatorics gives four symmetry types of the valence band eigenstate $|\psi_k\rangle$.

Now we switch to the real-space orbitals. Denote $|s\rangle$ and $|p\rangle$ the localized orbitals that transform under the trivial and the sign representation of the symmetry group, respectively. Where can we place them in the unit cell? There are only two possible positions: in the middle of the unit cell or at the unit cell boundary. Otherwise, we will need additional bands to preserve the inversion symmetry. Thus, we have four cases: $|s_0\rangle$, $|s_{\frac{1}{2}}\rangle$, $|p_0\rangle$, $|p_{\frac{1}{2}}\rangle$, where the subscript denotes the spatial position of the orbital in the unit cell. As a next step, we define lattices

$$
|s_0^n\rangle, \quad |p_0^n\rangle, \quad |s_{\frac{1}{2}}^n\rangle, \quad |p_{\frac{1}{2}}^n\rangle, \tag{9.33}
$$

obtained from the orbitals by discrete translations. Here, $n$ stands for the unit cell index. These lattices are shown in the bottom rows of Fig. 9.2. The orbitals transform under inversion as

$$
\hat{\mathcal{I}}|s_0^n\rangle = |s_0^{-n}\rangle, \quad \hat{\mathcal{I}}|p_0^n\rangle = -|p_0^{-n}\rangle, \quad \hat{\mathcal{I}}|s_{\frac{1}{2}}^n\rangle = |s_{\frac{1}{2}}^{-n-1}\rangle, \quad \hat{\mathcal{I}}|p_{\frac{1}{2}}^n\rangle = -|p_{\frac{1}{2}}^{-n-1}\rangle. \tag{9.34}
$$

Then we interpret them as sets of Wannier functions corresponding to the valence band of some two-band Hamiltonian. In this way, the symmetry representation acting on a single

orbital gives rise to the **induced representation** in the valence band subspace. Finally, we find the Bloch states as a Fourier transform of the Wannier functions. For example,

$$|\psi_k^{p\frac{1}{2}}\rangle = \frac{1}{\sqrt{N}}\sum_n e^{ikn}|p_{\frac{1}{2}}^n\rangle . \tag{9.35}$$

One checks that their inversion eigenvalues coincide with those listed in Fig. 9.2. For instance, at $k = \pi$ we have

$$\hat{\mathcal{I}}|\psi_\pi^{p\frac{1}{2}}\rangle = \frac{1}{\sqrt{N}}\sum_n (-1)^n\hat{\mathcal{I}}|p_{\frac{1}{2}}^n\rangle = \frac{1}{\sqrt{N}}\sum_n (-1)^{n+1}|p_{\frac{1}{2}}^{-n-1}\rangle = |\psi_\pi^{p\frac{1}{2}}\rangle , \tag{9.36}$$

so the corresponding eigenvalue is indeed $\lambda_\pi = +1$.

At this point, we can make two conclusions. First, it is possible to deduce certain properties of a model from symmetry considerations alone, without referring to a specific Hamiltonian. Second, there are no topological phases in two-band one-dimensional inversion-symmetric crystals. The reason is that all the sets of eigenvalues obtained from the band structure combinatorics can be reproduced by the states originating from the localized orbitals.

### 9.3.3 Inversion symmetry in two dimensions

Let us now apply the algorithm to a two-dimensional crystal with inversion symmetry. Here, we have four fixed points in the Brillouin zone

$$(k_x^\star, k_y^\star): \quad (0,0), \quad (\pi,0), \quad (0,\pi), \quad (\pi,\pi). \tag{9.37}$$

One can choose the parity of $|\psi_{\boldsymbol{k}}\rangle$ at these points independently, which results in 16 combinations. On the other hand, there are four high-symmetry positions in the real space:

$$(x,y): \quad (0,0), \quad (\tfrac{1}{2},0), \quad (0,\tfrac{1}{2}), \quad (\tfrac{1}{2},\tfrac{1}{2}), \tag{9.38}$$

where the origin $(0,0)$ is at the center of the unit cell, and both lattice constants are set to unity. We can populate these positions with either $|s\rangle$ or $|p\rangle$ orbitals, which gives only 8 types of eigenstates originating from the localized orbitals. Thus, even with such little effort we can conclude that some inversion-symmetric 2D crystals must be topological.

Following the pattern of the one-dimensional example, we denote $|s_{\frac{1}{2}0}^{\boldsymbol{n}}\rangle$ the $s$-type orbital with coordinates $x = \frac{1}{2}$ and $y = 0$ in the unit cell with index $\boldsymbol{n}$. The corresponding Bloch waves are obtained by the Fourier transform:

$$|\psi_{k_x k_y}^{s\frac{1}{2}0}\rangle = \frac{1}{N}\sum_{n_x,n_y} e^{i(k_x n_x + k_y n_y)}|s_{\frac{1}{2}0}^{n_x n_y}\rangle . \tag{9.39}$$

In this way, we define the Bloch states corresponding to both types of orbitals in four symmetric positions (9.38). Their inversion eigenvalues at the points (9.37) are listed in Table 1. The eigenvalues can be easily found graphically: for example, the lattices corresponding to the last line of the table are shown in Fig. 9.4.

What are the mysterious topological phases whose sets of the inversion eigenvalues do not appear in the table? Note that each set in the table contains even numbers of positive and negative values. Thus, the remaining 8 sets of eigenvalues have odd numbers of each eigenvalue. Let us consider implications of this fact for the Bloch Hamiltonian. To this end, we interpret the two-dimensional Bloch Hamiltonian

$$H_{\boldsymbol{k}} = H_{k_x k_y} = H_{k_x}(k_y), \tag{9.40}$$



| | $(0,0)$ | $(\pi,0)$ | $(0,\pi)$ | $(\pi,\pi)$ |
|---|:---:|:---:|:---:|:---:|
| $\lvert\psi_{\boldsymbol{k}}^{s00}\rangle$ | $+$ | $+$ | $+$ | $+$ |
| $\lvert\psi_{\boldsymbol{k}}^{s\frac{1}{2}0}\rangle$ | $+$ | $-$ | $+$ | $-$ |
| $\lvert\psi_{\boldsymbol{k}}^{s0\frac{1}{2}}\rangle$ | $+$ | $+$ | $-$ | $-$ |
| $\lvert\psi_{\boldsymbol{k}}^{s\frac{1}{2}\frac{1}{2}}\rangle$ | $+$ | $-$ | $-$ | $+$ |
| $\lvert\psi_{\boldsymbol{k}}^{p00}\rangle$ | $-$ | $-$ | $-$ | $-$ |
| $\lvert\psi_{\boldsymbol{k}}^{p\frac{1}{2}0}\rangle$ | $-$ | $+$ | $-$ | $+$ |
| $\lvert\psi_{\boldsymbol{k}}^{p0\frac{1}{2}}\rangle$ | $-$ | $-$ | $+$ | $+$ |
| $\lvert\psi_{\boldsymbol{k}}^{p\frac{1}{2}\frac{1}{2}}\rangle$ | $-$ | $+$ | $+$ | $-$ |

Table 1: Inversion eigenvalues of Bloch states $\lvert\psi_{\boldsymbol{k}}^{\alpha xy}\rangle$ at four inversion-symmetric points of the two-dimensional Brillouin zone. The states originate from lattices of localized orbitals of symmetry type $\alpha = s, p$ with coordinates $x = 0, \frac{1}{2}$ and $y = 0, \frac{1}{2}$ in the unit cell.

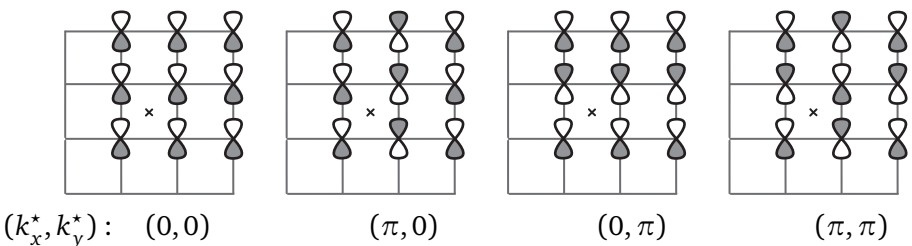

$(k_x^\star, k_y^\star):$    $(0,0)$         $(\pi,0)$         $(0,\pi)$         $(\pi,\pi)$

Figure 9.4: Typical shapes of the Bloch wave $\lvert\psi_{\boldsymbol{k}}^{p\frac{1}{2}\frac{1}{2}}\rangle$. The $p$-type orbital is placed at the top right corner of each unit cell. Four cases correspond to the crystal momenta $(k_x^\star, k_y^\star)$ that are fixed under inversion. The inversion center is marked by the cross.

as a family of one-dimensional Hamiltonians depending on $k_y$ as an external parameter. The inversion symmetry implies that

$$\mathcal{I}H_{\boldsymbol{k}}\mathcal{I}^{-1} = H_{-\boldsymbol{k}} \quad \Rightarrow \quad \mathcal{I}H_{k_x}(k_y)\mathcal{I}^{-1} = H_{-k_x}(-k_y). \tag{9.41}$$

It follows that for $k_y^\star = 0, \pi$, the Hamiltonian $H_{k_x}(k_y^\star)$ describes an inversion-symmetric chain. Depending on the combination of the eigenvalues, the chain can be in the trivial state, $\gamma_x = 0$, or in the polarized state, $\gamma_x = \pi$ (see Eq. (9.31)). Here, $\gamma_x$ denotes the Zak phase of the effective chain in the $x$ direction. Thus, for $k_y = k_y^\star$, the Wannier centers of the chain are pinned to the values $0$ or $\frac{1}{2}$.

For a general value of $k_y$, the symmetry requires that

$$\gamma_x(k_y) = -\gamma_x(-k_y), \tag{9.42}$$

since the Zak phase is odd under inversion. This condition restricts the possible trajectories of the Wannier centers. In particular, if $\gamma_x(0) \neq \gamma_x(\pi)$, the symmetry forces the Wannier center to shift by an odd number of unit cells per one "pumping cycle" in $k_y$. This is exactly the case when the set contains an odd number of negative eigenvalues. An example of such situation is shown in Fig. 9.5. We conclude that the sets $\{\lambda\}$ that are allowed by the band structure

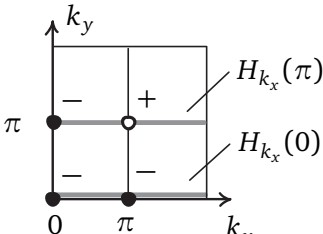 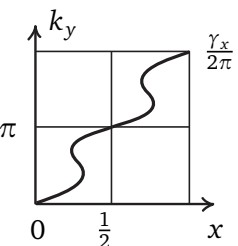

Figure 9.5: LEFT: One of the sets of inversion eigenvalues not appearing in the Table 1. The signs near the circles show inversion eigenvalues $\lambda_{k^\star}$. The Hamiltonians $H_{k_x}(k_y^\star)$ for $k_y^\star = 0, \pi$ describe effective one-dimensional inversion-symmetric chains in the $x$ direction. RIGHT: Typical trajectory of the zeroth Wannier center in the chain described by $H_{k_x}(k_y)$ as a function of $k_y$. According to the inversion eigenvalues and Eq. (9.31), the Zak phase $\gamma_x(k_y)$ must satisfy $\gamma_x(0) = 0$ and $\gamma_x(\pi) = \pi$.

combinatorics, but do not appear in the Table 1, correspond to inversion-symmetric Chern insulators. This can be expressed succinctly by introducing the product $\nu$ of all four symmetry eigenvalues:

$$\nu = \prod_{k \in \{k^\star\}} \lambda(k) = (-1)^{c(V^\nu)}. \tag{9.43}$$

The invariant $\nu$ detects the *parity* of the Chern number $c(V^\nu)$ of the valence band bundle. In the case when $\gamma_x(0) = \gamma_x(\pi)$, we have an even number of negative eigenvalues, so $\nu = 1$ and the Wannier center trajectory must traverse an even number of unit cells.

   This example both illustrates the power of the method and shows its limitations. Remarkably, we were able to deduce the existence of a topological insulator without any knowledge of vector bundles or the Berry curvature. We defined the invariant $\nu$, whose negative value necessarily means that the crystal is topologically non-trivial. On the other hand, it could be difficult to invent the concept of the Chern number $c(V^\nu)$ in this setting. Note also that any Chern insulator with an even Chern number has inversion eigenvalues from Table 1. This means that the set of symmetry labels is not a *complete* topological invariant, and some distinct phases can fall into the same category.

   **Exercise 9.4.** Consider a three-dimensional inversion-symmetric two-band Hamiltonian. There are eight points $k^\star$ in the Brillouin zone that are fixed under $\mathcal{I}$. Argue that if the product of the inversion eigenvalues at all $k^\star$ equals $-1$, the crystal must be gapless. [§10.3.2]

## 9.4 $T$-invariant topological insulators

Now we switch from the spatial crystalline symmetry $\mathcal{I}$ to the time reversal $T$. It is one of the internal symmetries that can be present in any quantum system. In contrast with spatial crystalline symmetries, the time-reversal symmetry is an anti-unitary operation, and does not have eigenvalues. The classification thus requires a different approach, which we briefly discuss in this section. We will give a qualitative description of two possible classes in 2D based on the stability of surface states. Then we will see how the presence of an additional symmetry (not included in the equivalence relation) can help with the classification problem.

### 9.4.1 Constructing representative models

We start by constructing a number of $T$-invariant models based on inversion-symmetric Chern insulators introduced above. To this end, we employ the physical degree of freedom we have ignored thus far: electron's spin. A minimal model of an insulator with spinful electrons contains four bands. Assume for a moment that the electronic system of the crystal consist of two independent, fully-polarized subsystems, which we will describe as "spin-up" and "spin-down". Now suppose that the Hamiltonian of an inversion-symmetric Chern insulator describes the spin-up subsystem. Then, by the time-reversal symmetry, the spin-down subsystem must contain the Chern insulator with the opposite Chern number.

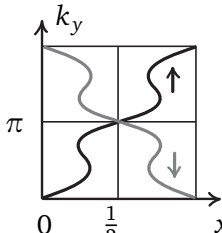 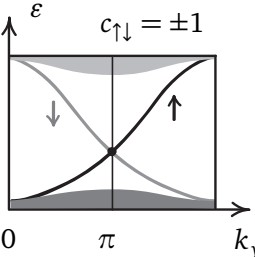

Figure 9.6: Features of a representative model for a two-dimensional $T$-invariant insulator with additional inversion symmetry. LEFT: Counter-propagating flows of two Wannier centers related by time reversal. RIGHT: Surface states of the two subsystems in the sample, which is finite in the $x$ direction. Only states at the edge with $x > 0$ are shown.

The time reversal acts on the Wannier center trajectory by reversing $k_y \rightarrow -k_y$ and by flipping the spin, which gives a typical picture shown in Fig. 9.6 on the left. Due to the inversion symmetry, Wannier centers at $k_y^\star = 0, \pi$ must have coordinates 0 or $\frac{1}{2}$. The points $k_y^\star = 0, \pi$ are also fixed under the time reversal, and are known as **time-reversal invariant momenta**, or TRIM. Since the time reversal commutes with the inversion operator, $[T, \hat{\mathcal{I}}] = 0$, the eigenstates $|\psi_{k^\star}\rangle$ and $T|\psi_{k^\star}\rangle$ have the same inversion eigenvalue.

Each of the two Chern insulators also has chiral edge states, which, by construction, intersect at the point $k_y^\star = \pi$ and are spin-polarized, as shown in Fig. 9.6 on the right. In this case, the Chern numbers of the two spin subsystems are $c_{\uparrow\downarrow} = \pm 1$. In a similar way, we construct models corresponding to other values of the Chern number. Our next goal is to determine which of these models are distinct under the equivalence relation $\sim_T$.

### 9.4.2 Lifting inversion symmetry

Here, we give a heuristic argument for the classification of $T$-invariant two-dimensional insulators. The argument is based on the stability of the edge states under the perturbations respecting the time-reversal symmetry. In general, a similar shape of the edge states in two phases does not imply that the phases belong to the same class (consider, for example, two trivial inversion-symmetric insulators with different sets of inversion eigenvalues). But in the present situation, the classification based on the surface states happens to give the complete picture. In any case, robust edge states do indicate the presence of non-trivial topology.

Let us start from the insulator described by Fig. 9.6. Now, we allow for any $T$-invariant perturbations. They can break the inversion symmetry and can violate the accidental conservation of $s_z$ by mixing the two spin species. At first glance, it is possible to open the gap at the

intersection of the two branches with opposite chiralities. However, this cannot happen, owing to the Kramers degeneracy discussed in Sec. 5.2.3. Two states at the intersection point are related by the time reversal and thus must have the same energy. On the other hand, opening the gap would create a pair of non-degenerate states at this TRIM. In this way, time-reversal symmetry for spinful electrons protects any level intersection at TRIM $k_i^\star = 0, \pi$ for $i = x, y$.

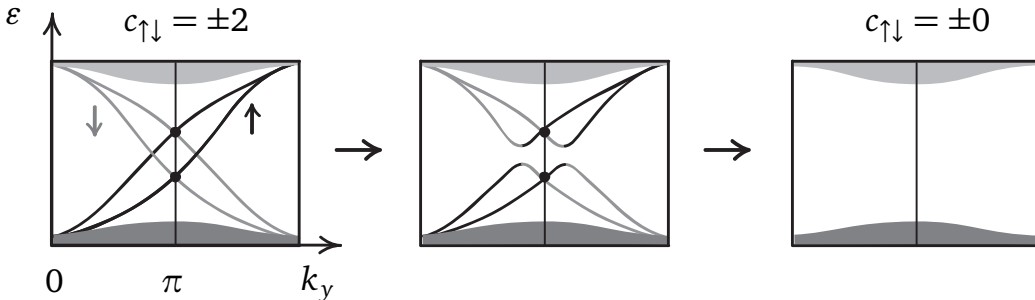

Figure 9.7: Deformation of the surface states of a $T$-invariant insulator. The middle panel shows opening the gaps at the points away from $k^\star$ by a time-reversal-invariant perturbation. All surface states shown are localized at one edge.

Now consider the situation when the original Chern insulators have an even Chern number. A typical picture of the edge states for the case $c_{\uparrow\downarrow} = \pm 2$ is shown in Fig. 9.7. While Kramers degeneracy protects intersections at $k^\star$, the intersections at any other point $k \neq k^\star$ can be gapped out by a perturbation mixing levels of the two spin subsystems. As shown in the figure, opening the gaps breaks the connectivity of the surface states, which allows one to push them into the bulk bands. Thus, the surface spectrum can be deformed to that of a trivial insulator.

Our analysis indicates that there are only two possible shapes of the surface states, up to $T$-invariant deformations: the stable spectrum shown in Fig. 9.6 on the right, or the trivial spectrum without any surface states. It turns out that this picture accurately reflects the classification of bulk Hamiltonians in the given symmetry class. The two-dimensional time-reversal invariant spinful insulator can be either trivial or topological. One says that these systems have $\mathbb{Z}_2$ classification (cf. Eq. (8.26)). A non-trivial system is called the **quantum spin Hall insulator**, which was introduced in Ref. [131] by upgrading the Haldane model to a spinful system with strong spin-orbit coupling.

There is a number of ways to determine whether a given Hamiltonian $\hat{H}(k_x, k_y)$ describes a non-trivial $T$-invariant insulator. Geometrically, we have a vector bundle of valence bands $V^v$ with a fiber isomorphic to $\mathbb{C}^2$. The (multi-band) Chern number $c(V^v)$ is necessarily zero, since otherwise we would have a Chern insulator, which breaks the time-reversal symmetry. However, there is a symmetry constraint on the sections of $V^v$, or eigenstates of $\hat{H}(k_x, k_y)$ over the Brillouin zone. If there is no non-vanishing section that satisfies the symmetry constraint, the insulator is non-trivial. This can be detected by several equivalent topological invariants. For a detailed description of these invariants and a proof of their equivalence, see Ref. [10]. Another way is to track coordinates of Wannier centers in one direction as functions of crystal momentum along the other. If there are counter-propagating flows of Wannier centers related by the time reversal (see Fig. 9.6, left), we have a non-trivial phase [132]. This method is especially convenient for a numerical implementation [133].

While the classification is based on the time-reversal symmetry, it can very well happen that the crystal of interest also has inversion symmetry. In this case, the Wannier center trajectories

must pass through the points $0, \frac{1}{2}$ at TRIM. Recall that the parity of the Chern number corresponding to one such trajectory is encoded in the invariant $\nu$, which is given by the product of the inversion eigenvalues (9.43). In turn, the parity of the Chern number determines the connectivity type of the surface states. Thus, the invariant $\nu$ provides a simple test telling whether an inversion-symmetric $T$-invariant Hamiltonian describes the quantum spin Hall phase [134].

Quantum spin Hall state was first studied experimentally in HgTe/CdTe quantum wells [135], following the theoretical prediction [136], as reviewed in Ref. [137]. Interestingly, spin-polarized surface states with linear dispersion relation were first predicted theoretically in Ref. [138]. It was shown that such robust states can arise at the boundary between two semiconductors, which have mutually inverted band structures. Only two decades later it was understood that a band inversion can change topology of eigenspace bundles, which underlies the stability of the boundary modes. See Ref. [139] for a discussion of these results in the context of topological insulators.

The early experiments focused on the measurements of the quantized transport. A characteristic feature of the quantum spin Hall state is its behavior in the external magnetic field, which breaks the time-reversal symmetry and thus makes it possible to open the gap in the edge states spectrum. As a result, the quantized conductance is suppressed. In a recent experimental study [140], the authors directly probed the conductance of a monolayer $WTe_2$ by a *local* measurement. They scanned the sample using a sharp conducting tip with applied voltage of microwave frequency, and measured the local response to the electric field. The resulting images show that the conductivity peaks in the narrow region outlining the irregular edge of the sample. The conductivity decreases in the presence of the magnetic field, indicative of the quantum spin Hall phase.

Finally, we note that the quantum spin Hall effect has a three-dimensional generalization. Recall from Sec. 8.4.1 that one can construct a three-dimensional topological phase by stacking Chern insulators. As one can expect, a similar construction is possible for the quantum spin Hall layers, which results in a **weak** topological insulator. What is more surprising, there is a three-dimensional $T$-invariant topological insulator, which *cannot* be obtained by stacking two-dimensional layers. This phase, called a **strong** topological insulator, generalizes to the three dimensions all aspects of the quantum spin Hall state we discussed above. Each surface of this crystal supports a single **Dirac cone** (or an odd number of them). It is a conical intersection of the spin-polarized surface states, which is a three-dimensional version of the level intersection shown in the right panel of Fig. 9.6. If the crystal has inversion symmetry, this phase can be diagnosed by the product of inversion eigenvalues at the eight TRIM. For more details and an overview of the early experimental results, see Ref. [1]. In particular, the experiments confirmed the presence of the Dirac cones. This was done by using ARPES, or angle-resolved photoemission spectroscopy, which directly probes the surface band structure. However, *transport* signatures of the surface states turned out to be elusive, due to the parasitic bulk conductivity. See Ref. [141] for a detailed technical discussion of related experimental challenges from the material growth point of view. For an overview of possible device applications of topological insulators, see Refs. [142] and [143].

## 9.5 Summary and outlook

In this section, we did not consider vector bundles directly, since it would have required more advanced mathematical techniques. However, we learned that in some cases one can detect a non-trivial topology of a vector bundle from the symmetry data alone, without even looking at the bundle itself. Classification of topological phases with spatial and internal symmetries is a vast topic. There at least two reasons for this: first, symmetries come in many types and combinations; second, the more restrictive is the equivalence relation, the richer will be the resulting classification. The first milestone was the discovery of the periodic table of

topological phases with internal symmetries. A recent breakthrough in the study of spatial symmetries is the development of topological quantum chemistry, which helped to identify thousands of candidate topological materials based on symmetry properties of band structure. An overview of the progress in this field, including material predictions and experimental realizations, is given in Ref. [144]. Here are the key points of our discussion:

- The classification of topological phases with symmetries is based on the equivalence relation (9.1): if two systems are inequivalent, any smooth deformation between them either closes the bulk gap or breaks the defining symmetry.

- Symmetry puts constraints on the Bloch Hamiltonian and its eigenstates in the momentum space and on the Wannier functions in the real space (in an appropriate gauge). The interplay of momentum- and real-space characterizations lies at the heart of topological quantum chemistry.

- Symmetry alters the definition of a topological phase: the phase is topological, if it cannot be described by exponentially localized and symmetric Wannier functions.

- In some cases, the topological nature of a phase can be deduced from the symmetry data alone. An example of such symmetry-indicated phase is the inversion-symmetric Chern insulator with an odd Chern number.

- If the phase is not symmetry-indicated, it can be detected by a topological invariant or by analysis of the Wilson loop spectrum (Wannier center flow). The presence of an additional symmetry can make the phase symmetry-indicated, as in the case of spinful $T$-invariant insulators with inversion symmetry.

- Inversion symmetry leads to the quantization of the Zak phase and restricts possible values of the electric polarization.

Below, we make two remarks on internal symmetries and discuss algebraic machinery of topological quantum chemistry.

▷ **SSH chain and chiral symmetry.** The inversion-symmetric diatomic chain may resemble the Su–Schreifer–Heeger model for polyacetylene [145], known as SSH chain. It is widely used in introductory texts on topological matter as a basic example of a topological phase — see, for example, textbooks [8] and [102]. Here, we comment on the differences between the SSH chain and the inversion-symmetric chain discussed above.

The SSH model describes a diatomic chain with non-zero nearest-neighbor hoppings ($t_{in}$ and $t_{ex}$ in our notation) and vanishing on-site potentials. Since the Hamiltonian contains only terms connecting orbitals from different sublattices, the matrix of $\hat{H}_k$ is off-diagonal, so it anti-commutes with $\sigma_z$. These are manifestations of **chiral** symmetry (also known as sublattice symmetry). Another chiral-symmetric model we encountered before is the Hamiltonian of graphene (5.41). In systems with chiral symmetry, the energy spectrum must be symmetric under sign change, so the surface states, if present, have exactly zero energy (cf. the spectrum of graphene in Fig. 8.5). In the SSH chain, the surface states are detected by the bulk topological invariant, the winding number of $\boldsymbol{h}_k$ vector around the origin. Because of the chiral symmetry, the vector lies in the $\sigma_x$-$\sigma_y$ plane, so the winding number cannot be changed without closing the bulk gap. In the trivial phase, the charge centers lie at the inner bond, while in the topological phase, they lie on the bond between two unit cells.

Two typical pictures of charge center distributions is the only common feature of the SSH model and the inversion-symmetric chain. The inversion symmetry allows for non-zero $h_z(k)$, so the winding number is not defined. At the boundary of a finite chain, the inversion symmetry

is broken, and the surface states are not protected. Also note that the "topological" state of the SSH chain is non-topological, in the sense that it is described by the exponentially-localized Wannier functions.

Another possible source of confusion is that discussion of the SSH chain often includes a computation of the following form:

$$\phi = \int_0^{2\pi} i\,\frac{1}{\sqrt{2}}\begin{pmatrix}-e^{ik} & 1\end{pmatrix}\partial_k \frac{1}{\sqrt{2}}\begin{pmatrix}-e^{-ik} \\ 1\end{pmatrix}dk = \pi\,, \tag{9.44}$$

which is intended to illustrate that in the "topological" state, the Zak phase has the value $\gamma = \pi$. However, the computation uses the components $\psi_{\alpha k}$ given by Eq. (9.24), and not $u_{\alpha k}$. The reason why this works is discussed in Sec. 9.2.1: due to the symmetry and because of the judicious choice of the spatial origin, the two potentials coincide $A_k = \widetilde{A}_k$, and we have $\gamma = \phi$. On the other hand, if we place the origin at the orbital of the type $a$, the quantized values of the Zak will become $\gamma = \frac{\pi}{2}, \frac{3\pi}{2}$, while the expression in Eq. (9.44) will remain unchanged.

▷ **The periodic table.**    Time-reversal symmetry (TR) is one of the fundamental quantum mechanical symmetries, along with particle-hole symmetry (PH) and chiral symmetry (C), which is the combination of TR and PH. There are three possibilities for TR and PH: the symmetry can be absent; if present, it can square to plus or minus identity operator. This gives nine combinations. When both symmetries are broken, it is also possible that the system is invariant under their combination. As a result, we obtain ten distinct **symmetry classes** determined by the internal quantum mechanical symmetries (see Ref. [146] for an overview). This classification is very general; in particular, it applies to systems without translational invariance, such as the electron gas in the quantum Hall effect. In fact, it has its roots in the study of ensembles of random matrices and universal properties of disordered systems [147].

The discoveries of the $T$-invariant topological insulators in two and three dimensions prompted the question: are there topological systems in other dimensions and symmetry classes? It was found that for each dimension, exactly five symmetry classes can contain topological phases, either with $\mathbb{Z}$ or $\mathbb{Z}_2$ classification [148]. Then it was noted that after an appropriate re-ordering of the list of symmetry classes, certain periodic pattern emerges [114], giving rise to the **periodic table** of topological insulators and superconductors. This pattern is directly related to the Bott periodicity in the groups formed by equivalence classes of vector bundles over spheres $S^n$, which are studied by the $K$-theory. The periodicity can also be understood in terms of the spinor representations of the orthogonal group [149]. The part of the table that describes the phases discussed above reads

|     | TR | PH | C | 1 | 2 | 3 |
|-----|----|----|---|---|---|---|
| A   | 0  | 0  | 0 |   | $\mathbb{Z}$ |   |
| AII | −1 | 0  | 0 |   | $\mathbb{Z}_2$ | $\mathbb{Z}_2$ |

.

The first row describes the symmetry class A, in which all symmetries are broken, as indicated by zeroes. A non-trivial phase is possible only in two spatial dimensions: this is the Chern insulator, which has $\mathbb{Z}$-classification. The second row corresponds to the spinful $T$-invariant systems, or the symmetry class AII. Here, we have non-trivial insulators with $\mathbb{Z}_2$-classification in two and three dimensions. Note that the table does not include the Hopf insulator and phases obtained by stacking 2D layers (such as 3D Chern insulators discussed in Sec. 8.4.1 and weak $T$-invariant insulators from Sec. 9.4.2). The reason is that the periodic table is related to the $K$-theory of spheres: first, the Hopf insulator is invisible for the $K$-theory (see Sec. 8.5); second, the layered constructions require that the base space be a torus rather than a sphere.

The situation becomes much more complicated when one takes spatial crystalline symmetries into account. The symmetries can be thought of as additional constraints, which relate fibers of the vector bundle at different points of the base space. Classification of vector bundles with symmetries by $K$-theoretic methods is a profoundly difficult problem that is to be solved on case-by-case basis. The topological quantum chemistry offers an indispensable alternative. Remarkably, the classification based on band structure combinatorics agrees with that given by $K$-theoretic methods whenever the latter is available (see, for example, Ref. [129]).

▷ **Linear combinations of band representations.** The topological quantum chemistry naturally incorporates the case of multiple occupied bands. Any set of bands originating from the localized symmetric Wannier functions is called a band representation. Of particular importance are *elementary* band representations (EBR), which are derived from the Wannier orbitals at certain high-symmetry positions in the real space. Recall that any group representation can be decomposed into a direct sum of irreducible representations. EBRs play a similar role for the band structures: any band representation can be constructed from EBRs as building blocks. In TQC, we are interested in the band structures that do not admit exponentially localized Wannier functions; thus, one cannot construct them from EBRs. If we are lucky, a topological band structure has a symmetry indicator, that is, a simple rule detecting the phase from the set of symmetry labels, like one given by Eq. (9.43).

It turns out that there is an algebraic way to find all possible symmetry indicators of topological bands structures for all space groups. The idea is to consider an abstract "vector space" formed by multiplicities of irreducible representations at high-symmetry points. Then the EBRs are used as basis vectors spanning the subspace of all band representations. Taking the quotient of the whole space by this subspace, one obtains the space of all topological band structures. The symmetry indicators can be extracted from the structure of the quotient. See Ref. [128] for an application of this method to the case of inversion symmetry in 2D, which was discussed in Sec. 9.3.3. Lecture [150] gives an introduction to TQC with the focus on symmetry indicators.

The linear structure of space of band representations allows for characterization of new kinds of topological phases. Suppose that we have a set of symmetry labels of some insulator. Then we can try to express it as a linear combination of EBRs with integer coefficients. If there is no such combination, we have a topological phase. If there exists a combination with non-negative coefficients, the phase is not topological. An interesting situation arises if all coefficients are integral, but some of them are negative: this would tell us that the system is topological, but it will become trivial after addition of some EBRs (which themselves are trivial). In this case, the original phase represents the **fragile topology** [151]. In contrast with stable topological phases, like Chern insulators, a fragile phase can be trivialized by addition of trivial bands. Both stable and fragile phases may or may not be symmetry-indicated; for details, see [128] and references therein.

# 10 Topological semimetals

The conical intersections in the band structure of graphene make it difficult to classify this material as an insulator or a metal. On the one hand, the graphene is gapless, so it is not an insulator. On the other hand, since the intersection points lie at the Fermi level, the Fermi surface consists of isolated points, and the corresponding density of states vanishes. Such systems are called **semimetals**. A minimal model of a semimetal can be constructed using a two-band Hamiltonian (8.3) without the term proportional to the identity matrix. The bands cross whenever the vector field describing the Hamiltonian vanishes, $|\boldsymbol{h}_{\boldsymbol{k}}| = 0$. The points of

degeneracy are called the **nodes** in the band structure.

In the previous sections, we were concerned with classifications defined by equivalence relations based on the gap-preserving deformations. The nodal points were associated with singular events accompanying a change of the equivalence class. Examples include the transition between a trivial and a topological insulator (Sec. 8.2.4) and a symmetry-preserving interpolation between two inversion-symmetric chains (Ex. 9.1). One might conclude that a gapless system cannot be topological; but this is not the case, as the title of the present section indicates. Below, we will see how topology affects the properties of semimetals in at least three different ways. First, we can study the familiar topology of band structure for the subspaces of the momentum space that avoid the points of degeneracy. Second, under certain conditions, the nodal points are stabilized against local perturbations due to continuity of the Hamiltonian. Third, there is a global constraint put on the (signed) number of nodal points, which can be understood by interpreting them as singularities in a section of a real vector bundle.

## 10.1 Zero locus of a vector field

At first sight, any degeneracy $|\boldsymbol{h_k}| = 0$ can be lifted by the avoided level crossing mechanism (see Sec. 7.2.1). However, it turns out that in some systems the nodes are stable against local perturbations. To see how this happens, we consider the question of local stability of a zero of a vector field $\boldsymbol{v}$. Here, we ignore the global topological aspects of the field, but for convenience, we will use the language of vector bundles. Let $\boldsymbol{v}$ be a section of a real vector bundle over a base space $\mathcal{B}$. Suppose that the section vanishes, $\boldsymbol{v}(p) = 0$, for some point $p \in \mathcal{B}$. Is it possible to remove the zero by a smooth deformation of the section $\boldsymbol{v}$ near the point $p$? Generically, the answer depends on the combination of two parameters: the dimension $d$ of the base space $\mathcal{B}$ and the dimension $n$ of the fiber. To see this, introduce a set of basis sections $\{\boldsymbol{e_i}\}$ near $p$, so that the section $\boldsymbol{v} = v_i \boldsymbol{e_i}$ can be described locally by $n$ functions of $d$ variables.

For $(n, d) = (1, 1)$, that is, a real line bundle over a one-dimensional base space, we have a single function of one variable. If such function crosses the zero, then the intersection is stable: a local perturbation will only shift the position of the crossing, but will not remove it. In the case $(n, d) = (2, 1)$, the situation is different. If a two-component vector field $\boldsymbol{v}$ vanishes at some point $p$, this means that there are *two* functions such that $v_1(p) = v_2(p) = 0$. One can slightly deform one of these functions, so that they will vanish at different points, and the zero of the section $\boldsymbol{v}$ will disappear. The same argument applies when the vector field has more than two components. Thus, only the bundles with one-dimensional fibers have stable point-like zeroes over one-dimensional base spaces.

Such analysis can be easily generalized to the base spaces of higher dimension. Consider the case $(n, d) = (1, 2)$. Here, the section $\boldsymbol{v}$ is described by a single function $v_1(x_1, x_2)$ of two variables. Generically, such a function vanishes along some one-dimensional curve. Now, if we increase the dimension of the fiber, $(n, d) = (2, 2)$, the section is described by a pair of these functions. Each vanishes along a curve, hence both functions will vanish at the points where these curves intersect. In general, the intersection point of two planar curves is stable against a small deformation of the curves. However, in the case $(n, d) = (3, 2)$ the zero $\boldsymbol{v}(p) = 0$ would require the intersection of three planar curves in one point, which can be destroyed by moving one of the curves away from the point of intersection.

Finally, we consider bundles over a three-dimensional base space. For $(n, d) = (1, 3)$, a section is described by a single function of three variables, which can vanish along a two-dimensional surface. If $(n, d) = (2, 3)$, the section $\boldsymbol{v}$ vanishes along the curves formed by intersection of pairs of such surfaces. In the case $(n, d) = (3, 3)$, the section $\boldsymbol{v}$ is zero where three surfaces intersect; generically, this occurs in isolated points. Our findings are summarized in Fig. 10.1, which shows typical shapes of spaces formed by points where $\boldsymbol{v} = 0$. Such a space is called a **zero locus** of the section $\boldsymbol{v}$. We conclude that the dimension of the zero

locus is given by $d - n$ if $d \geqslant n$ and there are no stable zeroes for $d < n$.

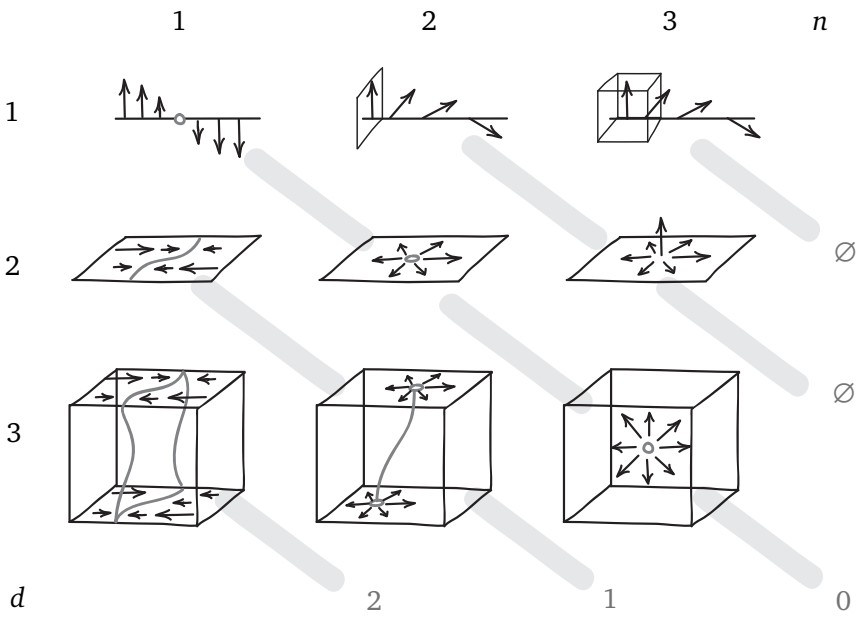

Figure 10.1: Zero loci of sections of real vector bundles. Numbers indicate dimension of the base space $d$, dimension of the fiber $n$, and dimension of the zero locus (gray numbers).

We have encountered some of these situations before. In the examples below we treat complex line bundles as real plane bundles.

$(n, d)$  Examples:
$(1, 1)$  Sec. 4.1.2: Singularity on a Möbius band in Fig. 4.1
$(1, 2)$  Sec. 4.3.3: The Jacobian $J(x_1, x_2)$ vanishes along one-dimensional curves
$(2, 2)$  Sec. 5.3.1: Dirac points in graphene, Sec. 4.1.2: Singularities in sections of $TS^2$
$(3, 2)$  Sec. 8.3.1: Opening the gap in graphene by breaking the symmetries
$(2, 3)$  Sec. 4.2.4: Line of zeroes of the wave function near the magnetic monopole
$(3, 3)$  Sec. 8.2.4: Berry monopole in Fig. 8.3.

## 10.2  Nodal line semimetals

Graphene is a semimetal, in which the vector field $\boldsymbol{h_k}$ belongs to the class $(n, d) = (2, 2)$: the momentum space is two-dimensional, and there are only two Pauli matrices in the Bloch Hamiltonian, due to its symmetry. Here, we extend this Hamiltonian into the third spatial dimension, while preserving the symmetry, which gives a model in the class $(n, d) = (2, 3)$.

### 10.2.1  Merging of Dirac points

Consider a modified graphene Hamiltonian, in which we change the value of the hopping amplitude inside the unit cell (the horizontal bonds in Fig. 5.2). We denote this internal hopping by $t' \in \mathbb{R}$, and the other two amplitudes by $t \in \mathbb{R}$. Then the function $f_{\boldsymbol{k}}$ from the Hamiltonian (5.41) becomes

$$f_{\boldsymbol{k}} = t' + t(e^{-ik_1} + e^{-ik_2}). \tag{10.1}$$

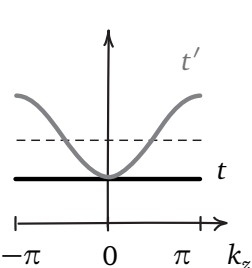 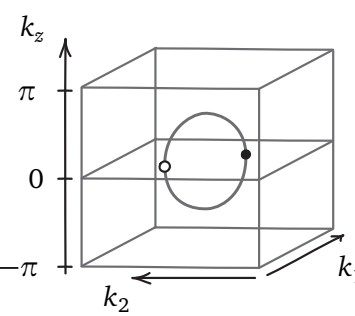 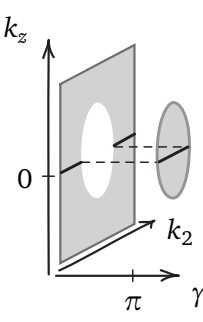

Figure 10.2: LEFT: Hopping amplitudes in the modified graphene Hamiltonian as functions of an external parameter $k_z$. MIDDLE: Nodal ring in the Brillouin zone of a three-dimensional crystal obtained by stacking graphene layers in the $z$ direction. Filled and empty circles show Dirac points in the graphene Brillouin zone at $k_z = 0$. RIGHT: The Zak phase $\gamma$ as a function of $k_2$ and $k_z$. The Zak phase takes value $\gamma = \pi$ inside the projection of the nodal ring, and vanishes outside. Black line at $k_z = 0$ corresponds to the graph $\gamma(k_2)$ in graphene shown in Fig. 8.5.

Following Ref. [152], we examine how this modification affects the positions of the Dirac points. These are solutions of $f_{\boldsymbol{k}} = 0$, which gives

$$e^{-ik_1} + e^{-ik_2} = -\frac{t'}{t}. \tag{10.2}$$

Since the amplitudes are real, this implies that $k_1 = -k_2$. Thus, the coordinate $k$ of a Dirac point must satisfy

$$\cos k = -\frac{t'}{2t}. \tag{10.3}$$

In graphene, we have $t' = t$, and there are two solutions: $k = \frac{2\pi}{3}$ and $k = \frac{4\pi}{3}$. If we increase the value of $t'$, the two solutions will move towards each other, until they merge at $k = \pi$ when $t' = 2t$. For $t' > 2t$, there are no solutions.

Now let us define a three-dimensional lattice model, which includes such merging process as a part of its band structure. To this end, consider a vertical stack of coupled graphene layers described by the Bloch Hamiltonian

$$H_{\boldsymbol{k}}(k_z) = \begin{pmatrix} 0 & f_{\boldsymbol{k}}(k_z) \\ \overline{f_{\boldsymbol{k}}(k_z)} & 0 \end{pmatrix}, \tag{10.4}$$

where $f_{\boldsymbol{k}}(k_z)$ is given by Eq. (10.1) with

$$t = t_0, \quad t' = (2 - \cos k_z)t_0, \quad t_0 \in \mathbb{R}. \tag{10.5}$$

Here, $\boldsymbol{k} = (k_1, k_2)$ are are the crystal momenta in the horizontal graphene lattice and $k_z$ is the momentum in the vertical direction.

The graphs of the model parameters as functions of $k_z$ are shown in Fig. 10.2 on the left. The middle panel of the figure shows the resulting **nodal ring** in the band structure. At $k_z = 0$, we have $t' = t = t_0$, and the two-dimensional Hamiltonian coincides with that of graphene, Eq. (5.41). As $|k_z|$ increases, the positions of the Dirac points become closer. For $|k_z| > \frac{\pi}{2}$, the two-dimensional Hamiltonian is gapped. By construction, the Hamiltonian contains only terms proportional to $\sigma_x$ and $\sigma_z$. As discussed in Sec. 5.3.2, this means that the model has combined $\mathcal{I} \circ T$ symmetry. It follows that the nodal ring is stable against any small $\mathcal{I} \circ T$-preserving perturbations. This (purely hypothetical) model gives an example of a **nodal line semimetal** [153].

### 10.2.2 Drumhead surface states

The surface signature of a nodal line semimetal are the **drumhead states**, or surface flat bands filling the projections of the bulk nodal loops. To understand their origin, recall from Sec. 9.2.1 that the Zak phase in graphene is quantized due to the $\mathcal{I} \circ T$ symmetry. As shown in Fig. 8.5 on the left, the Zak phase $\gamma = \pi$ between two projections of Dirac points, and vanishes elsewhere. This graph is shown by the black line in the right panel of Fig. 10.2. Since the Hamiltonian $H_{\boldsymbol{k}}(k_z)$ is again $\mathcal{I} \circ T$-symmetric, the Zak phase must be quantized. By continuity in $k_z$, the $\gamma = \pi$ value fills the whole projection of the nodal ring. In the absence of the surface potential, this will result in the mid-gap drumhead surface states. In a recent work [154], the authors study theoretically and probe experimentally such surface states in the nodal line semimetal ZrSiTe. Due to the intricate structure of the nodal loops, their surface projections overlap. The surface states are observed only in the regions containing an odd number of projections, in agreement with the possible values of the Zak phase $\gamma = 0, \pi$.

## 10.3 Weyl semimetals

A three-dimensional two-band Hamiltonian is described by a vector field $\boldsymbol{h}_{\boldsymbol{k}}$ which belongs to the class $(n, d) = (3, 3)$. We know that such a vector field can have stable point-like zeroes. This is the idea behind **Weyl semimetal** [155]: in a three-dimensional crystal, crossing points of an isolated pair of bands cannot be removed by a local perturbation. In this way, Weyl semimetal can be thought of as a three-dimensional generalization of graphene, where symmetry requires that $\boldsymbol{h}_{\boldsymbol{k}}$ lie in the plane and thus stabilizes Dirac points. In contrast with graphene, Weyl points do not rely on any symmetry, and thus can be expected to be a generic phenomenon. However, semimetallic physics requires an exact alignment of the Weyl points with the Fermi level, which is not guaranteed in general (but can be enforced by symmetry).

### 10.3.1 Weyl points

Let us construct a toy model of Weyl semimetal. Recall that we considered in Sec. 8.2.4 a process in which the change of the Chern number was accompanied by a gap closing. This led to creation of a degeneracy point in the three-dimensional mixed momentum-parameter space. Now let us extend the same Hamiltonian (8.15) to the parameter range $p \in [-\pi, \pi]$, so that the Hamiltonian becomes periodic in $p$. Another degeneracy point appears at $p = -\frac{\pi}{2}$. Finally, we interpret the external parameter $p$ as the crystal momentum $k_z$, just as we did when transforming a charge pump into a Chern insulator in Sec. 8.2. We obtain:

$$H_{\boldsymbol{k}}^{WSM} = \boldsymbol{\sigma} \cdot \boldsymbol{h}_{\boldsymbol{k}} = \begin{pmatrix} \sigma_x & \sigma_y & \sigma_z \end{pmatrix} \begin{pmatrix} t_0(1 + \cos k_x) + \delta \sin k_y \\ t_0 \sin k_x \\ \Delta_0(\cos k_y + \cos k_z - 1) \end{pmatrix} . \tag{10.6}$$

As a result, we have a Hamiltonian of a Weyl semimetal $H_{\boldsymbol{k}}^{WSM}$ with two point-like degeneracies, which are called **Weyl points** in this context (see the left panel of Fig. 10.3). Expansion to the linear order near a Weyl point $\boldsymbol{W}$ gives

$$H_{\boldsymbol{W}+\boldsymbol{q}} \approx \sum_{ij} \sigma_i A_{ij} q_j . \tag{10.7}$$

The sign of the determinant of the matrix $A$ is called the **chirality** $\chi$ of the Weyl point:

$$\chi = \text{sign}[\det A] . \tag{10.8}$$

From Sec. 8.2.4, we know that the point $W_+$ has chirality $\chi = \text{sign}[\delta t_0 \Delta_0]$.

**Exercise 10.1.** Compute the chirality of the other Weyl point $W_-$ with coordinates $(k_x, k_y, k_z) = (0, 0, -\frac{\pi}{2})$.

By the result of the exercise, two Weyl points have opposite chiralities. This is a manifestation of a general principle: for any Weyl semimetal, the sum of chiralities of all Weyl points $W_i$ vanishes,

$$\sum_i \chi[W_i] = 0, \tag{10.9}$$

which is closely related to the Nielsen–Ninomiya theorem [156] from particle physics. There, Weyl fermions are described by the Hamiltonian

$$\hat{H} = \pm \boldsymbol{\sigma} \cdot \boldsymbol{p}, \tag{10.10}$$

where $\boldsymbol{\sigma}$ is the vector of Pauli matrices and $\boldsymbol{p}$ is the momentum of the particle. The chirality is determined by the sign choice in the Hamiltonian (note that this agrees with the definition (10.8), if we set $A = \pm\mathbb{I}$ and $\boldsymbol{q} = \boldsymbol{p}$). When a field theory is put on a lattice, the infinite momentum space is replaced with a compact Brillouin zone. The theorem says that chiralities of massless Weyl fermions in any lattice model with odd spatial dimension must sum up to zero. It follows that any Weyl fermion must be accompanied by a fermion of opposite chirality, a phenomenon known as "fermion doubling".

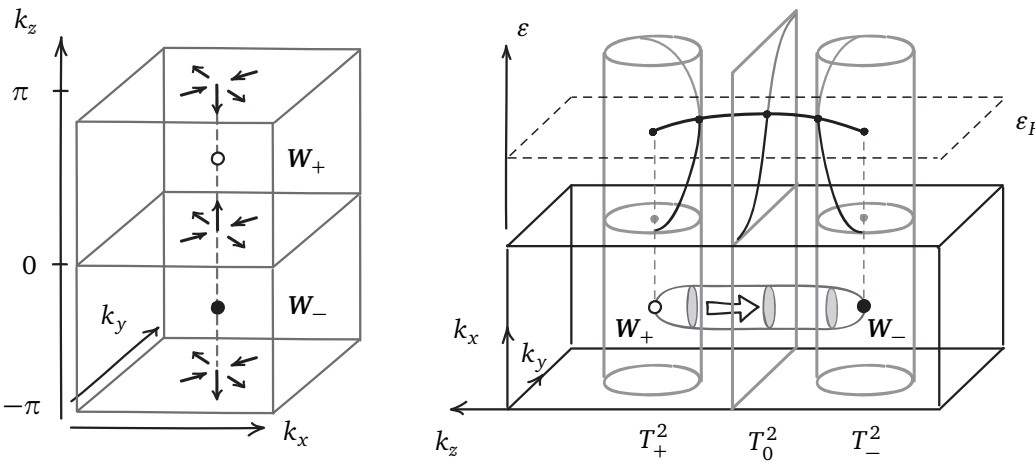

Figure 10.3: LEFT: Brillouin zone of the Weyl semimetal defined by Eq. (10.6), which extends the topological transition shown in Fig. 8.3 on the left. Filled and empty circles show positions of the Weyl points $W_\pm$. Arrows represent the vector field $\boldsymbol{h_k}$. RIGHT: Schematic picture of the Berry flux in the Brillouin zone of a Weyl semimetal and the spectrum of the surface states at the top surface of a crystal, which is finite in the $x$ direction. The arrow shows the Berry flux in the "flux tube" starting at $W_+$ and ending at $W_-$. The flux tube pierces three tori, $T_\pm^2$ and $T_0^2$, as indicated by gray discs. The Hamiltonian restricted to each of the tori describes a Chern insulator. The corresponding spectral branches of the edge states are shown in the upper part of the picture. Each of the branches intersect the Fermi level $\varepsilon_F$ in a point. Taken together, such points form a Fermi arc connecting the surface projections of the Weyl points.

For a Weyl semimetal, one can prove Eq. (10.9) by applying the divergence theorem to the Berry fluxes. Recall from Sec. 8.2.4 that each Weyl point acts as a source of the Berry flux, or

a monopole of the effective magnetic field. One can compute the number of such monopoles inside any closed two-dimensional surface by integrating the Berry curvature over the surface. This will give the total Berry flux, or $2\pi$ times the Chern number of the eigenspace bundle restricted to the surface. Now consider a sphere $S^2 \subset T^3$ inside the three-dimensional torus of the Brillouin zone. Suppose that the sphere $S^2$ is the boundary of a ball $B^3 \subset T^3$ that does not contain any Weyl points. Then the Berry flux though the sphere must vanish,

$$\int_{S^2} f_{12} dx_1 dx_2 = 0. \tag{10.11}$$

On the other hand, the sphere is also the boundary of the complement $T^3 \setminus B^3$ of the ball $B^3$ inside the torus $T^3$. It follows that the total charge of the Berry flux monopoles inside $T^3$ must be zero.

### 10.3.2 Symmetry considerations

Further constraints on the chiralities of the Weyl points are put by symmetries. Suppose that the crystal has inversion symmetry represented by $\sigma_x$ matrix. Then the conditions (9.5) imply that for each Weyl point $\mathbf{W}$ there is a Weyl point $-\mathbf{W}$ with the opposite momentum value, and two points have different chiralities:

$$\mathcal{I}: \qquad \chi[\mathbf{W}] = -\chi[-\mathbf{W}]. \tag{10.12}$$

On the other hand, the time reversal symmetry changes the sign of the $h_y$ component, as discussed in Sec. 5.2.2. When combined with the antipodal map $\mathbf{k} \to -\mathbf{k}$, this results in the Weyl point of the same chirality:

$$T: \qquad \chi[\mathbf{W}] = \chi[-\mathbf{W}]. \tag{10.13}$$

It follows that the minimal number of the Weyl points in a $T$-invariant crystal is four. As it turns out, the same rules apply in the case when electron's spin is taken into account.

> **Exercise 10.2.** Consider a Weyl point $\mathbf{W}$ with the outward-pointing vector field $\mathbf{h_k}$. Plot the corresponding vector fields for the symmetric images of the Weyl point at $-\mathbf{W}$ under inversion and time-reversal symmetries. Show that chiralities transform according to the rules stated above.

If a two-band Hamiltonian has both $\mathcal{I}$ and $T$ symmetries (or, at least, their combination), formation of Weyl points is prohibited: Eqs. (10.12) and (10.13) give together an impossible constraint. One checks that the model described by Eq. (10.6) breaks both symmetries; below, we consider examples of systems, in which one of the symmetries is preserved.

An inversion-symmetric Weyl semimetal must break time-reversal symmetry. We encountered an example of such system in Exercise 9.4. Consider two effective inversion-symmetric 2D Hamiltonians, $H_{k_x k_y}(k_z = 0)$ and $H_{k_x k_y}(k_z = \pi)$. The condition on the product of eight inversion eigenvalues $\nu = -1$ implies that the two Hamiltonians are characterized by the Chern numbers of different parity. It follows that the gap must close at some point $k_z \in (0, \pi)$. Due to the symmetry, there is another Weyl point at the opposite point of the Brillouin zone. This pair of Weyl points can mediate the process of changing inversion eigenvalues at the invariant momenta, as illustrated in Fig. 5 of Ref. [118].

To describe a $T$-invariant Weyl semimetal, we switch to a more physically relevant case of a spinful system. Recall that we constructed the Hamiltonian of a Weyl semimetal (10.6) by extending of the topological transition between a Chern insulator and a trivial insulator,

discussed in Sec. 8.2.4. Here, we consider a similar process, which starts from the quantum spin Hall state introduced in Sec. 9.4.2. There are two valence bands, which are degenerate at the TRIM due to the Kramers theorem (Sec. 5.2.3). A topological transition requires closing of the bulk gap, that is, creating a degeneracy between one of the valence bands and one of the conduction bands. As argued in Ref. [157], generically this happens at a pair of points $\pm\mathbf{k}$ away from TRIM. Then, adding a time-reversed copy of this process and interpreting the control parameter as $k_z$, we obtain a model of a $T$-invariant Weyl semimetal with four Weyl points.

### 10.3.3 Fermi arcs

The bulk structure of the Berry fluxes inside a Weyl semimetal results in the peculiar surface states, which assume the form of open one-dimensional Fermi surfaces. Right panel of Fig. 10.3 shows schematically the Brillouin zone of a Weyl semimetal with two Weyl points connected by a tube of the Berry flux (cf. right panel of Fig. 8.3). We are interested in the surface states on the top surface of the corresponding finite crystal. To find these states, consider a closed two-dimensional surface $\Sigma \subset T^3$ in the Brillouin zone. If the Hamiltonian is non-degenerate everywhere on $\Sigma$, it makes sense to compute the Chern number $c(V^v|_\Sigma)$ of the valence band bundle restricted to $\Sigma$; suppose that it is non-zero. If further the surface has the shape of a two-dimensional torus, $\Sigma = T^2$, we can interpret this torus as a Brillouin zone of a Chern insulator. Finally, if this effective Chern insulator terminates on the top surface of the original crystal, then there must be associated chiral edge states (recall that we used a similar reasoning when we considered surface states of a three-dimensional Chern insulator in Sec. 8.4.1).

With this in mind, we consider three closed surfaces. One is the torus $T_0^2$ defined by setting $k_z = 0$, and the other two are tori $T_\pm^2$ represented in Fig. 10.3 by the vertical cylinders surrounding two Weyl points. Since the Weyl points are connected by the Berry flux, the flux lines pierce all three surfaces. Thus, each of the surfaces supports a Hamiltonian of a Chern insulator with $|c| = 1$, with the sign depending on the orientation choices. Above the box of the Brillouin zone $T^3$, we plot the spectrum $\varepsilon(k_y, k_z)$ of the surface states. Each of the Chern insulators contributes a single chiral mode, which intersects the Fermi level $\varepsilon_F$ in a point. Now note that a similar argument applies to *any* cylinder containing a single Weyl point. By continuity, the intersection points form a one-dimensional line connecting the surface projections of the Weyl points. These lines are called **Fermi arcs**. There will be another Fermi arc at the bottom surface, and also a pair of arcs on the surfaces orthogonal to the $y$ direction. However, there need not be any arcs on the left and right surfaces, since projections of Weyl points to the $xy$ plane coincide.

The first Weyl semimetal discovered experimentally was TaAs [158, 159]. The Brillouin zone of this inversion-symmetry-breaking crystal contains 24 Weyl points of charge $\pm 1$, which gives an intricate structure of the Fermi arcs. In particular, some pairs of same-charge Weyl points project to a single point on the surface of interest. A thin cylinder containing such a pair is thus characterized by the Chern number $\pm 2$, so the projected point must be an origin of two Fermi arcs. Simultaneously with the discoveries in condensed matter, linear band intersections were found in a three-dimensional photonic crystal [160]. The structure of the crystal is inspired by gyroid, which is an example of triply-periodic minimal surface [161]. The crystal consists of two interpenetrating gyroid-like structures, one of which has defects breaking the inversion symmetry. It was found that the microwave-range transmission spectra of this structure contain four Weyl points.

## 10.4 From vector field $h_k$ to vector bundle

For two-band models, the information about band degeneracies is contained in the zeroes of the vector field $h_k$. Now let us consider this vector field as a section of a vector bundle, in the spirit of Sec. 1.1. In graphene, the fiber of the bundle in question is a two-dimensional real vector space with the basis $\{\sigma_x, \sigma_y\}$. For a Weyl semimetal, the fiber is three-dimensional, and the basis is $\{\sigma_x, \sigma_y, \sigma_z\}$. In both cases, we have a set of global basis sections, which makes either bundle trivial.

Dirac and Weyl points are the point-like singularities in the vector field $h_k$, and can be characterized by an index similar to one defined in Sec. 4.2.1. To this end, consider a sphere $S^n$ surrounding the point with a singularity ($n = 2$ for a Dirac point in graphene and $n = 3$ for a Weyl point). According to (2.32), the vector field $h_k$ on the sphere $S^n$ defines a map $h : S^n \to S^n$, where the second sphere belongs to the space of Pauli matrices. The index is defined as a degree of the map $h$. For $n = 2$, it is called a **winding number**, while for $n = 3$ it is known as a **topological charge**. If the index is $\pm 1$, the sign determines the chirality of the Dirac or Weyl point. The sum of indices of a section is again a topological invariant of the bundle and does not depend on the choice of the section. Since the bundles are trivial, the sum must be zero both for graphene and for a Weyl semimetal, which gives another argument for Eq. (10.9).

It is then natural to ask: is it possible that $h_k$ is a section of a *non-trivial* vector bundle? There are at least two examples of systems, in which this is the case; we will consider them below. In the first example, the non-triviality of the bundle forces the gap closing, while in the second case it renders the chirality of a Dirac point meaningless. Both examples are based on the geometric picture available only for simple two-band models. We will briefly discuss more general approaches in Sec. 10.5.

### 10.4.1 Bundle of symmetry-compatible Hamiltonians

Let us describe the bundle in more detail. Consider a two-band Bloch Hamiltonian

$$H_k = \sum_i h_i(k)\sigma_i, \tag{10.14}$$

and the corresponding vector field $h_k$ (we will use the two objects interchangeably). Suppose that the Bloch Hamiltonian has a symmetry $S$, which acts in the momentum space locally:

$$S_k H_k S_k^{-1} = H_k. \tag{10.15}$$

We are interested in the set of all Hamiltonians, which satisfy this symmetry constraint. First, choose a point $k$ and define the following set:

$$B_k^H = \{\text{all vectors } h_k \text{ satisfying the constraint (10.15) at } k\}. \tag{10.16}$$

Note that if $h_k, h_k' \in B_k^H$, then $h_k + h_k' \in B_k^H$; also, $\lambda h_k \in B_k^H$ for any real $\lambda$. Thus, for each $k$, we have a real vector space $B_k^H$, which is a subspace of the space of Pauli matrices. Taken together, these spaces form a vector bundle $B^H$ over the Brillouin zone:

$$B^H = \{B_k^H \mid k \in BZ\}. \tag{10.17}$$

We will call it the **bundle of symmetry-compatible Hamiltonians**.

For example, in graphene we have the symmetry $S = \mathcal{I} \circ T$, which acts in the momentum space locally, according to Eq. (5.49). As a result, the vector $h_k$ must lie in the $\sigma_x$-$\sigma_y$ plane. In other words, the space $B_k^H$ is spanned by the Pauli matrices $\sigma_x$ and $\sigma_y$. So, the bundle $B^H$ in this case is a real plane bundle, which is trivial, as noted above. The bundle $B^H$ can become non-trivial, if there is some twisting of the fibers, which requires the defining symmetry $S_k$ to be $k$-dependent.

### 10.4.2 Non-symmorphic symmetry group

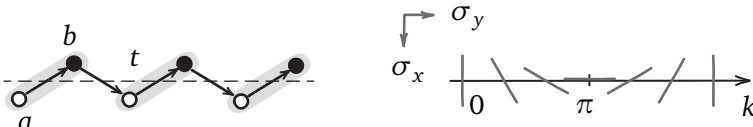

Figure 10.4: LEFT: Diatomic chain with a non-symmorphic symmetry group. Arrows show the hopping amplitude $t = t_1 + it_2$. RIGHT: Field of lines represents the real vector bundle $B^H$ of symmetry-compatible Hamiltonians.

A space group is called **non-symmorphic** if one cannot choose a point, which is fixed by all of its generating elements, not including lattice translations [162]. Examples of such groups include groups containing screw axes and glide planes. These operations combine a translation by a *fraction* of the lattice constant with a rotation or a reflection, respectively.

Consider the chain shown in Fig. 10.4 on the left. It is invariant under reflection with respect to the dashed line, combined with translation by a half of the unit cell. In the real space, the symmetry acts as

$$\hat{G}|a_m\rangle = |b_m\rangle, \qquad \hat{G}|b_m\rangle = |a_{m+1}\rangle. \tag{10.18}$$

In the momentum space, the matrix of the symmetry operator becomes

$$G_k = \begin{pmatrix} 0 & e^{-ik} \\ 1 & 0 \end{pmatrix}. \tag{10.19}$$

We rewrite it in the form of the unitary transformation (3.40):

$$G_k = e^{-ik/2}\begin{pmatrix} 0 & e^{-ik/2} \\ e^{ik/2} & 0 \end{pmatrix} = ie^{-ik/2}\left(\cos\frac{\pi}{2}\mathbb{I} + \sin\frac{\pi}{2}\,\boldsymbol{w}_k \cdot \frac{\boldsymbol{\sigma}}{i}\right), \tag{10.20}$$

where $\boldsymbol{w}_k = (\cos\frac{k}{2}, \sin\frac{k}{2}, 0)^T$.

Note that the symmetry $\hat{G}$ acts in the momentum space locally:

$$G_k H_k G_k^{-1} = H_k. \tag{10.21}$$

Conjugation of the Hamiltonian by $G_k$ describes the rotation through $\pi$ about the axis defined by $\boldsymbol{w}_k$. It follows that the vector $\boldsymbol{h}_k$ must belong to this line:

$$B_{\boldsymbol{k}}^H = \{\lambda \boldsymbol{w}_k \mid \lambda \in \mathbb{R}\}. \tag{10.22}$$

Now observe that when $k$ varies from 0 to $2\pi$, the line defined by $\boldsymbol{w}_k$ rotates by $\pi$. Thus, the bundle of symmetry-compatible Hamiltonians $B^H$ has the shape of the Möbius band, as shown in Fig. 10.4 on the right. One can consider $\boldsymbol{w}_k$ as its (discontinuous) basis section, and express the Hamiltonian as

$$\boldsymbol{h}_k = f_k \boldsymbol{w}_k, \tag{10.23}$$

where $f_k$ satisfies $f_{k+2\pi} = -f_k$.

**Exercise 10.3.** Find the function $f_k$ for the chain shown in Fig. 10.4 with complex nearest-neighbor hopping amplitude $t = t_1 + it_2$.

Recall from Sec. 4.1.2 that the Möbius band is an example of a non-trivial real line bundle: any section must have at least one zero. Here, a section $\boldsymbol{h}_k$ of a similar bundle $B^H$ describes a Hamiltonian compatible with the symmetry (10.21). It follows that for any such Hamiltonian there must be a point $k$ in the Brillouin zone, in which gap closes, $|\boldsymbol{h}_k| = 0$. In this way, a non-symmorphic symmetry group requires the presence of band degeneracy, but does not specify[15] the point $k$.

Alternatively, one can deduce this by observing the eigenvalues of $G_k$ (see Ref. [163] for a discussion in the context of topological semimetals). From $G_k^2 = \mathbb{I}e^{-ik}$ and $\operatorname{tr} G_k = 0$, one finds that the eigenvalues are $\lambda_{1,2} = \pm e^{-i\frac{k}{2}}$. As $k$ varies through the Brillouin zone, the eigenvalues switch places. Since one can label the Hamiltonian eigenstates by $\lambda_i$, the bands must also change their order in energy, and the gap must close somewhere.

### 10.4.3 Non-orientable real plane bundle

Consider now a two-band model with the combined $\mathcal{I} \circ T$ symmetry, which is discussed in Ref. [164]. The lattice is shown in Fig. 10.5 on the left. Let the inversion center lie on the $a$ site of the unit cell with coordinates $(m_x, m_y) = (0, 0)$. Then the inversion acts on the orbitals differently:

$$\hat{\mathcal{I}}|a_{m_x m_y}\rangle = |a_{-m_x, -m_y}\rangle, \qquad \hat{\mathcal{I}}|b_{m_x m_y}\rangle = |b_{-m_x-1, -m_y-1}\rangle. \qquad (10.24)$$

For site of type $a$ we have, as usual, $\mathcal{I}: \boldsymbol{m} \to -\boldsymbol{m}$, while for $b$ sites there is an additional shift.

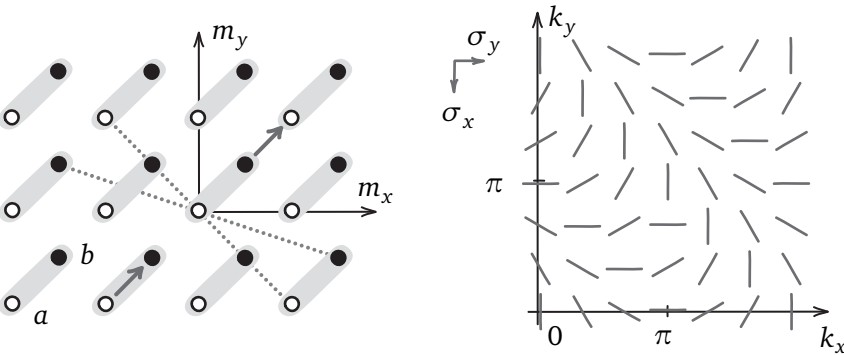

Figure 10.5: Two-dimensional crystal with the combined symmetry $\mathcal{I} \circ T$. LEFT: Checkerboard lattice. Dotted lines connect the sites related by the inversion symmetry. Arrows show purely imaginary hopping amplitudes, which break $\mathcal{I}$ and $T$ individually, but are invariant under $\mathcal{I} \circ T$. RIGHT: The corresponding bundle $B^H$ consists of real planes, each of which contains the vertical axis $z$ and intersects the horizontal $xy$ plane along one of the lines shown in the figure.

The matrix of the inversion operator in the Bloch wave basis reads

$$\mathcal{I}_k = \begin{pmatrix} 1 & 0 \\ 0 & e^{i(k_x+k_y)} \end{pmatrix} = e^{i\frac{k_x+k_y}{2}} \begin{pmatrix} e^{-i\frac{k_x+k_y}{2}} & 0 \\ 0 & e^{i\frac{k_x+k_y}{2}} \end{pmatrix}. \qquad (10.25)$$

Any Bloch Hamiltonian compatible with the $\mathcal{I} \circ T$ symmetry must satisfy

$$\mathcal{I}_k \overline{H_k} \mathcal{I}_k^{-1} = H_k. \qquad (10.26)$$

---

[15] In contrast with another type of symmetry-enforced degeneracies, which arise due to the presence of a high-dimensional irreducible representation of the symmetry group at a specific point $k$.

In the space of Pauli matrices, transformation on the left is the reflection in the $xz$ plane (complex conjugation) followed by the rotation through an angle $(k_x + k_y)$ about $z$ axis (conjugation by $\mathcal{I}_k$). It follows that at each $\boldsymbol{k}$, the vector $\boldsymbol{h_k}$ must lie in the plane that contains $z$ axis and a line in the $xy$ plane from the family shown in Fig. 10.5 on the right. The vector field $\boldsymbol{h_k}$ that describes the Hamiltonian is thus a section of a real plane bundle $B^H$ over a two-dimensional torus, and belongs to the class $(n, d) = (2, 2)$. Hence, one can expect appearance of locally stable degeneracy points. On the other hand, the Hamiltonian need not be gapless: the field $\boldsymbol{h_k} = (0, 0, 1)^T$ corresponds to a gapped Hamiltonian compatible with the $\mathcal{I} \circ T$ symmetry. Smooth deformations of the Hamiltonian can create and annihilate pairs of Dirac points, similar to the situation in graphene and Weyl semimetals. Still, the present case is different in one important aspect.

Recall that a vector bundle is non-trivial, if one cannot define $n$ nowhere-vanishing linearly independent sections, where $n$ is the dimension of the fiber. Here, the Möbius-like twists of the family of planes shown in the right panel of Fig. 10.5 prevent one from defining a global basis, so the bundle $B^H$ is non-trivial. In contrast with line bundles discussed in Sec. 4.1.2, for which $n = 1$, the non-triviality here does not force a section to have singularities. It manifests itself by making the bundle **non-orientable**. Due to the twists of the fibers by $\pi$, one cannot make a uniform choice of their orientation. In other words, we have a real plane bundle without a global field of plane normals $\boldsymbol{n}$. It follows that one cannot define *chirality* of Dirac points, or signs of winding numbers, in a global self-consistent way. The absence of the well-defined chirality can lead to apparent paradoxes of non-conservation of winding numbers in creation and annihilation processes for Dirac points. The authors of Ref. [164] consider such situation and resolve the problem by introducing "winding vector" as an additional parameter of a Dirac point. In our terms, this vector is the plane normal $\boldsymbol{n}$, which allows one to define orientation of fibers $B_k^H$ locally.

## 10.5 Summary and outlook

Above, we introduced basic examples of topological semimetallic systems, in which bands intersect at the Fermi level. The main takeaways of our discussion are:

- Depending on the dimensionality of the system, and on the number of components and symmetry of the Hamiltonian, a band structure can have nodal features of various types.

- These gapless objects are stable against local symmetry-preserving perturbations. Point-like nodes can be destroyed only by annihilation with a node of opposite chirality.

- Nodal line semimetals have drumhead surface states, which arise inside the projections of the nodal loops due to the quantized Zak phase.

- In Weyl semimetals, the Weyl points act as sources or sinks of the Berry flux, depending on their chirality. The sum of chiralities of all Weyl points is zero.

- In the surface spectrum, projections of Weyl points are connected by Fermi arcs. They originate from the chiral edge modes of the effective Chern insulators, which arise due to the Berry flux lines connecting the Weyl points.

- A non-symmorphic symmetry group can lead to the formation of globally stable band crossings.

Topological semimetals are in the focus of ongoing active research. A general overview of theoretical and experimental developments is given in Refs. [165] and [166]. The interplay of symmetries and nodal features leads to a rich variety of phases. Lecture [167] provides a

systematic discussion of the stable band intersection points in presence of fundamental symmetries and non-symmorphic spatial symmetry groups. This lecture also contains realistic models of nodal line semimetals. A completely different perspective on topological semimetals can be found in Ref. [168]. This review gives insights into chemist's intuitive approach to band structures, which is based on atomic orbitals and bonds rather than on the Bloch Hamiltonian.

Below, we briefly discuss several more aspects of topological semimetals. But before that, we make a general remark on a connection between different topological systems.

▷ **Dimensional hierarchy.** In many cases, models of topological matter can be linked to each other by extending or reducing dimensions. Let us review the chain of such extensions, which led to the tight-binding model of a Weyl semimetal given by the Bloch Hamiltonian (10.6). First, consider a single dimer, which consists of two atomic orbitals and is described by the real-space Hamiltonian $H$ given by Eq. (5.1). This is a zero-dimensional system. Suppose that the Hamiltonian depends periodically on a parameter $p$, so that $H(p) = H(p+2\pi)$. Next, interpret this parameter as the crystal momentum in a diatomic one-dimensional chain, which results in a Bloch Hamiltonian $H(k_x)$. Now, consider a periodic variation of this Hamiltonian $H(k_x, p)$, which describes a charge pump. By a similar identification, we turn it into a Hamiltonian of a Chern insulator $H(k_x, k_y)$. Finally, a periodic variation $H(k_x, k_y, p)$, which includes changing of the Chern number, leads to a model of a Weyl semimetal $H(k_x, k_y, k_z)$.

The process in the other direction is known as the dimensional reduction. It gives a useful tool for analysis of topological matter. For example, both two- and three-dimensional $T$-invariant insulators discussed in Sec. 9.4 can be thought of as a result of the dimensional reduction of a four-dimensional parent phase [169]. In the context of fundamental symmetries, dimensional reduction has been used to explain the periodicity in the tenfold-way classification [115].

▷ **Nodal points and Berry phase.** The first application of dimensional arguments to the accidental degeneracies between energy levels dates back to the work of Wigner and von Neumann [170]. In particular, they showed that a real two-band Hamiltonian depending on two parameters has locally stable point-like degeneracies. Later it was found that one cannot continuously define a *real* eigenstate of such Hamiltonian over a loop, which goes around a single degeneracy point [171]. Like a vector transported inside the Möbius band bundle, the eigenstate acquires the minus sign after going around a loop. In this way, topology of a real line bundle defined by the eigenstates detects the presence of a band degeneracy inside the contour.[16] This phenomenon played a key role in the discovery of the geometric phase by Berry [172].

Interestingly, this sign change can be interpreted in terms of the Berry phase, as follows. If we relax the reality condition on the eigenstate, it can be made continuous by adding an appropriate complex phase. What is the geometric phase associated with this eigenstate along the contour enclosing the degeneracy point? To find the answer, we adopt the discrete formulation of the geometric phase, which is especially convenient for numerical implementations [9]. In this setting, the parallel transport of an eigenstate over the discrete mesh of $k$-points is defined by the following condition: the inner product $\langle \psi_{k+\Delta k} | \psi_k \rangle$ must be real and positive. Now note that the real eigenstate satisfies this condition automatically. Thus, the state changes its sign as a result of the parallel transport along the contour. In other words, it acquires the $\pi$ Berry phase (see also Ref. [173]).

---

[16]In the light of discussion of tautological bundles in Sec. 4.5, this is analogous to how the topology of a *complex* eigenspace bundle over a sphere $S^2$ detects the presence of band degeneracies inside the region bounded by the sphere.

A familiar example of this situation occurs in graphene. If we go around any loop containing a single Dirac point, the vector $\boldsymbol{h_k}$ will make a full turn in the $\sigma_x$-$\sigma_y$ plane, which results in the Berry phase $\frac{\Omega}{2} = \pi$. This phase was experimentally observed by the momentum-space Aharonov–Bohm interfererometry in an "artificial graphene" made from ultracold atoms in an optical lattice [174].

▷ **Real vector bundles.**    In a spinless system with combined $\mathcal{I} \circ T$ symmetry, one can choose a basis in which the Hamiltonian will be real (this is also true for $C_2 \circ T$ symmetry in two dimensions, regardless of spin; here, $C_2$ is the $\pi$ rotation). Then there is a gauge in which the Hamiltonian eigenstates are also expressed as real vectors. The topology of real eigenstate bundles contains information about the band degeneracies, or nodal features. This approach is not limited to the two-band geometric picture used in Sec. 10.4 and allows for generalizations to the multi-band case.

Just as complex bundles are characterized by Chern numbers, the topology of real bundles is described by Stiefel–Whitney and Euler classes. The first Stiefel–Whitney class detects if the bundle is orientable. In the context of the topological band theory, it corresponds to the Berry phase along non-contractible loops in the Brillouin zone, which assumes quantized values 0 or $\pi$ in the real case, as discussed above. The geometric meaning of the second Stiefel–Whitney class is more subtle. In the physical terms, it determines the $\mathbb{Z}_2$ charge of a nodal ring, in the following way. A nodal ring considered in Sec. 10.2 can shrink to a point and disappear, leaving a trivial insulator. However, some nodal rings in multi-band systems cannot disappear in this manner: one says that they carry additional charge. The second Stiefel–Whitney class of the eigenstate bundle over the sphere containing a nodal ring corresponds to this charge. These and other applications of Stiefel–Whitney classes are reviewed in Ref. [175].

Euler class of a bundle generalizes the Euler characteristic $\chi(\mathcal{B})$ of a closed manifold $\mathcal{B}$. According to the Poincaré–Hopf theorem, $\chi(\mathcal{B})$ measures the number of zeroes in any tangent vector field on $\mathcal{B}$. In the band theory, the Euler class of an eigenspace bundle associated with a two-band subspace determines the number of nodal points between these bands. One example of a system with non-trivial Euler class is the twisted bilayer graphene: if we focus on a certain pair of bands in one valley, the sum of charges of Dirac points will be non-zero, effectively violating the Nielsen–Ninomiya theorem [176]. In semimetals with real Hamiltonians, the Euler class restricts the possibility of Weyl points to annihilate and to transform into nodal lines [177]. Three- and four-band representative models containing two-band subspaces with non-trivial Euler class are constructed in a recent work [178]. In wider context, Euler class has been used to characterize vector bundles associated with the spatial distribution of an order parameter, such as magnetization in ferromagnets and director field in cholesteric liquid crystals [179].

A brief physicist-oriented discussion of Stiefel–Whitney and Euler classes can be found in Ref. [5]. In mathematics, this topic belongs to the theory of characteristic classes, which is developed in textbooks [6] and [7].

▷ **Dirac semimetals.**    Note that if both time reversal and inversion are present in a *spinful* system, the combination $\mathcal{I} \circ T$ is an anti-unitary operator, which squares to minus identity and acts in the momentum space locally. It follows from the Kramers theorem (Sec. 5.2.3) that each band must be at least two-fold degenerate at each point of the momentum space. So, a band intersection leads to a four-fold degeneracy. According to the dimensional argument from Sec. 10.1, such band crossing is unstable, since there are more parameters of the Hamiltonian then there are dimensions of the momentum space. However, these intersection points can be stabilized by additional spatial symmetries, thus giving rise to the **Dirac semimetal** [180].

In a sense, a Dirac[17] point is a combination of two Weyl points of opposite chirality, which are mapped to each other by inversion (10.12) and time reversal (10.13), thus satisfying both constraints.

In contrast with Weyl semimetals, Dirac semimetals do not have *robust*, topologically protected surface states. The states analogous to the Fermi arcs may appear at the surface, but they can be deformed into closed rings and then removed, leaving only the surface projections of Dirac points [181]. Recently, it has been shown theoretically that certain Dirac semimetals do have robust surface signatures of non-trivial topology: the higher-order Fermi arcs [182]. Like ordinary Fermi arcs in Weyl semimetals, they connect the boundary projections of the bulk nodal points. However, these states are localized at the *hinges* of the crystal. This result connects the filed of topological semimetals with the notion of higher-order topology [183]. Informally, higher-order topological phases have the same kind of relationship with the theory of multipole moments (see Sec. 6.5), as Chern insulators have with the modern theory of electric polarization. For an overview of phenomenology of various topological phases, including semimetals and higher-order phases, see Ref. [184].

▷ **Chiral anomaly.** Nielsen–Ninomiya theorem is not the only result from high-energy physics, which finds its counterpart in Weyl semimetals. The same authors predicted that a pair of accidental degeneracies between two bands near the Fermi level can lead to the realization of the chiral anomaly [185].

Under an external magnetic field, the band structure of a Weyl semimetal turns into a set of Landau levels. Near each of the Weyl points, special levels appear: they connect the valence and conduction bands, similarly to the chiral branches of the spectrum at the edges of Chern insulators. Depending on the chirality of the original Weyl point, the resulting level goes up or down in energy as a function of crystal momentum parallel to the applied field. Now recall from Sec. 8.3.2, that the electric field shifts the crystal momentum of the occupied states, which changes the number of electrons on both edges of a Chern insulator (the electric field was described there in terms of the flux insertion). In a similar way, if we put a Weyl semimetal into the parallel electric and magnetic fields, this will lead to the redistribution of electrons between the Weyl points. If we focus on a single Weyl point, this process will look like (dis)appearance of fermions of certain chirality. In other words, locally this will break the conservation of the chiral charge. In particle physics, a similar effect is known as the chiral anomaly, which affects the lifetime of neutral pions. For an overview of transport properties of Weyl semimetals with the emphasis on the chiral anomaly, see Refs. [186] and [187]. A discussion of related experimental results can be found in a review [188].

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
