# Peer review of "Topological insulators and geometry of vector bundles"

_SciPost Physics Lecture Notes, doi:SciPost Phys. Lect. Notes 67 (2023)_

## Round 1 · Referee Report · Anonymous (Referee 1) · 2021-1-17

Strengths

See report.

Weaknesses

See report.

Report

This manuscript is a review of topological insulators from the formal perspective of vector bundles. The review is nicely illustrated. It starts from simple situations and introduces various concepts later used in the analysis of topological insulators. An effort has been made to focus on intuitive arguments instead of theorems and proofs, and to make the review self-contained.

On the whole, the manuscript would be strengthened if the target audience was clearly identified: Who are the readers? What are the prerequisites for reading this review? (I raise these questions because the answers provided in the introduction do not seem to me entirely compatible with the contents of the manuscript.) Indeed, there are already several reviews and books on topological insulators and related topics. What potential readers would benefit from reading this one? The current version appears to target an audience of physicists without expertise in topological insulators; this audience would benefit from more thorough and introductory discussions of the physical examples (Foucault pendulum, Aharonov-Bohm effect, polarization of crystals, TI, etc.) mentioned, as well as of the physics of TI themselves. The lack of formal statements (exact definitions, theorems, proofs) suggests that this is intended for physicists and not mathematicians. Yet, the manuscript mostly focus on formal aspects.

Overall, I found most of the physics discussions too superficial. A more precise, down to earth discussion might perhaps seem tedious, but would have a substantial pedagogical value. More generally, the formalism is often covered, but the background, context, significance only quickly mentioned. More discussions on the experimental side would also be welcome.

One aspect that could be discussed further is: why vector bundles? There are other approaches to TI, arguably as widely applicable as vector bundles.

There are several places in which more references would be useful: to support statements that might not be obvious to the readers, to provide more details, and to refer to more rigorous sources when intuitive arguments are used.

I also suggest to pay special attention to internal references. When a definition or a result is used, it is very helpful to refer as precisely as possible (equation number, section, page, etc.) to the location in which this definition or result is stated. This is particularly crucial in a long document.

The exercises are a neat way to get the readers involved with the material, but I strongly suggest providing solutions. This is particularly important as the results of these exercises are used in the remainder of the notes.

The figures would be improved by having more detailed captions. It can be useful to label the different panels of a figure so that one can refer to them individually in the caption.

Overall, the manuscript is indeed valid. My assessment is that the majority of the issues raised in the detailed comments below pertain to clarity, and can be corrected with relative ease. I however found a few statements (detailed below) that I don't understand, and that I believe to be at least very misleading. These need to be carefully corrected.

There are a few typos, some sentences are not grammatical, and there is a recurring issue with missing articles (like "a" or "the"). A global proofreading would be useful.

Detailed page-by-page comments follow.

  • page 3 Explain "which is known to be physically irrelevant".

What is the meaning of the comment "while the properties ... essentially quantum mechanical"? As illustrated by the discussion that follows (I would cite reviews on these topics), these properties are not "essentially quantum mechanical" at all.

  • page 4 Explain "wide range of perturbations"

  • page 5-6 The introduction of the first paragraph (before 1.1) requires rephrasing.

  • page 6 Note that bundles do not have to be vector bundles.

I suggest distinguishing local and global sections.

  • page 7 The definitions in 1.1.2 are not intrinsic (they require some kind of ambient bundle)... this might be OK for the sake of pedagogy, but could be clarified.

  • page 8 In 1.2.1, "as in the case of the flat bundle V, we can assume that these Cartesian bases are constant" requires further explanation. What allows one to do so?

  • page 9 Step 3 requires more explanation.

In 1.2.1 and later, why is $\phi$ singled out? What about $\theta$? It is not clear what is specific to the example, and what is general. A general definition of the covariant derivative and other notions, that can be applied besides this example, would be very welcome.

In 1.2.2, please define "the standard basis" when used.

  • page 10 In figure 1.3, the image suggests $\nabla_\phi e_\theta = \nabla_\phi 1$ and $e_\theta = 1$; is it intended?

  • page 11 Why not considering $\beta(\phi, \theta)$?

The notations bold 1 and 1' might be confusing. At the very least, an explanation of this choice and of how not to get confused would be necessary.

Is there a sum implied in (1.26)? (The same question occurs at several places.)

What does "now that we have a differential operator" mean? What need does the differential operator fulfill?

Please consider the following question: reading up to the end of page 11, I want to evaluate (1.27): what definition shall I use? To help the readers, especially those not familiar with the concepts discussed, similar questions should be considered throughout all the lecture notes, and addressed with comments and references to the definitions.

  • page 12-13 The paragraph 1.3.1 on physical interpretation would gain from more work.

In figure 1.5, what is the plane? What does it represents?

The gyroscope discussion is interesting, but precisely how the internal motion is restricted matters. This would be useful to discuss.

The paragraph on the Foucault pendulum is way too short. It would help a lot to carefully describe the situation, to explain what happens, how it is modeled, what are the results, and then how it relates to parallel transport. I advise targeting readers who are not already familiar with the Foucault pendulum.

What does $v_{PT} = \exp(i \alpha(\phi)) 1$ represent?

  • page 14 Is $\gamma$ a direction or a curve?

  • page 15 I'm unsure what the reader should understand and remember from 1.4.3, as it mostly consists in stating what will not be discussed. (References could be useful to interested readers, if this paragraph remains.)

  • page 16 An explanation of what a gauge field is would be very useful, especially in relation with bundles. Even if the electromagnetic case is well-known, defining and explaining the electromagnetic potentials would be useful from a pedagogical point of view.

  • page 17 The basis section $1$ can be chosen only if it exists, how is it guaranteed? It is not clear that the last paragraph before 2.2 can be understood without prior knowledge, it would gain from being more pedagogical.

  • page 19 The definition of the curvature is not clear. This is problematic as it is a crucial concept. I didn't find a clearly delineated definition such as "$f_{ij} = \partial_j \omega_i - \partial_i \omega_j$". Currently, it is implicit and the readers have to generalize themselves. In general, I suggest repeating the important definitions and results outside of the scope of illustrative examples, and emphasizing their level of generality.

  • page 20 It can be difficult for readers to address exercises such as "explain the discrepancy in the results", because they often don't follow the same path as the lecturer. I advise against this kind of exercise.

  • page 22 What is $\Sigma$? It is probably defined several paragraphs above, but this is not clear where at this point.

  • page 27 Examples of typos (suggested changes in capitals): "THE vector potential" "THE eigenstate wave functionS and THE corresponding momentum eigenvalueS are"

Please explain (and define) "winding number".

  • page 28 The comment "this is not a gauge transformation" requires precise definitions and a discussion.

  • page 29 The discussion in the last paragraph nicely illustrates the need of vector bundles to describe this situation. This could be emphasized.

  • page 30 The phrasing "according to Feynman" could give the impression that this is Feynman's (the person) opinion.

Why quoting "interferences"?

  • page 31 The example of Aharonov-Bohm interferences would gain from a more detailed analysis and discussion. For instance, one could compute the interference pattern, discuss how it changes, and explain the underlying physics. Currently, it is not enough.

  • page 34 What does "this electromagnetic interpretation of the Berry connection is taken literally by electrons themselves" mean? (The whole paragraph requires a bit of work.)

  • page 37 The "integer c(M)" (the first Chern number) could be named there already.

The paragraph starting with "this topological point of view..." is not clear.

  • page 41 Note that this is the first Chern number. There are other Chern numbers. It might be useful to emphasize the general hypotheses required for the first Chern number to be defined.

What are physical consequences in the case of $TS^2$?

  • page 44 The discussion between 4.29 and 4.30 is not clear. The footnote requires either more discussion, or less discussion (and references). Currently, it might be a bit misleading because there are various details involved (e.g. Chern classes and characters, and so on) that are not discussed.

  • page 50 A figure would be welcome to explain the polarization and corresponding notions. More generally, a more careful and detailed physical discussion would be very useful.

The last paragraph of 5.2.1 is not clear.

How is the "non-polar initial state" defined? What does "defining property" mean?

  • page 50-52 The paragraph 5.2.2 is confusing, and perhaps incorrect. As the unit cells are always disconnected, how can there be a net current? As far as I see, the electrons go back and forth, and there is no pumping.

In figures 5.2, 5.3, and so on, the meaning of all the lines in $e|\psi|^2$ should be explained. Also, something more precise than "color of circles indicate on-site potentials" is necessary (for instance, what do black/gray/white mean?).

  • page 52 The whole paragraph 5.3 is too long and not clear enough. It requires editing.

The notation $x^n$ looks like the n-th power of $x$.

  • page 54 In the discussion after 5.34, one can note that $\tilde{D}_k$ is directly related to the position operator, while this is not the case for $D_k$ (see for instance E. I. Blount 1962). It is likely the reason why it is used here. Can this be discussed?

  • page 55 Would it be possible to avoid phrasing such as "a little reflection shows"?

  • page 58 I do not understand the footnote 6.

  • page 60 In (6.1), what is the Hamiltonian? More generally, in the whole lecture notes, it would be useful to refer to equations (or other entities) that are not immediately close to the current discussion, when they are mentioned.

  • page 61 What are the unusual states? Eigenstates? Or something else?

  • page 62 What is the initial condition?

In figure 6.3, what is the meaning of the arrows? A more complete description of the figure is necessary.

  • page 63 The chains with different ends are different systems: adding one site is acceptable in itself, but then we get a different system; both systems could be described and analyzed.

  • page 64 The figure requires a detailed caption. It is not entirely clear what the symbols, different kinds of lines, and so on, mean.

  • page 65 A discussion of the physical implications and applications would be great.

  • page 66 Note that the first Chern number is not necessarily associated to momentum space: it is just a convenient way of computing it.

  • page 67 The total number of edge modes is on one edge only.

A picture of an edge mode (in physical space) would be welcome, as well as a discussion of the physical consequences (including, maybe, experiments).

  • page 68 The discussion about the localization of Wannier function requires references. This is also true for the bulk-boundary correspondence.

The remark about time-reversal symmetry is not clear.

  • page 70 The figure 7.3 is not clear.

What is the aim of paragraph 7.2?

It should be noted that the "unique chiral metal" at the edge of a Chern insulator can be realized as a 1D system, provided that some hypotheses are lifted (such as the unitarity of the evolution).

  • page 72 Can "direct sum" be explained a bit more?

A discussion of fragile phases would be interesting, and more literature could be mentioned.

  • page 74 The conclusion "Chern insulators can be classified by the number of the chiral edge modes" might be misleading: this is not a bulk quantity. It does not characterize a Chern insulator but the interface between two insulators.

I disagree with the statement: "the existence of Chern insulator does not require any symmetry, except the discrete translational symmetry of the crystal lattice" It is possible to define Chern insulators without any reference to a lattice, for instance through Laughlin argument, or by using a more general expression of the 1st Chern number, or by using equivalent invariants based on scattering matrices, or on response functions, and so on. The integer quantum Hall effect is characterized by a 1st Chern number, but it is a disordered 2D electron gas, not a system on lattice.

  • page 75 It might be more clear to write something like $\psi_\beta$ instead of just $\beta$, especially when a sum is involved.

  • page 76 Where should one set $\tau_a = \tau_b = 0$? (Again, this might not be the only instance in which a precise reference to previous equations/discussion is required. I did not take note of all of these.)

  • page 77 I don't understand the argument in "polarization of a finite chain". On the one hand, the symmetry is said to be broken, but on the other hand the symmetry is invoked to imply $v=0$.

  • page 78 What are "the classical reasons"?

  • page 79 Anti-linearity also requires $K(\psi+\chi) = K(\psi) + K(\chi)$.

References would be welcome.

  • page 80 What does "degeneracy follows from their orthogonality" mean?
    The degeneracy of $\psi$ and $T\psi$ follows from (8.18). It would simply be a trivial statement if $\psi$ and $T\psi$ were not linearly independent.

  • page 82 The discussions in 8.3.2 and 8.3.3 are very short, and leave out a lot.

It is also a bit unclear to what extent the point of view of vector bundles is present in this section. It is indeed possible to address the issue of topological insulators with symmetries from the point of view of vector bundles, by endowing them with an appropriate structure (see for instance works from Giuseppe De Nittis & Kiyonori Gomi and from Daniel S. Freed & Gregory W. Moore), but this is not discussed at all. I do not imply that it should necessarily be discussed, but again, I don't fully understand the perspective adopted in these notes.

Also, what would happen if inversion symmetry was not present? A key point of Kane-Mele topological insulators is that they only rely on time-reversal invariance, so it is still possible to define a $Z_2$ invariant even when inversion symmetry is absent. This is a rather important point, that I should be discussed.

  • page 84 The notion of Dirac cone was never introduced.

In 8.4, it might be interesting to explain the name of the "periodic table". Here, again, more references would be welcome.

The discussion at the end of 8 on pages 84-85 is interesting, but is probably too quick for a reader not somehow familiar with the topic.

  • page 85 The notions of crystallography introduced here are certainly useful (references would be beneficial for readers not familiar with these notions), but perhaps they would gain from being introduced earlier in the lecture notes, so that examples of the various topological systems discussed could be given in the corresponding parts. (This could also be useful for the section on polarization.)

It is a bit strange to see the Haldane model in the section on semimetals, but not on the section on Chern insulators. Again, my impression is that a careful assessment of the target audience and of the aims of the lecture notes would lead to a better organization.

Globally, a notable part of the paragraph on semimetals is not directly related to semimetals themselves.

  • page 86 A plot of the band structure of graphene would be useful.

  • page 87 The first paragraph of 9.1.2 is not clear. Details are required: it is very difficult to follow this kind of hand wavy discussion where any detail or actual computation is avoided. Graphene would be a nice occasion to discuss the Zak phase.

  • page 88 An explanation of why the Haldane model is so celebrated would be useful. One could also mention experimental realizations.

  • page 89 It would be useful to expand on the significance of the discussion in 9.1.3.

The section 9.2 on Weyl semimetals is globally unclear. My advice is to rework it completely.

Among other issues: I don't understand the point of the list on page 90. Type-II Weyl points are somehow defined, but not type-I (are there type III too?). The two band Hamiltonian describing a single Weyl point, either in the form $k_x \sigma_x + k_y \sigma_y + k_z \sigma_z$ and/or $v_{i j} k_i \sigma_j$ (as well as the link with the 1st Chern number) should probably be written somewhere in a section of Weyl points. The discussion of Fermi arcs is also not very clear, and the figure 9.4 is not very helpful in conveying what is going on.

  • page 91 The section 9.3 contains material that goes beyond mainstream topological insulators or semimetals, including novel interpretations (according to the introduction of the manuscript) that would be more suitable for a research article.

I do have concerns about this section and advise its removal.

First, it is not clear what is the purpose of this section. The topics are very briefly touched upon, without a sufficient discussion to gain an understanding of the topics discussed, which would all require a notably longer treatment. For instance, strictly speaking, the notion of symmorphic/nonsymmorphic applies to space groups, not to operations (see the IUCR Dictionary: https://dictionary.iucr.org/Symmorphic_space_groups). All of this requires more discussion than what is possible to convey in half a page.

Besides, as mentioned e.g. in Dániel Varjas et al 2018 New J. Phys. 20 093026, "nonsymmorphic symmetry operations only acquire k-dependence in the form of an overall phase factor" in an appropriate representation. This raises important issues about the discussion in this section. Perhaps one can address these issues, but again, this is in my view outside the scope of lecture notes.

Requested changes

I cannot recommend publication of the manuscript in its current state.

I believe that it is possible to bring this manuscript to a state in which I could wholly recommend publication, provided that both the high-level issues (about the audience, the perspective, and the goal of the lecture notes) and the more specific ones are both taken into account and addressed. This is likely to require a substantial amount of work, rather than only small corrections. I encourage the author to undertake this work.

  • validity: good
  • significance: ok
  • originality: low
  • clarity: ok
  • formatting: good
  • grammar: acceptable

Author:  Alexander Sergeev  on 2021-01-25  [id 1180]

(in reply to Report 1 on 2021-01-17)

I thank the Referee for the detailed report, which highlights many areas of possible improvement. I will do my best to resolve all the issues in the next version of the notes.

I would like to comment on several points made in the report. The source of these issues is probably the assumed context that is not explicitly described. It is now clear to me that a similar problem occurs at other places, too, and I will pay special attention to fix it.

page 50-52. The paragraph 5.2.2 is confusing, and perhaps incorrect. As the unit cells are always disconnected, how can there be a net current? As far as I see, the electrons go back and forth, and there is no pumping.

Here, the terms "current" and "pumping" refer to a single event of charge redistribution between the two sublattices.

page 74. I disagree with the statement: "the existence of Chern insulator does not require any symmetry, except the discrete translational symmetry of the crystal lattice" It is possible to define Chern insulators without any reference to a lattice, for instance through Laughlin argument, or by using a more general expression of the 1st Chern number, or by using equivalent invariants based on scattering matrices, or on response functions, and so on.

Earlier in the notes, the Chern insulator is defined as a crystal with a non-trivial bundle of valence band eigenspaces (with further implicit assumptions of no interactions and disorder).

page 77. I don't understand the argument in "polarization of a finite chain". On the one hand, the symmetry is said to be broken, but on the other hand the symmetry is invoked to imply $v=0$"

The bulk theory knows only about one unit cell with periodic boundary conditions. On this level, the chain is inversion-symmetric, which requires $P=0$ or $P=\pm e/2$. One can see that these values agree with the surface charges of the finite chain. The polarized finite chain as a whole has a well-defined dipole moment and must break inversion symmetry.

The section 9.2 on Weyl semimetals is globally unclear. My advice is to rework it completely.

The main idea of the section 9 on topological semimetals was to explain in which sense a Weyl semimetal can be thought of as a three-dimensional generalization of graphene. The reason is the "co-dimensional argument": when the number of vector field components coincides with the dimension of the base space, there are locally stable point-like singularities. Then it is natural to ask (at least in the context of the notes): is it possible that these vector fields are sections of non-trivial bundles? Now I see that this line of thought is obscure and that the discussion can be confusing from the physical point of view.

---

## Round 1 · Referee Report · Anonymous (Referee 2) · 2021-2-16

Strengths

self-contained, requires minimum prior knowledge (being familiar with TB approximation), for the most part (especially the math sections) it is very clearly written

Weaknesses

limited novelty and originality, several misconceptions in the theoretical presentation, unclear motivation for publishing a new lecture note in a journal when this is a largely established topics, insufficient referencing, the generalization beyond Chern number (Secs. 8 and 9) have rather unsatisfying quality, little reflection on the more recent developments

As for the three supposed "novelties": Sec. 3.1 is hard to comprehend, and Sec. 9.3 is most certainly not new -- see report.

Report

I have studied the lecture notes “Topological insulators and geometry of vector bundles” by Dr. Sergeev.

Given that there is already much introductory material either on topological insulators (e.g. the book by Bernevig) [14], theory of Berry phase in solids (e.g. book by Vanderbilt [11]) and introductory material on vector bundles (e.g. the book by Baez [7]), as well as plentiful review articles, it seems difficult to justify the publication of a yet another lecture note that combines these topics. (In contrast, accessible account of the various *advanced* topics in topological band theory, such as the K-theoretical classification, Floquet and non-Hermitian systems, cohomology description of SPTs, etc… is currently missing, but the submitted manuscript is very remote from all these topics. In fact, some of the comments esp. in the footnotes left with me questioning the author’s deeper understanding of the discussed material beyond that explicitly discussed in the text.)

I therefore fear that the “market” for this short lecture note is far more narrow than the author thinks (although I imagine master-level students entering the field of topological bands who could benefit from a single concise account of the concept of vector bundles within topological band theory).

I further find that the submitted material contains a very limited amount of original explanations which do not already appear in some of the cited references. For these reasons, I was lead to describe the scientific “significance” and “originality” both as “marginal”. Thus, although the lecture note is self-contained and accessibly written, I find it rather difficult to justify its publication in a journal. I should also remark that the work also contains many imprecise or misleading statements (I am sure that my long list below is still very incomplete), and left me with the feeling that the quality of the material would be greatly improved if the author were to repeat his lecture series in the future (and resubmitted to the journal only afterwards). To substantiate the value of this lecture note for publication, it should have at least one section that reflects on some recent developments, rather than providing a review of material that has fully established a decade ago. Finally, I also find the quality of the last two sections (Secs. 8 and 9) where the author generalizes from Chern insulators to other topological insulators and to topological semimetals very unsatisfactory.

I thus cannot recommend the present version of the manuscript for publication.

With the above points clarified from the start, let me focus on how the author could improve his material to make it more appropriate for publication in SciPost Physics Lecture Notes.

First, and foremost, clear credit should be given to the source material that is being adapted in the individual sections. This could conveniently be done by inserting one or two sentences in the intro paragraph to each (sub)section. Without a proper referencing of the source material, the lecture notes cannot be published in a journal simply because of the issue of originality.

Second, at the end of several (sub)sections I was left with the feeling “…so what?”. What I mean is that sometimes a few-paragraph (and sometimes even a few-pages) long sections are not at all motivated, and their exposition is closed without explaining the need of the derived results in the context of future sections. Overall, I found that the cross-referencing between the individual sections (especially alerting the readers how the early sections would be relevant to the later developments) is rather weak and should be improved.

Finally, the quality of presentations especially in Secs. 8 and 9 must be significantly improved, providing more justifications to the presented statements, and to offer a clear outlooks and suggestions for further reading.

It also seems to me that the text refers to colored figures, while they are all grayscale.

With the main general comments summarized above, let me now list point-by-point the places in the exposition which I found unclear. I hope the list below would help the author improve the clarity of the submitted material.

Page 2: To “any variations of the Hamiltonian that do not close the bulk gap” add “and preserve the symmetry of the system”. (I find that this important narrative is explicitly spelled too late in the lecture notes: only on the bottom of page 77.)

Page 3: the author uses the notion of “nodal line” to describe the zero of electron’s wave function. I find this terminology confusing in the present context, as “nodal line” so frequently refers to degeneracy of energy bands in k-space. A substitute phrase is needed, e.g. “lines of zero of the wave function”, or some other appropriate alternative.

Page 5: In the list of further reading, the author includes the online book on vector bundles by Hatcher [9]. I suggest to also include here the older but still very resourceful reference on vector bundles by Milnor and Stasheff, which approaches the classification of vector bundles from a less topological and more algebraic viewpoint.

Page 6: Above Eq. (1.2), replace “cutting” by “selecting”.

Page 7—8: The whole Sec. 1.1.2 is rather confusing. The author states that the differentiation should be performed “inside the vector bundle”, but at the end of 1.1.2 it remains unexplained what this phrase means.

Page 9 (and afterwards): The author uses [see e.g. Eq. (1.14)] Greek letters both to indicate both the direction of the derivative (phi) as well as the coordinates of the tangent plane (alpha and beta). I find this confusing since they really indicate different degrees of freedom (vector in manifold coordinates in the first case, elements of the Lie algebra of the gauge group in the second case). It’s a coincidence that for the tangent bundle the range of both of these indices is the same –this does not generalize when going to the Bloch bundle. Perhaps using Latin vs. Greek for the two would make some of the expressions appear more natural.

Page 9: The exposition should foreshadow in some way the connection coefficients “omega” are not uniquely defined, but depends on the choice of coordinates (gauge). Also, perhaps add a comment on the singularity of the chosen coordinates at the south pole.

Page 10—11: It appears to me that the discussion in Sec. 1.2.3 based on the “complexification” of the tangent plane is adapted from “Michael V. Berry. A. Shapere, F. Wilczek. Geometric Phases in Physics, chapter: The Quatum Phase, Five Years After, pages 7–28. World Scientific, 1989.” And should perhaps be acknowledged as such.

Page 10: It might be useful to compare (1.19) to the result (1.18) of the previous subsection.

Page 12: Typo in “physical terms by by employing”.

Page 12: The author discusses the parallel transport of a gyroscope. This discussion is well written, but it would be very interesting to add a comment explaining the “torque” the produced the change of angular momentum of the gyroscope after it is transported on a closed loop.

Page 12: In the last sentence of Sec. 1.3.1, the author writes that “angular momentum is locally conserved inside the tangent plane”. The precise meaning of “locally conserved” here is not clear. What would be the complementary notion of a “globally conserved” quantity? I find the language rather imprecise.

Page 14: On the top of the page, replace in “2pi discontinuity in beta” by “ambiguity”.

Page 14: In last paragraph of 1.4.1, the author discusses how declaring another basis in the ambient to be constant changes the connection. I find this rather unnatural – in a Euclidean ambient space it seems canonical which bases are constant. Rather, as an alternative, one could consider embedding the sphere into the ambient space differently (we do not equip the sphere with metric – only with topology; thus we have this freedom), e.g. by represented is as elongated/squashed/folded etc. Then the various embeddings would naturally lead to different connections, without the artificial step of declaring some curvilinear basis of R^3 as “constant”.

Page 15: The author writes “recall the we constructed the basis sections {…} for TS^2 from velocities […]”, but I think Sec. 1.2.2 has introduced the basis sections differently.

Page 15: In the end of 1.4.2, I feel a bit uneasy with the author insisting on interpreting the connection in terms of a projection from ambient space. The connection is a rule for parallel transport that can be defined independently of embedding and metric (this is partially resoled in Sec. 1.4.3).

Page 16: The meaning of q = |e| is not entirely clear to me. Note that later at Eq. (3.2) on page 27 an even more confusing version of this equation q = e = |e| appears.

Page 17: In the last paragraph of 2.1.2, the author comments that as we seem to live in a world without magnetic monopoles, the usefulness of the geometric description of electromagnetism to a physicist is unclear. Besides being appalled by such a nonchalant dismay of the geometric picture, I would like to urge the author to have a look at the book by Baez [7], where a beautiful relation between wormholes in space-time-geometry and the appearance of “emergent” monopole is discussed for a model with no true “microscopic“ monopoles.

Page 19: The result of Exercise 2.2 looks confusing, but this is probably because the author does not treat vector fields as differential operators (which is the canonical choice in essentially all modern books on differential geometry). Shouldn’t the left-hand side should contains an additional term $\nabla_{[e_1,e_2}$?

Pages 23—24: I find the way the author jumps between the geometric description of tangent bundle vs. magnetic field too disturbing and confusing, e.g. I was completely thrown off by the discussion following Eq.(2.30). What is “m” in the magnetic context? Certainly not magnetization. Why is “m” sometimes typesetted in bold and sometimes italicized?

Pages 24—25: The purpose of Secs. 2.4.3 and 2.4.4 is left unexplained at their end.

Page 28—29: The author discusses left panel of Fig. 3.2 as if it described the normal of a cone (with smooth tip), although strictly speaking only the case of $Phi_0/2$ can be interpreted as a cone. Even more confusing is the right panel, where after much thinking I was not able to figure out the meaning of the arrows on the annuli. I am simply unable to compare the right panel of this figure to the text of Secs. 3.1.2 and 3.1.3. Where in the figure is the acceleration referred to in the text? Whatever the author is trying to explain here, the illustration clearly does not serve its purpose.

Page 31: What is $\alpha$ in Eq. (3.16)? I find the notations in this and the next two formulas unclear.

Page 33: Below the third table, add “We conclude that the *adiabatic* evolution of the […]”.

Pages 34—36: I find that this discussion is very reminiscent of the first part of the review paper by Xiao et al., https://doi.org/10.1103/RevModPhys.82.1959, which should thus be cited in this context.

Page 36: Below Eq. (3.28), add “where w is a *normalized* three-dimensional vector”.

Page 35: At the top of the page, when writing “Note that this expression is conceptually similar to […]”. It would be helpful to remind the reader that we have in mind the interpretation via complex-line bundle, as discussed in Sec. 1.2.3.

Page 36: Below Eq. (3.39), we again switch unexpectedly to the magnetic field analogy, without a clear warning. Note that comparing this equation to the “other” magnetic-case results in Eq. (2.10), we find different numerical pre-factors (2pi vs. 4pi). The author should be more precise with his magnetic-field analogies. In the present case, the confusion is amplified even more by introducing the skyrmion structure in Fig. 3.5, which borrows bits of pieces of both of the two magnetic analogies mentioned above. Reading Sec. 3.3.5 without getting lost must be difficult for newcomers in the field.

Page 37: The last paragraph of Sec. 3.3.5 has multiple typos; (1) in “electron form the path perspective” change “form” to “from”, (2) in “all possible path appears” add plural to “paths”, and (3) in “the decomposition of the the unitary” there is a duplicity.

Page 37: in Sec. 4.1.1, if the author aims to define circle via its global properties, he should complete the definition as the “*connected* one-dimensional space such that […]”, else his definition includes a disjoint union of arbitrarily many circles.

Page 39: The last paragraph of Sec. 4.1.2 seems to contain a manifestly incorrect statements, namely that the section singularities are point-like if the dimensionality of the base space coincides with that of the fiber. Besides not being clear how the dimensionality of vector space is measured (is C-plane 1D or 2D?), this is also wrong. A 2D Chern insulator with N-occupied bands will have 2N-real-dimensional (i.e. N-complex-dimensional) fiber, but a 2-real-dimensional base space. Still, there would be a point-like singularity in the section. I am not sure what the author aims to communicate here, but the paragraph needs to be removed or completely changed.

Pages 41—42: The paragraph “Note that if a complex line bundle has an odd Chern number […]” seems oddly placed, neither within the discussion of TS^2 nor within the paragraph about torus. Furthermore, it seems the author misses the mathematical toolbox to discuss these relations: the tangent bundle of orientable 2-manifolds supports complex structure, and allows for its description via the Chern class. Nevertheless, since the complex structure is an additional structure that needs to be defined for such a description, such tangent bundles are more commonly described by the Euler class (so-called because it equals to Euler characteristic of the original 2-manifold). See a discussion of this relation in Nakahara [8]. (Note that non-orientable manifold with odd Euler characteristic do not support complex structure on their tangent bundle.)

Page 43—44: The motivation to introduce the pullback construction is unclear at the beginning of the corresponding section. For its whole duration is it very unclear where we are headed.

Page 43: In the last line of Sec. 4.3.1, perhaps explicitly expand (to reveal the triviality) in the product structure $m_0^*TS^2 = S^2 \times T_{m_0} S^2$.

Page 43: In Sec. 4.3.2, it is unclear what $\mathcal{S}$ stands for – sometimes it is explicitly called “sphere”, at other occasions we treat as a general closed 2-manifold.

Page 44: I don’t understand footnote 4. The Chern class is an element of the second cohomology of the manifold with integer coefficients, $H^2(M,\mathbb{Z})$ (For example, 3-torus has $H^2(T^3,\mathbb{Z}) = \mathbb{Z}^3$ -- three weak Chern numbers), whereas the author suggests they are integrands to be used in some further mathematical manipulations. Mentioning the cohomology ring structure of total Chern class and Chern numbers seems to be way beyond the level of the manuscript.

Page 50: In Sec. 5.2.1 the author describes some experiment to measure polarization of non-polar material by squeezing it and then measuring the surface charges. However, I do not understand the setup the author has in mind (it also sounds more like a description of piezoelectricity rather than of polarization measurement). Perhaps an illustration would be helpful.

Page 51: Captions to both Figs. 5.2 and 5.3 refer to some coloring, but the illustrations are in grayscale.

Page 53: I find the mathematical manipulations in Sec. 5.3.2 too dense; e.g. getting Eq. (5.29) already requires several steps, and the way the exponent $e^{ik\tau_\alpha}$ is being absorbed in the various expressions is also rather hard to follow. I suggest the author to expand this derivation. Furthermore, below Eq. (5.29) add ”where we used equations from *Sec. 5.1.3* and Sec. 5.1.4.“

Page 54: In Eqs. (5.36) and (5.37) the author introduces two different connections. It would be appropriate to announce which one is used throughout the rest of the section and the lecture note, and what is the convention concerning the use of the “tilde” (above orbitals, Hamiltonian, connection,…) in the subsequent text.

Pages 56—57: The author writes “the composition of charge density in terms of Wannier functions contains more information than that based on Bloch waves”. I am wondering what happens to the decomposition in Eq. (5.40) resp. Eq. (5.45) if a different (k-dependent) gauge for $\left|\psi_k\right>$ is adopted in the starting point in Eq. (5.15). Is the decomposition in the later equations invariant? What about the illustrations in Fig. 5.5? Is the observation compatible with the statement about “more information in Wannier functions than in Bloch waves”?

Pages 60—61: There is appears to be some confusion about the use of “tilde” in the Equations. The text writes “Since the Hamiltonian is periodic in both variables […]” (meaning $p$ and $k$), but this only true for the “non-tilde” version of the Bloch Hamiltonian in Eq. (5.10), while the “tilde” Hamiltonian in Eq. (5.53) is not periodic in momentum. Then, however, the equations (6.2—4) contain the “tilde” connection and curvature. I find the discussion in these paragraphs unclear.

Pages 61: I am wondering if the discussion on this page could be an appropriate place to remind of the statement from the end of Sec. 4.2.1, that “the integral of the curvature of a connection has the same value for any connection on M”. Thus, although one connection might be better suited to relate to the Berry phase to polarization, both (“tilde” vs. “non-tilde”) connections will agree on the value of the Chern number in Eq. (6).

Page 63: Typo above Eq. (6.8). There is extra “the” in “a type at the both sides”.

Page 63—64: I find that the construction of the boundary-located eigenstate residing solely on the “a” sublattice would be clearer if accompanied by an illustration. Also, I am wondering if one could indicate the left/right-boundary localization by adding a color scheme to Fig. 6.4.

Page 67: When introducing the “mixed space”, clarify what $m_x$ resp. $m_y$ mean. This is especially confusing since in $m$ is also used in the earlier sections to describe magnetization and surface normals, while in the immediately following Eq. (7.1) $m_i$ denotes the chirality of an edge mode. Perhaps re-labelling of some quantities might be appropriate.

Page 68: When discussing that the presence of Chern number leads to the absence of exponentially localized Wannier functions, an appropriate reference should be added.

Page 68: Typo in first sentence of Sec. 7.1.2, change “Chen” to “Chern”.

Pages 69—70: It seems that Sec. 7.1.3 relates to Fig. 3.2. from an earlier section. It would be nice to make the relation to that figure (which I unfortunately find very unclear) more explicit. Furthermore, concerning Fig. 7.3 in the present section: Could the author add some axis/axes to the top panels? Finally, one very confusing aspect of the top panels in Fig. 7.3. is the $\cap$-shaped line. This does not indicate any non-analyticity or singularity, so what it?

Page 71: The bottom panel of Fig. 7.4. is of course contradictory if one insists that the momentum space is finite (due to the existence of finite-size unit cells in real space), since in that case the edge band has to connect to itself on the Brillouin zone boundary. To avoid this contradiction, one must either assume (1) infinite k-space and no real-space lattice (i.e. use linear dispersion instead of cosine dispersion), or (2) add to the illustration a projection of the bulk bands into the which the top and bottom of the edge mode sinks.

Page 72: In the line right below Eq. (7.7) the author speaks of “the bulk gap”. One should perhaps write more precisely “the energy gap between occupied and unoccupied bands” (or some equivalent), because, by assumption of Sec. 7.3.1 each pair of consecutive bands are separated by a bulk energy gap.

Page 72: In the bottom paragraph, the author refers to fragile topological insulators which are trivialized when the dimension of the Hilbert space changes. I think that the Hopf insulator [https://doi.org/10.1103/PhysRevLett.101.186805] would be a more appropriate example here for two reasons. First, the references fragile topological phases are protected by crystalline symmetry – crystalline topological insulators are beyond the presented lecture notes. Second, fragile topological insulators care whether the extra band is added to the conduction or to the valence subspace; more closely to the author’s statement on the present page, the Hopf insulator is trivialized by adding a band to either conduction or to the valence subspace, i.e. any enlargement of the Hilbert space would do. [If Hopf insulator is too complicated an example, a simpler one corresponds to the winding number of real-valued Hamiltonians in one dimension. Two-band models (expressed via a pair of real Pauli matrices) support Z-valued winding number in 1D, while three and more bands lead to a reduction of the integer to a Z2-valued Berry phase.]

Pages 73—74: Besides lacking a reference to a material where the reader could find more details, I find that the presented discussion is too encyclopedic: There are no explanations (just statements), not examples, and no application follows. The presentation here has to be significantly improved.

Page 74: I find the reference to maximally localized Wannier functions in the context of Chern insulators (which do not support exponentially localized Wannier functions) rather odd. I don’t understand what the author aims to accomplish by adding the last two paragraphs of Sec. 7.3.2.

Page 74: In the intro to Sec. 8, the author should add a more explicit explanation to the statement: “In fact, such crystals must break time-reversal symmetry, since otherwise there would be no chiral edge modes.”

Page 76: When arguing that “symmetry constraints require that any possible deformation is symmetric, and the loop always encloses half of the Bloch sphere”, it is not enough to consider the very simple “representative” Hamiltonian. Instead, one should show how the reflection symmetry relates H(k) to H(-k) by a $\pi$-rotation around the $\sigma_x$-axis.

Page 77: When discussing the end states of the polarized chain, the author correctly states that the $\pi$ Berry phase generally does not lead to an in-gap topological state. However, the modern understanding is that there nevertheless is a topologically robust signature associated with the non-trivial quantized Berry phase, namely the filling anomaly: The system with open boundaries exactly at half-filling cannot simultaneously (1) respect the mirror symmetry (having the same boundary condition on both ends) and (2) be an insulator (there will always be the possibility of a zero-energy excitation, or two ground states at the same energy). It would be appropriate if the author could supplement his discussion by this point of view that developed over the last couple of years.

Page 78: The author switches from “reflection” to “inversion” symmetry. I don’t think these two can be distinguished in a one-dimensional system, therefore I am wondering why the author suddenly changed the terminology.

Page 78: When writing Eq. (8.10), the author presumably uses the convention where the Hamiltonian is periodic in the momentum space [i.e., not the convention of Eq. (5.53)]. This should be emphasized, because Eq. (8.10) looks different in the two conventions.

Pages 83—84: I find the discussion of 3D Tis too brief, in particular, it is not clarified why the strong Z2 invariant (so-far in the exposition only constructed via inversion eigenvalues) should remain stable if the inversion symmetry is broken.

Page 84: When referring to the “periodic table” of TIs and to the very mathematical work by Kitaev [35], the reader of the lecture note could perhaps benefit more by looking also at the more pedagogically oriented review of the tenfold way classification offered by Andreas Ludwig [https://doi.org/10.1088/0031-8949/2015/T168/014001].

Pages 83—84: I find the three-paragraph summary of the topological quantum chemistry too brief too communicate usual information to the reader. Note that several additional notions need to be introduced for this discussion to be understood, e.g. how the real-space orbitals induce the various irreducible representations at high-symmetry points in k-space, and on the compatibility relations, etc. Perhaps adding an example would make the discussion more useful (and also add value to the submitted script by presenting a bit more on the modern developments in the field).

Page 90: The author does not explicitly comment on the situation “(1,3)”, which would correspond to a “nodal sheet” in a 3D momentum space, perhaps considering them unphysical. While not very common, such a situation can actually be topologically stabilized in certain superconductors (nodal surfaces are generic for centrosymmetric, time-reversal breaking, multi-orbital superconductors). For a discussion of them, see e.g. the work by Agterberg and coauthors [https://doi.org/10.1103/PhysRevLett.118.127001]. The codimension analysis is also presented at length in [https://doi.org/10.1103/PhysRevB.96.155105].

Pages 90—91: The discussion of Weyl points in Sec. 9.2.2 is again too brief, and the construction illustrated in Fig. 9.4. is practically non-existent – as if the author grew too tired of the writing. With the high complexity of Fig. 9.4, in particular the right panel, some clarification should be added to the meaning of the plentiful species of lines and shapes. Also, while this is by no means the most recent summary, the early review of Weyl semimetal physics by Qi and Hosur [https://doi.org/10.1016/j.crhy.2013.10.010] could be a nice addition for a “further reading”.

Page 91: In intro to Sec. 9.3., the phrase “all compatible Hamiltonians” is rather confusing. Does the author mean the vector bundle (i.e. collection the information from all k-points)compatible with the crystal symmetries? The phrase “space of compatible Hamiltonians” also appears below Eq. (9.21) on the next page, with the precise meaning still unclear.

Page 92: The argument showing how the non-symmorphic “glide” symmetry enforces the appearance of a band node is much older than Ref. [42], and dates back to the work by Zak and collaborators from 20 years ago, see e.g. [https://doi.org/10.1103/PhysRevB.59.5998].

Pages 92—94: Although I am familiar with the result of Ref. [43], I find the exposition in Sec. 9.3.2 way too obscure to follow. Without a clear argument, the presentation eventually concedes that if the plane bundle is non-orientable, then the Dirac points are not classified by a Z-valued winding number but only by a Z2-valued quantity (Berry phase), leading to the seemingly paradoxical possibility to annihilate a pair of Dirac points with opposite winding number. However, the lecture note does not offer any justification or conclusion to this. The fact that Dirac points in space-time-inversion symmetric models only carry a Z2 Berry phase is at present well established, and even understood via the language of vector bundles.
As I already commented in the context of tangent bundles to 2-manifolds [pages 41—42], the plane bundle admits a complex structure only if it is orientable (in which chase it is described by the Z-valued Euler class). If the plane-bundle is non-orientable, the complex structure cannot be defined, nevertheless a Z2-valued “second Stiefel-Whitney” obstruction persists, and the correspondence between the “2SW” class and the appearance of band nodes is at present known, see e.g. the recent work by Bohm-Jung Yang [https://doi.org/10.1103/PhysRevX.9.021013]. In particular, the peculiar topology of space-time-inversion symmetric models results in an unusual non-Abelian “braiding” of the Dirac points in k-space, which has been discussed by Bzdusek and collaborators, see for example [https://doi.org/10.1038/s41567-020-0967-9]. I think the author could benefit from reading both works.

Finally, the lecture notes misses some sort of “outlooks” and “what next”, where the interested reader could read more details on the various particular topics.
  • validity: ok
  • significance: low
  • originality: poor
  • clarity: good
  • formatting: excellent
  • grammar: excellent

Author:  Alexander Sergeev  on 2021-02-27  [id 1268]

(in reply to Report 2 on 2021-02-16)

I thank Referee for the comprehensive report. I am grateful for many specific suggestions of corrections, examples, and references. All of these will be taken into account during the preparation of the next version.

I would like to respond to the following points:

Page 10—11: It appears to me that the discussion in Sec. 1.2.3 based on the “complexification” of the tangent plane is adapted from “Michael V. Berry. A. Shapere, F. Wilczek. Geometric Phases in Physics, chapter: The Quatum Phase, Five Years After, pages 7–28. World Scientific, 1989.” And should perhaps be acknowledged as such.

In the lecture notes, I introduce a complex structure $J$ on a real rank 2 oriented vector bundle, following Frankel [6]. Each fiber $V$ is then isomorphic to a complex line $\mathbb C$ (once a single basis vector is chosen). In the suggested reference, one defines a vector $\vec \psi = \frac{1}{\sqrt{2}}(\vec e_1 +i\vec e_2)$, which is a linear combination of the real vectors with complex coefficients. As such, it is an element of $V\otimes_{\mathbb R}\mathbb C$, a complex two-dimensional space. It is not immediately clear to me how this space is related to the geometry of $TS^2$.

Page 14: On the top of the page, replace in “2pi discontinuity in beta” by “ambiguity”

The real function $\beta$ describes a gauge transformation $e^{i\beta(\gamma)}$ on a closed contour $\mathcal{C}$ parameterized by $\gamma \in [0, 1]$. Then necessarily $\beta(1) = \beta(0) + 2\pi n$, so the function $\beta$ can be discontinuous (of course, the corresponding map $\mathcal{C} \to U(1)$ is continuous). But $\beta$ is not ambiguous, as it is related to a given gauge transformation.

Page 19: The result of Exercise 2.2 looks confusing, but this is probably because the author does not treat vector fields as differential operators (which is the canonical choice in essentially all modern books on differential geometry). Shouldn’t the left-hand side contain an additional term $\nabla_{[\vec e_1, \vec e_2]}$?

In the language of differential geometry, the operator $\nabla_j$ defined in the notes corresponds to the covariant derivative along a coordinate vector field $\nabla_{\vec{ \partial_j}}$. The Lie bracket of such coordinate vector fields vanishes. I decided against introducing expressions like $\nabla_{\vec L} \vec v$ for the two reasons. First, in the context of a tangent bundle, it can be very confusing to use different frames to decompose $\vec L$ and $\vec v$ (and I need a coordinate frame for $\vec L$ and an orthonormal frame for $\vec v$). Second, in the theory of topological insulators, one would not describe the Berry potential as a linear functional on the tangent vectors to the Brillouin torus.

Pages 34—36: I find that this discussion is very reminiscent of the first part of the review paper by Xiao et al., https://doi.org/10.1103/RevModPhys.82.1959, which should thus be cited in this context.

In this discussion, I follow the original paper by M.V.Berry, as many other authors do.

Page 35: At the top of the page, when writing “Note that this expression is conceptually similar to […]”. It would be helpful to remind the reader that we have in mind the interpretation via complex-line bundle, as discussed in Sec. 1.2.3.

Please note that the complex structure does not play any role in the correspondence discussed here. Eqs. (1.16) and (3.31) both describe decompositions of vectors in some subspace in terms of the basis vectors of an ambient space.

Page 39: The last paragraph of Sec. 4.1.2 seems to contain a manifestly incorrect statements, namely that the section singularities are point-like if the dimensionality of the base space coincides with that of the fiber. Besides not being clear how the dimensionality of vector space is measured (is C-plane 1D or 2D?), this is also wrong. A 2D Chern insulator with N-occupied bands will have 2N-real-dimensional (i.e. N-complex-dimensional) fiber, but a 2-real-dimensional base space. Still, there would be a point-like singularity in the section. I am not sure what the author aims to communicate here, but the paragraph needs to be removed or completely changed.

Regarding real/complex dimensionality: the paragraph says "..if the dimensionality of the base space coincides with that of the fiber (as a real space)".

The statement is essentially the "co-dimensional argument" discussed in Sec.9.2.1 in more detail. A scalar function on a two-dimensional surface generically vanishes along some curves. A pair of such functions vanish simultaneously at the points of intersection of these curves, and such intersection cannot be removed by a small perturbation. However, an intersection of three and more curves in a single point is unstable, as one can move one of the curves away from the intersection. A section of a rank $N$ complex vector bundle on a 2D base is locally described by $2N$ real functions of two variables. The section has zero in a point where all these functions vanish. Thus, one can expect point-like zeros in the section for $N=1$, and there is no stable zeros for $N>1$.

I suppose that in the context of a multiband Chern insulator, one tends to consider the bundle of the occupied eigenstates as a direct sum of complex line bundles corresponding to the individual energy levels (at least, away from the band degeneracies). Each of this line bundles can indeed have point-like singularities. However, a point-like zero in a section of the whole valence bundle would require that such singularities in all valence bands are located over the same point of the base space. Since this configuration is not stable against the local perturbations, there is no contradiction with the statement made in the text.

Pages 56—57: The author writes “the decomposition of charge density in terms of Wannier functions contains more information than that based on Bloch waves”. I am wondering what happens to the decomposition in Eq. (5.40) resp. Eq. (5.45) if a different ($k$-dependent) gauge for $|\psi_k\rangle$ is adopted in the starting point in Eq. (5.15). Is the decomposition in the later equations invariant? What about the illustrations in Fig. 5.5? Is the observation compatible with the statement about “more information in Wannier functions than in Bloch waves”?

It seems that there are two separate questions: one about the gauge freedom and the other about "more information".

First, let $|\psi_k\rangle$ be an eigenstate of a $k$-periodic Hamiltonian. Two standard basis choices give the components $\psi_{\alpha k}$ and $u_{\alpha k}$, and the corresponding Berry potentials. Now, suppose that we define yet another basis, so $|\psi_k\rangle$ has components $\phi_{\alpha k}$. Then one should not expect the corresponding geometric phase $\int_{BZ} i\sum_\alpha \overline{\phi_{\alpha k}} \partial_k \phi_{\alpha k} dk$ to have any physical meaning. However, this observation does not affect the Eq. (5.40). It is basis-dependent by construction: it relates two specific geometric phases obtained in the certain bases.

As for the information, it should be clear that since Fourier transform is invertible, Wannier functions cannot contain more information than the Bloch waves. However, suppose that we take the modulus squared of each Wannier function / Bloch wave to compute the total charge density of a charge pump. Then one can find the current from the charge distribution of each individual Wannier function, but one cannot find it from the charge density of the Bloch waves.

Page 78: The author switches from “reflection” to “inversion” symmetry. I don’t think these two can be distinguished in a one-dimensional system, therefore I am wondering why the author suddenly changed the terminology.

For simplicity, the quantization of the Zak phase is considered for the model with two orbitals at the same atomic site, which are related by the reflection operation.

Page 91: In intro to Sec. 9.3., the phrase “all compatible Hamiltonians” is rather confusing. Does the author mean the vector bundle (i.e. collection the information from all k-points)compatible with the crystal symmetries? The phrase “space of compatible Hamiltonians” also appears below Eq. (9.21) on the next page, with the precise meaning still unclear.

The section 9.3 will be rewritten, but now I would like to clarify this point. For a two-band model, the Bloch Hamiltonian is described by a three-component vector field $\vec h(k)$ (ignoring the term proportional to the identity matrix). If the symmetry $\hat G$ is local in $k$, the matrix equation $G_k H_k G_k^{-1} = H_k$ specifies at a each $k$ a vector subspace of all $\vec h(k)$ compatible with the symmetry. Taken together, these subspaces form a vector bundle over the Brillouin zone, dubbed "the space of compatible Hamiltonians" in the notes. The non-trivial topology of this bundle provides a way to explain the band-crossing in the non-symmorphic chain and the results of Ref. [43].

---

## Round 1 · Referee Report · Anonymous (Referee 3) · 2021-3-13

Strengths

  1. Self contained and pedagocical presentation.
  2. mix of introduction to mathematical tools, and use of them in physics.
  3. original choice of subjects, and order of presentation.

Weaknesses

  1. very few references, especially on the core of the notes. No mention of previous lectures notes or books, no mention of original papers.
  2. physical discussion disconnected from experimental reality (no mention of what can / has been measured).
  3. Almost no mention of history of the field, or of the context.
  4. Various sections should be introduced. Transitions with previous sections are almost non-existent. This could be the occasion for the author to justify his choice of subjects, and order of presentation.

Report

This manuscript constitute lecture notes on the vector bundle approach to topological insulator, topological semimetals and topological pumping. While the subject is rather old, and most recent aspects of topological condensed matter are barely touched upon, there is always room for new pedagogical presentations, if original.
In this spirit, I appreciated the effort of the author, but I think that some major work is needed for these lecture notes to be published.

The public targeted by the author is undergraduate students or beginning graduate students in theoretical condensed matter. Hence he focuses on well known and old aspects of the field, but barely scratches the surface of most of the recent developments. This makes sense. On the other hand he develops a self-contained expose, focusing on pedagogically introducing the necessary technical tools. His approach is mostly original by the choice of subjects. But what is extremely perturbing is the absence of references in the text (as opposed to the introduction), whether to other lecture notes or to the original work. As such, a critical reader cannot establish what part of the presentation is original, and what part is following previous introductions to the field. Moreover, a reader entering the field through these notes is neither guided to the relevant original papers, not to mote specialized treatments of various topics. This is my main criticism against this manuscript, and I view it strong enough that I cannot recommend a publication of the manuscript at this level.

On the style of writing, I have found the manuscript lacking explanations of « the general picture » in this field. Transitions are often abrupt, motivations of different sections and subsections are unclear. The author should motivate more clearly his choices of presentation. Why does he first discuss topological pump before Chern insulators, why first polarization, etc. This will help to clarify the originality of this presentation, and help walk reader through the steps of this introduction. The part devoted to physics lacks any reference to experimental results…

A lot of the figures need to be improved, mostly so that they can be understood without resorting to definitions in the core of the text.

Also, I have found a substantial number of typos that can easily be corrected (e.g. many form <-> from).

Let me now question some of the choices of subject made by the author. Most of these questions can be answered by mentioning the relevant litterature.

  • The section 1-4 focus on mathematics for physicists.

The author chooses to resort to an intuitive approach rather than a rigorous one. This is a justifiable choice, but further references to more rigorous (or exact) presentations should be provided throughout the section. Example : the definition of the vector bundle is practical, but not standard in mathematical books, which the author acknowledges : «Such a description is too vague to be useful for mathematicians, but it will suffice for our needs ». I would suggest to cite, whenever relevant, references for the interested reader who wants to move beyond this presentation. Moreover, I have found rather awkward in a presentation on vector bundles to avoid the use of differential forms. This is a personal choice, see in particular eq. (2.13) and (2.14). The author should at least mention the simplification he uses, and cite relevant references on that aspect.

  • The section 6 deals with charge pumping. The author effectively considers periodically driven Hamiltonian in this pumping process. Those have been recently considered within the Floquet formalism as topological quantum evolutions. Even if the author does not follow this route, he should mention its existence.

  • The section 8, on role of symmetry, is very synthetic. Probably too synthetic and very foused on D=2. Most real materials are in D=3. I didn’t find a mention of Dirac surface state, for example. This section can be expanded.

  • The section 9 is also very short. In D=3 the author focuses on Weyl semi-metals and not Dirac. The difference between both is not even stated. There is no mention of topological semimetals as critical topological phases between 3D topological insulators, no mention of nodal lines semi-metals, of the the Nielsen-Ninomiya theorem, of symmetry constraints on numbers of Weyl points. Yet two specific situations (non-symmorphic symmetries and winding number in the absence of chiral symmetry) are discussed. This section can also be reworked.

Let me finally mention a list of more specific remarks for the author.

p.7, eq.(1.3) : use different notation for R^3

P.7 section 1.1.2 « We wish the differentation to be performed inside the vector bundle  » 

Well, in physics we always differentiate the fields. This raises the questions of differentiation of this fields over a curved space, which naturally translates into a differentiation in the vector fibers. This is a (technical) question which should presented as natural.

P.8, after (1.7) : the author alludes to the hairy ball theorem. It should be mentioned, and references added.

P.8 : sec. 1.2.1 : the notion of « ambiant space » is … intuitive ? But it should be clarified. What is assumed here ? I guess that we can define a flat (trivial) vector bundle in which the studied vector bundle is embedded.

P.9 :, after eq. (1.15), « radius vector for a point on the sphere » -> coordinates of a point on the sphere

p.12, sec. 1.3.2 : a figure with the Foucault pendulum and definitions of notations would be welcome.

P.19, and whole section 2.2 : P.25, sec. 2.4.4 : why do you suddenly consider curvature of a cone ? You should explain the motivations.

Section 3 « Geometry of quantum states »

You should stress from the start that you consider now a base space which is real space. The single sentence of introduction is clearly insufficient. Considering wave function (quantum states over real space) is a very specific case (wave functions are single valued), and conceptually different than quantum states over a space of external parameters (magnetic field, etc).

P.31 : I felt frustrated to end sec. 3.2 by reading this sentence « This phenomenon is known as the Aharonov-Bohm effect ». This is a bit short, especially in physics lecture notes. You should mention that this is a measurable quantity, cite the relevant experiments or textbooks, and historical papers.

P.32, sec. 3.3.2 : the whole section on adiabatic transport lacks references. I think that condition of adiabaticity should be discussed (what does slow variation of parameters means ?).

P.34, after eq.(3.6) : « We are mostly interested in eigenstates, so we set h_0=0 » You should mention that eigenstate don’t depend on h_0. Otherwise the argument is unclear.

Similarly, just before eq.(3.27), you mention « Sometimes it is convenient to consider the restriction to the unit sphere S^2 » . Well, no ! Eigenstates don’t depend on |h|. So it is not a matter of convenience. Eigenstates are parametrized by h/|h| . Providing the eigenstates from the start would help.

P.36, after eq.(3.40) : « In some situations, this electromagnetic interpretation of the Berry connection is taken literally by electrons themselves. » What do you mean ?

P.36, before Fig. 3.5. You mention the notion of texture, which hasn’t been introduced, and fits nicely in the framework you use of vector bundles. You should define it.

P.39. « Such bundle cannot be represented as a product space, and its sections cannot be described as functions on the base space. » I didn’t understand this statement. Sections lie in the vector fibers. They cannot be represented on the base space.

P.39, after Fig. 4.2 : « To make such an identification, one needs n global linearly independent sections. » Actually, a section provides locally a basis of the vector space. It corresponds to the n vectors you mention. I think that one needs a single section.

Section 5 : Modern Theory of electric polarization

Probably the section which is best written.

P.47, section 5.1 : you consider from the start a system with a and b orbitals, that are only coupled one to the other. Why this choice ? Is this necessary ? This effectively amounts to consider a system with a chiral symmetry, which is important when discussing topological features : you consider a particular class of topological insulators.

P.49, eq.(5.13) : you seem to count all distances in units of unit cell size. This should be stated (and clarified).

P.50 : Similarly you make a choice of origin of space. Again this must be explained. This becomes apparent in eq. (5.18).

P.50, eq.(5.19) : a minus sign prior to the j(a) is missing

P.54, after eq.(5.34) : « but this product is not well-defined. As discussed in Sec. 3.3.2, it depends on the choice of the “constant basis” » This connection is well defined ! It depends on a choice of basis, but its definition is unambiguous ! You shouldn’t associate both notions.

Section 6 on Charge pumping

Section 6.4.1 : you relate the stability of pumps to Chern number on space x parameter manifold. This is fine of course. Yet you have considered pumps with a chiral symmetry (A/B decompositions of lattice), following your discussion of polarization. While chiral symmetry is necessary to quantize Zak phases, it is absolutely not necessary for Chern number to exist. This is a source of confusion. You are restricting yourself to special topological pumps, and you should clarify this point.

P.65, sec. 6..4.2 « There is a fact that any two maps of the same degree are connected by a smooth deformation. » Should be rewritten.

P.66, Fig. 6.7 : the caption is way too synthetic. Define the different quantities / notations appearing in the figure. At it, It is not understandable.

p.66 ; You suddenly switch from pumps to Chern insulators (section 7). You should write an introduction to this section, with a substantial transition. Where are you going ? What’s the relation with previously encountered notions, etc.

P.72 : Surprisingly, a reference appears at the end of the page. This is most welcome, but lacking elsewhere !

Section 8 : this section becomes much more qualitative than the previous parts. References should be added.

P.75 : you again consider alpha and beta orbitals, coupled between themselves, hence a chiral symmetry. Moreover, from the start two situations should be consider : whether parity exchanges alpha and beta, or leaves each orbital invariant (two typical models here).

P.78 : « In our case,the matrix of inversion happens to coincide, up to a scalar factor, with the Hamiltonian matrix Hk⋆. Thus the eigenstates |ψk⋆⟩ have inversion eigenvalues »  Not at all ! This is a consequence of eq.(8.10). It always holds.

P.87, eq.(9.8) « These points are known as Dirac points because of the conical form of the band touchings, reminiscent of the linear dispersion relation of massless relativistic particles. » This is not only reminiscent of the Dirac equation, this is an exact Dirac equation of motion for the low energy electrons. Hence the name.

P.87, section 9.1.2 « Since this model of graphene includes only hopping between different sublattices, the Hamiltonian does not have terms proportional to σz » This is the chiral symmetry I mentioned several times ! Hence absence of sigma_z terms can be attributed to the presence of chiral symmetry (or alternatively the TI combination).

P.89 : Weyl semimetals are 3D generalization of the 2D Dirac equation in that they correspond to 2 band crossing. But they differ from them at the fundamental level : they do not correspond to a Dirac equation. A pair of them of opposite chirality constitute a Dirac equation. A clarification on this point should be stated.

P.90 : paragraph sec. 9.2.2 is really too qualitative.

P.91 : again, a transition between sec.9.3 and the previous discussion is necessary. Also, the title of section 9.3 is misleading : are you still focusing on relativistic semi-metals ?

You then discuss some very particular cases of local symmetry constraints. This appears to me to be a personal choice, rather than a pedagogical one. You should motivate it.

Requested changes

See report.

  • validity: good
  • significance: ok
  • originality: ok
  • clarity: high
  • formatting: reasonable
  • grammar: reasonable

Author:  Alexander Sergeev  on 2021-04-09  [id 1348]

(in reply to Report 3 on 2021-03-13)

I am grateful to the Referee for the informative report. I appreciate the suggested corrections, as well as the general comments on the structure and presentation.

I would like to reply to the following remarks regarding the chiral symmetry:

P.47, section 5.1: you consider from the start a system with a and b orbitals, that are only coupled one to the other. Why this choice? Is this necessary? This effectively amounts to consider a system with a chiral symmetry, which is important when discussing topological features: you consider a particular class of topological insulators.

Please note that the model also includes the staggered on-site potentials. These contribute the term proportional to $\sigma_z$ in the Hamiltonian, thus the model does not have the chiral symmetry.

Section 6.4.1 : you relate the stability of pumps to Chern number on space x parameter manifold. This is fine of course. Yet you have considered pumps with a chiral symmetry (A/B decompositions of lattice), following your discussion of polarization. While chiral symmetry is necessary to quantize Zak phases, it is absolutely not necessary for Chern number to exist. This is a source of confusion. You are restricting yourself to special topological pumps, and you should clarify this point.

The charge pumping is accompanied by a continuous variation of the position of the Wannier charge center, which is given by the Zak phase. The chiral symmetry leads to the quantization of the Zak phase, hence the charge pump cannot have the chiral symmetry for all values of the pumping parameter.

P.75 : you again consider alpha and beta orbitals, coupled between themselves, hence a chiral symmetry. Moreover, from the start two situations should be considered: whether parity exchanges alpha and beta, or leaves each orbital invariant (two typical models here).

Here, the source of quantization of the Zak phase is a spatial symmetry, and not the chiral symmetry. The two representative Hamiltonians indeed happen to have chiral symmetry. However, general Hamiltonians obtained from them by smooth deformations preserving the spatial symmetry can include the terms proportional to $\sigma_z$, and thus break the chiral symmetry.

---

## Round 2 · Referee Report · David Carpentier (Referee 4) · 2022-11-29

Strengths

1. clarity of the discussion of the technical tools necessary to understand the simplest topological states of matter
2. wide range of technical aspects of this description.
3. accessible to a broad range of graduate students. The necessary background is minimal.

Weaknesses

1. does not cover the developments in the field of the last 15 years, i.e. the interplay between topology and symmetries.
2. Minor weakness: lack of mathematical rigor in some parts

Report

The manuscript has been largely improved after the initial review process. I believed that it now constitutes a very useful introduction to the basic aspects of topological states of matter. While there exists other documents (textbooks and review) that overlap with the present lectures notes, I am convinced that the variety of viewpoints is always beneficial to the community. I recommend the publication of this notes in their current state.

---

## Round 2 · Author Response

Dear Editor,

Please consider this re-submission of the manuscript "Topological insulators and geometry of vector bundles" for publication in the SciPost Physics Lecture Notes journal.

I am deeply grateful to the Referees for their numerous comments and recommendations, which showed many ways the notes can be improved. I apologize for the long time it took to prepare the new version. Most parts of the text have been heavily edited, some have been re-organized, and also there are several new topics (the additions mostly follow the suggestions). A detailed list of changes is given below.

One of the general changes aims to improve the connectivity and integrity of the text. A number of internal references have been added, both to the past and to the future sections. In many places, I have added or expanded introductory and concluding paragraphs, which explain motivation behind each section and its connection with other topics. Certain concepts and constructions are now listed as equations for easier referencing. After some exercises, there is a list of links to other exercises or sections that refer to this exercise.

Also, I have modified the choice of representative models to highlight the unity of the topological band theory. For example, a tight-binding model of a charge pump is introduced and extensively studied in Sec. 7. Then, this model is considered in the section on Chern insulators. Finally, a periodic evolution of the same system, which includes the closing of the bulk gap, gives a tight-binding model of a Weyl semimetal in Sec. 10.

Another major overall change is related to the context of discussion. I have added multiple paragraphs on the relevant experimental results and possible generalizations of the concepts discussed, as well as some historical remarks. Many of such paragraphs are collected in the new "Summary and outlook" sections. For example, these sections now contain all comments on the multi-band case, making the main text more focused on the two-band models.

Sec. 8 on Chern insulators and Sec. 9 on symmetry are re-organized around the concept of topological equivalence and the corresponding classification. This allows to put the material into the context of a general problem. Hopefully, this will give the reader a framework, which can help with navigating the original research articles. In particular, Sec. 9 starts with a formulation of the classification problem (9.2), which often underlies research works, but is rarely stated explicitly.

I have added numerous references to the literature, including:

  • Key original works (or reviews by the same authors).

  • Review articles, tutorials, and textbooks.

  • Relevant experimental results.

  • Historical studies and memoirs.

These references aim to guide an interested reader to additional information and provide further context.

There are also some new references to the sources of specific derivations. However, I have not inserted such references for all (sub)sections, as it was suggested by one of the Referees. In my view, the referencing in the lecture notes should follow the style adopted in textbooks, and not that of research articles. The material is often modified or does not follow any particular source, which makes such "local referencing" rather difficult. On the global level, the sources are listed in the "Sources and further reading" section in the Preface.

I decided against providing solutions of the exercises, for the following reasons. If the reader cannot solve an exercise immediately, it gives a good motivation to revisit the previous material or to consult other sources. A solution will provide an irresistible option of simply reading it and moving on without the firm understanding of the past material. On the other hand, the answers for the most important exercises are given, either implicitly or even explicitly, in the later sections.

I would also like to comment on Sec. 10.4 (former 9.3), which raised several concerns. This section provides a geometric interpretation of certain well-established results. I believe that it can serve as an interesting and simple illustration of the wide applicability of language of vector bundles, and so it is an appropriate part of these lecture notes. Unfortunately, the idea was poorly presented in the first version of the text. I hope that the new version gives a more clear and motivated exposition.

Please find the list of main changes below. Boldface numbers of subsections indicate substantial changes or new material. For the changes directly suggested in the reports, the number of the report is given in parentheses:

R1: Report of 2021-01-17

R2: Report of 2021-02-16

R3: Report of 2021-03-13

---

## Round 2 · List of Changes

### Preface

Mostly rewritten.

### 1 Connection on a vector bundle

New figures: 1.2 (left (R3)), 1.5, 1.6 (left), 1.7

New exercises: 1.2, 1.3, 1.4

**1.1.1**: rewritten with an emphasis on tangent vectors. Added distinction between local and global sections (R1). Added the definition of ambient bundle (R3).

**1.1.2**: new section introducing bundle metric and standard basis choices (R1)

1.1.3: added the construction of constant section of TR2 based on linear structure (instead of that based on angular momentum conservation) (R1)

1.2: switched to covariant derivative along an arbitrary (non-coordinate) curve (R1), added a remark on covariance (R3)

1.2.2: added a remark on base space / fiber indices (R2)

1.2.3: added details on complex structure

1.3: added the geometric picture of the parallel transport ("no in-plane rotation")

1.3.1: removed discussion of conservation law of angular momentum and incorrect statement about constrained gyroscopes (R1)

1.3.1: added discussion of parallel transport in TS2 along great circles

**1.3.2**: new section on the Foucault pendulum with a remark on inertial navigation systems based on constrained gyroscopes (R1, R2)

1.3.3: improved explanation of 2πn shift of Δα (R2)

1.4.1: extended discussion of ωτ as an angular velocity

Outlook: new comments on mathematical generalizations, added references (R1).

### 2 Electromagnetic field and curvature of connection

New figures: 2.7

New exercises: 2.3, 2.5

**2.1.1**: extended physical ("algebraic") discussion of gauge invariance, which is then contrasted with the geometric picture in 2.1.2 (R1)

2.1.2: improved description of the electromagnetic interpretation of plane bundles

2.2.1: added the definition of the curvature component (R1)

2.2.2: added a remark on covariantly constant vector fields on a sphere

2.2.3: improved description of the construction "vector field map" (R2)

2.3.3: added a remark on covariantly constant vector fields on a cone

Outlook: non-Abelian gauge fields, formalism of differential forms (R3), history of gauge invariance.

### 3 Geometry of quantum states

New figure: 3.5

3: emphasized the difference between wave functions over the real space and eigenstate bundles over a parameter space (R3)

3.1.1: added a paragraph on persistent currents

3.1.1: extended physical and geometric description of particle on a ring (R1)

**3.1.2, 3.1.3**: rewritten description of geometric construction (R2)

3.1.3: added emphasis on the need for the vector bundle picture (R1)

3.2.1: added a reference for the calculation of the interference pattern (R1)

3.2.2: added a paragraph on Aharonov-Bohm experiments (R3), mentioned synthetic gauge fields

3.3.3: added details on the spectrum and eigenstates of a two-level system (R3)

3.3.5: added a reference for magnetic skyrmions

3.3.5: extended discussion of emergent electrodynamics with an emphasis on the mixture of two geometric pictures (R1, R2, R3)

Outlook: Aharonov-Anandan phase, Zak phase, Wilczek-Zee phase.

### 4 Topology of vector bundles

New figures: 4.4, 4.7

New exercises: 4.1

**4.1.1**: rewritten with an emphasis on continuous functions

4.2.1: mentioned that c(M) is the first Chern number (R1)

**4.2.3**: new section discussing invariance of the Chern number under deformations

4.3.1: extended motivation and description of the pullback construction, added a figure (R2)

4.3.2: removed the mention of Whitney sum formula and a confusing footnote (R1, R2)

**4.4**: new section on the formalism of equivalence classes and topological classifications (preparation for charge pumps and Chern insulators)

Outlook: links to mathematical literature, Chern-Weil construction, idea of universal bundle, tautological bundles over projective spaces.

### 5 Tight-binding models and Bloch theory

This is a collection of several technical subsections, together with the discussion of graphene (all moved from other sections).

New figures: 5.2 (middle, right (R1))

New exercises: 5.1

5.1.3, 5.1.4: added remarks on the longer hopping amplitudes and the unit lattice constant (R3)

5.2.3: edited the proof of the Kramers theorem (R1)

### 6 Modern theory of electric polarization

New figures: 6.1

New exercises: 6.2

**6.1.1**: rewritten discussion of classical polarization, added a figure (R1)

6.1.2: replaced protocols in Fig. 6.3 with the full pumping cycles (R1)

6.2.1: added a remark on the gauge dependence of Wannier functions

6.2.2: expanded the derivation of the Zak phase (R2)

6.4.2: extended discussion of gauge-dependence (R1)

Outlook: Wannier functions and gauge freedom, multi-band case, multipole moments.

### 7 Charge pumping and topology

New figures: 7.3, 7.5, 7.6, 7.7, 7.8, 7.9, 7.10

New exercises: 7.1, 7.2

7.1.1: altered the continuous pumping protocol

**7.1.2**: added discussion of the charge quantization and of the choice of connection (R2)

7.2.1: added discussion of avoided level crossing

**7.2.2:** rewritten

* new introduction with an emphasis on what happens with the Bloch theory description once periodicity is broken

* added numerical spectrum for the continuous pump (Fig. 7.6)

* added an illustration of the accuracy of the ansatz (Fig. 7.7)

* added a plot of average positions of states illustrating localization of the end states (Fig. 7.8)

7.3.1: extended discussion of stability of end branches

**7.3.2**: new discussion of classification of pumps in periodic and finite settings

7.4: extended caption and discussion of Fig. 7.11 (R3)

Outlook: multi-band Chern number, Floquet theory (R3), experiments with cold atoms.

### 8 Chern insulators

This section is re-organized around the concept of topological equivalence of two-band Hamiltonians, which is studied in one, two, and three spatial dimensions. Qualitative discussion of the multi-band case is removed (R2).

New figures: 8.1, 8.3, 8.5, 8.7, 8.8

New exercises: 8.1, 8.2, 8.3, 8.4, 8.5

**8.1**: new section introducing the topological equivalence and corresponding classification problem (R3)

8.2.1: added the Bloch Hamiltonian of a Chern insulator derived from the continuous charge pump introduced in Sec. 7, and a plot of the corresponding vector field

8.2.2.: added a reference for the exponential localization of Wannier functions (R2)

8.2.3: specified an edge in the definition of nc (R1); mentioned local Chern marker

Ex. 8.2: new exercise about a boundary between two Chern insulators (R1)

**8.2.4**: rewritten the section on the topological phase transition (as a preparation for Weyl semimetals)

8.3: moved Haldane model here from the section on semimetals (R1)

**8.3.1**: added plots and discussion for edge states and Zak phase in graphene (R1), boron nitride, and Haldane model

8.3.2: edited edge spectra in Fig. 8.7 (R2)

**8.4**: new section introducing 3D Chern insulators and Hopf insulators (R2)

Outlook: multiple bands, Hopf insulator vs. K-theory; continuum limit; equatorial waves; experiments with amorphous lattices of gyroscopes; QAH experiments, including twisted bilayer graphene.

### 9 Role of symmetry

This section mostly consists of new material.

New figures: 9.2

New exercises: 9.1, 9.2, 9.3

9.1: added a remark on the other type of inversion, which preserves sublattices (R3)

**9.1.2**: new discussion of the interplay between gauge freedom and inversion symmetry

**9.1.3**: new section on symmetry-adapted Wannier functions

**9.2.1**: new derivation of the quantization of the Zak phase without using reflection symmetry (R2). Added discussion of the Zak phase quantization in graphene

9.2.2: extended discussion of polarization in periodic vs. finite crystals (R1), added a reference for the filling anomaly (R2)

**9.3**: new section on topological quantum chemistry (R1, R2)

**9.4**: edited to make better contact with the previous discussion (R1). Clarified the role of inversion symmetry. Added notes on experiments. Reduced discussion of 3DTI to a single paragraph.

Outlook: chiral symmetry (R3), comparison of inversion-symmetric chain with SSH model; periodic table for internal symmetries (R2); many bands in TQC, fragile topology (R1).

### 10 Topological semimetals

New figures: 10.2, 10.3 (left)

New exercises: 10.1, 10.2

**10.2**: new section on nodal line semimetals and drumhead surface states (R3)

**10.3.1**: new discussion of Weyl semimetals including

* a tight-binding model derived from the model for Chern insulator used before (R2)

* linearization of the Hamiltonian near a Weyl point (R1)

* Nielsen-Ninomiya theorem and symmetry constraints on Weyl points (R3)

* examples of inversion- and time-reversal-symmetric Weyl semimetals

10.3.2: added a link to the surface states of 3D Chern insulators

10.3.2: added discussion of experiments

**10.4**: extended introduction and motivation (R1, R2, R3)

10.4.2: corrected the definition of non-symmorphic symmetry group (R1)

Outlook: dimensional hierarchy; Berry phase detecting the nodal points; topology of real vector bundles; Dirac semimetals (R3); chiral anomaly in Weyl semimetals.

---

## Editorial Decision

published